# VOCSIM: A Training-free Benchmark for Zero-shot Content Identity in Single-source Audio

**Maris Basha** [1]  **Anja Zai** [1]  **Sabine Stoll** [2]  **Richard Hahnloser** [1]

## Abstract

General-purpose audio representations aim to map acoustically variable instances of the same event to nearby points, resolving content identity in a zero-shot setting. Unlike supervised classification benchmarks that measure adaptability via parameter updates, we introduce VOCSIM, a training-free benchmark probing the intrinsic geometric alignment of frozen embeddings, with no parameters updated and no labels used (a label-free PCA whitening is fit per subset to correct anisotropy). VOCSIM aggregates 125k single-source clips from 19 corpora spanning human speech, animal vocalizations, and environmental sounds, isolating content representation from source separation (polyphonic mixtures are out of scope). We evaluate embeddings with Precision@k for local purity and the Global Separation Rate (GSR) for point-wise class separation, calibrated by lift over an empirical permutation baseline. A simple pipeline of frozen Whisper features, time–frequency pooling, and label-free PCA yields strong zero-shot performance with stable GSR rankings across domains (Kendall's $\tau = 0.60$). However, on blind low-resource speech (Shipibo-Conibo, Chintang), local retrieval collapses while remaining above chance, exposing a cross-lingual speech generalization gap. As external validation, our top embeddings predict avian perceptual similarity, improve bioacoustic classification, and achieve state-of-the-art on the HEAR benchmark. We release data, code, and a public leaderboard.

[1]Institute of Neuroinformatics, University of Zurich and ETH Zurich, Zurich, Switzerland [2]Institute for the Interdisciplinary Study of Language Evolution, University of Zurich, Zurich, Switzerland. Correspondence to: Richard Hahnloser <rich@ini.ethz.ch>.

*Proceedings of the 43rd International Conference on Machine Learning*, Seoul, South Korea. PMLR 306, 2026. Copyright 2026 by the author(s).

## 1. Introduction

The ability to judge similarity between arbitrary sounds underpins fundamental behaviors in biological and artificial systems, from distinguishing phonetic contrasts in infancy (Kuhl, 2004) to song imitation in birds (Tchernichovski et al., 2001). Biological auditory systems achieve this with remarkable robustness by extracting a stable **content identity**, a core acoustic signature that defines a sound event (e.g., a specific phoneme, word, or bird call), while generalizing across nuisance variations such as speed, speaker identity, loudness, and recording conditions.

A core challenge for general audio intelligence is to develop embeddings that intrinsically organize sounds by their content identity without task-specific supervision. While foundation models like Whisper (Radford et al., 2022) and Wav2Vec 2.0 (Baevski et al., 2020) have achieved broad success, current evaluation paradigms focus primarily on *adaptability*. Benchmarks such as HEAR (Turian et al., 2022) and SUPERB (Yang et al., 2021) measure performance after training linear probes or fine-tuning parameters. This approach introduces a confounding variable: does a high score reflect the intrinsic quality of the representation or the effectiveness of the optimization recipe? Furthermore, relying on probing leaves no room to evaluate the representation's readiness for immediate, training-free deployment in retrieval or indexing tasks.

To fill this gap, we introduce **VOCSIM** (Figure 1), a benchmark designed to diagnose the intrinsic **zero-shot content identity** of frozen audio embeddings. VOCSIM aggregates 125,382 single-source clips from 19 corpora, spanning human speech, animal vocalizations, and environmental sounds.

**Generalization via Aggregation.** Validating true out-of-distribution (OOD) generalization requires testing models against a diverse array of recording conditions and acoustic environments. A benchmark constructed from a single data collection effort would inherently share specific channel characteristics (e.g., microphone type, room acoustics), allowing models to overfit to those latent variables. By aggregating 19 distinct corpora, VOCSIM forces models to generalize across uncorrelated nuisance variables, follow-

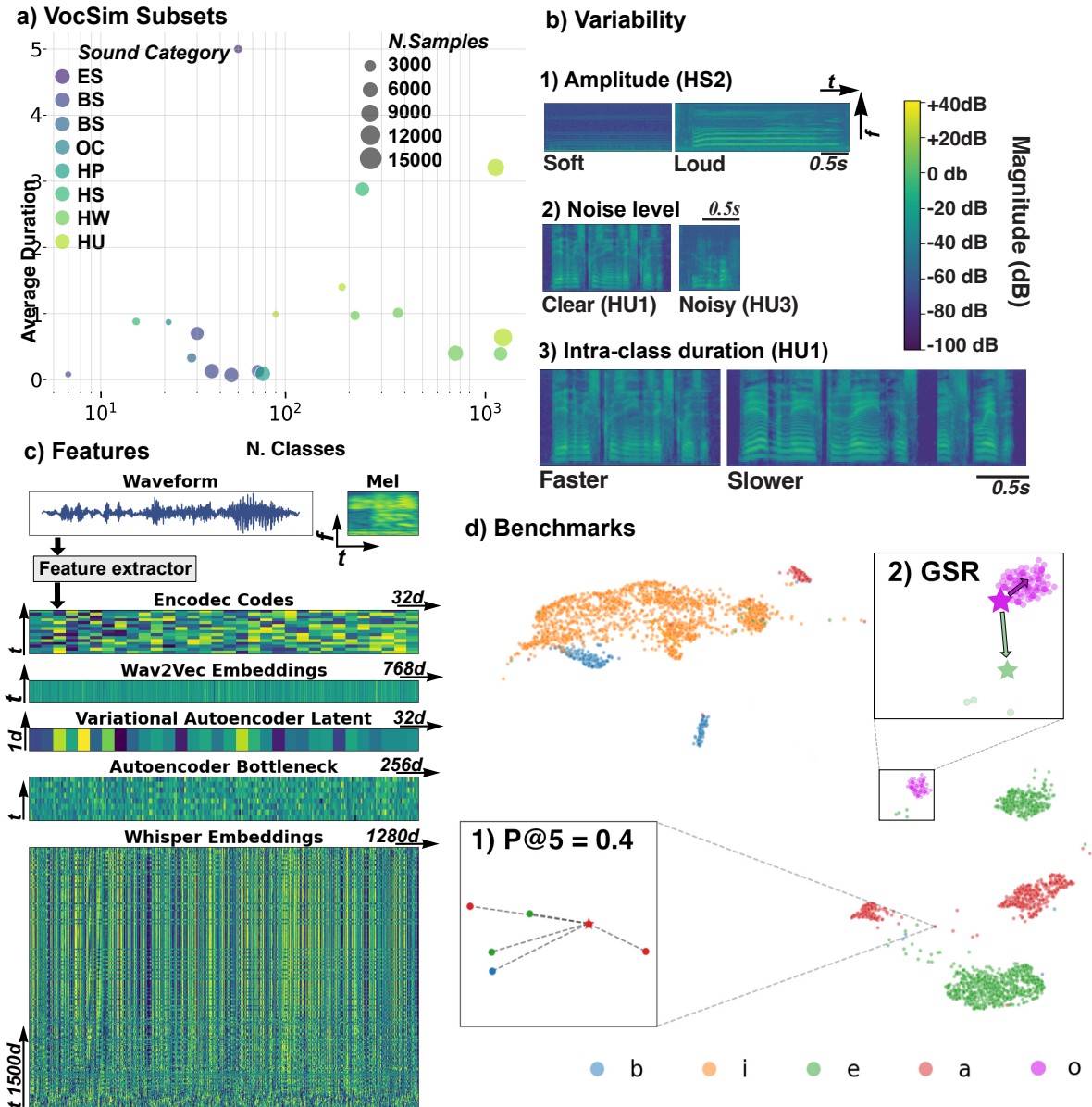

Figure 1. **Overview of the VOCSIM benchmark. (a) Dataset Composition:** VOCSIM aggregates 19 single-source corpora, with dot size indicating sample count. **(b) Acoustic Variability:** Spectrograms illustrate the engineered variation (duration, noise, production mechanism) used to stress-test generalization. **(c) Feature Representations:** We evaluate diverse frozen encoders, deriving fixed-length vectors via time-frequency pooling. **(d) Zero-Shot Metrics:** We use two training-free metrics: **Precision@k (P@k)** measures local neighborhood purity, while the **Global Separation Rate (GSR)** provides a point-wise measure of class separation by comparing each point's average intra-class distance (Avg_ID) to its nearest inter-class distance (NID). The UMAP projection illustrates distinct class clusters (colors) in the embedding space.

ing the standard set by generalization benchmarks in NLP (GLUE(Wang et al., 2019), MTEB (Muennighoff et al., 2023)) and Vision (VTAB (Zhai et al., 2020)).

**Isolating Content from Scene Analysis.** VOCSIM explicitly restricts evaluation to **single-source** audio. Just as computer vision benchmarks distinguish object classification (ImageNet (Deng et al., 2009)) from scene segmentation (COCO (Lin et al., 2015)), we isolate the geometry of con-

tent representation from the distinct problem of source separation. This geometric isolation is critical: in a zero-shot retrieval setting, evaluating polyphonic mixtures conflates the quality of the embedding with the model's ability to disentangle overlapping signals. By focusing on single-source content, VOCSIM provides a precise diagnostic of the representation itself.

**Defining the Zero-Shot Protocol.** We adhere to a **gradient-**

**free, label-free, transductive** definition of zero-shot evaluation: no parameters are updated via backpropagation, and no labels are used at any stage. However, raw embedding spaces from foundation models often exhibit high anisotropy (the "representation cone" problem). To evaluate the intrinsic geometry fairly, we apply **transductive whitening** via label-free PCA fitted on the statistics of *each evaluation subset independently* (not pooled across subsets). This is a standard non-parametric normalization technique in information retrieval (Song et al., 2018). This transductive step means our evaluation is *not* strictly single-sample plug-and-play inference; it is a label-free per-subset calibration that corrects known foundation-model anisotropy, allowing us to diagnose the **potential** of the intrinsic geometry. We report results both with and without PCA so that the strict-zero-shot reading is also directly recoverable from the tables.

Our extensive benchmarking reveals a simple, effective recipe: features from a frozen Whisper encoder, refined with time–frequency pooling and label-free PCA, yield strong zero-shot performance. However, VOCSIM also uncovers a critical **generalization gap**. While models perform well on public benchmarks, performance degrades substantially on our blind, low-resource speech test sets. We posit that the geometric alignment measured by VOCSIM serves as a strong proxy for utility in a wide range of downstream applications, including those not explicitly listed in this benchmark.

Our contributions are:

1. **VOCSIM:** A training-free benchmark for intrinsic audio geometry on 125k single-source clips.

2. **A new metric:** A point-wise Global Separation Rate (GSR) with permutation-based calibration to quantify lift-over-chance.

3. **Quantified Cross-Lingual Speech OOD Gap:** We demonstrate that on blind low-resource speech corpora (Shipibo-Conibo, Chintang), current SOTA models organize classes only marginally better than random chance. This conclusion pertains specifically to cross-lingual speech OOD, not to general audio.

4. **External Validation:** We show that VOCSIM performance predicts downstream utility: our top configuration achieves **state-of-the-art results on the HEAR benchmark** and accurately predicts avian perceptual similarity judgments.

## 2. Related Work

**Audio Evaluation Benchmarks.** The standard for evaluating general-purpose audio representations has largely focused on *task-specific adaptability*. Benchmarks such as HEAR (Turian et al., 2022) and SUPERB (Yang et al., 2021) evaluate models by training linear probes or fine-tuning parameters on a suite of downstream tasks (e.g., ASR, classification). While valuable for measuring transfer learning capacity, these benchmarks do not assess the intrinsic geometry of the frozen embedding space itself. In the bioacoustic domain, benchmarks like BirdSet (Rauch et al., 2025) and BIRB (Hamer et al., 2023) focus on robust detection and classification in complex, polyphonic soundscapes under domain shift. Similarly, challenges like DCASE (Mesaros et al., 2018) and datasets like AudioSet (Gemmeke et al., 2017) target scene analysis and event detection in multi-source recordings. In contrast, VOCSIM specifically isolates **single-source content identity**. By excluding polyphony and requiring no parameter updates, VOCSIM serves as a diagnostic for the foundational geometric alignment of representations, complementary to adaptability benchmarks.

**Acoustic Word Embeddings.** Prior to large-scale pre-training, significant research focused on learning fixed-dimensional Acoustic Word Embeddings (AWEs) for query-by-example search (Settle & Livescu, 2016; Kamper et al., 2016; Lim et al., 2018). While these works directly address content identity, they typically rely on specific supervision (e.g., word pairs) or limited domains. VOCSIM expands this evaluation to a zero-shot foundation model setting across biological and environmental domains.

**Foundation Audio Models.** The landscape of audio encoders has expanded rapidly. Self-supervised learning (SSL) models trained on raw waveforms, such as Wav2Vec 2.0 (Baevski et al., 2020), HuBERT (Hsu et al., 2021), and WavLM (Chen et al., 2022), learn via masked prediction or contrastive objectives. Spectrogram-based Transformers, including BEATs (Chen et al., 2023), AudioMAE (Huang et al., 2023), and EAT (Chen et al., 2024), utilize masked modeling on audio patches. Weakly supervised foundation models like Whisper (Radford et al., 2022) (trained on 680k hours of labeled audio) and CLAP (Elizalde et al., 2022) (trained on audio-text pairs) set new standards for robustness. While these models excel at their pre-training objectives (e.g., ASR, tagging), their zero-shot utility for generalized similarity retrieval, particularly on out-of-distribution (OOD) non-speech data, remains under-explored compared to their supervised performance.

**Zero-Shot Retrieval Evaluation.** Probing the intrinsic geometry of frozen embeddings is an established paradigm in other modalities. In Natural Language Processing, the Massive Text Embedding Benchmark (MTEB) (Muennighoff et al., 2023) evaluates off-the-shelf embeddings on retrieval and clustering. In Computer Vision, benchmarks like GeneCIS (Vaze et al., 2023) similarly test zero-shot generalization of attribute identity. VOCSIM brings this rigor to the audio domain. Crucially, studies on **representation anisotropy** (Ethayarajh, 2019; Li et al., 2020) have revealed that foundation model embeddings often occupy a narrow

cone, limiting their discriminative power. We adopt a **trans-ductive zero-shot** framework (Song et al., 2018): by employing label-free PCA to mitigate anisotropy, we utilize the test set's structure to diagnose the representation's readiness for retrieval without violating the constraint of zero-label supervision.

## 3. The VOCSIM Dataset

VOCSIM is a large-scale benchmark of 125,382 audio clips designed to evaluate zero-shot content identity. Its primary goal is to stress-test generalization by aggregating 19 distinct corpora spanning three production domains: human speech, animal vocalizations, and environmental sounds.

**Design Principles.** Our construction is guided by three principles:

1. **Isolation of Content:** We restrict the benchmark strictly to **single-source** recordings. This decouples the evaluation of representation quality from the distinct challenge of source separation. Consequently, we exclude polyphonic music and complex auditory scenes, as judging similarity in mixtures requires disentangling overlapping sources, a confound that obscures the intrinsic geometry of the embedding.

2. **Acoustic-to-Label Fidelity:** We require labels that denote specific acoustic units (e.g., phonemes, syllables) rather than abstract semantic tags. Crucially, this necessitates **precise temporal segmentation**: we select corpora with accurate start/end timestamps (e.g., forced alignments, hand-segmented boundaries) to ensure the embedding represents the target event rather than surrounding silence or background noise.

3. **Stress-Testing Generalization:** Aggregating diverse corpora introduces variability along four axes: duration (50ms to 5s), granularity (phones vs. utterances), production mechanism (human vocal tract vs. avian syrinx), and recording conditions (studio vs. field).

**Dataset Composition.** The benchmark comprises 19 subsets (Table 1). *Human Speech* includes phones, words, and utterances from standard corpora (LibriSpeech, VCTK, AMI) and vocal imitations. *Animal Vocalizations* cover six songbird corpora (Zebra finch, Bengalese finch, Canary) and Giant Otter calls, labeled by individual to preserve acoustic identity. *Environmental Sounds* are represented by ESC-50.

**The Blind Test Sets (Strict OOD).** A critical component of VOCSIM is the inclusion of held-out, non-public data to test true out-of-distribution (OOD) generalization. We reserve field recordings from two low-resource languages, **Shipibo-Conibo** and **Chintang** (Stoll & Bickel, 2013). As these datasets are not scraped from the web, they allow us to strictly evaluate models on data they have never seen during pre-training.

*Table 1.* Characteristics of the 19 VOCSIM subsets. **Status** indicates whether the data is Public (allowing potential pre-training transfer) or Blind (strict OOD test). **Duration** reports the average length in seconds followed by the (min-max) range.

| ID | Domain | Samples | Classes | Duration (s) | Status |
|---|---|---|---|---|---|
| *Animal Vocalizations* | | | | | |
| BS1 | Bird Syllables (Zebra Finch) | 473 | 6 | 0.08 (0.03-0.17) | Public |
| BS2 | Bird Syllables (Zebra Finch) | 10,001 | 36 | 0.13 (0.03-0.26) | Public |
| BS3 | Bird Syllables (Bengalese) | 9,988 | 46 | 0.07 (0.03-0.20) | Public |
| BS4 | Bird Syllables (Bengalese) | 7,035 | 64 | 0.13 (0.03-0.83) | Public |
| BS5 | Bird Syllables (Canary) | 8,244 | 30 | 0.70 (0.03-4.99) | Public |
| BC | Bird Calls (Zebra Finch) | 3,321 | 28 | 0.33 (0.03-2.99) | Public |
| OC | Otter Calls | 441 | 21 | 0.87 (0.28-5.32) | Public |
| *Environmental Sounds* | | | | | |
| ES1 | Env. Events (ESC-50) | 2,000 | 50 | 5.00 (5.00-5.00) | Public |
| *Human Speech (Public)* | | | | | |
| HP | Phones (LibriSpeech) | 10,687 | 68 | 0.09 (0.03-0.49) | Public |
| HW1 | Words (LibriSpeech) | 11,532 | 754 | 0.40 (0.07-1.10) | Public |
| HW2 | Words (AMI) | 8,827 | 1,324 | 0.39 (0.08-1.00) | Public |
| HU1 | Utterances (VCTK) | 14,463 | 1,245 | 3.21 (1.24-6.10) | Public |
| HU2 | Utterances (AMI) | 17,041 | 1,366 | 0.64 (0.03-1.49) | Public |
| HS1 | Non-Verbal (AMI) | 1,670 | 14 | 0.88 (0.04-2.98) | Public |
| HS2 | Vocal Imitations | 8,918 | 236 | 2.88 (0.04-6.00) | Public |
| *Human Speech (Blind / OOD)* | | | | | |
| **HW3** | **Words (Shipibo-Conibo)** | **4,540** | **368** | **1.01 (0.32-2.00)** | **Blind** |
| **HW4** | **Words (Chintang)** | **3,497** | **215** | **0.97 (0.46-2.00)** | **Blind** |
| **HU3** | **Utter. (Shipibo-Conibo)** | **1,703** | **183** | **1.40 (0.40-3.00)** | **Blind** |
| **HU4** | **Utter. (Chintang)** | **1,001** | **80** | **0.99 (0.47-1.98)** | **Blind** |

**Preprocessing.** All audio is standardized to 16 kHz mono. To ensure a rigorous evaluation of content identity, we apply consistent filtering of high-frequency stop words (e.g., 'the') across Public and Blind speech corpora. By removing these acoustically distinct but semantically ubiquitous tokens, we prevent Zipfian distribution artifacts from inflating retrieval scores. This ensures that our metrics probe the model's ability to distinguish complex lexical items rather than simply clustering the most frequent functional morphemes.

## 4. Benchmark and Methods

Our evaluation framework is designed to probe the intrinsic geometric structure of audio embeddings in a strictly zero-shot setting. To allow for rigorous comparisons between heterogeneous foundation models, which vary in architecture, parameter count, and pre-training data, we implement a standardized evaluation pipeline. This pipeline treats each model as a frozen feature extractor, funneled through uniform pooling and transductive dimensionality reduction, ensuring that performance differences reflect representational quality rather than feature engineering.

### 4.1. Feature Representations

We evaluate a broad suite of open-source models representing the current state-of-the-art in general-purpose audio intelligence. Consistent with established foundation model benchmarks in NLP (Enevoldsen et al., 2025) and Computer Vision (Vaze et al., 2023), our objective is to evaluate existing *artifacts* available to practitioners rather than to perform architectural ablations under controlled training regimes.

**Model Suite.** We categorize evaluated models by their supervision signal, (1) **Weakly Supervised:** We evaluate OpenAI's Whisper (Large-v3) (Radford et al., 2022), trained on 680k hours of labeled audio, and CLAP (Elizalde et al., 2022), trained on contrastive audio-text pairs, (2) **Self-Supervised (SSL):** We assess WavLM (Large) (Chen et al., 2022) and Wav2Vec 2.0. We select WavLM as the representative state-of-the-art for masked prediction, as it generally supersedes prior architectures like HuBERT on the SUPERB leaderboard, (3) **Spectrogram Transformers:** We include BEATs (Chen et al., 2023), EAT (Chen et al., 2024), and AudioMAE (Huang et al., 2023), which operate on masked spectrogram patches, and (4) **Domain-Specific Unsupervised Baselines:** To quantify the gain provided by large-scale pre-training, we train small VAEs and Autoencoders from scratch *independently on each subset, including each blind subset*, in a purely unsupervised manner. These are *not* directly comparable to the frozen general-purpose foundation models above (they are domain-adapted by construction); rather, they serve as an unsupervised upper bound for what a small model can learn topology-wise from each subset alone. As an indicator of this distinction, the per-subset VAE baseline scores below chance on the blind sets (Lift $-7.8$; Appendix), so any foundation model exceeding it on those sets is generalizing beyond what local unsupervised training can achieve.

**Standardized Extraction Pipeline.** Let an audio clip $x$ of variable duration be processed by an encoder $f(\cdot)$ to produce a sequence of feature vectors $Z \in \mathbb{R}^{T \times D}$, where $T$ varies with input length and $D$ is the model's hidden dimension. To resolve $Z$ into a fixed-length signature $v$ suitable for vector retrieval, we apply explicit statistical pooling. While we explored sequence-aware comparison methods (e.g., Dynamic Time Warping re-ranking), our ablations (see Appendix) reveal that simple statistical pooling yields competitive or superior retrieval accuracy at a fraction of the computational cost. Specifically, we compute $v = \text{Concat}(\mu_{time}(Z), \mu_{feat}(Z))$ where $\mu_{time}$ and $\mu_{feat}$ capture temporal and spectral summary statistics, respectively. We project all embeddings to $D' = 100$ using label-free PCA fitted on the target subset statistics. We acknowledge that relying on batch statistics distinguishes this approach from strict single-sample inference. However, we define this not as "test-time training" (as no gradient updates occur), but as **transductive whitening**. This is a standard non-parametric technique in information retrieval (Song et al., 2018) necessary to correct the well-documented anisotropy ("representation cone") of foundation models (Ethayarajh, 2019), effectively calibrating the embedding space's aspect ratio without altering the topology via supervision.

## 4.2. Distance Measures

To ensure our findings are robust to the choice of geometric space, we evaluate similarity using three distinct distance metrics, (1) **Cosine Distance:** $d_{cos}(\mathbf{u}, \mathbf{v}) = 1 - \frac{\mathbf{u} \cdot \mathbf{v}}{\|\mathbf{u}\|\|\mathbf{v}\|}$. Evaluates angular similarity, standard for normalized embedding spaces, (2) **Euclidean Distance:** $d_{euc}(\mathbf{u}, \mathbf{v}) = \|\mathbf{u} - \mathbf{v}\|_2$. Evaluates magnitude and position, sensitive to unnormalized scale, and (3) **Spearman Correlation Distance:** $d_{spr}(\mathbf{u}, \mathbf{v}) = 1 - \rho(\text{rank}(\mathbf{u}), \text{rank}(\mathbf{v}))$. A non-parametric rank-based metric. Because it compares the *relative ordering* of feature dimensions rather than their raw magnitudes, Spearman is invariant to per-dimension scale and largely insensitive to the anisotropy and "hub" effects that plague high-dimensional Euclidean/cosine spaces ("Radovanović et al., 2010), where a small number of points become disproportionately frequent nearest neighbors. This intuition is consistent with our empirical finding that Spearman dominates for pooled transformer embeddings (Whisper, CLAP), while Euclidean/cosine remain competitive for SSL speech models (WavLM, Wav2Vec 2.0) whose features are already closer to isotropic. Unless otherwise noted, we report results using the metric that maximizes performance for a given *model family* (not per subset), with full results for all three metrics provided in the Appendix.

## 4.3. Zero-Shot Similarity Metrics

We evaluate the geometry of the embedding space using two complementary training-free metrics.

**1. Precision@k (P@k) - Query-by-Example Utility.** This metric measures the purity of an item's immediate neighborhood. For a dataset of $N$ points with class labels $y$, let $\mathcal{N}_k(i)$ be the set of $k$ nearest neighbors to point $i$. P@k is defined as:

$$\text{P@k} = \frac{1}{N \cdot k} \sum_{i=1}^{N} \sum_{j \in \mathcal{N}_k(i)} \mathbb{I}(y_i = y_j) \qquad (1)$$

We report P@1 (Nearest Neighbor Accuracy) and P@5. In the context of deployment, **P@1 serves as a direct simulation of a Query-by-Example (QbE) search system**. A P@1 score of 60% implies that if a user queries a database with a sound clip, the top returned result will be semantically relevant 60% of the time, allowing us to quantify "readiness for retrieval" without building a custom application interface.

**2. Global Separation Rate (GSR) - Boundary Integrity.** GSR quantifies macroscopic separability via the point-wise score $(\text{NID}_i - \text{Avg\_ID}_i)/(\text{NID}_i + \text{Avg\_ID}_i + \epsilon)$, normalized across the dataset. Unlike Silhouette ($\rho \approx 0.8$), which compares intra-cluster cohesion to the *average* distance of the nearest cluster, GSR is deliberately asymmetric to prioritize retrieval reliability. We compare the *single nearest* inter-class neighbor (NID) against the *average* intra-class

spread (Avg_ID). Standard metrics like Silhouette use average inter-class distance, which can mask "leaks" where a confusing class merges with the target. By using the minimum distance (NID), GSR strictly penalizes *any* boundary breach, effectively simulating a false positive in a top-1 retrieval task. Conversely, we use the average (rather than maximum) intra-class distance to remain robust to singular outliers within the correct class, focusing on the density of the core manifold.

## 5. Results

We evaluate the intrinsic geometric alignment of audio representations across 19 diverse corpora. Our analysis focuses on identifying the most robust feature extraction pipeline and quantifying the gap between performance on public transfer tasks versus strictly out-of-distribution (OOD) blind test sets. Full per-subset results are provided in the Appendix.

### 5.1. Main Zero-Shot Performance

Table 2 summarizes the performance of the top configurations, while **Figure 2** visualizes the impact of dataset characteristics (e.g., class count, sample size) on these metrics. Across the public subsets, where models likely benefit from implicit transfer (e.g., overlapping pre-training data), a clear hierarchy emerges.

**The Whisper Encoder Dominates.** The configuration **EWMTF D100** (Whisper Large-v3 encoder features, aggregated via Time-Frequency pooling and reduced to 100 dimensions with transductive PCA) consistently achieves the highest performance. On public sets, it attains a Mean P@1 of **66.8%** and a raw GSR of **41.8%**. This suggests that weakly supervised pre-training on vast quantities of noisy audio-text pairs yields a more robust intrinsic geometry for content identity than self-supervised objectives (WavLM, Wav2Vec 2.0) or spectrogram masking (AudioMAE, BEATs), which often perform competitively only after fine-tuning. Note that our **VAE/AE baselines** are trained per-subset and are intended as a *domain-specific unsupervised upper bound*, not as a directly comparable frozen general-purpose baseline. Despite this favorable setup, they still underperform the transfer-learning capabilities of the frozen Whisper encoder on public sets, and fall *below* chance on the blind sets (Appendix).

**Effectiveness of Simple Pooling.** Despite the temporal complexity of audio, our results show that simple statistical pooling (Mean-Time + Mean-Freq) is highly effective. As shown in Table 2, raw pooled features perform competitively, though PCA (D100) provides a consistent improvement in neighborhood purity (P@1) by mitigating the curse of dimensionality and anisotropies in the raw embedding space.

**Pretraining-Overlap Caveat for Public Sets.** The public-set rankings above must be interpreted with care: our overlap audit (Appendix D.2, Table 5) identifies multiple subsets with confirmed or likely overlap with the pretraining corpora of the evaluated models. Public-set scores therefore reflect a mixture of intrinsic geometric quality and exposure to similar data during pretraining and should not be read as a pure measure of generalization. Importantly, our *animal* vocalization subsets have *no* confirmed overlap for any evaluated model, yet still produce strong, discriminative results (CLAP 88.4% P@1, Whisper 85.7% P@1; see below), which suggests that VOCSIM captures representational quality beyond direct corpus memorization. The blind low-resource speech sets remain the only strictly leakage-free evaluation in this work.

### 5.2. Domain-Disaggregated Performance

Because VOCSIM aggregates heterogeneous corpora, a single macro-average can obscure how rankings vary with pretraining domain. Following reviewer feedback, we report per-domain summaries (Table 3) over four groupings: **Animal** (7 subsets), **Environmental** (ESC-50, 1 subset), **Public Speech** (7 subsets), and **Blind Speech** (4 subsets, low-resource OOD).

**P@1 reflects domain alignment; GSR is more stable.** P@k1 is domain-sensitive: CLAP ranks highest on animal and environmental sounds (consistent with its broad audio-text pretraining), WavLM on public English speech (matching its speech-SSL objective), and Whisper on blind speech. By contrast, GSR is markedly more stable across domains. Computing pairwise Kendall's $\tau$ over the five models' GSR rankings across the four domain columns yields an average $\tau = 0.60$ (rising to 0.87 when the single-subset environmental column is excluded), despite per-subset class counts ranging from 6 to 1,366. This supports a more precise reading of the benchmark: *local retrieval purity reflects pretraining alignment, while global separation captures a more domain-invariant property of the embedding geometry.*

**The blind-speech collapse is a within-domain failure.** Whisper's drop from 46.5% P@k1 on public speech to 11.5% on Shipibo-Conibo / Chintang occurs within a single domain (human speech), and so cannot be explained as a coarse speech-vs.-non-speech domain mismatch. We restate accordingly: the headline OOD result is a *cross-lingual speech* generalization gap, not a claim about general-audio OOD.

### 5.3. The Generalization Gap

A critical finding of VOCSIM is the sharp divergence in performance between Public and Blind test sets.

*Table 2.* **Zero-Shot Content Identity Performance.** Values are macro-averages ± margin of error across subsets, derived from per-subset 95% bootstrap confidence intervals (300 resamples). **Public Sets** represent tasks where transfer learning is likely. **Blind Sets** are strict OOD low-resource speech (Shipibo-Conibo, Chintang) with no pre-training overlap. Dist: S = Spearman, E = Euclidean. *Note: On Blind Sets, transductive PCA yields negligible geometric adaptation due to sparse OOD manifold structure.*

| MODEL | METHOD | DIST | PUBLIC SETS (TRANSFER) | | | BLIND SETS (STRICT OOD) | | |
|---|---|---|---|---|---|---|---|---|
| | | | P@1 ↑ | P@5 ↑ | GSR ↑ | P@1 ↑ | P@5 ↑ | GSR ↑ |
| **WHISPER-L-V3** | **EWMTF D100 (PCA)** | S | **66.8**±0.8 | **57.4**±0.7 | **41.7**±0.2 | **11.5**±1.2 | **7.7**±1.1 | **39.4**±0.3 |
| WHISPER-L-V3 | EWMTF (RAW) | S | 61.5±0.8 | 53.0±0.8 | 40.2±0.3 | 11.5±1.3 | 7.7±0.9 | 39.4±0.3 |
| CLAP | D100 (PCA) | S | 63.7±0.7 | 55.6±0.6 | 38.1±0.2 | 8.1±1.4 | 5.2±1.0 | 36.2±0.2 |
| CLAP | RAW | S | 61.7±0.5 | 54.4±0.5 | 33.8±0.3 | 8.1±1.3 | 5.2±1.0 | 36.2±0.4 |
| WAVLM-LARGE | D100 (PCA) | E | 64.1±0.7 | 54.4±0.7 | 37.0±0.2 | 4.6±1.2 | 3.1±0.8 | 35.8±0.3 |
| WAVLM-LARGE | RAW | E | 62.8±0.6 | 51.4±0.7 | 35.2±0.3 | 4.6±1.1 | 3.1±0.8 | 35.8±0.2 |
| BEATS | D100 (PCA) | E | 64.3±0.9 | 55.4±0.9 | 31.4±0.2 | 11.4±1.2 | 7.0±0.7 | 34.7±0.2 |
| EAT | CLS TOKEN | E | 50.2±0.9 | 42.1±1.1 | 24.7±0.3 | 1.9±0.9 | 1.8±0.6 | 22.6±0.2 |
| *Baselines* | | | | | | | | |
| LOG-MEL | FLATTENED | S | 57.7±0.6 | 47.3±0.6 | 34.2±0.3 | 3.5±1.2 | 2.6±0.9 | 33.0±0.2 |
| VAE (SUBSET) | LATENT MEAN | E | 58.2±0.7 | 49.5±0.6 | 34.3±0.3 | 3.8±1.4 | 2.8±0.9 | 25.9±0.3 |

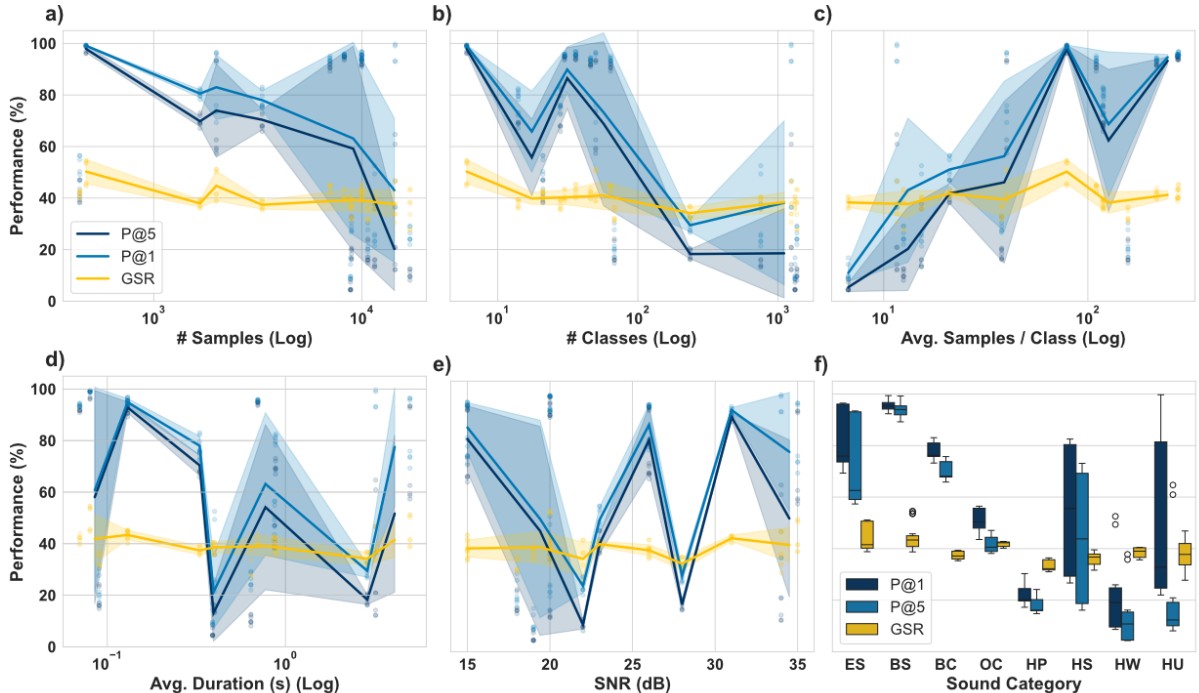

*Figure 2.* **Performance trends of top 5% of configurations across VOCSIM subsets. (a-e)** Each point represents a subset's performance. Trend lines show the mean performance against key subset properties. Local metrics like **P@1 and P@5** (blue lines) are highly sensitive to dataset structure, degrading significantly as the number of classes increases **(b)** while improving with more samples per class **(c)**. In contrast, the global metric **GSR** (yellow line) remains remarkably stable across these conditions, suggesting it captures a more intrinsic property of the embedding geometry. **(f)** Boxplots show performance distributions broken down by sound category, highlighting strengths (e.g., on Environmental Sounds) and weaknesses (e.g., on Human Utterances).

**Collapse of Local Structure (P@1).** While models achieve high local retrieval accuracy on public data (e.g., Whisper ≈ 67%), this collapses to ≈ 11.5% on the Blind OOD sets. This indicates that while the models are excellent at recognizing acoustic signatures similar to their training distribution (e.g., English speech, common environmental sounds), they struggle to form tight, pure neighborhoods

for the phonotactics of low-resource languages (Shipibo-Conibo, Chintang) they have never seen.

**Diagnosing the Gap: Geometry and Acoustics.** Although raw GSR appears comparable between Blind and Public sets (e.g., Whisper 39.4% vs 41.8%), permutation tests reveal this is an artifact of dataset density. The elevated permutation baseline on Blind sets (33.7% vs. 24.9%) reflects their

*Table 3.* **Domain-disaggregated P@1 and GSR.** P@k1 ranks shift by pretraining alignment (CLAP wins on Animal/Env., WavLM on Public Speech, Whisper on Blind Speech). GSR ranking is comparatively stable: Whisper leads in 3 of 4 domains, with CLAP leading only on the single environmental subset (ESC-50, where AudioSet-related overlap is expected).

| | **P@1** | | | |
|---|---|---|---|---|
| Model | Animal | Env. | Spch. Pub. | Spch. Blind |
| Whisper D100 | 85.7 | 75.9 | 46.5 | **11.5** |
| CLAP D100 | **88.4** | **95.7** | 34.4 | 8.1 |
| BEATs D100 | 87.3 | 95.1 | 37.0 | 11.4 |
| WavLM D100 | 80.6 | 34.0 | **51.8** | 4.6 |
| EAT CLS | 78.1 | 43.9 | 23.3 | 1.9 |
| | **GSR** | | | |
| Model | Animal | Env. | Spch. Pub. | Spch. Blind |
| Whisper D100 | **44.3** | 41.5 | **39.2** | **39.4** |
| CLAP D100 | 41.4 | **50.2** | 33.2 | 36.2 |
| BEATs D100 | 33.1 | 46.9 | 27.6 | 34.7 |
| WavLM D100 | 38.1 | 37.0 | 36.0 | 35.8 |
| EAT CLS | 30.7 | 25.7 | 18.5 | 22.6 |

smaller class counts and denser embedding-space packing, which increases the probability of chance boundary overlap under random label assignment. Whisper's lift over the random baseline drops from **+16.9** on Public sets to **+5.8** on Blind sets; while statistically significant ($p < 0.001$), this reduced lift corresponds to a collapse in retrieval utility (P@1 $\approx$ 11.5%). To confirm this reflects **phonotactic distance** (unseen phonetic inventories) rather than acoustic degradation, we analyzed the Dynamic Range Index (DRI), a proxy for SNR. Public 'Human Words' and Blind 'Shipibo-Conibo' share overlapping DRI distributions (15.2$_{\pm 3.1}$ dB vs 14.8$_{\pm 4.2}$ dB). This parity in DRI, contrasted with the extreme P@1 drop (from ∼66% to ∼11%), isolates the failure to resolve specific target phonemes as the primary driver of the generalization gap.

**Manifold Collapse and Baseline Anomalies.** This gap explains two geometric anomalies. First, while transductive PCA yields large gains on Public sets (+2-5% P@1), it yields negligible improvement on Blind sets. We interpret this as a geometric diagnosis: on OOD data, the model fails to map the signal to a structured phonotactic manifold; without a coherent structure, whitening operations merely rotate the noise.

### 5.4. Ablation Studies

To validate that our findings reflect intrinsic model quality rather than artifacts of our evaluation pipeline, we conducted rigorous ablations (detailed in Appendix).

**Pooling & Sequence Matching.** Counter-intuitively, our ablations show that including padding tokens in the mean pooling often outperforms masking them out (e.g., BC: 65.5% vs

41.1%). We hypothesize that because the Whisper encoder is trained to process 30-second chunks, the "silence" embeddings produced for padding tokens are well-structured within the manifold. We compared our statistical pooling against a computationally expensive Dynamic Time Warping (DTW) re-ranking baseline. DTW did not improve P@1 or GSR scores on average, and significantly increased inference time. This confirms that the fixed-length statistical vectors used in our pipeline capture the necessary content identity information efficiently.

**Robustness to Label Noise.** A potential critique of zero-shot benchmarks is sensitivity to label noise. We simulated synthetic label noise (flipping 1-20% of labels). We found that while P@1 degrades precipitously (as expected for a local metric), GSR degrades linearly and gracefully. Even at 20% noise, GSR remains well above the random baseline, confirming it is a robust measure of global geometric structure.

**Layer Sensitivity.** A sweep of all 32 layers of the Whisper encoder shows performance is stable across the network, thus justifies our standard use of the final layer embeddings, as no layer drastically alters the ranking of models.

**Model Scale (Whisper-Small vs. Large-v3).** To probe the cost–performance trade-off, we re-ran the full pipeline with Whisper-Small (best configuration: EWMTF + cosine). Smaller capacity leads to a substantial drop in both local and global geometric quality, particularly on the blind sets, indicating that the geometry exposed by VOCSIM scales materially with model size within the Whisper family (Table 4).

*Table 4.* **Whisper-Small vs. Large-v3** under the same pipeline.

| | Public | | Blind | |
|---|---|---|---|---|
| Model | P@1 | GSR | P@1 | GSR |
| Whisper-Large-v3 | **66.8** | **41.7** | **11.5** | **39.4** |
| Whisper-Small | 57.4 | 28.7 | 5.7 | 29.1 |
| Δ | −9.4 | −13.0 | −5.8 | −10.3 |

## 6. External Validation

A benchmark is only valuable if its metrics proxy real-world utility. We posit that a well-structured zero-shot manifold (VOCSIM) is a prerequisite for efficient adaptation. We test the hypothesis that **intrinsic geometric alignment** is a strong predictor of downstream success in two distinct regimes: **supervised adaptability** (HEAR, Mouse Classification) and **biological alignment** (Avian Perception).

### 6.1. State-of-the-Art on the HEAR Benchmark

We evaluated our frozen Whisper embeddings on the **HEAR benchmark** (Turian et al., 2022) using the official linear evaluation protocol. As detailed in Table 5, our pipeline es-

*Table 5.* **HEAR Benchmark Validation.** Linear probe accuracy (%) $\pm$ std dev. Our frozen Whisper embedding achieves **SOTA on 5/7 tasks** (bolded) and outperforms the CLAP baseline across all tasks.

| HEAR TASK | FOLDS | WHISPER D100 (OURS) | CLAP D100 | HEAR SOTA | SOTA MODEL REF. |
|---|---|---|---|---|---|
| BEIJING OPERA | 5 | **97.65 $\pm$ 4.13** | 97.03 $\pm$ 3.95 | 97.46 | OPENL3 |
| GTZAN MUSIC | 10 | **99.23 $\pm$ 2.31** | **99.23 $\pm$ 2.31** | **99.23** | CP-JKU |
| GUNSHOT TRIANG. | 7 | **97.92 $\pm$ 3.34** | 94.05 $\pm$ 5.83 | 94.94 | OPENL3 |
| MRIDANGAM STROKE | 5 | 96.86 $\pm$ 0.93 | 96.42 $\pm$ 0.47 | **97.53** | GURA |
| MRIDANGAM TONIC | 5 | 89.71 $\pm$ 0.76 | 87.73 $\pm$ 1.04 | **96.55** | REDRICE |
| CREMA-D EMOTION | 5 | **79.28 $\pm$ 0.56** | 58.42 $\pm$ 0.95 | 75.21 | GURA |
| SPEECH CMDS (5H) | TVT | **98.61** | 68.24 | 97.63 | IUT-CSE |

tablishes new **State-of-the-Art (SOTA) results on 5 out of 7 tasks**, consistently outperforming the strong CLAP baseline. Notably, we surpass specialized systems on **Speech Commands** (**98.6%**) and **CREMA-D** emotion recognition (**79.3%**). This confirms that the zero-shot content identity measured by VOCSIM translates to high-performance supervised adaptation.

## 6.2. Alignment with Avian Perception

We validated our embeddings against behavioral data from Zebra finches (Zandberg et al., 2024) on a Triplet discrimination task (is sound $X$ closer to $A$ or $B$?). On the high-consistency subset (derived from 2AFC), the frozen Whisper embedding achieves **80.9%** accuracy in predicting bird choices, within the estimated inter-bird consistency range (80–90%) and thus comparable to between-conspecific variability (Zandberg et al., 2024).

## 6.3. Bioacoustic Classification Utility

We assessed utility in a specialized scientific domain: classifying Mouse Ultrasonic Vocalizations (USVs) by genetic strain and individual identity (Goffinet et al., 2021). Using our fixed feature extractor with a high-frequency spectrogram frontend (Appendix) and a simple MLP classifier, we obtain **99.5%** Top-1 on Strain Classification (vs. $\sim 90\%$ for spectrogram baselines) and **53.1%** Top-1 on Identity (chance $\approx 2.8\%$, 36 individuals), showing that the geometry favored by VOCSIM translates directly to fine-grained scientific classification.

## 7. Discussion and Conclusion

**A Blueprint for Zero-Shot Similarity.** Our evaluation identifies a robust recipe for content identity: large weakly supervised encoders (e.g., Whisper), statistical time-frequency pooling, and transductive label-free PCA. This fine-tuning-free pipeline serves as an ideal default for retrieval and indexing.

**The Cross-Lingual Speech Generalization Ceiling.** VOCSIM quantifies a cross-lingual speech gap: geometric "lift" over chance collapses on blind, low-resource speech, even

though this collapse occurs entirely within the speech domain. This suggests SOTA models interpolate high-resource languages rather than learning universal phonotactics. We do not extend this to general-audio OOD; blind animal/environmental evaluation is left to future work. Our permutation-calibrated GSR provides the diagnostic to measure progress toward closing this gap.

**Predictive Validity.** VOCSIM performance predicts downstream utility: embeddings dominating our zero-shot leaderboard also achieve SOTA results on the supervised HEAR benchmark and align with biological perceptual judgments, validating VOCSIM as an effective early-stage filter for model selection.

**Limitations and Scope.** VOCSIM is, by design, a diagnostic of *single-source* content-identity geometry; polyphonic mixtures, noisy scenes, and music are *out of scope*, as raw embeddings on overlapping sources would conflate content representation with source separation. Our protocol is *gradient-free, label-free, transductive* rather than single-sample (per-subset label-free PCA; results reported with and without PCA). Environmental audio is one small subset (ESC-50) and acts as a robustness check rather than the basis for environmental claims. Blind sets are restricted to low-resource speech (Shipibo-Conibo, Chintang); the headline OOD result is therefore a *cross-lingual speech* gap, not a general-audio claim. Animal subsets have no confirmed pre-training overlap (Appendix D.2), providing indirect no-leakage evidence rather than strict blind OOD; strict zero-leakage requires offline bioacoustic data we will pursue in future iterations.

**Conclusion.** We release VOCSIM as a living benchmark for intrinsic audio geometry, aiming to catalyze representations with the robust content identity found in biology.

## Impact Statement

This paper presents a benchmark for evaluating general-purpose audio representations. The immediate societal impact of our work lies in exposing the **performance disparities** inherent in current foundation models.

**Bias in Low-Resource Languages.** Our results on the blind

test sets (Shipibo-Conibo and Chintang) reveal that state-of-the-art models, which excel on English and high-resource languages, organize the acoustic space of low-resource languages only marginally better than random chance. This highlights a risk that deploying current "universal" audio models in global contexts may perpetuate a digital divide, offering degraded performance for under-represented linguistic communities. VOCSIM provides a metric to track and incentivize improvements in true cross-lingual generalization.

**Data Sovereignty and Ethical Evaluation.** The blind test sets used in this benchmark are derived from field documentation of indigenous languages. To uphold data sovereignty and ethical commitments to these communities, we do not release the raw audio or labels publicly. Instead, we have established a secure, server-side evaluation protocol. This ensures that the data is used strictly for benchmarking out-of-distribution generalization without being harvested for model training, respecting the rights and privacy of the data originators.

# Acknowledgements

We thank the ICML 2026 reviewers (dgdh, f1M9, b75m, zhTd) for thorough and constructive feedback that materially improved this paper. We thank the language documentation teams who recorded and curated the Shipibo-Conibo and Chintang corpora, and the communities whose speech they preserve. We also acknowledge the maintainers of the open datasets and pretrained checkpoints aggregated by VOCSIM, without whom this benchmark would not exist. This work was supported by the Institute of Neuroinformatics, University of Zurich and ETH Zurich, by the Institute for the Interdisciplinary Study of Language Evolution, University of Zurich, and by the NCCR Evolving Language.

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

## LLM Usage

Large Language Models were used as a writing aid and code assistant throughout the preparation of this manuscript. Specific uses included: (1) Rephrasing sentences and paragraphs to improve clarity and flow. (2) Correcting grammar and spelling. (3) Suggesting alternative phrasings for technical concepts. (4) Providing code snippet suggestions. The core scientific ideas, experimental design, and interpretation of results are entirely our own. LLMs were not used for research ideation or generating substantive content.

## A. Appendix: Supplemental Details

This appendix provides supplemental information to support the main text. Appendix B details the public URLs for all code, data, and leaderboards. Appendix D details the sources and preprocessing of the 19 corpora aggregated in VOCSIM and provides justification for our aggregation methodology. Appendix F provides detailed descriptions of our evaluation metrics, including justification for the GSR metric's design and robustness, our empirical baseline methodology, and an analysis of metric correlations. Full, detailed results for all configurations are provided in Appendix P, followed by further details on applications and benchmark extensibility.

## B. Code, Data, and Leaderboard

All the code to reproduce the results of the paper is available at the following url: https://github.com/vocsim/benchmark. The code contains feature extraction, model training (AE/VAE), distance computation, benchmark evaluation (P@k, GSR), clustering analysis, and application benchmarks (avian perception, mouse classification).

The VOCSIM dataset is available at the following url: https://huggingface.co/datasets/vocsim/public.

A leaderboard for comparing results on the VOCSIM benchmark is maintained at: https://huggingface.co/spaces/vocsim/VocSim. The leaderboard ranks submissions primarily by their performance on the blind test sets, reporting P@1, P@5, and GSR as separate columns rather than collapsing them into a single scalar score. This design reflects our view that local retrieval (P@k) and global separability (GSR) are complementary aspects of embedding quality; for the configurations evaluated so far, the same Whisper-based model is top-ranked across all three metrics on the blind sets.

The processed Avian Perception dataset is available at: https://huggingface.co/datasets/vocsim/avian-perception-benchmark.

The processed Mouse Strain dataset is available at: https://huggingface.co/datasets/vocsim/mouse-strain-classification-benchmark.

The processed Mouse Identity dataset is available at: https://huggingface.co/datasets/vocsim/mouse-identity-classification-benchmark.

## C. Glossary of Terms and Abbreviations

This section provides a comprehensive legend for all abbreviations and technical terms used throughout the paper to ensure clarity.

*Table 6.* Comprehensive glossary of abbreviations.

| Abbreviation | Description |
|---|---|
| **Dataset Subset Categories** | |
| BS | **B**ird **S**yllables: Individual, stereotyped acoustic units from birdsong. |
| BC | **B**ird **C**alls: Shorter, often simpler vocalizations than song syllables. |
| OC | **O**tter **C**alls: Vocalizations from giant otters. |
| HP | **H**uman **P**hones: The smallest units of sound in human speech (e.g., /a/, /k/). |
| HW | **H**uman **W**ords: Spoken words from various speakers and contexts. |
| HU | **H**uman **U**tterances: Complete spoken phrases or sentences. |
| HS | **H**uman **S**ounds: Non-verbal human vocalizations (e.g., coughs, laughter). |
| ES | **E**nvironmental **S**ounds: Sounds from the environment (e.g., siren, rain). |
| **Base Feature Extractors (Models)** | |
| EW | **E**ncoder of **W**hisper (large-v3 model). |
| W2V | **W**av**2Vec** 2.0 (Base model). |
| WLM | **W**av**LM** (Large model). |
| CLP | **CLAP** (Contrastive Language-Audio Pre-training) model. |
| EAT | **E**fficient **A**udio **T**ransformer; we use its utterance-level [CLS] token embedding. |
| BEATs | Transformer-based audio encoder pretrained on AudioSet-2M with masked audio modeling. |
| MAE | Masked **A**uto**E**ncoder (AudioMAE). |
| VC / AC | Our custom-trained per-subset **V**ariational Autoen**c**oder / **A**utoen**c**oder. |
| M | Baseline Log-**M**el spectrograms. |
| CC | **C**odebook **C**odes from the EnCodec neural audio codec (concatenated codebook indices). |
| **Pooling and Post-Processing Variants** | |
| MTF | **M**ean over **T**ime and **M**ean over **F**. (F = frequency for spectrograms; feature/channel for hidden states.) EWMTF concatenates these two vectors. |
| TF | **T** first time-step and **F** first frequency/feature. EWTF concatenates these two vectors. |
| T / F | Use only the first time-step vector (T) or the first frequency/feature trajectory (F). |
| ET / EF | For Whisper/WhisperSeg: ET = first frame embedding; EF = values of the first feature/channel across time. |
| EWTF | Whisper encoder features concatenating ET and EF (first-time + first-F). |
| EWMTF | Whisper encoder features concatenating mean-over-time and mean-over-F. |
| ETF / EMTF | Analogous to EWTF/EWMTF for WhisperSeg. |
| D100 / D30 | PCA to 100 or 30 dimensions (label-free, per-subset). |
| *Note: For spectrograms, F denotes frequency (Mel bins); for hidden-state sequences, F denotes the feature/channel axis.* | |
| **Metrics and Technical Terms** | |
| P@k | **P**recision@**k**: A local metric measuring neighborhood purity. |
| GSR | **G**lobal **S**eparation **R**ate: Our global metric for class separability, normalized to [0,1] (reported as %). |
| NID | **N**earest **I**nter-class **D**istance: The distance to the closest point of a different class. |
| Avg_ID | **Avg**. **I**ntra-class **D**istance: The average distance to all points of the same class. |
| OOD | **O**ut-**o**f-**D**istribution: Data from a different distribution than the training/public data. |
| DRI | **D**ynamic **R**ange **I**ndex: A heuristic for the signal-to-noise ratio within a clip. |

# D. Dataset Details

This section provides specifics about the datasets aggregated within VOCSIM and the preprocessing steps applied.

## D.1. On the Methodology of Benchmark Aggregation

VOCSIM's construction by aggregating multiple corpora is a deliberate methodological choice and a principled strategy shared by many of the most influential benchmarks in machine learning, which are designed to test robust generalization across diverse tasks and conditions. Prominent examples span multiple domains: in natural language processing, benchmarks like GLUE (Wang et al., 2019), SuperGLUE (Wang et al., 2020), and the Massive Text Embedding Benchmark (MTEB) (Enevoldsen et al., 2025) aggregate numerous existing datasets to form a comprehensive evaluation suite. In computer vision, the WILDS collection (Koh et al., 2021) curates datasets specifically to test in-the-wild generalization. This is also a standard practice in the audio domain itself, with both the HEAR (Turian et al., 2022) and SUPERB (Yang et al., 2021) benchmarks being composed of aggregated tasks from various sources.

Creating a novel, monolithic dataset with VOCSIM's engineered variability would be prohibitively difficult and resource-intensive. Such an effort would require an immense, multi-disciplinary data collection campaign spanning both pristine lab

environments and noisy, uncontrolled field settings; capturing a vast dynamic range of signal durations from milliseconds to seconds; and encompassing fundamentally different vocal production systems, from human phonetics to the avian syrinx. Therefore, aggregation is the only feasible and principled approach to construct a benchmark that can systematically probe the limits of generalization in the way VOCSIM is designed to do.

### D.2. Pre-training Data Overlap Analysis

To ensure a fair and transparent zero-shot evaluation, we conducted a systematic audit to identify potential overlaps between VOCSIM's source corpora and the known pre-training datasets of the foundation models we benchmarked. The analysis distinguishes between confirmed overlap (source explicitly listed in training data), likely overlap (e.g., a common public dataset likely scraped for a web-scale corpus that lacks a manifest), and no evidence of overlap. The results are summarized in Table 7.

*Table 7.* Pre-training data overlap matrix for key models and VOCSIM's public subsets.

| VOCSIM Subset | | Foundation Model | | | | | | |
|---|---|---|---|---|---|---|---|---|
| ID | Source Type | Whisper-L-v3 | WavLM-Large | Wav2Vec 2.0 | CLAP | AudioMAE | EAT-Large | BEATs |
| *Animal Vocalizations* | | | | | | | | |
| BC, BS1-5, OC1 | Bird/Otter calls | ○ | ○ | ○ | ○ | ○ | ○ | ○ |
| *Environmental Sounds* | | | | | | | | |
| ES1 | ESC-50 (env. sounds) | ∼ | ○ | ○ | ● | ● | ● | ● |
| *Human Speech & Non-Speech* | | | | | | | | |
| HP, HW1 | LibriSpeech-derived | ∼ | ● | ● | ○ | ○ | ○ | ○ |
| HW2, HU2 | AMI Meeting Corpus | ∼ | ○ | ○ | ○ | ○ | ○ | ○ |
| HU1 | VCTK Corpus | ∼ | ○ | ○ | ○ | ○ | ○ | ○ |
| HS1, HS2 | Vocal imitations (AudioSet-like) | ∼ | ○ | ○ | ● | ● | ● | ● |

**Legend:**
● **Confirmed overlap (explicitly listed in pre-training corpus)**
∼ **Likely overlap (indirect/content-based)**
○ **No evidence of overlap**

**Wav2Vec 2.0:**

- **Base Model**: Trained on LibriSpeech (Panayotov et al., 2015) (960 hours), leading to confirmed overlap with VOCSIM subsets using LibriSpeech-derived data.

- **Large Model**: Utilizes Libri-Light (Kahn et al., 2020) (53,200 hours), which also overlaps with LibriSpeech-derived subsets.

**WavLM:** Involves LibriSpeech, Libri-Light, GigaSpeech (Chen et al., 2021), and VoxPopuli (Wang et al., 2021), causing likely overlap with any English-based speech datasets within VOCSIM.

**Whisper and WhisperSeg** Pre-trained on a broad multilingual dataset ( 680,000 hours, including  563,000 hours of English) scraped from the web, suggesting potential overlap with diverse human speech subsets. WhisperSeg's fine-tuning with animal vocalizations introduces specialized capabilities but does not use specific subsets.

**CLAP** Likely overlap in environmental sounds like ESC-50, given its use of AudioSet and data from sources such as Freesound (Font et al., 2013), which include public audio clips.

**AudioMAE, BEATs, and EAT:** All these models were trained using AudioSet-2M, creating likely overlaps with environmental sounds represented in VOCSIM from general audio sources.

**EnCodec:** Utilizes datasets like DNS Challenge (Dubey et al., 2023), Common Voice (Ardila et al., 2020), AudioSet, and others, suggesting potential, though indirect, overlap with general audio categories.

## D.3. Comparison to Existing Datasets and Benchmarks

*Table 8.* Comparison of VOCSIM with other major audio benchmarks and datasets. VOCSIM is uniquely positioned as a diagnostic tool for the intrinsic, zero-shot geometric quality of single-source audio representations, complementing benchmarks that focus on downstream adaptability, complex scene analysis, or high-level semantic tasks like music tagging.

| Resource | Primary Purpose | Evaluation Paradigm | Key Challenge |
|---|---|---|---|
| *Evaluation Benchmarks for Model Adaptability* | | | |
| SUPERB (Yang et al., 2021) / HEAR (Turian et al., 2022) | Benchmark Model Adaptability | Fine-Tuning or Linear Probing | Adapting a frozen or fine-tuned encoder to succeed on a wide suite of supervised downstream tasks (e.g., ASR, SID). |
| **BirdSet** (Rauch et al., 2025) / **BIRB** (Hamer et al., 2023) / **BEANS** (Hagiwara et al., 2023) | Bioacoustic Scene Analysis | Supervised (Weak) | Detecting calls in polyphonic soundscapes under domain shift. |
| DCASE Challenge (Mesaros et al., 2018) | Evaluate Scene Analysis Systems | Supervised Training (System Competition) | Detecting and classifying overlapping sound events within complex, polyphonic real-world acoustic scenes. |
| **VOCSIM (Ours)** | **Benchmark Intrinsic Representation** | **Training-Free (Zero-Shot)** | **Generalizing acoustic signature matching across diverse, single-source sounds with varying recording conditions.** |
| *Large-Scale Datasets for Training/Evaluation* | | | |
| AudioSet (Gemmeke et al., 2017) | Train/Evaluate Classifiers | Supervised Training | Multi-label classification of events in complex, often polyphonic, video soundtracks. |
| WavCaps (Mei et al., 2024), LAION-Audio (Wu et al., 2024), Fusion-Audio (Chen et al., 2025) | Pre-train Foundation Models | N/A (Training Data) | Learning robust multimodal representations from vast quantities of noisy, unstructured audio-text pairs scraped from the web. |
| MagnaTagATune (Law et al., 2009) | Train/Evaluate Music Taggers | Supervised Training | Predicting high-level semantic and musical tags (e.g., "rock", "guitar", "fast") from complex polyphonic music clips. |

## D.4. VOCSIM Aggregation Sources

**BC − The Vocal Repertoire of the Domesticated Zebra Finch (Elie).** Source: (Elie & Theunissen, 2020), based on (Elie & Theunissen, 2015). Contains 3,433 calls from 50 birds across 11 types. License: **CC BY 4.0**. URL: `https://doi.org/10.6084/m9.figshare.11905533.v1`.

**BS1 − Deep Audio Segmenter Dataset (DAS).** Source: (Clemens, 2021), used for ("Steinfath et al., 2021). Contains 473 syllables from 1 bird across 6 types. License: **CC0 1.0**. URL: `https://data.goettingen-research-online.de/citation?persistentId=doi:10.25625/ZXJJJY`.

**BS2 − Clustered Subset of "Benchmarking Nearest Neighbor Retrieval..." (Tomka).** Source: (Tomka, Tomas et al., 2024). Contains 48,411 vocalizations from 4 birds across 36 types. License: **CC BY 4.0**. URL: `http://hdl.handle.net/20.500.11850/673918`.

**BS3 − Bengalese Finch Song Repository (Nicholson).** Source: (Nicholson et al., 2022), used by (Cohen et al., 2022). Contains >245,000 syllables from 4 birds. License: **CC BY 4.0**. URL: `https://figshare.com/articles/dataset/Bengalese_Finch_song_repository/4805749`.

**BS4 − Automated annotation of birdsong with a neural network that segments spectrograms.** Source: (Cohen et al., 2022). Contains Bengalese finch syllables. License: **CC BY 4.0**. URL: `https://doi.org/10.7554/eLife.63853`.

**BS5 − Labeled songs of domestic canary M1-2016-spring (Serinus canaria)** Source: (Giraudon et al., 2021). Contains canary syllables. License: **CC BY 4.0**. URL: `https://doi.org/10.5281/zenodo.6521932`.

**ES1 − ESC50: Dataset for Environmental Sound Classification.** Source: (Piczak, 2015). Contains 2,000 recordings across 50 classes (40 clips/class). License: **CC BY 4.0**. URL: `https://doi.org/10.1145/2733373.2806390`.

**HP, HW1 − LibriSpeech Corpus w/ Alignments.** Source: Core data from (Panayotov et al., 2015), with alignments generated via MFA (McAuliffe et al., 2017). Contains segmented phones (HP) and words (HW1) derived from read English speech. License: CC-BY-4.0. URL: `https://huggingface.co/datasets/gilkeyio/librispeech-alignments`.

**HU1 − CSTR VCTK Corpus.** Source: (Yamagishi et al., 2019). Contains segmented utterances (HU1) from English speakers with various accents. License: CC-BY-4.0. URL: `https://huggingface.co/datasets/CSTR-Edinburgh/vctk`.

**HS2 − Vocal Sketch & Vocal Imitation Set (VocImSet).** Source: (Cartwright et al., 2018; Kim & Pardo, 2018), used in (Cartwright & Pardo, 2015; Kim & Pardo, 2018). Contains 10,690 vocal imitations after filtering. Licenses: **Open** (Vocal Sketch), **CC BY 4.0** (VocImSet). URLs: `https://doi.org/10.5281/zenodo.1251982`, `https://doi.org/10.5281/zenodo.1340763`.

**HS1, HW2, HU2 − AMI Meeting Corpus.** Source: (Carletta et al., 2005). Contains 100 hrs of meetings with segmented vocal sounds (HS1), words (HW2), utterances (HU2). License: **CC-BY-4.0**. URL: `https://groups.inf.ed.ac.uk/ami/corpus/`.

**HW3, HU3 − Shipibo-Conibo Language Corpus (ACQDIV).** Source: (Stoll & Bickel, 2013). Contains 75,000 transcribed utterances (words HW3, utterances HU3). Status: **Blind Test Set (Non-public)**. URL: `https://www.acqdiv.uzh.ch/en/resources.html`.

**HW4, HU4 − Chintang Language Corpus (ACQDIV).** Source: (Stoll & Bickel, 2013). Contains >1M transcribed words (words HW4, utterances HU4) from 90 sessions used here. Status: **Blind Test Set (Non-public)**. URL: `https://www.acqdiv.uzh.ch/en/resources.html`.

**OC1 − The Vocal Repertoire of Adult and Neonate Giant Otters (Pteronura brasiliensis).** Source: ("Mumm & Knórnschild, 2014). Contains 441 recordings across 21 call types. License: **CC BY 4.0**. URL: `https://doi.org/10.1371/journal.pone.0112562`.

### D.5. Domain Coverage.

While our suite includes environmental sounds (ESC-50), the benchmark is weighted toward bioacoustics and speech. This imbalance is unavoidable: strictly **single-source** environmental corpora suitable for zero-shot content evaluation effectively **do not exist** in the current open data landscape. Major datasets like AudioSet are inherently polyphonic, violating the core geometric isolation principle of VOCSIM.

### D.6. VOCSIM Preprocessing

The following steps were applied after aggregating the source datasets:

1. **Resampling:** All audio waveforms were resampled to 16 kHz using 'torchaudio'.

2. **Outlier Removal:** For each subset, audio samples with durations in the top 2% (98th percentile) were removed to exclude extreme outliers that might skew results.

3. **Minimum Class Size:** Classes (defined by the 'label' column) containing fewer than six samples were removed. This ensures that metrics like P@k (especially k=5) and GSR are meaningful and that classes have sufficient examples for robust comparison.

4. **Subset Size Capping (for AE/VAE training):** If a subset contained more than 10,000 samples after the above steps, a random selection of classes was performed to bring the sample count to approximately 10,000-17,000, while maintaining class distributions as much as possible. This step was primarily to manage training time for the subset-specific AE/VAE models. The benchmark evaluation itself uses the data after steps 1-3.

5. **Class Balancing strategy:** To ensure a rigorous evaluation of content identity, we aim for a balanced distribution of classes across all subsets to prevent Zipfian artifacts.

   - **Public Speech Corpora (LibriSpeech, AMI, VCTK):** These datasets, being pre-defined standard splits, naturally exhibit extreme class imbalance (e.g., function words). To harmonize these with the benchmark standards, we explicitly filter out the top-N most frequent words/utterances (e.g., top 50 in LibriSpeech HW1).
   - **Blind Test Sets (Shipibo-Conibo, Chintang):** For these subsets, we **actively selected** classes from the raw ACQDIV corpus to ensure a balanced distribution during the dataset construction phase. Because we controlled the sampling procedure to maximize phonetic diversity and minimize imbalance, no post-hoc frequency filtering was required.
   - **Bioacoustic Sets:** Specific outlier removal (e.g., 'TRASH' classes) was applied to ensure only valid vocalizations remained.

The overall preprocessing pipeline is summarized in Algorithm 1.

### D.7. Harmonization of Class Distributions

A critical requirement for VOCSIM is that the evaluation rewards robust acoustic matching rather than the statistical exploitation of class imbalances.

Large, general-language corpora (like LibriSpeech or AMI) are typically provided as fixed splits that exhibit a heavy-tailed Zipfian distribution. Without intervention, a model could achieve high retrieval scores simply by clustering the most frequent stop words (e.g., "the", "is"). Consequently, for these public subsets, we apply a strict frequency filter to remove these dominant heads and flatten the distribution.

For the **Blind Test Sets** (Shipibo-Conibo and Chintang), we adopted a **balanced selection strategy** at the source. Rather than taking an imbalanced slice and filtering it, we constructed these subsets by sampling classes from the raw ACQDIV corpus specifically to ensure broad phonetic coverage and uniform class distribution.

---

**Algorithm 1** Data Preprocessing Procedure

---

**Require:** Aggregated dataset $D$ containing multiple subsets.
**Ensure:** Processed dataset $D_{proc}$.
1:  $D_{proc} \leftarrow \emptyset$
2: **for** each subset $S$ in $D$ **do**
3:     Resample all audio in $S$ to 16 kHz.
4:     Remove samples with duration > 98th percentile duration in $S$.
5:     Identify classes $C_S$ in $S$.
6:     Remove classes $c \in C_S$ where $|c| < 6$.
7:     Apply source-specific frequency/class filtering (e.g., LibriSpeech-aligned, VCTK, AMI).
8:     **if** $S$ is a birdsong subset **then**
9:        Make labels unique per bird (e.g., $bird1\_syllA$).
10:    **end if**
11:    Let $S_{filtered}$ be the resulting subset.
12:    **if** $|S_{filtered}| > 17000$ **then**
13:       Rank classes in $S_{filtered}$ by size descending. {Optional step for AE/VAE training efficiency}
14:       Uniformly select classes to form $S'_{filtered}$ with $|S'_{filtered}| \approx 10k - 17k$.
15:       $S_{final} \leftarrow S'_{filtered}$
16:    **else**
17:       $S_{final} \leftarrow S_{filtered}$
18:    **end if**
19:    Add $S_{final}$ to $D_{proc}$.
20: **end for**
21: **return** $D_{proc}$

---

## D.8. VAE and AE Training Details

Our evaluation includes custom unsupervised models to serve as strong, domain-specific baselines. All models were trained without access to any labels.

**Per-Subset Models (Domain-Specific Baselines)**   The primary custom baselines reported in our results (labeled **VC** for VAE and **AC** for AE) were trained on a per-subset basis. For each of the 19 subsets in VOCSIM, including the blind test sets, a separate AE and VAE model was trained exclusively on the unlabeled audio from that specific subset. This approach does not create a single generalist model; instead, it establishes a strong, domain-specific performance baseline for each task. This allows us to directly quantify the benefit of large-scale pre-training by comparing foundation models against a simple unsupervised model that is perfectly adapted to the target domain's acoustic statistics.

**VAE Training**   The VAE compresses $128 \times 128$ log-Mel spectrogram patches into a 32-dimensional Gaussian latent $(\mu, \sigma^2)$ and reconstructs them via a symmetric decoder. Its loss is the negative Evidence Lower Bound (ELBO), based on (Goffinet et al., 2021):

$$\mathcal{L}_{\text{VAE}} = \underbrace{\mathbb{E}_{q(z|x)}[-\log p(x|z)]}_{\text{reconstruction error}} + \underbrace{D_{\text{KL}}\big[q(z|x) \,\|\, \mathcal{N}(0, I)\big]}_{\text{latent regularization}}.$$

Training hyperparameters:

- Optimizer: Adam, $\alpha = 1 \times 10^{-3}$, betas $(0.9, 0.999)$

- Batch size: 64 spectrogram chunks (50% overlap)

- Epochs: up to 50, with early stopping after 10 epochs without ELBO improvement

- Spectrogram frontend: 512-sample FFT, 256-sample hop, 128-band Mel filter, log-scaling

Because the KL term enforces a smoothly varying latent space, reconstructions preserve overall patterns but exhibit modest smoothing of fine spectral detail (Figure 3).

**AE Training** The AE encodes full-length Mel spectrograms into a 256-dimensional bottleneck and decodes them back. Its objective is pure reconstruction with L1 loss plus a small sparsity penalty on the bottleneck, based on (Best et al., 2023):

$$\mathcal{L}_{\mathrm{AE}} = \|X - \widehat{X}\|_1 \; + \; 0.01 \, \|Z\|_1.$$

Training hyperparameters: Optimizer: AdamW, $\alpha = 3 \times 10^{-4}$, weight decay $1 \times 10^{-2}$. Batch size: 128 full-spectrograms. Epochs: up to 50, with ReduceLROnPlateau (factor 0.5, patience 5) and early stopping after 10 epochs without loss improvement. Mixed-precision training enabled for GPU. Because it optimizes only reconstruction fidelity, the AE reproduces fine spectro-temporal details very closely, resulting in sharp, high-fidelity outputs (Figure 4).

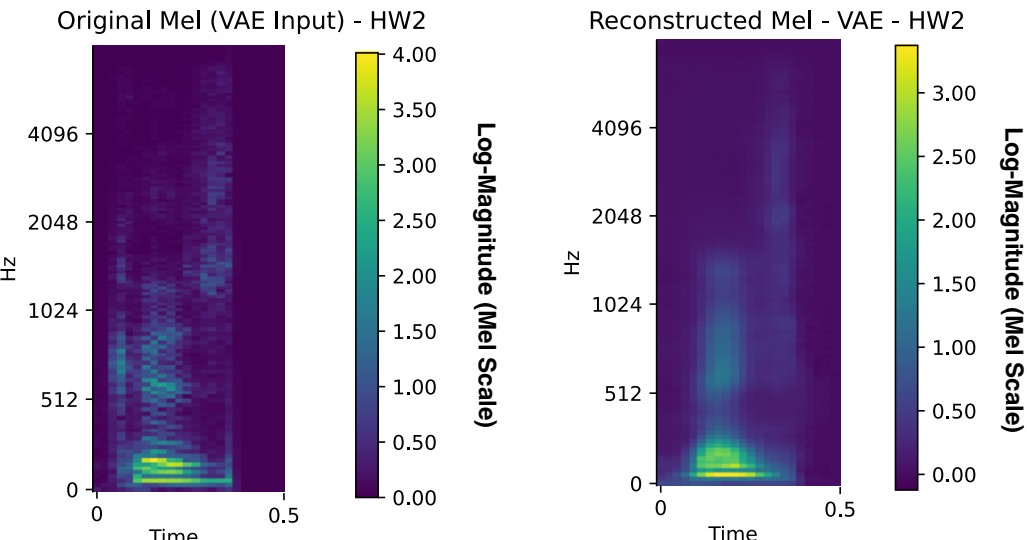

*Figure 3.* VAE reconstruction (HW2). **Left:** Original log-Mel input. **Right:** Reconstructed output, showing smoothness due to KL regularization.

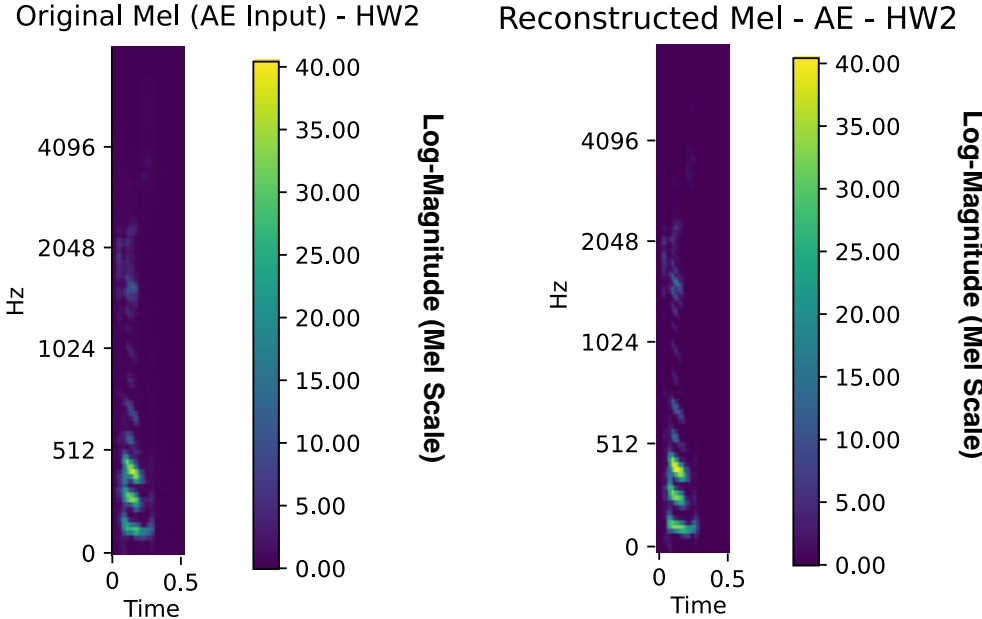

*Figure 4.* AE reconstruction (HW2). **Left:** Original log-Mel input. **Right:** Decoded output, closely matching fine spectral details.

The domain-specific VAE baselines perform *worse* than random chance on Blind sets (Lift -7.8). This indicates that training

from scratch on small, low-resource datasets ($N < 5k$) leads to overfitting on channel artifacts rather than learning content identity, reinforcing the necessity of transfer learning from large foundation models

### D.9. Feature Extraction Procedure

All audio clips in VOCSIM were first resampled to 16 kHz (except where models internally require a different rate) and amplitude-normalized. We then extracted a variety of feature embeddings, some learned directly on VOCSIM, others borrowed from large pretrained networks, and, where necessary, applied pooling or dimensionality reduction to obtain fixed-length vectors.

**Log-Mel Spectrograms (M)** We compute 128-band log-Mel spectrograms with a 512-sample FFT window and 256-sample hop (Hann window), followed by $\log(1 + x)$ compression (Davis & Mermelstein, 1980). Each clip yields a $128 \times T$ time-frequency matrix, which we flatten to a single vector for distance calculations.

**VAE Embeddings (VC)** Using our convolutional variational autoencoder trained on all VOCSIM sounds (Appendix D.8), we split each clip into overlapping $128 \times 128$-frame spectrogram patches. Each patch is mapped to a 32-dimensional latent mean vector by the encoder. We obtain 32xT-D representation per clip (Goffinet et al., 2021).

**AE Embeddings (AC)** Our convolutional autoencoder compresses full-length Mel spectrograms into a 256-D bottleneck. After feeding a clip's spectrogram through the encoder, yielding a 256xT-D vector per clip (Best et al., 2023).

**EAT (EAT)** We include the Efficient Audio Transformer (EAT) (Chen et al., 2024), a recent spectrogram-based model, as a strong baseline. We use the `worstchan/EAT-large` version fine-tuned on AudioSet-2M. The model processes a normalized log-Mel spectrogram of the input audio. To obtain a single, fixed-length vector representing the entire clip, we take the output embedding of the special `[CLS]` token from the final Transformer layer. This provides a 1024-dimensional utterance-level representation.

**BEATs (BEATs)** We use the BEATs model (Chen et al., 2023), specifically the `BEATs_iter3_plus_AS2M.pt` checkpoint pretrained on AudioSet-2M. The model internally processes raw audio by first computing a 128-bin log-Mel spectrogram (fbank), which is then normalized using the official preset mean (15.41663) and standard deviation (6.55582). This spectrogram is divided into patches, which are fed into a Transformer encoder. We extract the output from the final encoder layer, which produces a sequence of frame-level embeddings that are then handled by our standard pooling and optional PCA pipeline.

**Whisper Encoder (EW)** We employ the encoder of OpenAI's Whisper model (`openai/whisper-large-v3`) (Radford et al., 2022). Input audio is padded or truncated to its required 30-second context window. The encoder produces hidden states of shape $[1500 \times 1280]$ ($T \times D$). **Pooling Strategy:** To derive a fixed-length representation, we apply the model's native attention mask to identify valid audio frames. We compute the mean over the time axis strictly on these non-padding tokens, ensuring the embedding represents the active signal content rather than silence. We do not run external VAD, relying on the model's internal masking.

**1. Encoder Self-Attention:** The `transformers` library implementation correctly uses an attention mask. This ensures that when the model computes the representation for a valid audio frame, its self-attention mechanism *only* attends to other valid audio frames and ignores the padded regions. This is critical for generating a meaningful contextual representation of the audio content itself.

**2. Post-Hoc Pooling:** Our simple post-hoc pooling operations (e.g., mean over the time axis) are applied to the full $[1500 \times 1280]$ output sequence without an explicit mask. The output vectors corresponding to padded input frames are not zero; they are valid "padding embeddings" that the model learns to produce for silent or non-audio inputs. Including these vectors in a simple mean pooling operation has a mild regularizing effect, pulling the final vector slightly towards a neutral, "no-audio" representation based on the clip's duration. While masked pooling (which would involve explicitly excluding these padding embeddings from the mean calculation) is a valid alternative, our approach serves as a simple, common, and empirically effective heuristic for creating fixed-length representations from the variable-length encoder outputs.

**WhisperSeg Encoder (E)**   A CTranslate2-converted Whisper variant with a custom voice-activity detection front end (Gu et al., 2023) processes each clip into a sequence of $[750 \times 1280]$ (T × D) embeddings. This model is tuned for animal sounds and delivers frame-level features comparable to standard Whisper. Unless otherwise noted, we use the same default mean variant as in our main tables and report both **Mean (incl. pad)** and **Mean (mask-excluding)** in ablations.

**Wav2Vec 2.0 (W2V) and WavLM (WLM)**   We extract contextualized features from pretrained Wav2Vec 2.0 (768-D per frame) (Baevski et al., 2020) and WavLM (768-D per frame) (Chen et al., 2022). Raw waveforms are fed directly, and the final Transformer layer's activations are used as our sequence features.

**CLAP (CLP)**   From the Contrastive Language–Audio Pretraining model (Elizalde et al., 2022), we take the final audio embedding (typically 512-D) produced for each clip after audio–text joint training on web data.

**AudioMAE (MAE)**   A masked-spectrogram autoencoder (Huang et al., 2023) yields a fixed 768×512 feature map per clip. We flatten this map into a single vector.

**EnCodec Codes (CC)**   Using a pretrained neural audio codec (Défossez et al., 2022), we extract discrete codebook indices (e.g., 8 codebooks × $T$ frames), which we concatenate into a long integer-valued vector representation.

We include EnCodec as a representative discrete-token baseline to situate neural codec approaches within our framework. Our primary metrics operate in continuous vector spaces to probe embedding geometry directly; a comprehensive, training-free evaluation tailored to discrete sequences (e.g., token-level edit or alignment distances) entails different design choices and is orthogonal to VocSim's focus on continuous content-identity geometry. We therefore treat EnCodec as a reference point rather than a dedicated sequence-similarity track.

For all encoder models, we extract features from the final Transformer layer's activations, a standard practice for obtaining the most contextually-rich representations in frozen-feature evaluations.
For models like EAT and CLAP that already produce a single utterance-level vector per clip (e.g., the [CLS] token in EAT or the final audio embedding in CLAP), we treat this as the fixed-length embedding and optionally apply PCA, without applying any additional temporal pooling.

**Derived Fixed-Length Vectors**   For models that produce a sequence of embeddings, we adopt a canonical shape of [T x D] (Time frames x Feature dimensions). We derive fixed-length vectors using one or more of the following explicit pooling operations:

- **mean_time**: Mean pooling across the time axis (axis 0), yielding a single vector of length **D**.

- **mean_feat**: Mean pooling across the feature axis (axis 1), yielding a vector of length **T**.

- **first_time**: Taking the full feature vector from the first time frame (t=0), yielding a vector of length **D**.

- **first_feat**: Taking the activation values of the first feature dimension (d=0) across all time frames, yielding a vector of length **T**.

- **Concatenated Pooling**: Concatenating vectors derived from two of the above methods, such as concat_mean_time_feat (length D+T) or concat_first_time_feat (length D+T).

After pooling, we optionally apply Principal Component Analysis (PCA) fit on each VocSim subset to produce the final embeddings. Below are two concise tables: one for the base extractors and one for the pooled+PCA variants. Unless otherwise noted, we use **Mean (incl. pad)** as the default pooling for sequential encoders to obtain fixed-length vectors.

*Table 9.* Base Feature Extractors and Their Raw Output Shapes (canonical $[T \times D]$ format)

| Short | Extractor | Raw Output Shape [T x D] |
|---|---|---|
| M | Log-Mel spectrogram | [T x 128] |
| VC | VAE latent mean | [T x 32] |
| AC | AE bottleneck | [T x 256] |
| EW | Whisper encoder | [1500 x 1280] |
| E | WhisperSeg encoder | [750 x 1280] |
| WLM | WavLM encoder | [T x 768] |
| W2V | Wav2Vec 2.0 encoder | [T x 768] |
| EAT | EAT [CLS] token | [1024] |
| CLP | CLAP audio embedding | [512] (Already fixed-length) |
| MAE | AudioMAE masked-spectrogram features | [512 x 768] |
| CC | EnCodec discrete codes | [T x 8] |

*Table 10.* Derived Fixed-Length Embeddings via Pooling and PCA

| Base (Shape [T x D]) | Pooling / Variant (Explicit Name) | PCA Dims | Final Length |
|---|---|---|---|
| **Whisper (EW)** [1500x1280] | first_time (ET) | – | 1280 |
| | + PCA (ET D30/D100) | 30/100 | 30 / 100 |
| | first_feat (EF) | – | 1500 |
| | + PCA (EF D30/D100) | 30/100 | 30 / 100 |
| | concat_first_time_feat (EWTF) | – | 2780 |
| | + PCA (EWTF D30/D100) | 30/100 | 30 / 100 |
| | concat_mean_time_feat (EWMTF) | – | 2780 |
| | + PCA (EWMTF D30/D100) | 30/100 | 30 / 100 |
| **WhisperSeg (E)** [750x1280] | first_time (ET) | – | 1280 |
| | + PCA (ET D30/D100) | 30/100 | 30 / 100 |
| | first_feat (EF) | – | 750 |
| | + PCA (EF D30/D100) | 30/100 | 30 / 100 |
| | concat_first_time_feat (ETF) | – | 2030 |
| | + PCA (ETF D30/D100) | 30/100 | 30 / 100 |
| | concat_mean_time_feat (EMTF) | – | 2030 |
| | + PCA (EMTF D30/D100) | 30/100 | 30 / 100 |
| **CLAP / MAE / CC / WavLM / W2V** | + PCA (D30/D100) | 30/100 | 30 / 100 |

**Notes:**

- $T$ = number of time frames;

- "–" under PCA indicates the raw pooling (no dimensionality reduction).

- "first_row_col" concatenates the first time-step and first F; "mean_row_col" concatenates the mean over time and mean over F. F is frequency for spectrograms and feature/channel for hidden-state sequences.

## E. Notes on Spearman dissimilarity

Spearman's rank-based dissimilarity is computationally heavier than Cosine or Euclidean, but offers two advantages in our setting: (i) reduced sensitivity to feature scaling and marginal distributions induced by pooling and PCA; and (ii) mitigation of hubness in high-D spaces by emphasizing relative order rather than absolute magnitudes ("Radovanović et al., 2010). Its consistent gains across foundation models suggest rank-order relationships are a good fit for content-identity geometry.

# F. Benchmark Methods Details

Our evaluation pipeline is deterministic: for a given dataset, feature extractor, and distance metric, the resulting distance matrix and all subsequent benchmark scores are identical on every run. Therefore, run-to-run stochasticity is zero, and we report point estimates. This section details the algorithms for the metrics used in our analysis.

## F.1. Precision@k (P@k) Algorithm

Precision@k is a standard local metric that measures the purity of an item's immediate neighborhood. It is a direct and intuitive measure of the local coherence of the embedding space.

1. **Input:** A square pairwise distance matrix $D$ of size $N \times N$, a list of class labels $L$ of length $N$, and a set of integers $k$ (e.g., $\{1, 5\}$).

2. **Data Filtering:** Items with no valid labels are excluded from the evaluation.

3. **Per-Point Precision Calculation:** For each evaluable data point $i$:

   3.a. Let $C(i)$ be the class of point $i$.
   3.b. Identify the set $N_k(i)$ of the $k$ nearest neighbors to point $i$ (excluding $i$ itself) based on the distances in $D$.
   3.c. Count the number of neighbors in $N_k(i)$ that share the same class as point $i$.

   $$\text{CorrectNeighbors}_k(i) = |\{j \mid j \in N_k(i), C(j) = C(i)\}|$$

4. **Aggregation:** The final P@k score is the total number of correct neighbors across all points, divided by the total number of neighbors considered ($N_{\text{evaluable}} \times k$). This is equivalent to the average proportion of correct neighbors.

   $$\text{P@k} = \frac{\sum_i \text{CorrectNeighbors}_k(i)}{N_{\text{evaluable}} \times k}$$

5. **Output:** The final P@k score, a value in $[0, 1]$.

## F.2. Global Separation Rate (GSR) Algorithm

The Global Separation Rate (GSR) is a robust, point-wise global metric that averages the local separation of every point in the dataset. It provides a more continuous and outlier-resistant measure than binary, percentile-based methods.

The algorithm is implemented as follows:

1. **Input:** A square pairwise distance matrix $D$ of size $N \times N$, a list of class labels $L$ of length $N$, and a minimum class size parameter `min_class_size` (e.g., 2).

2. **Data Filtering:** Items with no valid labels or belonging to classes with fewer than `min_class_size` samples are excluded from the evaluation.

3. **Per-Point Score Calculation:** For each evaluable data point $i$:

   3.a. Let $C(i)$ be the class of point $i$.
   3.b. Find the **Average Intra-class Distance (Avg_ID)**: The mean of distances from point $i$ to all other points within its own class.
   $$\text{Avg\_ID}_i = \text{Mean}\left(\{d(i, j) \mid j \neq i, C(j) = C(i)\}\right)$$

   3.c. Find the **Nearest Inter-class Distance (NID)**: The distance from point $i$ to the closest point from any other class.
   $$\text{NID}_i = \min_{k: C(k) \neq C(i)} d(i, k)$$

   3.d. Calculate the **Local Separation Score** for point $i$, which ranges from -1 (total overlap) to +1 (perfect local separation).
   $$\text{Local\_Score}_i = \frac{\text{NID}_i - \text{Avg\_ID}_i}{\text{NID}_i + \text{Avg\_ID}_i + \epsilon}$$

4. **Calculate Final GSR Score:** The final score is the average of all local scores, normalized to the range [0, 1].

$$\text{GSR}_{\text{norm}} = \frac{1}{2}\left(\frac{\sum_i \text{Local\_Score}_i}{\text{Number of evaluable points}} + 1\right)$$

This score is then multiplied by 100 to be presented as a percentage in all result tables.

$$\text{GSR} = \text{GSR}_{\text{norm}} \times 100$$

5. **Output:** The final normalized GSR score.

### F.3. Class Separation Ratio (CSR) Algorithm

The Class Separation Ratio (CSR) is a point-wise metric, similar in spirit to GSR, but it compares the nearest inter-class distance to the *furthest* intra-class distance. It evaluates how well a class is separated from others relative to its own internal spread.

1. **Input:** A square pairwise distance matrix $D$ of size $N \times N$, a list of class labels $L$ of length $N$, and a minimum class size parameter `min_class_size`.

2. **Data Filtering:** Items with no valid labels or belonging to classes with fewer than `min_class_size` samples are excluded.

3. **Per-Point Score Calculation:** For each evaluable data point $i$:

   3.a. Let $C(i)$ be the class of point $i$.

   3.b. Find the **Maximum Intra-class Distance (MID)**: The distance from point $i$ to the *furthest* point within its own class.
   $$\text{MID}_i = \max_{j: j \neq i, C(j) = C(i)} d(i, j)$$

   3.c. Find the **Nearest Inter-class Distance (NID)**: The distance from point $i$ to the *closest* point from any other class.
   $$\text{NID}_i = \min_{k: C(k) \neq C(i)} d(i, k)$$

   3.d. Calculate the **Local Class Separation Score** for point $i$, ranging from -1 to +1.
   $$\text{Local\_CSR}_i = \frac{\text{NID}_i - \text{MID}_i}{\text{NID}_i + \text{MID}_i + \epsilon}$$

4. **Aggregation:** The local scores are first averaged within each class. The final score is the weighted average of these per-class scores, weighted by class size.

$$\text{CSR}_{\text{raw}} = \frac{\sum_{c \in \text{Classes}} |c| \times \text{Mean}(\{\text{Local\_CSR}_i \mid i \in c\})}{\text{Total number of evaluable points}}$$

5. **Normalization:** The final score is normalized to the range [0, 1], where 1.0 is best.

$$\text{CSR} = \frac{1}{2}(\text{CSR}_{\text{raw}} + 1)$$

### F.4. F-Value Benchmark (CS) Algorithm

The F-Value, which we abbreviate as CS (Class Separation), is a class-pair-wise metric that measures the ratio of inter-class separation to intra-class compactness. A higher score indicates better separation.

1. **Input:** A distance matrix $D$ and labels $L$.

2. **Per-Class Statistics:** For each valid class $C_i$ (with at least `min_class_size` samples):

- Calculate the **Average Intra-class Distance**:

$$\text{AvgIntra}(C_i) = \text{Mean}(\{d(a, b) \mid a, b \in C_i, a \neq b\})$$

3. **Per-Class-Pair Calculation:** For every ordered pair of distinct classes $(C_i, C_j)$:

- Calculate the **Average Inter-class Distance**:

$$\text{AvgInter}(C_i, C_j) = \text{Mean}(\{d(a, b) \mid a \in C_i, b \in C_j\})$$

- Calculate a separation ratio, where a larger value indicates better separation between the class pair.

$$S_{ij} = \frac{\text{AvgInter}(C_i, C_j)}{\text{AvgIntra}(C_i) + \epsilon}$$

- To create a bounded score in the range [0, 1] where higher is better, we transform this ratio:

$$F_{\text{transformed},ij} = \frac{S_{ij}}{1 + S_{ij}}$$

4. **Aggregation:** The final CS score is the mean of all transformed F-Values over all $M \times (M - 1)$ ordered class pairs.

### F.5. Class Separation Confusion Fraction (CSCF) Algorithm

CSCF is an intuitive, class-pair-wise metric that counts the fraction of "confused" class pairs. A lower raw score is better.

1. **Input:** A distance matrix $D$ and labels $L$.

2. **Per-Class Statistics:** As with the F-Value, calculate $\text{AvgIntra}(C_i)$ for each class $C_i$.

3. **Per-Class-Pair Calculation:** For every ordered pair of distinct classes $(C_i, C_j)$:

- Calculate $\text{AvgInter}(C_i, C_j)$.
- A **confusion event** occurs if the average distance between the classes is smaller than the average internal distance of the anchor class $C_i$.

$$\text{IsConfused}(i, j) = \begin{cases} 1 & \text{if } \text{AvgInter}(C_i, C_j) < \text{AvgIntra}(C_i) \\ 0 & \text{otherwise} \end{cases}$$

4. **Aggregation:** The final CSCF score is the total number of confusion events divided by the total number of ordered class pairs.

$$\text{CSCF} = \frac{\sum_{i \neq j} \text{IsConfused}(i, j)}{M \times (M - 1)}$$

### F.6. Clustering Purity Benchmark Algorithm

This benchmark evaluates how well an embedding's intrinsic structure aligns with the ground-truth labels in a completely unsupervised setting.

1. **Data Preparation:** Start with the fixed-length embeddings for all samples in a subset.

2. **Dimensionality Reduction (UMAP):** Project the high-dimensional embeddings into a 2D space using UMAP (McInnes et al., 2020). This step helps preserve both local and global structure in a space that is more amenable to density-based clustering.

3. **Clustering (HDBSCAN):** Apply the HDBSCAN algorithm (McInnes & Healy, 2017) to the 2D UMAP projection. HDBSCAN is used for its ability to find clusters of varying shapes and densities, and for its robustness in identifying points that do not belong to any cluster (noise).

4. **Weighted Purity Calculation:** The quality of the resulting clusters is measured against the ground-truth labels.

    4.a. For each cluster discovered by HDBSCAN (excluding noise points), identify the majority ground-truth class among its members.

    4.b. The purity of that cluster is the fraction of its members belonging to that majority class.

    4.c. The final **Weighted Purity** is the size-weighted average of these individual cluster purities:

$$\text{Purity}_{\text{weighted}} = \frac{\sum_j |C_j| \times \text{purity}(C_j)}{\sum_j |C_j|}$$

    where $C_j$ is the set of items in the $j$-th cluster found by HDBSCAN.

**Silhouette Score**    The Silhouette Score is a metric that quantifies the quality of clusters by measuring how similar an object is to its own group (cohesion) compared to other groups (separation). In our benchmark, we compute this score not on clusters discovered by an algorithm, but directly on the ground-truth classes. This "supervised" use of the metric serves as a well-established measure of the geometric coherence of the true classes within the embedding space. For each data point, it compares its average distance to other points in its own class with its average distance to points in the nearest neighboring class. A high score indicates that the ground-truth classes are dense and well-separated.

1. **Intra-Cluster Cohesion** ($a(i)$)**:** The average distance between point $i$ and all other points within the same ground-truth class. A low value for $a(i)$ indicates that the point is well-matched to its own cluster.

2. **Inter-Cluster Separation** ($b(i)$)**:** The average distance from point $i$ to all points in the single nearest neighboring cluster (i.e., the cluster that is closest to point $i$, to which $i$ does not belong). A high value for $b(i)$ indicates that the point is well-separated from neighboring clusters.

The silhouette score for point $i$ is then given by the formula:

$$s(i) = \frac{b(i) - a(i)}{\max\{a(i), b(i)\}}$$

The overall score is the mean of $s(i)$ for all points. The score ranges from -1 to +1, where a value near +1 indicates dense and well-separated clusters, a value near 0 indicates overlapping clusters, and a value near -1 suggests that points have likely been assigned to the wrong cluster.

### F.7. P@k Baseline Calculation

To rigorously evaluate whether the observed Precision@k (P@k) scores reflect true class structure or are simply an artifact of the embedding's intrinsic geometry, we established an empirical random baseline using permutation tests. This procedure allows us to determine the P@k score that would be expected by chance for a given embedding space and to test the statistical significance of the observed score.

**Procedure**    The test was conducted for the top-performing configuration (Whisper 'EWMTF D100' embeddings with Spearman distance) on each of the 19 VOCSIM subsets.

1. **Calculate Observed Score:** For a given subset, the pairwise distance matrix is computed once. The true P@k scores (for k=1 and k=5) are then calculated using this matrix and the ground-truth labels.

2. **Create Null Distribution:** The core of the method involves creating a null distribution of scores that could occur by chance. To do this, we hold the distance matrix fixed and randomly shuffle the ground-truth labels 1,000 times. For each of these permutations, we recalculate the P@k scores. This process generates a distribution of P@k scores expected under the null hypothesis that there is no relationship between the embedding geometry and the class labels.

3. **Calculate Statistics:** From this null distribution, we compute:

    • **The Baseline Mean P@k:** The average of the 1,000 permuted P@k scores, which serves as our empirical random baseline.

- **The 95% Confidence Interval (CI):** The range containing 95% of the permuted scores (from the 2.5th to the 97.5th percentile), indicating the expected spread of random scores.
- **The p-value:** The proportion of permuted scores that were greater than or equal to the originally observed P@k score. A low p-value (e.g., $p < 0.001$) indicates that the observed score is highly unlikely to have occurred by chance.

This analysis provides a robust statistical foundation for interpreting the P@k results. The aggregated results are summarized in Table 11 and Table 12.

*Table 11.* Empirical Permutation Test for P@1 Significance. Results compare the observed P@1 of the top-performing embedding against a baseline derived from 1000 label permutations per subset.

| Set Type | Observed P@1 (Mean %) | Baseline P@1 (Mean %) | Baseline 95% CI (Mean) | p < 0.001 (Count) |
|---|---|---|---|---|
| Public | 66.80 | 5.80 | [5.36, 6.13] | 15/15 |
| Blind | 11.45 | 0.92 | [0.70, 1.18] | 4/4 |

*Note: Baseline for EWMTF D100 (PCA) with Spearman distance.*

*Table 12.* Empirical Permutation Test for P@5 Significance. Results compare the observed P@5 of the top-performing embedding against a baseline derived from 1000 label permutations per subset.

| Set Type | Observed P@5 (Mean %) | Baseline P@5 (Mean %) | Baseline 95% CI (Mean) | p < 0.001 (Count) |
|---|---|---|---|---|
| Public | 57.35 | 5.80 | [5.62, 6.01] | 15/15 |
| Blind | 7.67 | 0.91 | [0.80, 1.02] | 4/4 |

*Note: Baseline for EWMTF D100 (PCA) with Spearman distance.*

## G. GSR Baseline Calculation

To rigorously evaluate whether the observed Global Separation Rate (GSR) scores reflect true class structure or are simply an artifact of the embedding's intrinsic geometry, we established an empirical random baseline using permutation tests. This procedure allows us to determine the GSR score that would be expected by chance for a given embedding space and to test the statistical significance of the observed score.

**Procedure**   The test was conducted for the top-performing configuration (Whisper 'EWMTF D100' embeddings with Spearman distance) on each of the 19 VOCSIM subsets.

1. **Calculate Observed Score:** For a given subset, the pairwise distance matrix is computed once. The true GSR score is then calculated using this matrix and the ground-truth labels.

2. **Create Null Distribution:** The core of the method involves creating a null distribution of scores that could occur by chance. To do this, we hold the distance matrix fixed and randomly shuffle the ground-truth labels 1,000 times. For each of these permutations, we recalculate the GSR score. This process generates a distribution of GSR scores expected under the null hypothesis that there is no relationship between the embedding geometry and the class labels.

3. **Calculate Statistics:** From this null distribution, we compute:

   - **The Baseline Mean GSR:** The average of the 1,000 permuted GSR scores, which serves as our empirical random baseline.
   - **The 95% Confidence Interval (CI):** The range containing 95% of the permuted scores (from the 2.5th to the 97.5th percentile), indicating the expected spread of random scores.
   - **The p-value:** The proportion of permuted scores that were greater than or equal to the originally observed GSR score. A low p-value (e.g., $p < 0.001$) indicates that the observed score is highly unlikely to have occurred by chance.

This analysis provides a robust statistical foundation for interpreting the GSR results. The aggregated results are summarized in Table 13.

*Table 13.* Empirical Permutation Test for GSR Significance. Results compare the observed GSR of the top-performing embedding against a baseline derived from 1000 label permutations per subset.

| Set Type | Observed GSR (Mean %) | Baseline GSR (Mean %) | Baseline 95% CI (Mean) | p < 0.001 (Count) |
|---|---|---|---|---|
| Public | 41.76 | 24.90 | [24.82, 24.98] | 15/15 |
| Blind | 39.52 | 33.74 | [33.69, 33.80] | 4/4 |

*Note: Baseline for EWMTF D100 (PCA) with Spearman distance.*

### G.1. Interpretation of Metric Correlations

The correlation matrix (Table 14) reveals the relationships between different evaluation metrics. For this analysis, all metrics were transformed such that **higher scores indicate better performance** (e.g., CSCF becomes $1 - \text{CSCF}_{raw}$). The correlations are calculated using Spearman's $\rho$ and are averaged across all public VOCSIM subsets.

*Table 14.* Spearman Rank Correlation ($\rho$) of Key Performance Metrics on Public Subsets. All metrics are transformed such that higher values indicate better performance.

| | GSR | Silhouette | P@1 | P@5 | CSR | CS | CSCF |
|---|---|---|---|---|---|---|---|
| GSR | 1.00 | 0.82 | 0.77 | 0.83 | 0.95 | 0.15 | 0.77 |
| Silhouette | 0.82 | 1.00 | 0.80 | 0.85 | 0.72 | 0.40 | 0.82 |
| P@1 | 0.77 | 0.80 | 1.00 | 0.95 | 0.71 | 0.46 | 0.85 |
| P@5 | 0.83 | 0.85 | 0.95 | 1.00 | 0.75 | 0.49 | 0.90 |
| CSR | 0.95 | 0.72 | 0.71 | 0.75 | 1.00 | -0.05 | 0.74 |
| CS | 0.15 | 0.40 | 0.46 | 0.49 | -0.05 | 1.00 | 0.32 |
| CSCF | 0.77 | 0.82 | 0.85 | 0.90 | 0.74 | 0.32 | 1.00 |

To generate the correlation matrix in Table 14, we followed a two-step process. First, for each feature-distance configuration (e.g., 'EWMTF D100 - Spearman'), we calculated its average score on each metric (P@1, GSR, etc.) across all 15 public VOCSIM subsets. This yielded a single summary value per metric for each of the configurations evaluated. Second, we computed the Spearman rank correlation ($\rho$) between the vectors of these summary scores. This analysis reveals how the ranking of different embedding methods according to one metric corresponds to their ranking according to another.

**High Correlations ($\rho > 0.8$):** The analysis reveals a cluster of highly inter-correlated metrics, suggesting they measure a similar underlying quality of embedding geometry.

- **GSR vs. CSR (0.95):** This very high correlation is expected. Both metrics assess the integrity of class boundaries relative to internal class spread, with GSR operating on a point-wise basis and CSR on a class-wise basis. Their strong agreement confirms they capture the same fundamental property.

- **P@1 vs. P@5 (0.95):** This is also an intuitive result. An embedding that correctly identifies the single nearest neighbor (high P@1) is highly likely to have multiple correct neighbors within its top five (high P@5).

- **Silhouette, P@5, and CSCF ($\rho \geq 0.82$):** These three metrics exhibit strong positive correlations with each other and with GSR. This indicates that embeddings with good cluster cohesion versus separation (Silhouette) also tend to have pure local neighborhoods (P@5). However, in practice, we observe cases (e.g., noisy bioacoustics) where Silhouette remains high due to cluster density, while GSR drops significantly because it detects subtle leakage between nearest neighbors. This makes GSR a safer conservative metric for retrieval systems where false positives are costly.

**Moderate Correlations ($0.7 < \rho < 0.8$):** This group shows solid relationships, reinforcing the connections between different aspects of a well-structured embedding space.

- **Boundary Metrics (GSR/CSR) vs. Neighborhood Purity (P@k):** The correlations here range from 0.71 to 0.83. This demonstrates that embeddings with clear, well-defined class boundaries (high GSR/CSR) reliably produce pure local neighborhoods where a point's nearest neighbors share its class.

- **Boundary Metrics (GSR/CSR) vs. Centroid Separation (CSCF):** With correlations around 0.74 to 0.77, the data shows a strong tendency for embeddings with sharp class boundaries to also have well-separated class averages.

**Low or Near-Zero Correlations:** The most significant insight comes from the CS (Class Separation / F-Value) metric, which behaves largely independently from the other metrics. This highlights that CS measures a distinct geometric property.

- **Consensus Metrics (GSR, CSR, P@k, Silhouette, CSCF):** This large group is moderately to highly inter-correlated, collectively rewarding embeddings that produce distinct, internally consistent clusters with sharp boundaries and pure local neighborhoods.

- **Outlier Metric (CS):** This metric is based on the ratio of the average distance between all inter-class pairs to all intra-class pairs. It is sensitive to the global placement of clusters' "centers of mass" but less so to the integrity of their boundaries.

This distinction explains the uniquely low correlations involving CS:

- **CS vs. CSR ($\rho = -0.05$):** This near-zero correlation is a crucial finding. It demonstrates that achieving sharp class boundaries (high CSR) has no systematic relationship with maximizing the average distance between clusters (high transformed CS). An embedding can produce extremely tight, compact clusters (which is favorable for CSR) while these clusters remain geometrically close to one another, resulting in a modest CS score.

- **CS vs. All Other Metrics ($\rho \approx 0.15 - 0.49$):** CS shows only weak positive correlations with the entire consensus group. This indicates that knowing an embedding's CS score is a poor predictor of its performance on metrics that measure boundary integrity (GSR, CSR), local neighborhood purity (P@k), or cluster cohesion (Silhouette).

In summary, the analysis reveals a strong agreement among six key performance metrics. The CS metric stands apart, measuring a different aspect of embedding quality that is not strongly correlated with the properties measured by the rest.

### G.2. Details on Global Separation Rate (GSR)

The formulation of GSR establishes a theoretical neutral point at 50%, corresponding to the mathematical case where the nearest inter-class distance ('NID') equals the average intra-class distance ('Avg_ID'). However, this theoretical point does not represent the performance of a random baseline on a real-world embedding space. Because GSR compares a *minimum* ('NID') with an *average* ('Avg_ID'), its expected value on a structured but randomly labeled space is non-obvious and depends entirely on the embedding's geometry.

A high GSR score requires both clear class boundaries (a large 'NID') and high intra-class compactness (a small 'Avg_ID'). Its significance is therefore best measured by its improvement over an **empirical random baseline**, which must be calculated for each dataset via permutation testing (Appendix G). As our results show, this baseline varies (e.g., $24.9\%$ for public sets vs. $33.7\%$ for blind sets), as it is sensitive to the intrinsic geometric structure of the point cloud for each data subset.

This formulation gives GSR two key advantages over the Silhouette Score. First, by using the distance to the single nearest inter-class neighbor (NID), GSR provides a much stricter, **point-wise** test of the class **boundary** for every single sample. Second, this reliance on the NID makes GSR inherently more robust to the non-convex manifold shapes common in audio, as it does not assume a coherent "neighboring cluster." While each local score is sensitive to outliers, the final GSR metric achieves robustness by averaging thousands of these scores across the dataset, yielding a stable, global measure of class separability.

## H. Ablation Studies and Robustness

**Methodological Robustness.** Our findings are supported by rigorous validations detailed in this appendix. An exhaustive sweep of all 32 Whisper encoder layers reveals remarkable performance stability across model depth, validating our standard use of the final layer. Further ablations confirm our conclusions are insensitive to the choice of pooling strategy and that our results are strongly corroborated by alternative clustering metrics (NMI, Purity, ARI). This comprehensive validation ensures our results reflect intrinsic embedding properties, not artifacts of our experimental design.

### H.1. Robustness to Label Noise

To evaluate the robustness of our primary metrics to potential annotation errors, we performed a label-noise sensitivity analysis. We systematically introduced noise by randomly flipping a fraction $y\%$ of the class labels, for $y \in \{1, 5, 10, 20\}$,

and then recomputed GSR and P@1. The experiment was run on three representative subsets (HP, BC, ES1) using the embeddings from our top-performing models (Whisper, CLAP, and WavLM). The results, shown in Figure 5, demonstrate that GSR is considerably more robust to label noise than P@1. As a local metric, P@1 degrades sharply as even a small fraction of incorrect labels corrupts the immediate neighborhoods of many points. In contrast, GSR, which averages a global signal across all points, degrades more smoothly. For instance, with the Whisper model on the BC subset, a 10% label noise rate causes P@1's performance to drop by 19.2% from its baseline, whereas GSR's performance decreases by only 8.8%. This trend holds across all tested models and subsets. This analysis validates that GSR is not merely a proxy for the mislabeled rate; it provides a more stable measure of an embedding's geometric integrity that is less susceptible to moderate levels of annotation noise, a common challenge in real-world datasets.

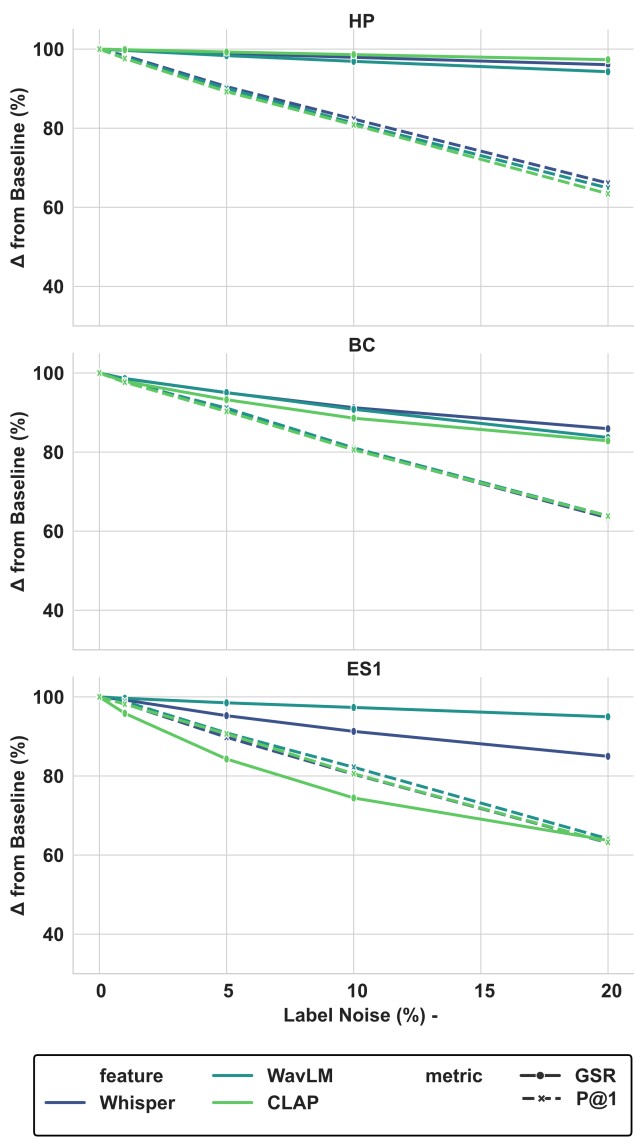

*Figure 5.* **Robustness to Label Noise: GSR vs. P@1.** Performance of GSR and P@1 as a function of the percentage of randomly flipped labels. Scores are normalized to the performance at 0% noise (100%) to show relative degradation. Across all models (Whisper, CLAP, WavLM) and subsets (HP, BC, ES1), GSR shows a much more graceful decay compared to the steep drop-off of P@1, highlighting its superior robustness to annotation errors.

## H.2. Sequence Pooling Ablation

To test sensitivity to the pooling method used to aggregate frame-level features, we compared six strategies for the Whisper encoder. The evaluation was run on three representative subsets, sampling their top 100 classes with 6 clips each. **Main-results default used throughout the paper: Whisper EWMTF D100 with Spearman distance**, where EWMTF = concat(mean_time [mask-excluding] and mean_F), with label-free per-subset PCA to 100D. The strategies are:

- **Mean (mask-excluding):** Mean over valid frames only.
- **Mean (incl. pad):** Mean over all output frames (including padding embeddings).
- **Max Pooling:** Element-wise maximum over frames.
- **First Time:** Only the first frame embedding.
- **Last Time:** Only the last frame embedding.
- **Attention Pooling:** Magnitude-weighted average of frame embeddings.

Both mean variants are shown below for direct comparison.

**Results.** Table 15 shows that while performance varies slightly, the overall conclusions are stable. Masked mean and max pooling, which consider the entire sequence, are effective and consistent. Time-only strategies (first/last) generally underperform. The relatively narrow performance range across methods indicates that our main findings are not an artifact of a specific pooling choice. Best scores within each subset/column are highlighted in **bold**.

*Table 15.* Pooling ablation results per subset (P@k and GSR shown as percentages).

| Subset | Method | P@1 (%) | P@5 (%) | GSR (%) |
|---|---|---|---|---|
| HP | Masked Mean (mask-excl.) | 14.46 | 9.41 | **39.78** |
| | Mean (incl. pad) | **15.9** | **12.1** | 39.4 |
| | Max Pool | 5.15 | 3.97 | 31.60 |
| | First Time | 5.64 | 4.17 | 20.70 |
| | Last Time | 3.92 | 3.14 | 14.37 |
| | Attention Pool | 5.88 | 3.77 | 19.53 |
| BC | Masked Mean (mask-excl.) | 41.07 | 22.02 | **40.50** |
| | Mean (incl. pad) | **65.5** | **58.9** | 36.5 |
| | Max Pool | 42.26 | 24.76 | 37.44 |
| | First Time | 27.98 | 18.21 | 29.69 |
| | Last Time | 10.71 | 11.31 | 22.24 |
| | Attention Pool | 26.79 | 18.45 | 28.78 |
| BS1 | Masked Mean (mask-excl.) | 97.22 | 89.44 | 61.64 |
| | Mean (incl. pad) | **99.2** | **96.1** | 53.3 |
| | Max Pool | 97.22 | 86.67 | **62.17** |
| | First Time | 97.22 | 82.78 | 60.29 |
| | Last Time | 94.44 | 68.33 | 48.70 |
| | Attention Pool | 94.44 | 78.89 | 57.82 |

## H.3. Sequence-Aware Distance via DTW Re-ranking

We also tested whether temporal pooling discards important sequence information by implementing a sequence-aware baseline using Dynamic Time Warping (DTW).

**Methodology.** We evaluated this baseline on five representative VOCSIM subsets (HP, BC, BS3, ES1, HU2), each sampled to include 5 clips from the 50 most frequent classes. The process was as follows:

1. **Feature Extraction:** We extracted frame-level features $[T \times 1280]$ from the final layer of a frozen Whisper-large-v3 encoder.
2. **Preprocessing:** Each sequence was truncated based on its true audio duration to remove padding, each frame was L2-normalized, dimensionality was reduced to $D=64$ using label-free PCA, and the sequence was temporally subsampled with a stride of 3.

3. **Candidate Shortlisting:** For efficiency, we first computed a full distance matrix using Spearman distance on the mean-pooled, PCA-reduced vectors.

4. **DTW Re-ranking:** For each query, we identified the top $M{=}200$ nearest candidates from the pooled distance matrix. We then computed the true sequence distance for only these candidate pairs using multi-dimensional DTW with a Sakoe–Chiba band (radius $r{=}0.1$) and path-length normalization.

This re-ranking approach preserves the zero-shot protocol by modifying only the distance function, while keeping the encoder frozen and computation tractable.

**Results.** As shown in Table 16, DTW re-ranking did not improve performance on average, and in most cases, it significantly degraded both local precision (P@k) and global separation (GSR). On very short clips (e.g., HP, BS3), the effective sequence length after subsampling was often too short (∼1–2 frames) for alignment to be meaningful. On longer clips (ES1), DTW slightly improved the global metric (GSR +1.1) but drastically hurt local precision (P@1 –20.8). These results strongly suggest that our simple pooling of contextualized frame embeddings is a highly effective and efficient strategy for this task.

*Table 16.* Average results of the DTW re-ranking ablation across five representative subsets.

| Method | P@1 (%) | P@5 (%) | GSR (%) |
|---|---|---|---|
| Pooled (Spearman) | 38.57 | 22.87 | 41.45 |
| DTW Re-rank (M=200) | 23.66 | 16.16 | 38.59 |
| **Delta (DTW − Pooled)** | **−14.91** | **−6.70** | **−2.86** |

**Per-Subset Breakdown and Computational Cost.** The full per-subset results are provided in Table 17. The average time for a single DTW comparison was 0.6 ms on an NVIDIA RTX 3090 GPU, with a total computation time of ∼2.1 minutes for the entire experiment.

*Table 17.* Per-subset results for the DTW re-ranking ablation. Metrics are P@1 / P@5 / GSR, all in percent.

| Subset | Pooled Baseline | DTW Re-rank | Delta (DTW - Pooled) |
|---|---|---|---|
| HP | 14.8 / 9.3 / 36.8 | 7.2 / 5.5 / 33.4 | −7.6 / −3.8 / −3.4 |
| BC | 35.7 / 19.9 / 41.4 | 12.9 / 7.6 / 38.1 | −22.8 / −12.3 / −3.3 |
| BS3 | 67.1 / 42.3 / 47.0 | 66.2 / 45.2 / 43.4 | −0.9 / +2.9 / −3.6 |
| ES1 | 47.6 / 29.4 / 42.7 | 26.8 / 17.8 / 43.8 | −20.8 / −11.6 / +1.1 |
| HU2 | 27.6 / 13.5 / 39.3 | 5.2 / 4.6 / 34.2 | −22.4 / −8.9 / −5.1 |

## H.4. Clustering-based Evaluation: NMI, Purity, and ARI

To provide an alternative, clustering-based view of representation quality, we followed the protocol of prior work like HuBERT (Hsu et al., 2021). We ran k-means clustering (with k set to the number of true classes) on the frozen embeddings of our top models. The resulting clusters were then evaluated against the ground-truth labels using Normalized Mutual Information (NMI), Purity, and Adjusted Rand Index (ARI).

**Results.** The results, summarized in Table 18, corroborate the findings from our primary P@k and GSR metrics. The top-performing embeddings according to our main metrics, **Whisper and CLAP**, also achieve the highest scores here, yielding clusters that align well with the true class structure. This demonstrates that the strong geometric separation captured by our main metrics translates directly to meaningful and coherent clusters in a fully unsupervised setting.

*Table 18.* Average clustering metrics across three representative subsets (HP, BC, BS3).

| Configuration (Average) | NMI (%) | Purity (%) | ARI (%) |
|---|---|---|---|
| Whisper EWMTF D100 | **64.73** | **57.59** | **31.05** |
| CLAP D100 | 63.95 | 55.34 | 28.74 |
| WavLM D100 | 59.86 | 49.43 | 22.45 |

## H.5. Layer Dependence Analysis

To assess the sensitivity of our results to the choice of encoder layer, we performed an exhaustive sweep of all 32 layers of the Whisper encoder on sampled versions of all 19 VOCSIM subsets.

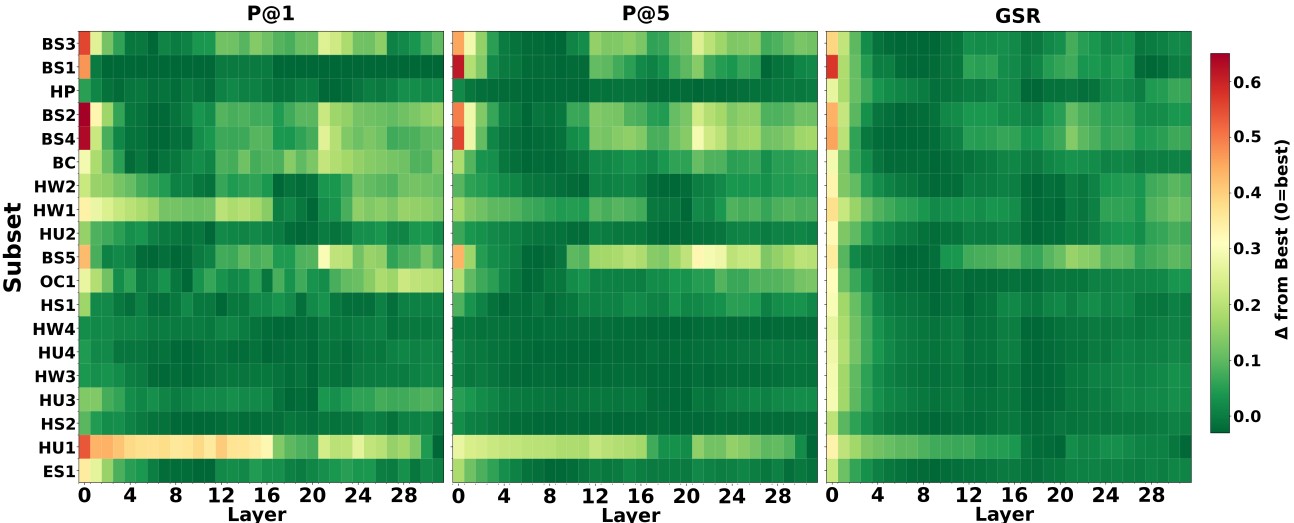

*Figure 6.* Heatmaps showing the performance drop ($\Delta$) from the best-performing layer (0=best, red=worse) for each metric across all 32 Whisper layers on the sampled subsets. The deltas are consistently small, indicating low sensitivity to layer choice.

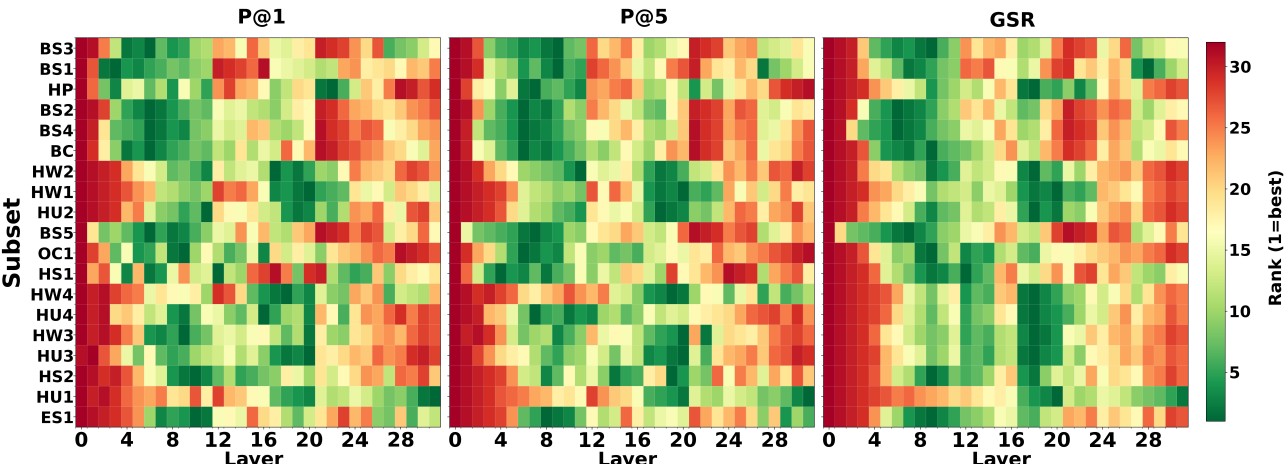

*Figure 7.* Layer rankings (1=best, 32=worst) for each subset and metric. While middle layers often rank highly, the overall pattern is consistent across metrics.

**Results.**   Our analysis reveals that performance is remarkably stable across the encoder's depth. As shown in Figure 6, the performance drop ($\Delta$) from the best-performing layer is uniformly small across all subsets and metrics. While middle layers (approximately 8-20) often rank highest (Figure 7), the potential performance gain from an optimal layer choice is marginal (typically $<1\%$ P@1).

We emphasize that this exhaustive 32-layer sweep was conducted for the Whisper encoder as a representative case study. Extending this analysis to *every* model, pooling strategy, and distance metric combination would be computationally prohibitive, requiring thousands of additional evaluations, and is beyond the scope of this paper's primary contribution, which is to establish the benchmark itself. Given the observed stability across Whisper layers, we adopt the standard practice of using final-layer embeddings for all encoders. We release our evaluation framework to enable the community to conduct model-specific layer ablations where warranted.

# I. Computational Cost and Feasibility

To address the practical feasibility of using different embeddings, we analyzed the computational requirements for feature extraction. While large foundation models like Whisper offer the best performance, they are computationally intensive and best suited for post-hoc analysis in a server or lab environment rather than real-time, resource-constrained deployment. Our analysis, shown in Table 19, provides a guide to the trade-offs between zero-shot performance and computational cost. The success of PCA compression on large model embeddings offers a valuable pathway to creating smaller, more efficient representations for downstream tasks.

*Table 19.* Computational analysis of benchmarked models. MACs were estimated using the 'fvcore' library for a 1-second, 16kHz audio input. Peak memory is for inference with batch size 1 on an NVIDIA RTX 3090 GPU. Model names and parameter counts are verified against their official sources.

| Model Pipeline | Parameters (M) | MACs (G/s) | Peak Memory (GB) |
|---|---|---|---|
| *Large-Scale Pretrained Models* | | | |
| Whisper-L-v3 | 635.05 | 953.06 | 6.41 |
| WavLM-Large | 206.30 | 377.19 | 8.10 |
| Wav2Vec2-Base | 89.65 | 201.65 | 8.28 |
| AudioMAE | 85.25 | 43.71 | 0.38 |
| CLAP | 68.55 | 14.94 | 0.83 |
| Smaller & Custom Models | | | |
| EnCodec (24kHz) | 7.43 | 44.73 | 0.50 |
| Paper VAE (Custom) | 8.73 | 0.79 | 0.19 |
| Paper AE (Custom) | 1.96 | 1.35 | 0.10 |
| *Baselines* | | | |
| Mel Spectrogram | 0.00 | 0.00 | Minimal |

# J. Full Performance Trend Visualizations

The main text analyzes performance trends using the top 5% of configurations for clarity (Figure 2). This appendix provides the corresponding visualizations for broader selections of the data.

Figure 8 illustrates the performance trends for the top 50% of all evaluated feature–distance configurations. Each subplot shows the relationship between performance (P@1, P@5, and GSR) and a specific structural property of the VOCSIM subsets.

Figure 9 presents the identical analysis but includes all (100%) of the configurations, incorporating the full performance range from the best models down to the weakest baselines.

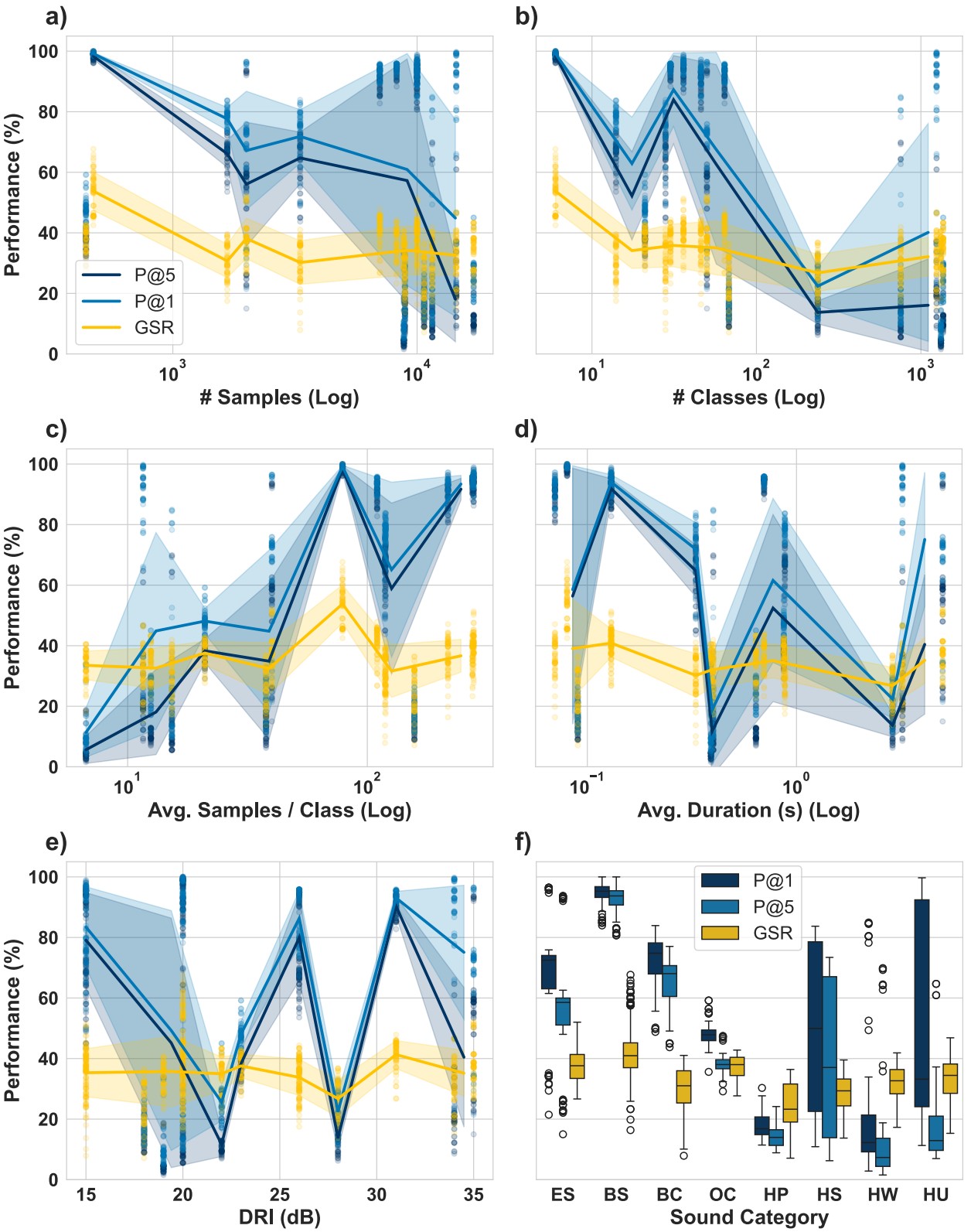

*Figure 8.* **Generalization trends for the top 50% of configurations.** The patterns observed, such as the sensitivity of P@k to class structure and the stability of GSR, are consistent with the analysis of the top 5% in the main text, albeit with more variance due to the inclusion of less optimal configurations.

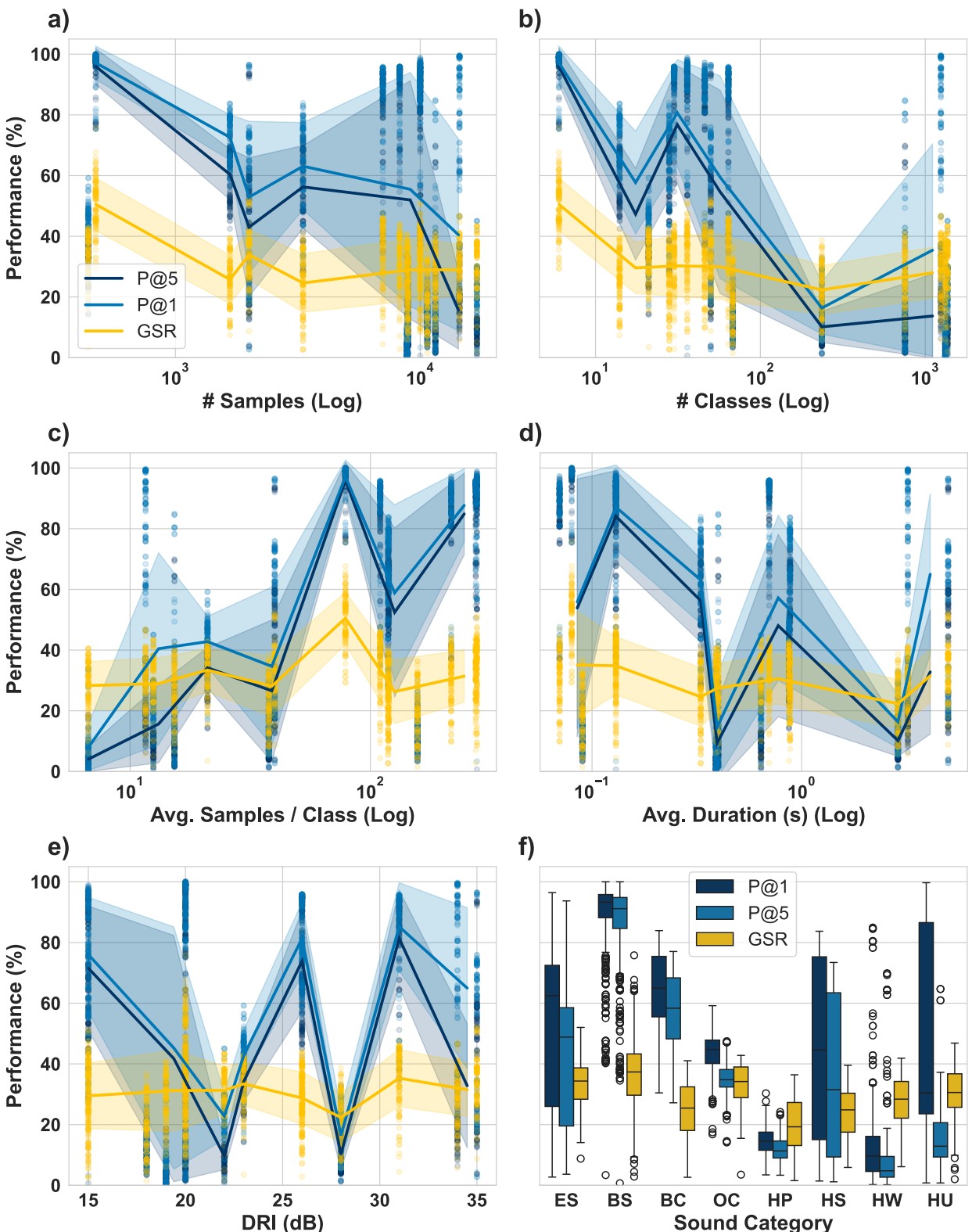

*Figure 9.* **Generalization trends for all (100%) configurations.** While the inclusion of all baseline and suboptimal methods introduces significant noise and lowers the average performance, the fundamental trends remain visible. This supports our decision to focus on the top 5% for a clearer presentation in the main text.

# K. Evaluating Custom Models on VOCSIM

VOCSIM's evaluation framework is designed to accommodate new audio embedding methods in three main steps: (1) implement a compatible feature extractor, (2) register it in the VOCSIM configuration, and (3) execute the existing zero-shot pipeline.

**1. Define a Custom Extractor**  Create a class that adheres to the VOCSIM extractor interface:

- It must accept as input a raw waveform (array or tensor) and its sampling rate.

- It must output either

    - A fixed-length vector (1D embedding), or
    - A sequence of frame/patch embeddings (2D array of shape [frames × dim]).

- Optionally, it may produce higher-dimensional structures (e.g., codebook indices, spectrogram patches) that VOCSIM's pooling routines can reduce.

- The extractor's constructor should handle loading any pretrained weights or model files and placing the model on the correct device.

**2. Configure VOCSIM to Use the Extractor**  In your VOCSIM YAML configuration:

- Add an entry under `feature_extractors` with

    - A unique `name` and `short_name` for reporting,
    - The Python import path and class name of your extractor,
    - Any constructor parameters (e.g., model checkpoint path),
    - A list of distance metrics to compute (e.g., `cosine`, `euclidean`).

- To define pooled or PCA-compressed variants, add additional entries that reference the base extractor, specify pooling (e.g., `mean_time`, `first_frame`), and the target PCA dimension.

**3. Run the Zero-Shot Pipeline**  Invoke the VOCSIM runner with the updated configuration, selecting the steps you wish to execute. This will:

1. *Extract features:* Apply all registered extractors to each VOCSIM subset.

2. *Compute distances:* Build pairwise distance matrices for each feature–metric pair.

3. *Evaluate benchmarks:* Compute P@1, P@5, and GSR (and any additional benchmarks configured).

**4. Inspect and Share Results**  After completion, VOCSIM produces CSV/JSON summaries of all metrics per feature and subset. Compare your custom model's scores against existing baselines to assess its zero-shot generalization. We encourage publishing these results on the public VOCSIM leaderboard to facilitate community comparison and progress.

**5. Blind Test Set Evaluation via Secure Protocol**  To strictly uphold data sovereignty for the indigenous language corpora (Shipibo-Conibo, Chintang), the raw audio and labels for the Blind Test Sets are never released publicly. To evaluate a custom model on these sets:

1. Fork the VOCSIM repository and implement your feature extractor.

2. Submit a Pull Request.

3. We will run your code in a secure, offline server environment that hosts the blind data.

4. Only the aggregate metrics (P@k, GSR) will be returned and posted to the leaderboard; no raw data or fine-grained predictions are exposed.

# L. Principled Scope and Limitations

VOCSIM's design involves deliberate, principled choices to ensure a focused and interpretable evaluation. Its primary scope is to measure zero-shot content identity in single-source audio. This focus means other important aspects of audio understanding are explicitly outside its scope.

**Exclusion of Polyphony and Scene Analysis.** By design, VOCSIM does not evaluate performance on overlapping sources. This is a geometric necessity: in a vector space, a single point representing a mixture (e.g., "Dog + Siren") cannot be uniquely mapped to a "Dog" cluster or a "Siren" cluster without an intermediate source-separation step. Evaluating raw embeddings on polyphony conflates the quality of the representation with the distinct capability of disentanglement.

**Exclusion of Abstract Semantics and Music.** We restrict the benchmark to classes with strong **acoustic-to-label fidelity**. We exclude music tasks that rely on high-level semantic tags (e.g., Genre: "Rock" vs. "Pop") because the intra-class acoustic variance is extremely high and ill-defined. A failure on such a task is ambiguous: it is unclear if the model failed to represent the audio or failed to infer the cultural context. We aim to include music in future releases only via datasets of isolated notes or instruments, where labels correspond to physical acoustic properties.

**Standardization.** The decision to resample all audio to 16 kHz is a pragmatic standard for evaluating foundation models, ensuring compatibility with architectures like Whisper and Wav2Vec 2.0. While this may discard some high-frequency bioacoustic information (e.g., ultrasonic harmonics), it ensures that all models are evaluated on a level playing field regarding input resolution.

# M. Applications: Further Details

## M.1. Unsupervised Clustering

To understand whether embeddings naturally partition sounds by their true categories without any label supervision, we apply a two-stage unsupervised pipeline, UMAP for dimensionality reduction followed by HDBSCAN for clustering, and evaluate the resulting clusters against the ground-truth labels using weighted purity.

**Procedure**

1. **Data Preparation.**

   - Remove any samples lacking valid class labels.
   - If using raw embeddings, replace any NaNs or infinities with finite values (e.g., global min/max) to ensure numerical stability.
   - If using a precomputed distance matrix, confirm its diagonal is zero.

2. **Dimensionality Reduction (UMAP).**

   - Project high-dimensional features (or precomputed distances) into 2D (or another low-dimensional space) using UMAP (McInnes et al., 2020).
   - UMAP preserves local and global structure, facilitating clustering in a compact space.

3. **Clustering (HDBSCAN).**

   - Run HDBSCAN (McInnes & Healy, 2017) on the UMAP embedding to discover clusters of variable shape and density.
   - Parameters such as `min_cluster_size` ensure clusters have meaningful support; points not belonging to any dense region are labeled as noise.

4. **Weighted Purity Calculation.**

   - For each non-noise cluster, identify the majority true class among its members.
   - Compute the cluster's purity as the fraction of members belonging to that majority class.
   - The *weighted purity* is the size-weighted average of these cluster purities:

$$\text{Purity}_{\text{weighted}} = \frac{\sum_j |C_j| \times \text{purity}(C_j)}{\sum_j |C_j|},$$

where $C_j$ is the set of items in cluster $j$.

A high weighted purity (near 1) indicates that the embedding space, when clustered without labels, aligns closely with the true class structure. Conversely, low purity suggests that same-class items are scattered across multiple clusters or mixed with other classes, revealing weaknesses in the embedding's global organization. This unsupervised clustering benchmark complements local metrics (such as P@k) by evaluating the global geometry of the embedding space.

### M.2. Alignment with Avian Perceptual Similarity

This benchmark tests whether zero-shot audio embeddings mirror zebra finches' own judgments of song-syllable similarity, using behavioral data from Zandberg et al. (2024) (Zandberg et al., 2024). In that study, finches performed a two-alternative-forced-choice (2AFC) task, associating each probe syllable $X$ with one of two training sounds ($A$ or $B$), and also yielded derived triplets $(A, P, N)$ where $P$ was judged closer to anchor $A$ than $N$. Their measurements establish an empirical ceiling ($\approx 80$–$90\%$ accuracy) based on bird–bird consistency.

We evaluate our embeddings against the high-consistency subset of these finch judgements in two tasks:

**Probe (2AFC) Task** For each trial $(X, A, B)$ with bird decision $D \in \{A, B\}$:

1. Look up distances $d(X, A)$ and $d(X, B)$ in the precomputed distance matrix.

2. The model "chooses" the closer sound.

3. *Accuracy* is the fraction of trials where model choice matches $D$.

4. A binomial test (chance=50%) checks significance.

5. Optionally, compute Spearman's $\rho$ and Kendall's $\tau$ between the signed distance difference $d(X, A) - d(X, B)$ and the bird's choice encoded as $+1$ (chose $A$) or $-1$ (chose $B$).

**Triplet Task** For each derived triplet $(A, P, N)$ drawn from high-consistency trials:

1. Retrieve $d(A, P)$ and $d(A, N)$ from the distance matrix.

2. The model "agrees" if $d(A, P) < d(A, N)$.

3. *Accuracy* is the percentage of triplets where the inequality holds.

4. A binomial test assesses significance against 50% chance.

5. Optionally, correlate $d(A, P) - d(A, N)$ with a constant bird-choice indicator (e.g., +1 for each triplet) to quantify rank-order alignment.

In Zandberg et al. (2024), zebra finches themselves agree on the same similarity judgment only about 80–90% of the time, both when different birds are compared (inter-bird consistency, around 80%) and when the same bird is retested on identical probes (intra-bird consistency, up to about 90%). These figures set a practical "ceiling" for any model attempting to mimic avian perception:

- An **80%** model accuracy matches the average agreement level one bird's choices have with another's, indicating the model performs as well as a typical zebra finch on this task.

- A **90%** model accuracy approaches the repeatability of a single bird's own judgments, and so represents a near-maximal alignment with avian perception given the natural variability in behavior.

Thus, a computational embedding that achieves $\sim 80\%$ accuracy is within the expected range of bird–bird agreement, while pushing toward 90% suggests the model captures nearly all of the reliably perceived distinctions.

## M.3. Methodology for Ultrasonic Vocalization (USV) Analysis

The state-of-the-art results for the mouse USV tasks were achieved by feeding the standard Whisper encoder model a specialized input representation suitable for high-frequency audio.

**Spectrogram Generation Method.** The log-Mel spectrogram was computed directly from the original 250 kHz audio waveforms. This was accomplished by dynamically adjusting the Short-Time Fourier Transform (STFT) parameters to be appropriate for the high sample rate, thereby preserving the spectral information in the ultrasonic range (>20 kHz).

**Implementation.** In our framework, this high-frequency spectrogram generation is implemented within the 'WhisperSegExtractor'. The embeddings used for the downstream classification tasks reported in the main text were generated using this extractor.

## M.4. Downstream Classification: Mouse Strain

To assess the practical value of our embeddings in bioacoustic applications, we predict the genetic strain of laboratory mice from their ultrasonic vocalizations (USVs) (Van Segbroeck et al., 2017; Goffinet et al., 2021). This task tests whether embeddings capture strain-specific acoustic cues beyond generic similarity.

**Dataset and Preprocessing** We use a publicly available USV dataset in which each syllable is labeled by mouse strain (e.g., C57 vs. DBA) and the identity of the individual mouse. Audio is segmented into individual syllables and preprocessed as described in Appendix M.3 to generate fixed-length embeddings (e.g., Whisper-based, VAE latents, log-Mel+PCA).

**Classification Protocol** To ensure a rigorous evaluation of generalization to unseen individuals and prevent data leakage, syllables from the same mouse never appear in both training and testing sets of a fold. We employ **group-stratified 5-fold cross-validation** by mouse identity. For each embedding set, we evaluate three off-the-shelf classifiers:

- Standardize features per fold (zero mean, unit variance).

- **k-Nearest Neighbors** (k=3,10,30).

- **Random Forest** (max depth=10,15,20; balanced class weights).

- **Multi-Layer Perceptron** (one hidden layer, L2 regularization $\alpha \in \{0.1, 0.01, 0.001\}$).

Hyperparameters are selected by grid search within each training fold.

**Metrics and Baselines** We report mean Top-1 and Top-5 accuracies (±standard deviation) across folds. As baselines, we reproduce results for spectrogram+PCA and VAE latents from (Goffinet et al., 2021) under the same evaluation protocol. Higher accuracy signals that the embedding encodes subtle spectral and temporal markers distinctive of mouse strains.

## M.5. Downstream Classification: Mouse Identity

We further evaluate embeddings by testing whether they capture fine-grained individual signatures in mouse ultrasonic vocalizations (USVs). Each syllable is labeled by the emitting mouse's identity (36 individuals), making this a challenging multi-class task that probes subtle, consistent vocal traits.

**Dataset and Preprocessing** We use a publicly available USV dataset in which each syllable is tagged with the individual mouse identity. Audio is segmented into discrete syllables and preprocessed as described in Appendix M.3 to generate fixed-length embeddings (e.g., Whisper-based, Mel+PCA, VAE latents).

### Classification Setup

- **Classifier:** A Multi-Layer Perceptron (MLP) with one or two hidden layers. We sweep L2 regularization strengths ($\alpha \in \{0.01, 0.001, 0.0001\}$) and hidden-layer sizes (e.g., 400 or [200,200] neurons).

- **Cross-Validation:** For the closed-set task of identifying a mouse from a known set of 36 individuals, we used 5-fold stratified cross-validation. This strategy partitions the syllables for each mouse across the folds, ensuring that the training set in each fold contains examples from every identity, and the test set contains held-out syllables from those same identities. This approach correctly tests the model's ability to learn a discriminative signature for each of the known mice and classify new vocalizations from that same set, which is the appropriate methodology for a closed-set identification task.

- **Feature Scaling:** Within each training fold, embeddings are standardized to zero mean and unit variance; the same scaling is applied to the test fold.

**Performance Metrics**

- *Top-1 Accuracy*: Percentage of syllables correctly assigned to their true individual.

- *Top-5 Accuracy*: Fraction of cases where the correct identity appears among the classifier's top five predictions.

**Baselines for Comparison**   We compare against the results from Goffinet et al. (2021) (Goffinet et al., 2021), who evaluated spectrogram+PCA, MUPET, and VAE latent features under similar MLP settings. These published accuracies serve as direct reference points. High Top-1 and Top-5 accuracies, substantially above the chance level of $1/36 \approx 2.8\%$, indicate that embeddings encode idiosyncratic vocal characteristics unique to individual mice.

**M.6. HEAR Benchmark Evaluation Details**

To provide extensive external validation, we evaluated our top-performing embeddings on the HEAR 2021 benchmark (Turian et al., 2022). **Protocol.** We strictly followed the official `heareval` evaluation protocol. For each task, we used the provided K-fold cross-validation splits. **Solver.** We trained a linear Logistic Regression classifier on the frozen embeddings. **Standardization.** Embeddings were standardized (z-score normalization) per dimension, with statistics computed on the training split and applied to validation/test splits. **Hyperparameter Tuning.** The regularization parameter $C$ was grid-searched ($10^{-4}$ to $10^4$) on the **validation split** for each fold. The final accuracy is reported on the held-out test split using the optimal $C$.

**Results.**   The results, summarized in Table 20, demonstrate state-of-the-art performance for the Whisper EWMTF D100 embedding. It consistently outperforms the strong CLAP D100 baseline and meets or exceeds the top scores from specialized systems on the official HEAR leaderboard. For Speech Commands (5h), we followed the official train/validation/test split, so the accuracy is reported as a single value. We assume the 7% lower accuracy on Mridangam tonic identification is due to pitch precision requirements, which are beyond the Whisper encoder's design objective

*Table 20.* Detailed performance on HEAR benchmark tasks. We report mean accuracy (%) $\pm$ standard deviation over the official K-fold splits.

| HEAR Task | Folds | Whisper D100 | CLAP D100 | HEAR SOTA | SOTA Model Ref. |
|---|---|---|---|---|---|
| Beijing Opera (Tian et al., 2014) | 5 | **97.65 $\pm$ 4.13** | 97.03 $\pm$ 3.95 | 97.46 | OpenL3 (Cramer et al., 2019) |
| GTZAN Music/Speech (Tzanetakis, 1999) | 10 | **99.23 $\pm$ 2.31** | **99.23 $\pm$ 2.31** | **99.23** | CP-JKU (Koutini et al., 2022) |
| Gunshot Triangulation (Cooper & Shaw, 2020) | 7 | **97.92 $\pm$ 3.34** | 94.05 $\pm$ 5.83 | 94.94 | OpenL3 (Cramer et al., 2019) |
| Mridangam Stroke (Akshay Anantapadmanabhan et al., 2020a) | 5 | 96.86 $\pm$ 0.93 | 96.42 $\pm$ 0.47 | **97.53** | GURA (Turian, 2022) |
| Mridangam Tonic (Akshay Anantapadmanabhan et al., 2020b) | 5 | 89.71 $\pm$ 0.76 | 87.73 $\pm$ 1.04 | **96.55** | RedRice (Dinkel et al., 2023) |
| CREMA-D Emotion (Cao et al., 2014) | 5 | **79.28 $\pm$ 0.56** | 58.42 $\pm$ 0.95 | 75.21 | GURA (Turian, 2022) |
| Speech Commands (5h) (Warden, 2018) | TVT | **98.61** | 68.24 | 97.63 | IUT-CSE (Turian, 2022) |

# N. Compute Resources

All experiments were run on a single workstation with an NVIDIA RTX 3090 GPU (24 GB VRAM), an AMD Ryzen 9 5950X CPU, and 128 GB DDR4 RAM. Reproducing the full pipeline end-to-end requires roughly 6 days ($\approx 144$ h) of wall-clock time, broken down as: *Feature extraction* ($\approx 72$ h), *Distance matrix computation* ($\approx 48$ h), and *Benchmark evaluations (P@k, GSR, clustering, applications)* ($\approx 24$ h).

# O. Limitations and future extensions.

VOCSIM excludes polyphonic mixtures by design; complementary training-free benchmarks could target mixture-aware similarity or separation-invariant matching. Our PCA adaptation is deliberately simple; future work may compare other label-free normalizations (e.g., whitening, ICA) under a training-free constraint. Finally, expanding single-source coverage (e.g., isolated musical notes, percussion) would further probe cross-mechanism generalization.

# P. Full Results

*Table 21.* P@1 Results Across Subsets and Distances (↑ better)

| Method | Dist | BC | BS1 | BS2 | BS3 | BS4 | BS5 | ES1 | HP | HS1 | HS2 | HU1 | HU2 | HU3 | HU4 | HW1 | HW2 | HW3 | HW4 | OC1 | Avg | Avg (Blind) |
|---|---|---|---|---|---|---|---|---|---|---|---|---|---|---|---|---|---|---|---|---|---|---|
| EWMTF D100 | S | 76.2 | 99.2 | 95.8 | 93.6 | 93.4 | 95.1 | 75.9 | **30.3** | 79.5 | 30.5 | **99.7** | 22.0 | 16.3 | **13.3** | 49.4 | 14.3 | **8.6** | 7.6 | 46.7 | **66.8** | 11.5 |
| EWMTF | S | 65.5 | 99.2 | 92.2 | 89.0 | 93.4 | 95.1 | 75.9 | 15.9 | 78.7 | 20.0 | **99.7** | 22.0 | 16.3 | **13.3** | 15.1 | 14.3 | **8.6** | 7.6 | 46.7 | 61.5 | 11.5 |
| BEATs | E | 75.5 | 99.4 | 95.8 | 91.7 | 92.3 | 95.7 | 95.1 | 17.2 | 83.6 | 40.4 | 55.9 | 20.7 | **17.0** | 13.1 | 31.1 | 10.1 | 7.6 | **7.9** | **60.5** | 64.3 | 11.4 |
| BEATs | C | 75.6 | 99.4 | 95.8 | 91.7 | 92.4 | 95.6 | 95.5 | 17.2 | **86.4** | **40.7** | 55.7 | 25.6 | 16.4 | 12.9 | 31.2 | 10.0 | 7.4 | 7.3 | **60.5** | 64.9 | 11.0 |
| EWTF D100 | S | 80.5 | 98.5 | 97.2 | 95.3 | 95.6 | 95.8 | 74.8 | 18.9 | 77.6 | 30.0 | 92.3 | 25.8 | 15.3 | 10.6 | 19.4 | 9.3 | 7.8 | 6.1 | 51.5 | 64.2 | 9.9 |
| EWTF | S | 73.3 | 98.5 | 97.2 | 92.4 | 95.6 | 95.8 | 74.8 | 11.4 | 77.4 | 21.7 | 92.3 | 25.8 | 15.3 | 10.6 | 9.7 | 9.3 | 7.8 | 6.1 | 51.5 | 61.8 | 9.9 |
| EMTF | S | 61.7 | 97.5 | 95.4 | 84.9 | 92.3 | 95.0 | 69.3 | 15.7 | 76.4 | 15.1 | 99.0 | 26.5 | 15.4 | 10.0 | 16.1 | 16.8 | 8.2 | 5.9 | 43.5 | 60.3 | 9.9 |
| EMTF D100 | S | 73.2 | 98.5 | 95.4 | 92.9 | 92.3 | 95.0 | 69.3 | 27.7 | 78.2 | 26.7 | 99.0 | 26.5 | 15.4 | 10.0 | 52.5 | 16.8 | 8.2 | 5.9 | 43.5 | 65.8 | 9.9 |
| EWT | S | 73.2 | 98.5 | 97.2 | 92.1 | 95.7 | 95.8 | 74.1 | 11.6 | 76.9 | 22.4 | 91.9 | 27.7 | 14.9 | 10.0 | 9.8 | 9.3 | 7.8 | 6.4 | 49.2 | 61.7 | 9.8 |
| EWT D100 | S | 79.7 | 98.5 | 97.0 | 95.4 | 95.7 | 95.8 | 74.1 | 18.7 | 79.4 | 29.4 | 91.9 | 27.7 | 14.9 | 10.0 | 19.1 | 9.3 | 7.8 | 6.4 | 49.2 | 64.1 | 9.8 |
| EWTF D100 | E | 78.2 | 99.4 | 96.8 | 95.0 | 95.5 | 95.8 | 73.1 | 15.6 | 80.5 | 29.4 | 88.6 | 23.7 | 13.5 | 11.2 | 17.1 | 7.9 | 7.2 | 5.6 | 47.2 | 62.9 | 9.4 |
| EWTF | E | 75.0 | 99.4 | 95.7 | 93.5 | 95.5 | 95.8 | 73.1 | 11.7 | 79.5 | 24.4 | 88.6 | 23.7 | 13.5 | 11.2 | 10.2 | 7.9 | 7.2 | 5.6 | 47.2 | 61.4 | 9.4 |
| EWTF D100 | C | 79.2 | 99.4 | 96.7 | 95.0 | 95.5 | 95.7 | 72.5 | 15.8 | 80.0 | 29.3 | 88.5 | 24.6 | 13.4 | 11.1 | 17.2 | 7.9 | 7.2 | 5.8 | 47.8 | 63.0 | 9.4 |
| EWTF | C | 75.8 | 99.6 | 97.1 | 93.4 | 95.5 | 95.7 | 72.5 | 11.9 | 79.5 | 24.6 | 88.5 | 24.6 | 13.4 | 11.1 | 10.4 | 7.9 | 7.2 | 5.8 | 47.8 | 61.6 | 9.4 |
| EWT D100 | C | 78.7 | 99.4 | 96.6 | 94.7 | 95.2 | 95.8 | 72.4 | 15.7 | 79.6 | 28.7 | 88.0 | 27.3 | 13.3 | 11.1 | 17.1 | 7.9 | 7.0 | 5.9 | 47.8 | 63.0 | 9.3 |
| EWT | C | 75.3 | 99.4 | 97.1 | 93.3 | 95.2 | 95.7 | 72.4 | 11.8 | 79.8 | 24.1 | 88.0 | 27.3 | 13.3 | 11.1 | 10.3 | 7.9 | 7.0 | 5.9 | 47.8 | 61.7 | 9.3 |
| EWT D100 | E | 78.2 | 99.4 | 96.8 | 94.8 | 95.3 | 95.8 | 73.1 | 15.3 | 80.2 | 28.7 | 88.1 | 18.1 | 13.2 | 10.9 | 17.0 | 7.9 | 6.8 | 5.8 | 47.4 | 62.4 | 9.2 |
| EWT | E | 74.6 | 99.4 | 95.8 | 93.2 | 95.3 | 95.8 | 73.1 | 11.6 | 78.9 | 24.0 | 88.1 | 18.1 | 13.2 | 10.9 | 10.1 | 7.9 | 6.8 | 5.8 | 47.4 | 60.9 | 9.2 |
| ETF D100 | S | 77.7 | 98.9 | 95.8 | 94.4 | 93.7 | 95.7 | 74.3 | 21.9 | 78.2 | 28.4 | 95.4 | 29.7 | 14.3 | 8.7 | 25.2 | 10.9 | 7.3 | 5.1 | 48.5 | 64.6 | 8.9 |
| ETF | S | 69.4 | 98.9 | 95.8 | 89.9 | 93.7 | 95.7 | 74.3 | 15.3 | 76.7 | 20.3 | 95.4 | 29.7 | 14.3 | 8.7 | 11.5 | 10.9 | 7.3 | 5.1 | 48.5 | 61.8 | 8.9 |
| ET | S | 68.6 | 98.9 | 91.4 | 94.5 | 93.9 | 95.7 | 74.0 | 14.7 | 76.3 | 20.0 | 95.3 | 29.6 | 14.0 | 9.0 | 11.6 | 10.6 | 7.2 | 5.2 | 48.5 | 61.6 | 8.8 |
| ET D100 | S | 76.8 | 98.9 | 95.6 | 94.5 | 93.9 | 95.6 | 74.0 | 22.1 | 78.7 | 27.8 | 95.3 | 29.6 | 14.0 | 9.0 | 25.3 | 10.6 | 7.2 | 5.2 | 48.5 | 64.5 | 8.8 |
| EWMTF D100 | C | 68.3 | 98.9 | 93.9 | 91.1 | 91.3 | 94.3 | 69.3 | 20.0 | 79.8 | 23.1 | 95.3 | 28.0 | 11.3 | 10.8 | 26.7 | 6.0 | 5.8 | 5.9 | 46.9 | 62.2 | 8.4 |
| EWMTF | C | 64.9 | 98.7 | 94.6 | 88.7 | 91.3 | 94.3 | 69.3 | 14.0 | 79.2 | 18.8 | 95.3 | 28.0 | 11.3 | 10.8 | 13.2 | 6.0 | 5.8 | 5.9 | 46.9 | 60.2 | 8.4 |
| EW | C | 64.9 | 99.6 | 97.1 | 88.7 | 91.3 | 93.2 | 69.3 | 14.0 | 79.2 | 18.8 | 95.3 | 28.0 | 11.3 | 10.8 | 13.2 | 6.0 | 5.8 | 5.9 | 46.9 | 60.4 | 8.4 |
| EW | E | 65.0 | 99.6 | 94.0 | 88.6 | 91.3 | 93.2 | 67.8 | 13.5 | 78.6 | 18.4 | 95.8 | 17.2 | 11.0 | 10.5 | 12.3 | 6.0 | 5.8 | 5.5 | 45.8 | 59.1 | 8.2 |
| EWMTF D100 | E | 67.8 | 99.2 | 94.0 | 90.9 | 91.3 | 94.1 | 67.8 | 19.7 | 78.6 | 22.6 | 95.8 | 17.2 | 11.0 | 10.5 | 25.9 | 6.0 | 5.8 | 5.5 | 45.8 | 61.1 | 8.2 |
| EWMTF | E | 65.0 | 99.2 | 94.0 | 88.6 | 91.3 | 94.1 | 67.8 | 13.5 | 78.6 | 18.4 | 95.8 | 17.2 | 11.0 | 10.5 | 12.3 | 6.0 | 5.8 | 5.5 | 45.8 | 59.2 | 8.2 |
| ETF D100 | C | 75.9 | 99.2 | 95.6 | 93.9 | 93.5 | 95.8 | 73.0 | 21.0 | 80.1 | 27.5 | 92.9 | 29.4 | 13.7 | 7.6 | 23.5 | 9.7 | 6.5 | 4.9 | 49.4 | 64.0 | 8.2 |
| ETF | C | 71.6 | 99.2 | 95.6 | 91.1 | 93.5 | 95.8 | 73.0 | 16.8 | 78.3 | 22.1 | 92.9 | 29.4 | 13.7 | 7.6 | 13.6 | 9.7 | 6.5 | 4.9 | 49.4 | 62.1 | 8.2 |
| ET | C | 71.1 | 99.2 | 95.4 | 91.0 | 93.3 | 95.8 | 72.5 | 16.6 | 78.4 | 21.9 | 92.8 | 29.2 | 13.7 | 7.6 | 13.6 | 9.5 | 6.4 | 4.7 | 46.9 | 61.8 | 8.1 |
| ET D100 | C | 75.4 | 99.2 | 95.4 | 93.8 | 93.3 | 95.6 | 72.5 | 20.9 | 79.8 | 27.0 | 92.8 | 29.2 | 13.7 | 7.6 | 23.4 | 9.5 | 6.4 | 4.7 | 46.9 | 63.6 | 8.1 |
| CLP D100 | S | 83.0 | 98.9 | 96.9 | 94.4 | 95.2 | 95.8 | 95.7 | 21.3 | 81.3 | 33.0 | 46.6 | 24.1 | 13.5 | 7.9 | 25.3 | 8.9 | 5.8 | 5.2 | 54.9 | 63.7 | 8.1 |
| CLP | S | 76.1 | 98.9 | 96.9 | 91.2 | 95.2 | 95.9 | 95.7 | 15.9 | 81.6 | 26.0 | 46.6 | 24.1 | 13.5 | 7.9 | 17.8 | 8.9 | 5.8 | 5.2 | 54.9 | 61.7 | 8.1 |
| CLP D100 | C | 83.0 | 99.4 | 96.4 | 94.5 | 94.2 | 95.7 | **96.5** | 20.0 | 83.6 | 32.7 | 36.2 | 24.1 | 12.3 | 8.9 | 26.0 | 9.3 | 5.5 | 4.9 | 59.2 | 63.4 | 7.9 |
| CLP | C | 79.8 | 99.2 | 95.3 | 93.0 | 94.2 | **95.9** | **96.5** | 17.7 | 83.7 | 28.9 | 36.2 | 24.1 | 12.3 | 8.9 | 21.3 | 9.3 | 5.5 | 4.9 | 59.2 | 62.3 | 7.9 |
| ETF D100 | E | 76.0 | 99.2 | 95.5 | 93.9 | 93.4 | 95.8 | 73.8 | 20.2 | 80.6 | 27.5 | 93.2 | 29.2 | 12.5 | 7.5 | 23.3 | 9.7 | 6.3 | 5.0 | 50.3 | 64.1 | 7.8 |
| ETF | E | 71.6 | 99.2 | 96.2 | 91.3 | 93.4 | 95.8 | 73.8 | 16.5 | 76.5 | 22.4 | 93.2 | 29.2 | 12.5 | 7.5 | 13.4 | 9.7 | 6.3 | 5.0 | 50.3 | 62.1 | 7.8 |
| ET D100 | E | 75.5 | 99.2 | 95.3 | 93.7 | 93.2 | 95.6 | 73.5 | 20.2 | 79.6 | 27.2 | 93.0 | 28.9 | 12.3 | 7.6 | 23.1 | 9.6 | 6.3 | 4.9 | 49.2 | 63.8 | 7.8 |
| ET | E | 71.7 | 99.2 | 95.3 | 91.0 | 93.2 | 95.8 | 73.5 | 16.3 | 77.7 | 22.1 | 93.0 | 28.9 | 12.3 | 7.6 | 13.3 | 9.6 | 6.3 | 4.9 | 49.2 | 62.0 | 7.8 |
| CLP D100 | E | 82.3 | 99.4 | 96.6 | 94.5 | 94.4 | 95.7 | **96.5** | 19.2 | 82.5 | 32.4 | 36.4 | 24.0 | 11.7 | 8.6 | 26.0 | 8.7 | 5.3 | 4.4 | 56.5 | 63.0 | 7.5 |
| CLP | E | 79.7 | 99.4 | 96.7 | 93.3 | 94.4 | **95.9** | **96.5** | 17.3 | 82.0 | 28.9 | 36.4 | 24.0 | 11.7 | 8.6 | 21.3 | 8.7 | 5.3 | 4.4 | 56.5 | 62.1 | 7.5 |
| EWTF D30 | C | 75.8 | 99.6 | 95.8 | 93.4 | 94.1 | 95.2 | 67.3 | 11.9 | 79.5 | 24.6 | 64.8 | 20.9 | 9.9 | 10.9 | 10.4 | 4.5 | 4.7 | 4.4 | 45.4 | 58.9 | 7.5 |
| EWT D30 | C | 75.3 | 99.4 | 95.8 | 93.3 | 94.1 | 95.2 | 67.6 | 11.8 | 79.8 | 24.1 | 64.1 | 27.3 | 9.9 | 10.7 | 10.3 | 4.6 | 4.8 | 4.8 | 44.9 | 59.2 | 7.4 |
| EMTF | C | 62.1 | 98.1 | 92.5 | 85.1 | 89.1 | 93.9 | 62.6 | 14.6 | 75.1 | 14.4 | 80.6 | 30.1 | 10.5 | 8.9 | 15.0 | 7.6 | 5.4 | 4.0 | 45.8 | 57.8 | 7.2 |
| E | C | 62.1 | 98.1 | 3.4 | 85.1 | 89.1 | 94.7 | 62.6 | 14.6 | 75.1 | 14.4 | 80.6 | 30.1 | 10.5 | 8.9 | 15.0 | 7.6 | 5.4 | 4.0 | 45.8 | 51.9 | 7.2 |
| EMTF D100 | C | 66.3 | 98.7 | 92.5 | 88.9 | 89.1 | 93.6 | 62.6 | 20.1 | 76.3 | 18.6 | 80.6 | 30.1 | 10.5 | 8.9 | 29.6 | 7.6 | 5.4 | 4.0 | 45.8 | 60.0 | 7.2 |
| EWT D30 | E | 74.6 | 99.4 | 95.8 | 93.2 | 94.3 | 95.3 | 67.5 | 11.6 | 78.9 | 24.0 | 65.5 | 18.1 | 9.9 | 9.3 | 10.1 | 4.5 | 4.8 | 4.6 | 42.6 | 58.4 | 7.1 |
| EWTF D30 | E | 75.0 | 99.4 | 95.7 | 93.5 | 94.3 | 95.2 | 67.8 | 11.7 | 79.5 | 24.4 | 66.1 | 19.0 | 10.0 | 9.1 | 10.2 | 4.6 | 4.8 | 4.4 | 43.5 | 58.7 | 7.1 |
| EMTF | E | 61.5 | 99.2 | 93.6 | 84.8 | 89.7 | 93.7 | 61.6 | 13.8 | 74.0 | 13.8 | 80.5 | 19.3 | 10.1 | 8.9 | 13.8 | 7.9 | 5.2 | 3.9 | 42.0 | 56.6 | 7.0 |
| E | E | 61.5 | 99.2 | 95.3 | 84.8 | 89.7 | 94.7 | 61.6 | 13.8 | 74.0 | 13.8 | 80.5 | 19.3 | 10.1 | 8.9 | 13.8 | 7.9 | 5.2 | 3.9 | 42.0 | 56.8 | 7.0 |
| EMTF D100 | E | 66.4 | 99.2 | 93.0 | 88.9 | 89.7 | 93.7 | 61.6 | 19.9 | 74.2 | 18.0 | 80.5 | 19.3 | 10.1 | 8.9 | 29.3 | 7.9 | 5.2 | 3.9 | 42.0 | 58.9 | 7.0 |
| EWTF D30 | S | 73.3 | 98.9 | 95.4 | 92.4 | 93.1 | 94.3 | 65.1 | 11.4 | 77.4 | 21.7 | 59.0 | 20.2 | 9.5 | 9.3 | 9.7 | 4.1 | 4.2 | 3.7 | 47.8 | 57.6 | 6.7 |
| EWT D30 | S | 73.2 | 98.7 | 95.4 | 92.1 | 93.0 | 94.5 | 64.9 | 11.6 | 76.9 | 22.4 | 60.8 | 27.7 | 9.6 | 8.2 | 9.8 | 4.1 | 4.2 | 4.2 | 43.5 | 57.9 | 6.6 |
| ET D30 | C | 71.1 | 99.2 | 93.2 | 91.0 | 91.0 | 94.8 | 68.1 | 16.6 | 78.4 | 21.9 | 76.7 | 29.2 | 9.5 | 7.7 | 13.6 | 5.2 | 4.5 | 3.5 | 44.4 | 59.6 | 6.3 |
| ETF D30 | C | 71.6 | 99.2 | 93.2 | 91.1 | 91.3 | 94.8 | 68.8 | 16.8 | 78.3 | 22.1 | 77.2 | 29.4 | 9.6 | 7.7 | 13.6 | 5.4 | 4.4 | 3.4 | 47.4 | 60.0 | 6.3 |
| CLP D30 | C | 79.8 | 99.2 | 95.3 | 93.0 | 92.7 | 95.5 | 95.8 | 17.7 | 83.7 | 28.9 | 23.7 | 24.1 | 9.7 | 6.9 | 21.3 | 6.9 | 4.6 | 4.4 | 55.3 | 60.9 | 6.3 |
| EWMTF D30 | C | 64.9 | 98.7 | 92.4 | 88.7 | 89.4 | 93.0 | 63.0 | 14.0 | 79.2 | 18.8 | 85.7 | 28.0 | 7.6 | 8.6 | 13.2 | 3.0 | 3.4 | 4.8 | 44.0 | 58.4 | 6.1 |
| ETF D30 | E | 71.6 | 99.2 | 93.4 | 91.3 | 91.6 | 95.1 | 68.8 | 16.5 | 76.5 | 22.4 | 77.9 | 29.2 | 9.3 | 7.0 | 13.4 | 5.1 | 4.4 | 3.3 | 44.4 | 59.7 | 6.0 |
| EWMTF D30 | E | 65.5 | 97.7 | 92.6 | 89.0 | 89.0 | 92.3 | 63.7 | 15.9 | 78.7 | 20.0 | 86.1 | 22.0 | 8.5 | 7.6 | 15.1 | 3.5 | 3.6 | 4.2 | 42.9 | 58.3 | 6.0 |
| ET D30 | E | 71.7 | 99.2 | 93.2 | 91.0 | 91.3 | 95.1 | 68.3 | 16.3 | 77.7 | 22.1 | 77.3 | 28.9 | 9.4 | 6.9 | 13.3 | 5.1 | 4.1 | 3.2 | 42.6 | 59.5 | 5.9 |
| CLP D30 | E | 79.7 | 99.4 | 95.4 | 93.3 | 93.0 | 95.5 | 96.0 | 17.3 | 82.0 | 28.9 | 23.9 | 24.0 | 9.7 | 6.2 | 21.3 | 6.7 | 3.9 | 3.8 | 52.4 | 60.6 | 5.9 |
| CLP D30 | S | 76.1 | 98.9 | 94.0 | 91.2 | 91.6 | 94.9 | 94.8 | 15.9 | 81.6 | 26.0 | 22.1 | 24.1 | 8.7 | 7.0 | 17.8 | 5.3 | 3.6 | 3.8 | 48.5 | 58.9 | 5.8 |
| EWMTF D30 | E | 65.0 | 99.2 | 92.5 | 88.6 | 89.6 | 92.9 | 61.5 | 13.5 | 78.6 | 18.4 | 86.7 | 17.2 | 6.6 | 8.7 | 12.3 | 2.8 | 3.1 | 4.3 | 43.8 | 57.5 | 5.7 |
| EMTF D30 | C | 62.1 | 98.1 | 90.4 | 85.1 | 86.2 | 92.5 | 56.7 | 14.6 | 75.1 | 14.4 | 60.1 | 30.1 | 8.3 | 7.9 | 15.0 | 3.5 | 3.2 | 3.0 | 43.8 | 55.2 | 5.6 |
| ET D30 | S | 68.6 | 98.3 | 91.4 | 89.4 | 89.5 | 94.2 | 65.8 | 14.7 | 76.3 | 20.0 | 70.7 | 29.6 | 8.3 | 7.3 | 11.6 | 4.0 | 3.7 | 3.0 | 43.5 | 57.9 | 5.6 |
| ETF D30 | S | 69.4 | 98.3 | 91.8 | 89.9 | 89.4 | 94.2 | 65.0 | 15.3 | 76.7 | 20.3 | 70.9 | 29.7 | 8.6 | 7.1 | 11.5 | 4.0 | 3.6 | 2.7 | 45.4 | 58.1 | 5.5 |
| EMTF D30 | E | 61.5 | 98.5 | 90.8 | 84.8 | 86.8 | 92.3 | 56.1 | 13.8 | 74.0 | 13.8 | 60.0 | 19.3 | 8.0 | 7.6 | 13.8 | 3.4 | 3.1 | 3.1 | 40.4 | 54.0 | 5.5 |
| EMTF D30 | S | 61.7 | 97.5 | 90.4 | 84.9 | 85.2 | 92.1 | 56.5 | 15.7 | 76.4 | 15.1 | 68.3 | 26.5 | 7.5 | 6.4 | 16.1 | 4.0 | 3.7 | 3.7 | 43.3 | 55.6 | 5.3 |
| WLM | S | 48.8 | 98.9 | 87.8 | 84.0 | 90.0 | 93.2 | 34.4 | 24.1 | 73.0 | 10.9 | 99.4 | 11.3 | 6.8 | 6.9 | 78.3 | 33.7 | 3.0 | 4.4 | 50.1 | 61.2 | 5.3 |
| WLM D100 | S | 58.2 | 98.9 | 92.3 | 89.1 | 90.0 | 92.1 | 34.4 | 26.3 | 75.7 | 14.8 | 99.4 | 42.0 | 6.8 | 6.9 | 84.5 | 33.7 | 3.0 | 4.4 | 50.1 | 65.4 | 5.3 |
| WLM | C | 50.3 | 99.2 | 89.2 | 85.5 | 89.5 | 93.3 | 34.4 | 25.1 | 74.8 | 12.3 | **99.2** | **45.0** | 5.5 | 7.6 | 80.3 | **33.9** | 2.4 | 4.1 | 49.0 | 64.5 | 4.9 |
| WLM D100 | C | 56.6 | 99.2 | 92.2 | 89.0 | 89.5 | 92.2 | 34.4 | 26.3 | 75.8 | 14.7 | **99.2** | **45.0** | 5.5 | 7.6 | **84.9** | **33.9** | 2.4 | 4.1 | 49.0 | 65.5 | 4.9 |
| WLM D100 | E | 55.6 | 99.4 | 92.2 | 89.1 | 89.6 | 92.2 | 34.0 | 26.3 | 74.5 | 14.9 | **99.2** | 30.1 | 4.9 | 7.2 | 84.7 | 33.1 | 2.5 | 3.9 | 46.3 | 64.1 | 4.6 |
| WLM | E | 49.8 | 99.2 | 89.3 | 85.7 | 89.6 | 93.4 | 34.0 | 24.9 | 74.0 | 12.4 | **99.2** | 30.1 | 4.9 | 7.2 | 80.5 | 33.1 | 2.5 | 3.9 | 46.3 | 62.8 | 4.6 |
| M | C | **83.9** | **100.0** | 98.8 | **97.2** | 95.6 | 95.0 | 31.8 | 22.2 | 69.5 | 12.5 | 98.7 | 24.6 | 5.8 | 6.2 | 29.6 | 10.2 | 3.1 | 3.3 | 50.8 | 61.4 | 4.6 |

*Table 21.* (Continued) P@1 Results Across Subsets and Distances (↑ better)

| Method | Dist | BC | BS1 | BS2 | BS3 | BS4 | BS5 | ES1 | HP | HS1 | HS2 | HU1 | HU2 | HU3 | HU4 | HW1 | HW2 | HW3 | HW4 | OC1 | Avg | Avg (Blind) |
|---|---|---|---|---|---|---|---|---|---|---|---|---|---|---|---|---|---|---|---|---|---|---|
| M | E | 82.4 | 99.8 | **98.9** | 97.2 | **95.9** | 95.1 | 31.4 | 21.3 | 66.9 | 12.4 | 98.5 | 23.5 | 5.1 | 5.8 | 31.2 | 10.5 | 2.5 | 2.9 | 49.2 | 60.9 | 4.1 |
| VC | C | 77.1 | **100.0** | 97.5 | 96.1 | 94.7 | 95.6 | 34.5 | 17.8 | 70.8 | 14.4 | 88.1 | 23.7 | 3.9 | 7.0 | 22.3 | 8.2 | 2.2 | 3.2 | 46.7 | 59.2 | 4.1 |
| VC | E | 76.8 | **100.0** | 97.6 | 95.9 | 94.3 | 95.4 | 35.1 | 17.4 | 67.4 | 14.5 | 81.3 | 19.2 | 4.0 | 6.0 | 21.3 | 7.9 | 2.1 | 3.1 | 48.1 | 58.2 | 3.8 |
| AC | S | 80.0 | **100.0** | 98.0 | 93.1 | 93.8 | 94.7 | 34.2 | 12.0 | 69.3 | 15.3 | 98.6 | 21.1 | 4.8 | 5.6 | 15.5 | 5.7 | 2.3 | 1.8 | 35.6 | 57.8 | 3.6 |
| WLM D30 | C | 50.3 | 99.2 | 89.2 | 85.5 | 86.8 | 90.8 | 30.2 | 25.1 | 74.8 | 12.3 | 94.6 | **45.0** | 3.5 | 5.9 | 80.3 | 27.8 | 1.8 | 3.3 | 45.1 | 62.5 | 3.6 |
| M | S | 78.2 | **100.0** | 97.6 | 96.0 | 94.1 | 93.6 | 21.4 | 14.6 | 67.5 | 11.1 | 98.5 | 22.1 | 3.9 | 5.0 | 19.4 | 8.1 | 2.3 | 2.7 | 43.5 | 57.7 | 3.5 |
| WLM D30 | E | 49.8 | 99.2 | 89.3 | 85.7 | 86.7 | 90.8 | 29.4 | 24.9 | 74.0 | 12.4 | 95.2 | 30.1 | 3.2 | 5.7 | 80.5 | 27.9 | 1.7 | 2.8 | 45.4 | 61.4 | 3.4 |
| WLM D30 | S | 48.8 | 98.9 | 87.8 | 84.0 | 83.8 | 88.6 | 26.7 | 24.1 | 73.0 | 10.9 | 91.3 | 11.3 | 3.4 | 4.7 | 78.3 | 25.6 | 1.7 | 3.2 | 45.6 | 58.6 | 3.2 |
| EF D100 | C | 58.7 | 98.5 | 91.3 | 88.3 | 87.7 | 91.8 | 23.8 | 14.8 | 65.9 | 7.3 | 61.4 | 19.1 | 4.1 | 5.1 | 10.1 | 2.1 | 1.2 | 2.1 | 41.0 | 50.8 | 3.1 |
| EF | C | 56.1 | 98.3 | 91.3 | 85.1 | 87.7 | 91.8 | 23.8 | 12.5 | 66.8 | 6.9 | 61.4 | 19.1 | 4.1 | 5.1 | 7.2 | 2.1 | 1.2 | 2.1 | 41.0 | 50.1 | 3.1 |
| EF | S | 55.6 | 97.5 | 88.4 | 84.3 | 88.2 | 92.3 | 19.1 | 11.6 | 65.2 | 5.8 | 65.0 | 21.7 | 3.1 | 5.1 | 6.7 | 2.6 | 1.5 | 2.5 | 38.8 | 49.5 | 3.0 |
| EF D100 | S | 58.9 | 97.5 | 92.2 | 89.5 | 88.2 | 92.3 | 19.1 | 17.0 | 66.5 | 6.2 | 65.0 | 21.7 | 3.1 | 5.1 | 13.2 | 2.6 | 1.5 | 2.5 | 38.8 | 51.2 | 3.0 |
| AC | C | 77.4 | **100.0** | 97.8 | 93.3 | 93.1 | 94.2 | 33.3 | 11.2 | 68.3 | 13.6 | 94.2 | 20.4 | 3.5 | 4.4 | 15.1 | 5.1 | 1.8 | 1.9 | 35.1 | 56.8 | 2.9 |
| AC | E | 77.3 | **100.0** | 97.7 | 93.3 | 93.1 | 94.1 | 29.8 | 11.0 | 61.5 | 11.6 | 90.1 | 13.1 | 3.6 | 4.3 | 14.6 | 4.9 | 1.8 | 1.9 | 35.6 | 55.2 | 2.9 |
| EF | E | 55.0 | 98.7 | 91.9 | 84.8 | 87.9 | 91.8 | 20.8 | 12.3 | 64.6 | 6.8 | 61.2 | 19.1 | 3.6 | 4.7 | 6.8 | 2.0 | 1.1 | 2.1 | 40.1 | 49.6 | 2.9 |
| EF D100 | E | 57.6 | 98.7 | 91.2 | 88.3 | 87.9 | 91.8 | 20.8 | 14.5 | 64.7 | 7.0 | 61.2 | 19.1 | 3.6 | 4.7 | 9.8 | 2.0 | 1.1 | 2.1 | 40.1 | 50.3 | 2.9 |
| EF D30 | E | 55.0 | 98.5 | 88.8 | 84.8 | 85.6 | 90.5 | 23.2 | 12.3 | 64.6 | 6.8 | 37.5 | 19.1 | 3.3 | 4.7 | 6.8 | 1.6 | 1.0 | 1.8 | 37.9 | 47.5 | 2.7 |
| EF D30 | S | 55.6 | 97.5 | 88.4 | 84.3 | 84.3 | 89.9 | 19.4 | 11.6 | 65.2 | 5.8 | 30.7 | 21.7 | 2.7 | 4.8 | 6.7 | 1.6 | 1.2 | 1.8 | 37.2 | 46.7 | 2.6 |
| EF D30 | C | 56.1 | 98.3 | 88.9 | 85.1 | 85.4 | 90.4 | 24.1 | 12.5 | 66.8 | 6.9 | 37.4 | 19.1 | 3.4 | 4.2 | 7.2 | 1.6 | 1.0 | 1.7 | 37.6 | 47.8 | 2.6 |
| VC | S | 64.5 | 99.6 | 93.9 | 92.4 | 91.5 | 92.0 | 25.0 | 13.3 | 63.8 | 7.7 | 58.7 | 19.2 | 2.6 | 4.1 | 12.0 | 3.5 | 1.0 | 1.8 | 39.7 | 51.8 | 2.4 |
| EWF | S | 56.0 | 98.9 | 92.8 | 86.1 | 87.9 | 88.8 | 21.0 | 10.0 | 67.1 | 5.4 | 89.9 | 19.3 | 2.3 | 3.7 | 3.9 | 1.4 | 1.2 | 1.8 | 42.4 | 51.4 | 2.3 |
| EWF D100 | S | 61.1 | 98.9 | 92.8 | 90.5 | 87.9 | 88.8 | 21.0 | 14.4 | 64.3 | 6.0 | 89.9 | 19.3 | 2.3 | 3.7 | 8.1 | 1.4 | 1.2 | 1.8 | 42.4 | 52.4 | 2.3 |
| MAE | S | 43.8 | 98.7 | 85.2 | 81.1 | 74.8 | 78.1 | 64.6 | 7.0 | 62.5 | 6.9 | 77.1 | 0.8 | 1.9 | 4.1 | 2.2 | 0.9 | 0.8 | 1.9 | 34.7 | 47.9 | 2.2 |
| MAE D100 | S | 50.6 | 98.7 | 85.2 | 89.5 | 74.8 | 80.0 | 64.6 | 9.2 | 64.3 | 11.4 | 77.1 | - | 1.9 | 4.1 | 3.5 | 0.9 | 0.8 | 1.9 | 34.7 | 53.2 | 2.2 |
| W2V D100 | S | 36.0 | 82.0 | 61.1 | 46.6 | 47.6 | 74.4 | 20.1 | 6.5 | 61.4 | 7.1 | 99.3 | 22.5 | 2.3 | 4.2 | 56.7 | 12.6 | 0.9 | 1.3 | 27.7 | 44.1 | 2.2 |
| W2V | S | 32.8 | 75.5 | 56.7 | 41.4 | 47.6 | 74.2 | 20.1 | 5.2 | 59.2 | 5.5 | 99.3 | 22.5 | 2.3 | 4.2 | 39.7 | 12.6 | 0.9 | 1.3 | 27.7 | 41.3 | 2.2 |
| EAT | S | 64.0 | 98.7 | 91.1 | 88.5 | 88.3 | 92.0 | 48.9 | 10.8 | 67.5 | 12.4 | 76.1 | 17.1 | 2.7 | 3.7 | 7.1 | 1.4 | 0.8 | 1.4 | 35.4 | 53.3 | 2.2 |
| EAT | C | 63.4 | 99.2 | 91.1 | 88.1 | 87.7 | 92.1 | 48.8 | 10.4 | 68.3 | 12.6 | 75.7 | 17.1 | 3.0 | 3.7 | 7.0 | 1.4 | 0.8 | 1.4 | 33.8 | 53.1 | 2.2 |
| EWF | C | 55.9 | 98.9 | 88.0 | 86.4 | 85.6 | 88.2 | 25.4 | 10.1 | 65.9 | 6.2 | 75.9 | 17.4 | 2.6 | 3.2 | 4.4 | 1.1 | 1.2 | 1.2 | 42.4 | 50.1 | 2.1 |
| EWF D100 | C | 58.1 | 98.5 | 90.9 | 89.3 | 85.6 | 88.2 | 25.4 | 12.4 | 65.9 | 6.7 | 75.9 | 17.4 | 2.6 | 3.2 | 6.0 | 1.1 | 1.2 | 1.2 | 42.4 | 50.9 | 2.1 |
| EWF D30 | C | 55.9 | 98.5 | 88.0 | 86.4 | 81.6 | 86.3 | 26.2 | 10.1 | 65.9 | 6.2 | 55.0 | 16.8 | 2.2 | 3.7 | 4.4 | 1.0 | 1.1 | 1.1 | 43.5 | 48.4 | 2.0 |
| W2V | C | 33.6 | 78.9 | 56.9 | 42.0 | 45.2 | 74.8 | 20.1 | 5.3 | 61.2 | 5.7 | 99.3 | 24.4 | 1.6 | 4.1 | 42.8 | 11.6 | 1.0 | 1.2 | 26.8 | 41.9 | 2.0 |
| W2V D100 | C | 35.0 | 81.2 | 59.6 | 44.9 | 45.2 | 74.6 | 20.1 | 6.0 | 62.7 | 7.0 | 99.3 | 24.4 | 1.6 | 4.1 | 55.7 | 11.6 | 1.0 | 1.2 | 26.8 | 43.6 | 2.0 |
| EAT | E | 60.6 | 98.3 | 90.3 | 87.9 | 86.3 | 91.0 | 43.9 | 9.9 | 65.4 | 10.5 | 54.4 | 16.7 | 2.6 | 3.1 | 5.6 | 0.9 | 0.7 | 1.2 | 32.0 | 50.2 | 1.9 |
| EWF | E | 55.1 | 98.9 | 87.9 | 86.2 | 85.3 | 88.1 | 22.4 | 9.8 | 65.0 | 6.3 | 75.8 | 17.3 | 2.2 | 2.9 | 4.3 | 0.9 | 1.2 | 1.4 | 42.9 | 49.7 | 1.9 |
| EWF D100 | E | 58.0 | 98.9 | 90.7 | 89.0 | 85.3 | 88.1 | 22.4 | 12.0 | 62.9 | 6.6 | 75.8 | 17.3 | 2.2 | 2.9 | 6.0 | 0.9 | 1.2 | 1.4 | 42.9 | 50.4 | 1.9 |
| MAE | E | 43.5 | 97.0 | 82.7 | 80.9 | 71.0 | 80.9 | 59.3 | 6.6 | 63.3 | 7.3 | 61.5 | 2.5 | 1.7 | 3.7 | 2.3 | 0.8 | 0.8 | 1.4 | 30.4 | 46.0 | 1.9 |
| MAE D100 | E | 47.7 | 98.1 | 82.7 | 86.7 | 71.0 | 76.8 | 59.3 | 8.0 | 63.4 | 10.3 | 61.5 | 2.5 | 1.7 | 3.7 | 3.1 | 0.8 | 0.8 | 1.4 | 30.4 | 46.8 | 1.9 |
| MAE | C | 44.0 | 97.7 | 77.4 | 80.9 | 71.4 | 80.8 | 62.3 | 6.6 | 64.5 | 7.4 | 62.2 | 16.3 | 1.9 | 3.5 | 2.3 | 0.7 | 0.7 | 1.5 | 30.2 | 47.0 | 1.9 |
| MAE D100 | C | 48.8 | 97.7 | 82.8 | 87.0 | 71.4 | 77.2 | 62.3 | 8.2 | 65.6 | 10.3 | 62.2 | 16.3 | 1.9 | 3.5 | 3.4 | 0.7 | 0.7 | 1.5 | 30.2 | 48.3 | 1.9 |
| EWF D30 | S | 56.0 | 98.5 | 88.0 | 86.1 | 80.7 | 85.8 | 22.0 | 10.0 | 67.1 | 5.4 | 56.5 | 16.6 | 1.5 | 3.8 | 3.9 | 0.7 | 0.9 | 1.4 | 41.3 | 47.9 | 1.9 |
| EWF D30 | E | 55.1 | 98.9 | 87.9 | 86.2 | 81.3 | 86.2 | 25.1 | 9.8 | 65.0 | 6.3 | 55.4 | 14.7 | 2.0 | 3.3 | 4.3 | 1.0 | 1.0 | 1.1 | 44.4 | 48.1 | 1.9 |
| W2V D100 | E | 34.1 | 79.9 | 59.3 | 43.7 | 44.6 | 73.7 | 18.1 | 5.9 | 59.2 | 6.6 | 99.3 | 8.5 | 1.5 | 3.6 | 52.5 | 9.5 | 0.9 | 1.2 | 27.7 | 41.5 | 1.8 |
| W2V | E | 33.5 | 78.0 | 56.8 | 40.9 | 44.6 | 73.4 | 18.1 | 5.3 | 58.8 | 5.6 | 99.3 | 8.5 | 1.5 | 3.6 | 41.2 | 9.5 | 0.9 | 1.2 | 27.7 | 40.1 | 1.8 |
| W2V D30 | S | 32.8 | 75.5 | 56.7 | 41.4 | 40.6 | 68.6 | 18.0 | 5.2 | 59.2 | 5.5 | 99.0 | 22.5 | 1.3 | 3.7 | 39.7 | 7.0 | 0.8 | 1.1 | 27.0 | 39.9 | 1.7 |
| W2V D30 | C | 33.6 | 78.9 | 56.9 | 42.0 | 41.1 | 70.3 | 18.4 | 5.3 | 61.2 | 5.7 | 99.1 | 24.4 | 1.3 | 3.8 | 42.8 | 8.3 | 0.8 | 1.0 | 27.7 | 41.0 | 1.7 |
| MAE D30 | C | 44.0 | 96.0 | 77.4 | 80.9 | 62.8 | 66.4 | 58.7 | 6.6 | 64.5 | 7.4 | 37.6 | 16.3 | 1.5 | 3.5 | 2.3 | 0.5 | 0.7 | 1.1 | 27.4 | 43.2 | 1.7 |
| MAE D30 | S | 43.8 | 95.8 | 77.3 | 81.1 | 61.4 | 66.5 | 56.5 | 7.0 | 62.5 | 6.9 | 37.6 | - | 1.3 | 3.8 | 2.2 | 0.5 | 0.6 | 1.1 | 29.7 | 44.9 | 1.7 |
| W2V D30 | E | 33.5 | 78.0 | 56.8 | 40.9 | 40.2 | 69.7 | 17.2 | 5.3 | 58.8 | 5.6 | 99.1 | 8.5 | 1.5 | 3.3 | 41.2 | 7.7 | 0.8 | 1.0 | 28.6 | 39.4 | 1.6 |
| MAE D30 | E | 43.5 | 95.6 | 77.4 | 80.9 | 62.8 | 66.2 | 56.8 | 6.6 | 63.3 | 7.3 | 37.1 | 2.5 | 1.4 | 3.4 | 2.3 | 0.6 | 0.6 | 1.1 | 28.3 | 42.1 | 1.6 |
| CC | E | 33.8 | 89.6 | 72.7 | 47.6 | 58.6 | 68.1 | 2.7 | 3.9 | 57.4 | 1.6 | 18.7 | 15.1 | 0.8 | 3.8 | 1.8 | 0.2 | 0.6 | 1.0 | 16.8 | 32.6 | 1.5 |
| CC | S | 30.4 | 90.7 | 70.0 | 40.7 | 47.5 | 61.2 | 4.2 | 3.6 | 56.5 | 1.9 | 46.4 | 15.1 | 1.0 | 3.3 | 1.7 | 0.2 | 0.4 | 0.9 | 17.5 | 32.5 | 1.4 |
| CC | C | 31.4 | 89.0 | 71.9 | 44.8 | 55.1 | 62.2 | 2.8 | 3.4 | 56.2 | 1.4 | 25.6 | 15.1 | 0.7 | 3.2 | 1.9 | 0.2 | 0.4 | 0.7 | 18.6 | 32.0 | 1.3 |

*Table 22.* P@5 Results Across Subsets and Distances (↑ better)

| Method | Dist | BC | BS1 | BS2 | BS3 | BS4 | BS5 | ES1 | HP | HS1 | HS2 | HU1 | HU2 | HU3 | HU4 | HW1 | HW2 | HW3 | HW4 | OC1 | Avg | Avg (Blind) |
|---|---|---|---|---|---|---|---|---|---|---|---|---|---|---|---|---|---|---|---|---|---|---|
| EWMTF D100 | S | 68.0 | 96.1 | 94.5 | 91.6 | 90.7 | 93.5 | 62.6 | **24.2** | 68.7 | 19.7 | 60.9 | 8.1 | **10.9** | **9.1** | 35.8 | 7.3 | **5.4** | **5.3** | 38.5 | **57.4** | **7.7** |
| EWMTF | S | 58.9 | 96.1 | 89.4 | 86.0 | 90.7 | 93.5 | 62.6 | 12.1 | 67.7 | 12.9 | 60.9 | 8.1 | **10.9** | **9.1** | 9.6 | 7.3 | **5.4** | **5.3** | 38.5 | 53.0 | **7.7** |
| BEATs | E | 67.1 | 97.8 | 93.8 | 89.1 | 88.6 | 94.5 | 88.5 | 13.6 | 76.3 | 24.8 | 15.3 | 9.4 | 10.0 | 8.2 | 18.8 | 4.7 | 4.6 | 5.0 | 48.3 | 55.4 | 7.0 |
| BEATs | C | 67.1 | 97.7 | 93.8 | 89.1 | 88.6 | 94.4 | 88.8 | 13.5 | 77.3 | 25.0 | 15.3 | 10.4 | 9.8 | 8.8 | 19.0 | 4.7 | 4.4 | 4.9 | **49.9** | 55.6 | 7.0 |
| EMTF | S | 55.1 | 96.2 | 93.6 | 81.3 | 89.2 | 94.0 | 57.3 | 12.7 | 64.6 | 10.0 | **64.7** | 11.4 | 10.1 | 7.6 | 10.9 | 8.3 | 5.1 | 4.3 | 38.1 | 52.5 | 6.8 |
| EMTF D100 | S | 65.8 | 96.2 | 93.6 | 90.9 | 89.2 | 94.0 | 57.3 | 23.4 | 67.0 | 16.8 | **64.7** | 11.4 | 10.1 | 7.6 | 37.8 | 8.3 | 5.1 | 4.3 | 38.1 | 57.0 | 6.8 |
| EWTF | S | 66.6 | 98.0 | 96.1 | 90.3 | 94.3 | 95.1 | 59.7 | 9.0 | 66.0 | 13.5 | 20.0 | 10.6 | 9.8 | 7.6 | 5.5 | 4.5 | 4.8 | 4.1 | 39.0 | 51.2 | 6.6 |
| EWTF D100 | S | 73.2 | 98.0 | 96.1 | 93.9 | 94.3 | 95.1 | 59.7 | 14.6 | 67.3 | 17.7 | 20.0 | 10.6 | 9.8 | 7.6 | 10.5 | 4.5 | 4.8 | 4.1 | 39.0 | 53.0 | 6.6 |
| EWT D100 | S | 72.7 | 97.5 | 96.1 | 93.8 | 94.2 | 95.0 | 60.0 | 14.4 | 66.8 | 17.4 | 19.8 | 11.4 | 9.7 | 7.3 | 10.4 | 4.5 | 4.7 | 4.0 | 38.0 | 52.8 | 6.4 |
| EWT | S | 66.5 | 97.5 | 96.1 | 90.0 | 94.2 | 95.1 | 60.0 | 9.2 | 66.2 | 13.6 | 19.8 | 11.4 | 9.7 | 7.3 | 5.5 | 4.5 | 4.7 | 4.0 | 38.0 | 51.2 | 6.4 |
| EWTF D100 | C | 71.9 | 99.1 | 95.3 | 93.2 | 93.6 | 95.0 | 58.7 | 11.7 | 69.0 | 17.2 | 19.2 | 9.7 | 8.7 | 7.1 | 9.0 | 3.6 | 4.1 | 3.8 | 39.3 | 52.4 | 5.9 |
| EWTF | C | 69.0 | 98.9 | 95.7 | 91.3 | 93.6 | 95.0 | 58.7 | 9.0 | 67.8 | 14.7 | 19.2 | 9.7 | 8.7 | 7.1 | 5.7 | 3.6 | 4.1 | 3.8 | 39.3 | 51.4 | 5.9 |
| ETF | S | 61.8 | 97.1 | 94.4 | 87.6 | 91.6 | 94.8 | 61.1 | 12.3 | 64.5 | 12.5 | 22.2 | 12.9 | 8.9 | 6.6 | 7.3 | 5.0 | 4.3 | 3.7 | 38.8 | 50.9 | 5.9 |
| ETF D100 | S | 69.9 | 97.1 | 94.4 | 92.8 | 91.6 | 94.8 | 61.1 | 17.9 | 66.9 | 17.0 | 22.2 | 12.9 | 8.9 | 6.6 | 15.5 | 5.0 | 4.3 | 3.7 | 38.8 | 53.2 | 5.9 |
| EWT | C | 68.7 | 98.8 | 95.7 | 91.1 | 93.5 | 95.0 | 58.5 | 9.0 | 67.6 | 14.5 | 19.1 | 11.2 | 8.5 | 6.9 | 5.6 | 3.6 | 4.1 | 3.8 | 38.0 | 51.3 | 5.8 |
| EWT D100 | C | 71.7 | 99.1 | 95.2 | 93.0 | 93.5 | 95.0 | 58.5 | 11.6 | 69.0 | 17.0 | 19.1 | 11.2 | 8.5 | 6.9 | 9.0 | 3.6 | 4.1 | 3.8 | 38.0 | 52.3 | 5.8 |
| ET D100 | S | 69.0 | 96.7 | 94.1 | 92.6 | 91.1 | 94.7 | 60.6 | 17.9 | 65.9 | 16.7 | 22.0 | 12.9 | 8.8 | 6.3 | 15.3 | 5.0 | 4.4 | 3.7 | 37.9 | 52.8 | 5.8 |
| ET | S | 61.8 | 97.1 | 89.4 | 92.6 | 91.1 | 94.8 | 60.6 | 12.2 | 64.6 | 12.3 | 22.0 | 12.9 | 8.8 | 6.3 | 7.3 | 5.0 | 4.4 | 3.7 | 37.9 | 50.8 | 5.8 |
| EWTF D100 | E | 70.9 | 98.9 | 95.4 | 93.1 | 93.8 | **95.1** | 58.6 | 11.3 | 69.5 | 17.3 | 19.2 | 9.5 | 8.4 | 7.2 | 9.1 | 3.5 | 3.9 | 3.7 | 37.9 | 52.2 | 5.8 |
| EWTF | E | 68.4 | 99.0 | 94.4 | 91.3 | 93.8 | **95.1** | 58.6 | 9.0 | 67.8 | 14.8 | 19.2 | 9.5 | 8.4 | 7.2 | 5.6 | 3.5 | 3.9 | 3.7 | 37.9 | 51.2 | 5.8 |
| EWT D100 | E | 70.7 | 98.8 | 95.3 | 92.9 | 93.7 | 95.0 | 58.4 | 11.2 | 69.3 | 17.0 | 19.2 | 9.1 | 8.2 | 7.2 | 9.0 | 3.5 | 4.0 | 3.6 | 37.1 | 52.0 | 5.7 |
| EWT | E | 68.3 | 99.0 | 94.3 | 91.2 | 93.7 | **95.1** | 58.4 | 8.9 | 67.6 | 14.6 | 19.2 | 9.1 | 8.2 | 7.2 | 5.6 | 3.5 | 4.0 | 3.6 | 37.1 | 51.0 | 5.7 |
| ETF | C | 65.1 | 98.9 | 93.8 | 89.3 | 90.4 | 94.8 | 59.7 | 13.1 | 66.6 | 13.5 | 21.0 | 12.8 | 8.4 | 6.5 | 8.2 | 4.4 | 3.9 | 3.5 | 41.3 | 51.5 | 5.6 |
| ETF D100 | C | 68.7 | 99.2 | 93.8 | 92.0 | 90.4 | 94.8 | 59.7 | 16.5 | 68.3 | 16.3 | 21.0 | 12.8 | 8.4 | 6.5 | 13.6 | 4.4 | 3.9 | 3.5 | 41.3 | 52.9 | 5.6 |
| ET D100 | C | 68.2 | 99.1 | 93.6 | 91.7 | 90.1 | 94.8 | 59.4 | 16.4 | 67.9 | 16.0 | 20.9 | 12.6 | 8.3 | 6.2 | 13.4 | 4.4 | 3.9 | 3.5 | 40.3 | 52.6 | 5.5 |
| ET | C | 64.4 | 99.1 | 93.6 | 89.0 | 90.1 | 94.8 | 59.4 | 13.0 | 66.8 | 13.2 | 20.9 | 12.6 | 8.3 | 6.2 | 8.1 | 4.4 | 3.9 | 3.5 | 40.3 | 51.3 | 5.5 |
| EW | C | 58.7 | 98.6 | 95.7 | 85.6 | 87.2 | 90.3 | 55.8 | 10.5 | 66.5 | 11.8 | 36.1 | 11.8 | 7.1 | 7.3 | 8.0 | 2.8 | 3.5 | 3.8 | 33.2 | 50.2 | 5.4 |
| EWMTF | C | 58.7 | 97.5 | 92.1 | 85.6 | 87.2 | 92.1 | 55.8 | 10.5 | 66.5 | 11.8 | 36.1 | 11.8 | 7.1 | 7.3 | 8.0 | 2.8 | 3.5 | 3.8 | 33.2 | 50.0 | 5.4 |
| EWMTF D100 | C | 61.0 | 97.5 | 91.4 | 87.9 | 87.2 | 92.1 | 55.8 | 14.7 | 67.1 | 14.3 | 36.1 | 11.8 | 7.1 | 7.3 | 15.5 | 2.8 | 3.5 | 3.8 | 33.2 | 51.2 | 5.4 |
| ETF D100 | E | 68.1 | 98.9 | 94.0 | 91.9 | 90.7 | 95.0 | 59.4 | 16.0 | 69.4 | 16.3 | 20.9 | 12.6 | 8.0 | 6.5 | 13.4 | 4.4 | 3.7 | 3.4 | 40.5 | 52.8 | 5.4 |
| ETF | E | 64.3 | 98.9 | 94.4 | 89.4 | 90.7 | 95.0 | 59.4 | 12.8 | 66.3 | 13.5 | 20.9 | 12.6 | 8.0 | 6.5 | 7.8 | 4.4 | 3.7 | 3.4 | 40.5 | 51.4 | 5.4 |
| ET | E | 63.9 | 98.9 | 93.8 | 89.1 | 90.3 | 95.0 | 58.7 | 12.7 | 66.8 | 13.3 | 20.8 | 12.4 | 7.9 | 6.3 | 7.8 | 4.3 | 3.7 | 3.4 | 39.3 | 51.1 | 5.3 |
| ET D100 | E | 67.6 | 98.8 | 93.8 | 91.7 | 90.3 | 95.0 | 58.7 | 15.9 | 68.9 | 16.1 | 20.8 | 12.4 | 7.9 | 6.3 | 13.3 | 4.3 | 3.7 | 3.4 | 39.3 | 52.5 | 5.3 |
| CLP D100 | C | 75.4 | 99.1 | 95.4 | 92.9 | 91.9 | 94.8 | **93.7** | 16.6 | **73.4** | 20.5 | 12.2 | 9.7 | 7.9 | 6.4 | 16.4 | 4.6 | 3.4 | 3.3 | 47.4 | 56.3 | 5.3 |
| CLP | C | 72.5 | 98.9 | 94.2 | 91.5 | 91.9 | 94.8 | **93.7** | 15.2 | 73.0 | 18.4 | 12.2 | 9.7 | 7.9 | 6.4 | 14.3 | 4.6 | 3.4 | 3.3 | 47.4 | 55.5 | 5.3 |
| CLP D100 | S | 75.7 | 96.8 | 95.6 | 93.2 | 93.0 | 94.7 | 92.7 | 17.0 | 69.3 | **20.5** | 14.2 | 9.6 | 7.8 | 6.2 | 15.8 | 4.7 | 3.5 | 3.4 | 41.7 | 55.6 | 5.2 |
| CLP | S | 69.0 | 98.1 | 95.6 | 89.5 | 93.0 | 94.8 | 92.7 | 14.0 | 70.0 | 16.6 | 14.2 | 9.6 | 7.8 | 6.2 | 11.9 | 4.7 | 3.5 | 3.4 | 41.7 | 54.4 | 5.2 |
| EWMTF | E | 58.0 | 97.7 | 91.5 | 85.3 | 87.4 | 91.8 | 55.2 | 10.2 | 68.2 | 11.5 | 37.2 | 9.3 | 6.5 | 7.1 | 7.4 | 2.7 | 3.4 | 3.6 | 32.4 | 49.7 | 5.2 |
| EWMTF D100 | E | 60.4 | 97.7 | 91.5 | 87.6 | 87.4 | 91.8 | 55.2 | 14.1 | 68.2 | 14.0 | 37.2 | 9.3 | 6.5 | 7.1 | 14.8 | 2.7 | 3.4 | 3.6 | 32.4 | 50.9 | 5.2 |
| EW | E | 58.0 | 98.6 | 91.5 | 85.3 | 87.4 | 90.3 | 55.2 | 10.2 | 68.2 | 11.5 | 37.2 | 9.3 | 6.5 | 7.1 | 7.4 | 2.7 | 3.4 | 3.6 | 32.4 | 49.7 | 5.2 |
| CLP | E | 72.5 | 99.2 | 95.4 | 91.6 | 91.9 | 94.8 | 93.4 | 14.8 | 72.1 | 18.4 | 12.2 | 9.7 | 7.3 | 6.2 | 14.0 | 4.4 | 3.2 | 3.0 | 47.1 | 55.4 | 4.9 |
| CLP D100 | E | 75.0 | 99.2 | 95.3 | 93.0 | 91.9 | 94.8 | 93.4 | 16.1 | 73.1 | 20.1 | 12.2 | 9.7 | 7.3 | 6.2 | 16.2 | 4.4 | 3.2 | 3.0 | 47.1 | 56.1 | 4.9 |
| EWTF D30 | C | 69.0 | 99.2 | 94.3 | 91.3 | 91.8 | 94.4 | 53.9 | 9.0 | 67.8 | 14.7 | 16.6 | 7.4 | 6.6 | 6.2 | 5.7 | 2.2 | 3.0 | 3.2 | 37.6 | 50.3 | 4.8 |
| EWT D30 | C | 68.7 | 99.1 | 94.2 | 91.1 | 91.5 | 94.3 | 53.5 | 9.0 | 67.6 | 14.5 | 16.5 | 11.2 | 6.6 | 6.2 | 5.6 | 2.2 | 3.0 | 3.2 | 36.6 | 50.4 | 4.7 |
| EWTF D30 | E | 68.4 | 99.0 | 94.4 | 91.3 | 92.2 | 94.5 | 53.6 | 9.0 | 67.8 | 14.8 | 16.8 | 7.0 | 6.4 | 6.4 | 5.6 | 2.2 | 2.9 | 3.1 | 37.3 | 50.2 | 4.7 |
| EWTF D30 | S | 66.6 | 98.6 | 94.0 | 90.3 | 90.7 | 93.6 | 52.0 | 9.0 | 66.0 | 13.5 | 15.9 | 7.0 | 6.5 | 6.4 | 5.5 | 2.2 | 2.9 | 3.1 | 37.2 | 49.5 | 4.7 |
| EMTF D100 | C | 57.2 | 97.2 | 89.8 | 85.0 | 84.8 | 92.5 | 48.8 | 15.6 | 63.8 | 11.1 | 33.4 | 13.6 | 6.9 | 6.0 | 18.1 | 3.6 | 3.1 | 2.9 | 35.3 | 50.0 | 4.7 |
| E | C | 53.3 | 97.0 | 0.7 | 80.7 | 84.8 | 93.0 | 48.8 | 11.5 | 63.1 | 8.9 | 33.4 | 13.6 | 6.9 | 6.0 | 9.6 | 3.6 | 3.1 | 2.9 | 35.3 | 42.5 | 4.7 |
| EMTF | C | 53.3 | 97.0 | 89.8 | 80.7 | 84.8 | 92.5 | 48.8 | 11.5 | 63.1 | 8.9 | 33.4 | 13.6 | 6.9 | 6.0 | 9.6 | 3.6 | 3.1 | 2.9 | 35.3 | 48.4 | 4.7 |
| EWT D30 | S | 66.5 | 98.4 | 94.1 | 90.0 | 90.2 | 93.5 | 52.0 | 9.2 | 66.2 | 13.6 | 16.1 | 11.4 | 6.4 | 6.4 | 5.5 | 2.2 | 2.8 | 3.2 | 36.2 | 49.7 | 4.7 |
| EWT D30 | E | 68.3 | 98.7 | 94.3 | 91.2 | 92.0 | 94.5 | 53.4 | 8.9 | 67.6 | 14.6 | 16.7 | 9.1 | 6.3 | 6.4 | 5.6 | 2.1 | 2.9 | 3.1 | 36.6 | 50.2 | 4.7 |
| CLP D30 | C | 72.5 | 98.9 | 94.2 | 91.5 | 89.9 | 94.4 | 93.2 | 15.2 | 73.0 | 18.4 | 9.3 | 9.7 | 6.9 | 5.8 | 14.3 | 3.9 | 2.8 | 2.8 | 46.5 | 55.0 | 4.6 |
| EMTF D100 | E | 56.3 | 97.4 | 90.2 | 85.2 | 85.1 | 92.4 | 48.0 | 15.2 | 64.6 | 10.9 | 33.7 | 10.7 | 6.7 | 5.6 | 17.9 | 3.5 | 2.8 | 2.7 | 34.6 | 49.7 | 4.5 |
| E | E | 53.0 | 97.4 | 93.8 | 80.7 | 85.1 | 92.9 | 48.0 | 11.1 | 63.4 | 8.5 | 33.7 | 10.7 | 6.7 | 5.6 | 9.0 | 3.5 | 2.8 | 2.7 | 34.6 | 48.4 | 4.5 |
| EMTF | E | 53.0 | 97.4 | 90.9 | 80.7 | 85.1 | 92.4 | 48.0 | 11.1 | 63.4 | 8.5 | 33.7 | 10.7 | 6.7 | 5.6 | 9.0 | 3.5 | 2.8 | 2.7 | 34.6 | 48.1 | 4.5 |
| CLP D30 | E | 72.5 | 99.2 | 94.2 | 91.6 | 90.1 | 94.4 | 93.0 | 14.8 | 72.1 | 18.4 | 9.4 | 9.7 | 6.2 | 5.7 | 14.0 | 3.6 | 2.8 | 2.6 | 45.9 | 54.9 | 4.3 |
| ET D30 | C | 64.4 | 99.1 | 91.3 | 89.0 | 87.6 | 94.0 | 54.5 | 13.0 | 66.8 | 13.2 | 18.4 | 12.6 | 6.5 | 5.3 | 8.1 | 2.6 | 2.7 | 2.7 | 38.7 | 50.2 | 4.3 |
| ETF D30 | C | 65.1 | 99.2 | 91.6 | 89.3 | 87.9 | 94.0 | 55.0 | 13.1 | 66.6 | 13.5 | 18.5 | 12.8 | 6.5 | 5.2 | 8.2 | 2.6 | 2.7 | 2.8 | 39.7 | 50.5 | 4.3 |
| EWMTF D30 | C | 58.7 | 97.5 | 89.6 | 85.6 | 85.1 | 90.5 | 51.2 | 10.5 | 66.5 | 11.8 | 25.1 | 11.8 | 5.4 | 6.3 | 8.0 | 1.6 | 2.3 | 3.1 | 32.7 | 48.4 | 4.3 |
| EWMTF D30 | S | 58.9 | 96.3 | 90.3 | 86.0 | 84.7 | 90.3 | 52.7 | 12.1 | 67.7 | 12.9 | 26.4 | 8.1 | 5.7 | 5.9 | 9.4 | 2.0 | 2.3 | 3.1 | 33.3 | 48.8 | 4.2 |
| EWMTF D30 | E | 58.0 | 97.8 | 89.6 | 85.3 | 85.3 | 90.0 | 50.3 | 10.2 | 67.6 | 11.5 | 25.7 | 9.3 | 4.9 | 6.4 | 7.4 | 1.5 | 2.3 | 3.0 | 31.6 | 48.1 | 4.1 |
| ET D30 | E | 63.9 | 98.8 | 91.4 | 89.1 | 88.0 | 94.1 | 54.3 | 12.7 | 66.8 | 13.3 | 18.6 | 12.4 | 6.0 | 5.1 | 7.8 | 2.5 | 2.6 | 2.6 | 39.1 | 50.2 | 4.1 |
| ETF D30 | E | 64.3 | 98.8 | 91.6 | 89.4 | 88.2 | 94.1 | 54.8 | 12.8 | 66.3 | 13.5 | 18.6 | 12.6 | 5.9 | 5.1 | 7.8 | 2.5 | 2.6 | 2.6 | 39.3 | 50.3 | 4.1 |
| CLP D30 | E | 69.0 | 98.1 | 92.4 | 89.5 | 88.7 | 93.9 | 14.0 | 70.0 | 16.6 | 8.6 | 9.6 | 5.9 | 5.4 | 11.9 | 3.1 | 2.5 | 2.8 | 41.0 | 53.2 | 4.1 |
| EMTF D30 | S | 55.1 | 96.2 | 88.2 | 81.3 | 81.0 | 90.6 | 44.7 | 12.7 | 64.6 | 10.0 | 25.2 | 11.4 | 5.5 | 5.6 | 10.9 | 2.3 | 2.2 | 2.6 | 34.8 | 47.3 | 4.0 |
| ET D30 | S | 61.8 | 97.8 | 89.4 | 87.3 | 85.9 | 93.2 | 52.2 | 12.2 | 64.6 | 12.3 | 17.7 | 12.9 | 5.8 | 5.0 | 7.3 | 2.3 | 2.6 | 2.5 | 36.6 | 48.9 | 4.0 |
| ETF D30 | S | 61.8 | 97.7 | 89.8 | 87.6 | 86.2 | 93.4 | 52.0 | 12.3 | 64.5 | 12.5 | 17.7 | 12.9 | 5.7 | 5.0 | 7.3 | 2.2 | 2.5 | 2.5 | 37.5 | 49.0 | 3.9 |
| EMTF D30 | C | 53.3 | 97.0 | 87.5 | 80.7 | 81.9 | 90.7 | 44.0 | 11.5 | 63.1 | 8.9 | 21.0 | 13.6 | 5.4 | 5.3 | 9.6 | 2.0 | 2.1 | 2.3 | 34.1 | 46.6 | 3.8 |
| WLM | S | 43.7 | 97.8 | 86.2 | 81.5 | 86.4 | 90.7 | 25.9 | 21.1 | 58.8 | 7.2 | 37.0 | 9.3 | 4.2 | 5.3 | 62.7 | **18.7** | 1.9 | 2.9 | 34.2 | 50.8 | 3.6 |
| WLM D100 | S | 50.4 | 97.1 | 90.5 | 87.0 | 86.4 | 90.4 | 25.9 | 23.1 | 62.3 | 9.3 | 37.0 | 22.6 | 4.2 | 5.3 | 69.7 | **18.7** | 1.9 | 2.9 | 34.2 | 53.6 | 3.6 |
| EMTF D30 | E | 53.0 | 97.5 | 87.8 | 80.7 | 81.9 | 90.6 | 43.3 | 11.1 | 63.4 | 8.5 | 21.0 | 10.7 | 5.2 | 4.8 | 9.0 | 1.9 | 2.0 | 2.1 | 33.3 | 46.2 | 3.5 |
| WLM | C | 44.9 | 98.6 | 87.5 | 83.0 | 85.6 | 91.3 | 26.2 | 21.5 | 60.1 | 8.0 | 30.9 | **25.9** | 3.5 | 5.2 | 64.5 | 18.2 | 1.6 | 2.7 | 37.6 | 52.3 | 3.2 |
| WLM D100 | C | 49.5 | 98.5 | 90.5 | 86.6 | 85.6 | 90.2 | 26.2 | 22.8 | 61.7 | 9.2 | 30.9 | **25.9** | 3.5 | 5.2 | 69.9 | 18.2 | 1.6 | 2.7 | 37.6 | 53.5 | 3.2 |
| WLM | E | 45.1 | 98.5 | 87.5 | 83.1 | 85.5 | 91.2 | 24.7 | 21.4 | 59.6 | 8.0 | 30.3 | 20.1 | 3.2 | 5.1 | 64.2 | 17.5 | 1.6 | 2.6 | 34.7 | 51.4 | 3.1 |
| WLM D100 | E | 49.1 | 98.5 | 90.4 | 86.6 | 85.5 | 90.2 | 24.7 | 22.7 | 62.0 | 9.0 | 30.3 | 20.2 | 3.2 | 5.1 | 69.3 | 17.5 | 1.6 | 2.6 | 34.7 | 52.7 | 3.1 |
| M | C | **77.0** | 99.9 | 98.2 | 96.4 | 94.2 | 93.1 | 22.8 | 17.1 | 56.2 | 7.2 | 23.5 | 9.0 | 3.4 | 5.0 | 17.9 | 4.7 | 1.8 | 2.2 | 37.5 | 50.3 | 3.1 |

*Table 22.* (Continued) P@5 Results Across Subsets and Distances (↑ better)

| Method | Dist | BC | BS1 | BS2 | BS3 | BS4 | BS5 | ES1 | HP | HS1 | HS2 | HU1 | HU2 | HU3 | HU4 | HW1 | HW2 | HW3 | HW4 | OC1 | Avg | Avg (Blind) |
|---|---|---|---|---|---|---|---|---|---|---|---|---|---|---|---|---|---|---|---|---|---|---|
| VC | C | 70.7 | 99.9 | 97.0 | 95.3 | 93.5 | 94.6 | 24.8 | 15.0 | 56.8 | 9.2 | 20.8 | 9.4 | 3.2 | 4.9 | 15.3 | 4.5 | 1.6 | 2.2 | 39.0 | 49.7 | 3.0 |
| VC | E | 70.2 | 99.8 | 96.9 | 95.2 | 93.4 | 94.3 | 26.5 | 14.8 | 56.7 | 9.2 | 19.6 | 8.5 | 3.3 | 4.6 | 14.4 | 4.5 | 1.5 | 1.9 | 38.0 | 49.5 | 2.8 |
| WLM D30 | C | 44.9 | 98.6 | 87.5 | 83.0 | 82.8 | 88.2 | 23.3 | 21.5 | 60.1 | 8.0 | 22.9 | **25.9** | 2.8 | 4.8 | 64.5 | 14.7 | 1.3 | 2.3 | 37.2 | 50.9 | 2.8 |
| WLM D30 | E | 45.1 | 98.5 | 87.5 | 83.1 | 82.8 | 88.3 | 22.9 | 21.4 | 59.6 | 8.0 | 22.8 | 20.2 | 2.7 | 4.6 | 64.2 | 14.5 | 1.2 | 2.2 | 36.1 | 50.3 | 2.7 |
| M | S | 70.7 | **100.0** | 96.7 | 94.8 | 92.2 | 91.2 | 15.0 | 11.3 | 53.6 | 6.3 | 22.8 | 7.7 | 2.7 | 4.3 | 11.1 | 3.8 | 1.5 | 2.0 | 33.0 | 47.3 | 2.6 |
| M | E | 75.5 | 99.7 | **98.2** | **96.4** | **94.4** | 93.2 | 22.1 | 17.1 | 55.1 | 6.9 | 24.3 | 8.7 | 2.6 | 4.5 | 18.8 | 4.7 | 1.5 | 1.9 | 36.9 | 50.1 | 2.6 |
| WLM D30 | S | 43.7 | 97.8 | 86.2 | 81.5 | 80.3 | 86.7 | 21.0 | 21.1 | 58.8 | 7.2 | 22.7 | 9.3 | 2.6 | 4.0 | 62.7 | 13.7 | 1.2 | 2.3 | 35.6 | 48.6 | 2.5 |
| EF D100 | C | 50.8 | 97.8 | 88.9 | 85.0 | 83.7 | 89.1 | 18.4 | 11.8 | 53.7 | 4.6 | 16.4 | 6.0 | 2.6 | 4.2 | 6.5 | 1.2 | 1.0 | 1.8 | 34.6 | 43.2 | 2.4 |
| EF | C | 48.6 | 97.5 | 88.9 | 81.3 | 83.7 | 89.1 | 18.4 | 10.1 | 53.1 | 4.6 | 16.4 | 6.0 | 2.6 | 4.2 | 5.0 | 1.2 | 1.0 | 1.8 | 34.6 | 42.6 | 2.4 |
| AC | S | 71.7 | **100.0** | 97.2 | 92.1 | 92.1 | 93.5 | 22.5 | 9.5 | 56.7 | 9.3 | 21.4 | 7.3 | 2.8 | 3.9 | 9.5 | 2.7 | 1.4 | 1.5 | 29.2 | 47.6 | 2.4 |
| EF | E | 48.3 | 98.4 | 89.5 | 80.9 | 83.7 | 88.9 | 17.2 | 9.9 | 52.9 | 4.5 | 16.2 | 6.1 | 2.6 | 4.1 | 4.7 | 1.2 | 0.9 | 1.7 | 32.8 | 42.4 | 2.3 |
| EF D100 | E | 50.0 | 98.4 | 88.7 | 84.6 | 83.7 | 88.9 | 17.2 | 11.6 | 53.7 | 4.6 | 16.2 | 6.1 | 2.6 | 4.1 | 6.1 | 1.2 | 0.9 | 1.7 | 32.8 | 42.9 | 2.3 |
| EF D30 | E | 48.3 | 97.8 | 86.4 | 80.9 | 81.1 | 87.3 | 18.2 | 9.9 | 52.9 | 4.5 | 11.7 | 6.1 | 2.5 | 4.0 | 4.7 | 1.0 | 0.7 | 1.7 | 32.0 | 41.5 | 2.2 |
| EF D100 | S | 51.8 | 95.9 | 89.6 | 87.0 | 84.5 | 89.7 | 14.7 | 13.8 | 52.1 | 4.3 | 17.0 | 7.8 | 2.4 | 3.9 | 8.6 | 1.4 | 0.9 | 1.8 | 31.3 | 43.3 | 2.2 |
| EF | S | 48.2 | 97.4 | 86.2 | 80.6 | 84.5 | 89.7 | 14.7 | 9.9 | 52.4 | 4.3 | 17.0 | 7.8 | 2.4 | 3.9 | 4.8 | 1.4 | 0.9 | 1.8 | 31.3 | 42.0 | 2.2 |
| EF D30 | C | 48.6 | 97.5 | 86.5 | 81.3 | 81.1 | 87.2 | 18.7 | 10.1 | 53.1 | 4.6 | 11.7 | 6.0 | 2.6 | 3.9 | 5.0 | 1.1 | 0.8 | 1.6 | 33.7 | 41.8 | 2.2 |
| EF D30 | S | 48.2 | 97.4 | 86.2 | 80.6 | 79.9 | 87.1 | 15.3 | 9.9 | 52.4 | 4.3 | 10.1 | 7.8 | 1.9 | 3.9 | 4.8 | 1.1 | 0.9 | 1.6 | 32.6 | 41.2 | 2.1 |
| VC | S | 57.2 | 99.0 | 91.5 | 89.7 | 88.8 | 88.8 | 17.2 | 11.3 | 51.4 | 5.2 | 14.8 | 6.4 | 2.2 | 3.7 | 8.1 | 2.0 | 0.8 | 1.5 | 30.5 | 44.1 | 2.1 |
| AC | C | 69.0 | 99.1 | 96.9 | 91.9 | 91.6 | 92.7 | 22.1 | 9.1 | 54.4 | 8.0 | 20.7 | 6.8 | 2.3 | 3.6 | 8.9 | 2.4 | 1.0 | 1.4 | 25.3 | 46.6 | 2.1 |
| EWF | C | 48.9 | 97.9 | 84.6 | 82.4 | 79.2 | 83.2 | 19.5 | 8.1 | 51.5 | 4.1 | 18.5 | 5.0 | 2.3 | 3.4 | 2.9 | 0.7 | 0.9 | 1.4 | 34.8 | 41.4 | 2.0 |
| EWF D100 | C | 50.7 | 97.8 | 87.8 | 85.5 | 79.2 | 83.2 | 19.5 | 9.1 | 51.3 | 4.3 | 18.5 | 5.0 | 2.3 | 3.4 | 3.7 | 0.7 | 0.9 | 1.4 | 34.8 | 42.1 | 2.0 |
| AC | E | 68.1 | 99.1 | 96.9 | 91.7 | 91.5 | 92.5 | 19.3 | 9.0 | 52.3 | 6.9 | 19.9 | 5.2 | 2.1 | 3.5 | 8.7 | 2.4 | 1.0 | 1.3 | 24.9 | 45.9 | 2.0 |
| EWF D100 | S | 51.9 | 95.6 | 90.7 | 87.4 | 82.8 | 84.1 | 17.5 | 11.5 | 51.2 | 3.7 | 20.5 | 6.4 | 1.8 | 3.5 | 5.1 | 0.8 | 0.8 | 1.6 | 32.9 | 42.8 | 1.9 |
| EWF | S | 48.1 | 95.6 | 90.7 | 82.3 | 82.8 | 84.1 | 17.5 | 8.2 | 50.6 | 3.7 | 20.5 | 6.4 | 1.8 | 3.5 | 3.0 | 0.8 | 0.8 | 1.6 | 32.9 | 41.8 | 1.9 |
| EWF | E | 48.3 | 98.0 | 84.5 | 82.2 | 79.0 | 82.9 | 17.6 | 8.1 | 51.7 | 4.1 | 18.4 | 5.0 | 2.1 | 3.3 | 2.8 | 0.7 | 0.9 | 1.4 | 33.7 | 41.1 | 1.9 |
| EWF D100 | E | 50.0 | 98.0 | 87.7 | 85.4 | 79.0 | 82.9 | 17.6 | 9.1 | 51.3 | 4.2 | 18.4 | 5.0 | 2.1 | 3.3 | 3.5 | 0.7 | 0.9 | 1.4 | 33.7 | 41.8 | 1.9 |
| EWF D30 | S | 48.1 | 97.2 | 84.9 | 82.3 | 73.7 | 81.3 | 17.2 | 8.2 | 50.6 | 3.7 | 15.3 | 4.6 | 1.8 | 3.4 | 3.0 | 0.6 | 0.8 | 1.4 | 32.9 | 40.2 | 1.9 |
| EAT | C | 54.2 | 97.4 | 88.7 | 85.1 | 82.7 | 89.0 | 36.5 | 8.5 | 53.3 | 7.8 | 17.5 | 4.7 | 2.1 | 3.5 | 4.7 | 0.8 | 0.7 | 1.3 | 27.3 | 43.9 | 1.9 |
| EAT | E | 50.8 | 96.7 | 87.5 | 84.6 | 79.7 | 86.9 | 32.2 | 8.2 | 50.4 | 6.4 | 13.4 | 4.5 | 1.7 | 3.4 | 3.8 | 0.6 | 0.7 | 1.3 | 26.1 | 42.1 | 1.8 |
| EAT | S | 54.4 | 97.3 | 88.9 | 85.8 | 83.2 | 89.2 | 36.4 | 8.6 | 52.7 | 7.9 | 17.5 | 4.7 | 2.0 | 3.1 | 4.7 | 0.8 | 0.7 | 1.3 | 27.6 | 44.0 | 1.8 |
| EWF D30 | E | 48.3 | 97.9 | 84.5 | 82.2 | 74.0 | 81.2 | 19.0 | 8.1 | 51.7 | 4.1 | 14.7 | 4.2 | 1.9 | 3.3 | 2.8 | 0.6 | 0.8 | 1.4 | 32.8 | 40.4 | 1.8 |
| EWF D30 | C | 48.9 | 97.8 | 84.6 | 82.4 | 74.5 | 81.6 | 20.0 | 8.1 | 51.5 | 4.1 | 14.6 | 4.6 | 2.0 | 3.3 | 2.9 | 0.6 | 0.8 | 1.3 | 33.3 | 40.6 | 1.8 |
| W2V | C | 28.7 | 77.2 | 53.7 | 37.4 | 38.4 | 68.4 | 14.5 | 4.7 | 47.3 | 4.1 | 32.2 | 9.2 | 1.5 | 3.7 | 29.6 | 5.9 | 0.7 | 1.2 | 24.2 | 31.7 | 1.8 |
| W2V D100 | C | 29.7 | 77.8 | 55.9 | 39.9 | 38.4 | 68.8 | 14.5 | 5.1 | 48.5 | 4.5 | 32.2 | 9.2 | 1.5 | 3.7 | 40.0 | 5.9 | 0.7 | 1.2 | 24.2 | 33.0 | 1.8 |
| W2V | S | 28.6 | 75.4 | 53.4 | 37.2 | 40.8 | 68.4 | 15.0 | 4.7 | 46.7 | 3.7 | 34.8 | 7.8 | 1.8 | 3.0 | 27.0 | 6.7 | 0.8 | 1.2 | 23.8 | 31.6 | 1.7 |
| W2V D100 | S | 30.4 | 76.2 | 57.5 | 41.6 | 40.8 | 69.7 | 15.0 | 5.6 | 48.8 | 4.4 | 34.8 | 7.8 | 1.8 | 3.0 | 41.3 | 6.7 | 0.8 | 1.2 | 23.8 | 33.6 | 1.7 |
| W2V | E | 28.2 | 76.8 | 53.4 | 36.9 | 37.9 | 67.1 | 12.6 | 4.5 | 46.4 | 3.9 | 31.5 | 4.7 | 1.3 | 3.6 | 27.8 | 4.8 | 0.7 | 1.1 | 23.1 | 30.6 | 1.7 |
| W2V D100 | E | 29.5 | 77.5 | 55.5 | 38.9 | 37.9 | 67.8 | 12.6 | 4.9 | 46.5 | 4.3 | 31.5 | 4.7 | 1.3 | 3.6 | 36.7 | 4.8 | 0.7 | 1.1 | 23.1 | 31.8 | 1.7 |
| W2V D30 | S | 28.6 | 75.4 | 53.4 | 37.2 | 35.3 | 64.1 | 12.9 | 4.7 | 46.7 | 3.7 | 27.5 | 7.8 | 1.4 | 3.5 | 27.0 | 4.0 | 0.6 | 1.0 | 24.6 | 30.2 | 1.6 |
| W2V D30 | C | 28.7 | 77.2 | 53.7 | 37.4 | 35.3 | 64.9 | 14.0 | 4.7 | 47.3 | 4.1 | 28.2 | 9.2 | 1.4 | 3.4 | 29.6 | 4.3 | 0.7 | 1.1 | 24.2 | 30.8 | 1.6 |
| W2V D30 | E | 28.2 | 76.8 | 53.4 | 36.9 | 34.5 | 64.4 | 13.4 | 4.5 | 46.4 | 3.9 | 28.1 | 4.7 | 1.3 | 3.4 | 27.8 | 3.9 | 0.7 | 1.0 | 23.2 | 30.0 | 1.6 |
| MAE D100 | S | 45.0 | 95.6 | 81.5 | 86.4 | 67.7 | 73.1 | 49.2 | 7.8 | 50.8 | 7.1 | 18.5 | - | 1.2 | 3.2 | 2.7 | 0.6 | 0.6 | 1.3 | 26.4 | 43.7 | 1.6 |
| MAE | S | 38.8 | 95.6 | 81.5 | 76.2 | 67.7 | 68.7 | 49.2 | 6.2 | 49.1 | 4.6 | 18.5 | 0.8 | 1.2 | 3.2 | 1.8 | 0.6 | 0.6 | 1.3 | 26.4 | 39.0 | 1.6 |
| MAE D100 | C | 41.7 | 93.9 | 78.9 | 82.6 | 64.6 | 68.9 | 48.1 | 7.1 | 50.9 | 6.1 | 15.0 | 4.3 | 1.3 | 3.4 | 2.4 | 0.5 | 0.6 | 1.1 | 24.0 | 39.3 | 1.6 |
| MAE | C | 38.5 | 93.9 | 72.6 | 75.3 | 64.6 | 72.3 | 48.1 | 6.1 | 50.0 | 4.6 | 15.0 | 4.3 | 1.3 | 3.4 | 1.8 | 0.5 | 0.6 | 1.1 | 24.0 | 38.1 | 1.6 |
| MAE | E | 38.0 | 93.8 | 78.7 | 75.0 | 64.3 | 72.4 | 44.2 | 6.1 | 49.5 | 4.6 | 14.9 | 1.5 | 1.4 | 3.2 | 1.7 | 0.5 | 0.6 | 1.1 | 23.8 | 37.9 | 1.6 |
| MAE D100 | E | 41.0 | 93.7 | 78.7 | 82.3 | 64.3 | 68.5 | 44.2 | 7.1 | 50.6 | 5.9 | 14.9 | 1.5 | 1.4 | 3.2 | 2.3 | 0.5 | 0.6 | 1.1 | 23.8 | 38.6 | 1.6 |
| MAE D30 | S | 38.8 | 92.6 | 73.1 | 76.2 | 56.1 | 58.6 | 42.5 | 6.2 | 49.1 | 4.6 | 10.8 | - | 1.2 | 3.0 | 1.8 | 0.4 | 0.5 | 1.1 | 23.0 | 38.1 | 1.4 |
| MAE D30 | C | 38.5 | 92.3 | 72.6 | 75.3 | 56.2 | 57.6 | 44.4 | 6.1 | 50.0 | 4.6 | 10.3 | 4.3 | 1.2 | 3.1 | 1.8 | 0.4 | 0.5 | 0.9 | 22.7 | 35.8 | 1.4 |
| MAE D30 | E | 38.0 | 92.5 | 72.4 | 75.0 | 56.1 | 57.2 | 42.9 | 6.1 | 49.5 | 4.6 | 10.2 | 1.5 | 1.3 | 2.9 | 1.7 | 0.4 | 0.5 | 0.9 | 21.6 | 35.3 | 1.4 |
| CC | E | 30.6 | 85.8 | 68.9 | 42.4 | 51.2 | 60.6 | 3.6 | 3.5 | 44.9 | 1.4 | 6.4 | 3.4 | 0.8 | 3.0 | 1.6 | 0.2 | 0.5 | 0.9 | 14.4 | 27.9 | 1.3 |
| CC | C | 28.3 | 85.0 | 68.5 | 40.0 | 47.5 | 54.7 | 3.6 | 3.4 | 45.3 | 1.1 | 8.2 | 3.4 | 0.8 | 2.9 | 1.5 | 0.2 | 0.4 | 0.8 | 14.6 | 27.0 | 1.2 |
| CC | S | 27.2 | 89.0 | 67.8 | 38.3 | 42.2 | 50.6 | 3.8 | 3.3 | 42.5 | 1.4 | 12.9 | 3.4 | 0.9 | 2.8 | 1.5 | 0.2 | 0.5 | 0.7 | 14.1 | 26.5 | 1.2 |

*Table 23.* GSR Results Across Subsets and Distances (↑ better)

| Method | Dist | BC | BS1 | BS2 | BS3 | BS4 | BS5 | ES1 | HP | HS1 | HS2 | HU1 | HU2 | HU3 | HU4 | HW1 | HW2 | HW3 | HW4 | OC1 | Avg | Avg (Blind) |
|---|---|---|---|---|---|---|---|---|---|---|---|---|---|---|---|---|---|---|---|---|---|---|
| EWMTF D100 | S | 39.3 | 53.7 | 43.5 | 42.5 | 45.1 | 43.4 | 41.5 | 36.4 | 39.5 | 36.8 | 37.8 | **43.3** | 40.3 | 39.9 | 40.4 | **40.5** | 38.9 | 38.4 | 42.3 | **41.7** | **39.4** |
| EWMTF | S | 36.5 | 53.3 | 44.5 | 39.7 | 45.1 | 43.4 | 41.5 | 32.0 | 35.7 | 32.5 | 37.8 | **43.3** | 40.3 | 39.9 | 34.8 | **40.5** | 38.9 | 38.4 | 42.3 | 40.2 | **39.4** |
| ET | E | 36.5 | 53.3 | 43.3 | 39.6 | 44.9 | 43.3 | 41.4 | 32.1 | 35.9 | 32.5 | 37.7 | 43.2 | 40.3 | 39.9 | 34.8 | 40.5 | 38.9 | 38.4 | 42.3 | 40.1 | 39.3 |
| ET D100 | E | 39.3 | 53.5 | 43.3 | 42.4 | 44.9 | 43.3 | 41.4 | **36.4** | **39.5** | 36.7 | 37.7 | 43.2 | 40.3 | 39.9 | 40.4 | 40.5 | 38.9 | 38.4 | 42.3 | 41.7 | 39.3 |
| M | E | **41.0** | 58.6 | 50.8 | **47.1** | **47.9** | 39.3 | 36.6 | 31.9 | 32.0 | 34.4 | 39.4 | 35.4 | 40.3 | 39.3 | 38.4 | 38.3 | **39.4** | 38.3 | 39.1 | 40.7 | 39.3 |
| CLP D100 | E | 38.7 | 54.6 | 45.1 | 43.9 | 44.8 | 39.9 | 51.1 | 35.0 | 38.9 | 36.6 | 33.9 | 38.2 | 39.6 | 39.1 | 39.2 | 38.8 | 38.2 | 37.4 | **42.8** | 41.4 | 38.6 |
| CLP | E | 36.0 | 54.4 | 45.3 | 42.1 | 44.8 | 40.3 | 51.1 | 31.8 | 36.0 | 33.2 | 33.9 | 38.2 | 39.6 | 39.1 | 36.3 | 38.8 | 38.2 | 37.4 | **42.8** | 40.3 | 38.6 |
| EWTF D100 | E | 38.3 | 56.3 | 44.5 | 43.2 | 46.1 | 41.7 | 41.4 | 32.9 | 38.7 | 35.9 | 34.5 | 38.4 | 39.8 | 38.8 | 37.1 | 38.7 | 38.4 | 37.2 | 41.6 | 40.6 | 38.6 |
| EWTF | E | 35.8 | 57.5 | 43.1 | 41.0 | 46.1 | 41.7 | 41.4 | 28.5 | 35.3 | 31.7 | 34.5 | 38.4 | 39.8 | 38.8 | 31.3 | 38.7 | 38.4 | 37.2 | 41.6 | 39.1 | 38.6 |
| EWT D100 | E | 38.2 | 56.0 | 44.4 | 43.0 | 45.9 | 41.7 | 41.4 | 32.9 | 38.8 | 35.8 | 34.4 | 41.7 | 39.8 | 38.8 | 37.1 | 38.6 | 38.3 | 37.1 | 41.5 | 40.8 | 38.5 |
| EWT | E | 35.8 | 57.5 | 43.0 | 40.8 | 45.9 | 41.7 | 41.4 | 28.5 | 35.4 | 31.6 | 34.4 | 41.7 | 39.8 | 38.8 | 31.3 | 38.6 | 38.3 | 37.1 | 41.5 | 39.3 | 38.5 |
| EMTF | S | 24.7 | 42.5 | 40.4 | 28.9 | 42.5 | 40.1 | 39.0 | 19.7 | 26.2 | 20.2 | **46.9** | 33.6 | 37.7 | 37.4 | 25.3 | 37.5 | 35.1 | 35.0 | 40.2 | 33.8 | 36.3 |
| EMTF D100 | S | 36.0 | 44.7 | 40.4 | 39.7 | 42.5 | 40.1 | 39.0 | 31.6 | 36.8 | 32.4 | **46.9** | 33.6 | 37.7 | 37.4 | 40.3 | 37.5 | 35.1 | 35.0 | 40.2 | 38.8 | 36.3 |
| CLP | S | 27.3 | 45.6 | 41.8 | 34.2 | 44.0 | 30.7 | 50.2 | 21.2 | 28.7 | 21.6 | 30.6 | 27.8 | 37.3 | 37.5 | 25.5 | 35.8 | 35.2 | 34.8 | 41.5 | 33.8 | 36.2 |
| CLP D100 | S | 37.1 | 45.3 | 41.8 | 41.3 | 44.0 | 38.7 | 50.2 | 32.2 | 38.3 | 31.9 | 30.6 | 27.8 | 37.3 | 37.5 | 35.6 | 35.8 | 35.2 | 34.8 | 41.5 | 38.1 | 36.2 |
| ETF | E | 24.3 | 45.7 | 28.5 | 29.8 | 42.1 | 39.7 | 38.8 | 18.5 | 26.3 | 20.3 | 46.3 | 27.7 | 37.2 | 37.3 | 23.6 | 36.3 | 34.7 | 34.4 | 40.2 | 32.5 | 35.9 |
| ETF D100 | E | 35.3 | 45.7 | 40.0 | 39.8 | 42.1 | 39.7 | 38.8 | 31.0 | 36.7 | 31.6 | 46.3 | 27.7 | 37.2 | 37.3 | 39.5 | 36.3 | 34.7 | 34.4 | 40.2 | 38.1 | 35.9 |
| WLM D100 | E | 26.2 | 53.5 | 42.8 | 34.4 | 37.0 | 35.0 | 37.0 | 24.1 | 33.6 | 33.7 | 40.2 | 40.8 | 36.5 | 37.0 | **41.9** | 37.6 | 34.7 | 35.0 | 37.5 | 37.0 | 35.8 |
| WLM | E | 20.4 | 53.0 | 39.2 | 29.4 | 37.0 | 40.6 | 37.0 | 19.7 | 29.4 | 27.3 | 40.2 | 40.8 | 36.5 | 37.0 | 38.2 | 37.6 | 34.7 | 35.0 | 37.5 | 35.2 | 35.8 |
| EMTF D100 | E | 33.1 | 49.0 | 39.5 | 37.3 | 40.9 | 36.3 | 37.1 | 31.6 | 32.6 | 30.4 | 39.0 | 38.0 | 36.6 | 36.1 | 38.4 | 36.6 | 35.2 | 34.2 | 38.0 | 37.2 | 35.5 |
| EMTF | E | 30.3 | 49.0 | 40.6 | 34.5 | 40.9 | 36.3 | 37.1 | 27.6 | 29.6 | 26.7 | 39.0 | 38.0 | 36.6 | 36.1 | 33.3 | 36.6 | 35.2 | 34.2 | 38.0 | 35.8 | 35.5 |
| E | E | 30.3 | 49.0 | 43.3 | 34.5 | 40.9 | **44.3** | 37.1 | 27.6 | 29.6 | 26.7 | 39.0 | 38.0 | 36.6 | 36.1 | 33.3 | 36.6 | 35.2 | 34.2 | 38.0 | 36.5 | 35.5 |
| ETF | S | 27.4 | 45.4 | 40.9 | 31.9 | 43.7 | 40.7 | 38.5 | 20.8 | 26.9 | 21.0 | 33.2 | 38.2 | 36.7 | 36.6 | 23.4 | 36.1 | 34.6 | 34.1 | 40.6 | 33.9 | 35.5 |
| ETF D100 | S | 36.7 | 45.4 | 40.9 | 40.6 | 43.7 | 40.7 | 38.5 | 31.7 | 37.0 | 31.4 | 33.2 | 38.2 | 36.7 | 36.6 | 35.6 | 36.1 | 34.6 | 34.1 | 40.6 | 38.0 | 35.5 |
| ET | S | 27.4 | 45.4 | 33.9 | 40.6 | 43.4 | 40.5 | 38.4 | 20.8 | 27.0 | 20.9 | 33.1 | 37.4 | 36.7 | 36.6 | 23.3 | 36.1 | 34.6 | 34.0 | 40.4 | 33.9 | 35.5 |
| ET D100 | E | 36.8 | 45.4 | 40.7 | 40.6 | 43.4 | 40.5 | 38.4 | 31.7 | 37.1 | 31.4 | 33.1 | 37.4 | 36.7 | 36.6 | 35.6 | 36.1 | 34.6 | 34.0 | 40.4 | 37.9 | 35.5 |
| CLP D30 | E | 36.0 | 55.5 | 43.7 | 42.1 | 43.0 | 37.7 | 51.0 | 31.8 | 36.0 | 33.2 | 30.1 | 38.2 | 36.5 | 35.9 | 36.3 | 35.8 | 34.7 | 33.9 | 40.9 | 39.4 | 35.2 |
| EWMTF | E | 31.4 | 54.2 | 37.5 | 35.0 | 40.8 | 35.0 | 37.1 | 23.3 | 31.8 | 27.1 | 40.8 | 34.8 | 36.3 | 35.8 | 28.1 | 32.0 | 34.9 | 33.9 | 38.0 | 35.1 | 35.2 |
| EWMTF D100 | E | 33.7 | 54.2 | 37.5 | 37.3 | 40.8 | 35.0 | 37.1 | 27.2 | 31.8 | 30.7 | 40.8 | 34.8 | 36.3 | 35.8 | 33.4 | 32.0 | 34.9 | 33.9 | 38.0 | 36.3 | 35.2 |
| EW | E | 31.4 | 55.9 | 37.5 | 35.0 | 40.8 | 35.0 | 37.1 | 23.3 | 31.8 | 27.1 | 40.8 | 34.8 | 36.3 | 35.8 | 28.1 | 32.0 | 34.9 | 33.9 | 38.0 | 35.2 | 35.2 |
| ETF D30 | E | 36.5 | 54.7 | 41.2 | 39.7 | 43.0 | 41.3 | 38.8 | 32.0 | 35.7 | 32.5 | 32.2 | **43.3** | 36.1 | 36.2 | 34.8 | 35.6 | 34.3 | 34.3 | 39.9 | 38.7 | 35.2 |
| ET D30 | E | 36.5 | 54.4 | 41.0 | 39.6 | 42.8 | 41.2 | 38.7 | 32.1 | 35.9 | 32.5 | 32.1 | 43.2 | 36.1 | 36.2 | 34.8 | 35.6 | 34.3 | 34.2 | 39.8 | 38.7 | 35.2 |
| EF | E | 28.0 | 51.8 | 42.3 | 35.2 | 42.2 | 36.9 | 39.9 | 24.9 | 25.9 | 24.9 | 36.9 | 35.3 | 35.6 | 35.6 | 30.3 | 34.4 | 34.1 | 33.7 | 38.3 | 35.1 | 34.8 |
| EF D100 | E | 33.7 | 51.9 | 41.2 | 39.4 | 42.2 | 36.9 | 39.9 | 30.0 | 31.6 | 31.0 | 36.9 | 35.3 | 35.6 | 35.6 | 35.6 | 34.4 | 34.1 | 33.7 | 38.3 | 37.2 | 34.8 |
| EWT | S | 26.9 | 45.6 | 41.4 | 34.1 | 45.2 | 39.6 | 37.6 | 18.8 | 26.8 | 19.7 | 28.5 | 35.3 | 36.1 | 35.9 | 19.3 | 34.2 | 34.0 | 33.0 | 40.0 | 32.9 | 34.7 |
| BEATs | E | 24.0 | 58.6 | 31.2 | 17.9 | 29.0 | 32.0 | 46.9 | 9.1 | 34.8 | 30.0 | 27.6 | 28.8 | 36.2 | 35.5 | 33.2 | 29.9 | 34.5 | 32.7 | 38.8 | 31.4 | 34.7 |
| EWT D100 | S | 35.7 | 45.6 | 41.3 | 41.5 | 45.2 | 39.6 | 37.6 | 29.7 | 36.4 | 29.8 | 28.5 | 35.3 | 36.1 | 35.9 | 30.9 | 34.2 | 34.0 | 33.0 | 40.0 | 36.8 | 34.7 |
| EWTF D100 | S | 35.7 | 45.6 | 41.4 | 41.5 | 45.4 | 39.5 | 37.6 | 29.7 | 36.4 | 29.9 | 28.7 | 32.9 | 36.1 | 35.9 | 30.9 | 34.2 | 34.0 | 32.9 | 40.0 | 36.6 | 34.7 |
| EWTF | S | 26.9 | 45.6 | 41.4 | 34.0 | 45.4 | 39.5 | 37.6 | 18.8 | 26.8 | 19.8 | 28.7 | 32.9 | 36.1 | 35.9 | 19.2 | 34.2 | 34.0 | 32.9 | 40.0 | 32.7 | 34.7 |
| EWTF D30 | E | 35.8 | 57.5 | 43.1 | 41.0 | 44.1 | 39.6 | 38.5 | 28.5 | 35.3 | 31.7 | 28.7 | 33.6 | 35.9 | 35.2 | 31.3 | 38.8 | 34.1 | 33.2 | 39.3 | 37.5 | 34.6 |
| EWT D30 | E | 35.8 | 57.2 | 43.0 | 40.8 | 43.9 | 39.8 | 38.5 | 28.5 | 35.4 | 31.6 | 28.6 | 41.7 | 35.8 | 35.2 | 31.3 | 33.8 | 34.1 | 33.1 | 39.2 | 37.9 | 34.6 |
| EWF | E | 28.1 | 52.4 | 31.8 | 32.5 | 38.2 | 35.7 | 38.3 | 21.5 | 25.5 | 22.7 | 33.3 | 33.1 | 34.9 | 34.3 | 23.9 | 30.8 | 33.2 | 31.9 | 37.9 | 32.4 | 33.6 |
| EWF D100 | E | 32.6 | 52.4 | 35.9 | 36.4 | 38.5 | 35.7 | 38.3 | 26.4 | 30.9 | 28.7 | 33.3 | 33.1 | 34.9 | 34.3 | 30.7 | 30.8 | 33.2 | 31.9 | 37.9 | 34.8 | 33.6 |
| WLM | S | 10.1 | 46.6 | 29.3 | 19.8 | 30.8 | 35.2 | 34.9 | 8.7 | 21.3 | 16.1 | 37.0 | 24.2 | 34.1 | 34.8 | 28.2 | 31.7 | 31.9 | 31.9 | 33.4 | 27.1 | 33.2 |
| WLM D100 | S | 18.0 | 46.6 | 35.9 | 31.9 | 30.8 | 30.2 | 34.9 | 15.9 | 31.4 | 28.5 | 37.0 | 36.0 | 34.1 | 34.8 | 36.6 | 31.7 | 31.9 | 31.9 | 33.4 | 31.9 | 33.2 |
| EF D100 | S | 31.6 | 44.8 | 37.9 | 38.4 | 39.7 | 35.9 | 38.8 | 30.1 | 33.7 | 30.2 | 31.8 | 29.0 | 33.9 | 34.9 | 33.2 | 31.2 | 31.8 | 31.9 | 39.0 | 35.0 | 33.2 |
| EF | S | 18.7 | 44.0 | 28.7 | 28.0 | 39.7 | 35.9 | 38.8 | 16.4 | 21.2 | 15.7 | 31.8 | 29.0 | 33.9 | 34.9 | 19.0 | 31.2 | 31.8 | 31.9 | 39.0 | 29.1 | 33.2 |
| M | S | 35.9 | 65.8 | 50.2 | 44.2 | 44.2 | 29.9 | 30.5 | 17.5 | 22.9 | 25.7 | 29.1 | 25.7 | 34.0 | 33.4 | 27.0 | 29.6 | 32.8 | 31.6 | 34.0 | 34.2 | 33.0 |
| AC | S | 32.4 | 64.8 | 47.6 | 40.4 | 41.1 | 35.8 | 29.2 | 22.8 | 30.6 | 24.9 | 35.0 | 28.8 | 35.5 | 32.8 | 31.7 | 30.0 | 33.1 | 30.4 | 35.6 | 35.4 | 32.9 |
| M | C | 35.7 | 67.7 | **51.3** | 44.4 | 45.5 | 31.7 | 27.9 | 20.0 | 24.9 | 25.2 | 28.3 | 26.5 | 34.9 | 32.8 | 28.4 | 30.1 | 33.0 | 30.6 | 29.8 | 34.5 | 32.8 |
| BEATs | C | 22.6 | 59.5 | 30.4 | 17.1 | 28.2 | 30.9 | 44.5 | 7.9 | 32.5 | 28.4 | 26.6 | 26.9 | 34.1 | 33.4 | 31.9 | 28.3 | 32.5 | 30.7 | 37.9 | 30.2 | 32.7 |
| W2V D100 | E | 18.8 | 39.9 | 25.8 | 28.8 | 29.6 | 29.5 | 34.0 | 21.2 | 28.0 | 29.3 | 41.2 | 34.0 | 33.3 | 33.9 | 37.9 | 33.2 | 31.0 | 31.2 | 34.2 | 31.0 | 32.4 |
| W2V | E | 12.4 | 37.8 | 22.0 | 26.2 | 29.6 | 33.6 | 34.0 | 17.0 | 22.3 | 21.9 | 41.2 | 34.0 | 33.3 | 33.9 | 33.0 | 33.2 | 31.0 | 31.2 | 34.2 | 28.8 | 32.4 |
| EMTF D30 | E | 30.3 | 48.9 | 37.1 | 34.5 | 38.7 | 33.5 | 34.3 | 27.6 | 29.6 | 26.7 | 33.7 | 38.0 | 32.9 | 32.7 | 33.3 | 32.0 | 31.2 | 30.5 | 35.7 | 34.3 | 31.8 |
| EWMTF D30 | E | 31.4 | 54.5 | 35.4 | 35.0 | 38.7 | 32.6 | 34.1 | 23.3 | 29.1 | 27.1 | 35.6 | 34.8 | 32.7 | 32.4 | 28.1 | 27.0 | 31.1 | 30.2 | 35.8 | 33.5 | 31.6 |
| W2V D100 | S | 13.2 | 38.4 | 22.5 | 23.9 | 22.5 | 25.5 | 34.4 | 18.5 | 28.5 | 28.1 | 39.4 | 30.1 | 33.5 | 33.0 | 34.5 | 30.0 | 30.0 | 29.3 | 32.2 | 28.1 | 31.4 |
| W2V | S | 5.5 | 31.2 | 12.6 | 16.0 | 22.5 | 29.7 | 34.4 | 8.7 | 16.4 | 12.9 | 39.4 | 30.1 | 33.5 | 33.0 | 24.5 | 30.0 | 30.0 | 29.3 | 32.2 | 23.1 | 31.4 |
| EWF D100 | S | 30.8 | 44.7 | 36.2 | 36.0 | 36.7 | 33.9 | 38.7 | 27.8 | 32.0 | 27.4 | 27.7 | 23.0 | 31.9 | 32.8 | 28.9 | 27.5 | 29.5 | 29.7 | 38.7 | 32.7 | 31.0 |
| EWF | S | 18.6 | 44.7 | 36.2 | 24.4 | 36.7 | 33.9 | 38.7 | 12.8 | 20.1 | 13.5 | 27.7 | 23.0 | 31.9 | 32.8 | 12.4 | 27.5 | 29.5 | 29.7 | 38.7 | 27.3 | 31.0 |
| CC | E | 23.0 | 32.7 | 23.4 | 13.8 | 18.9 | 31.1 | 39.1 | 10.7 | 26.2 | 30.3 | 40.6 | 25.3 | 29.6 | 33.3 | 28.8 | 26.0 | 28.4 | 31.5 | 35.0 | 27.0 | 30.7 |
| WLM D30 | E | 20.4 | 53.0 | 39.2 | 29.4 | 34.4 | 30.4 | 31.2 | 19.7 | 29.4 | 27.3 | 31.0 | 40.8 | 31.1 | 32.6 | 38.2 | 33.9 | 28.7 | 29.7 | 32.1 | 32.7 | 30.5 |
| ETF | C | 25.0 | 55.6 | 37.8 | 30.5 | 39.9 | 36.7 | 33.4 | 18.3 | 23.8 | 18.8 | 26.5 | 36.7 | 31.4 | 31.2 | 22.1 | 31.5 | 28.9 | 28.2 | 35.4 | 31.5 | 30.0 |
| ETF D100 | C | 30.1 | 58.5 | 37.8 | 35.7 | 39.9 | 36.7 | 33.4 | 25.0 | 30.3 | 25.3 | 26.5 | 36.7 | 31.4 | 31.2 | 31.4 | 31.5 | 28.9 | 28.2 | 35.4 | 34.3 | 30.0 |
| ET | C | 25.1 | 58.0 | 37.4 | 30.4 | 39.6 | 36.4 | 33.3 | 18.4 | 24.1 | 18.7 | 26.3 | 36.5 | 31.4 | 31.2 | 22.1 | 31.5 | 29.0 | 28.2 | 35.4 | 31.5 | 29.9 |
| ET D100 | C | 30.1 | 58.0 | 37.4 | 35.6 | 39.6 | 36.4 | 33.3 | 25.1 | 30.5 | 25.2 | 26.3 | 36.5 | 31.4 | 31.2 | 31.4 | 31.5 | 29.0 | 28.2 | 35.4 | 34.2 | 29.9 |
| EF D30 | E | 28.0 | 52.4 | 37.1 | 35.2 | 38.3 | 31.3 | 34.5 | 24.9 | 25.9 | 24.9 | 30.6 | 35.3 | 30.2 | 30.3 | 30.3 | 29.1 | 28.3 | 28.1 | 34.1 | 32.8 | 29.2 |
| CLP | C | 23.8 | 61.9 | 38.1 | 34.4 | 39.2 | 30.3 | **52.0** | 18.3 | 24.5 | 20.0 | 20.5 | 27.3 | 30.4 | 29.7 | 24.3 | 29.0 | 27.9 | 26.8 | 36.5 | 32.0 | 28.7 |
| CLP D100 | C | 28.5 | 60.5 | 40.6 | 37.8 | 39.2 | 30.4 | **52.0** | 23.1 | 29.3 | 25.5 | 20.5 | 27.3 | 30.4 | 29.7 | 29.4 | 29.0 | 27.9 | 26.8 | 36.5 | 34.0 | 28.7 |
| EWTF | C | 24.1 | 60.2 | 40.0 | 32.7 | 42.0 | 33.8 | 33.1 | 14.0 | 23.3 | 17.8 | 21.4 | 27.8 | 30.5 | 29.3 | 17.3 | 28.5 | 28.0 | 26.3 | 34.1 | 30.0 | 28.5 |
| EWTF D100 | C | 28.5 | 62.5 | 39.8 | 36.8 | 42.0 | 33.8 | 33.1 | 20.0 | 29.2 | 24.0 | 21.4 | 27.8 | 30.5 | 29.3 | 25.9 | 28.5 | 28.0 | 26.3 | 34.1 | 32.5 | 28.5 |
| EWT | C | 24.1 | 59.7 | 40.0 | 32.5 | 41.7 | 33.7 | 33.1 | 14.1 | 23.4 | 17.6 | 21.2 | 34.3 | 30.4 | 29.2 | 17.3 | 28.5 | 28.0 | 26.2 | 33.9 | 30.3 | 28.5 |
| EWT D100 | C | 28.4 | 61.8 | 39.5 | 36.6 | 41.7 | 33.7 | 33.1 | 20.1 | 29.3 | 23.8 | 21.2 | 34.3 | 30.4 | 29.2 | 25.9 | 28.5 | 28.0 | 26.2 | 33.9 | 32.4 | 28.5 |
| MAE | E | 20.7 | 45.4 | 28.2 | 24.3 | 28.6 | 36.2 | 37.7 | 16.2 | 24.9 | 22.0 | 32.7 | 37.0 | 29.4 | 29.3 | 24.7 | 29.2 | 27.5 | 26.7 | 36.2 | 29.6 | 28.2 |
| MAE D100 | E | 25.8 | 44.0 | 28.2 | 29.4 | 28.6 | 27.3 | 37.7 | 20.7 | 30.3 | 28.9 | 32.7 | 37.0 | 29.4 | 29.3 | 30.3 | 29.2 | 27.5 | 26.7 | 36.2 | 31.1 | 28.2 |
| EWF D30 | E | 28.1 | 52.4 | 31.8 | 32.5 | 33.9 | 30.5 | 32.6 | 21.5 | 25.5 | 22.7 | 27.1 | 24.3 | 29.3 | 29.0 | 23.9 | 24.6 | 27.4 | 26.0 | 34.0 | 29.7 | 27.9 |
| MAE D100 | S | 23.1 | 43.3 | 24.7 | 27.9 | 22.9 | 26.0 | 37.1 | 17.6 | 29.0 | 25.6 | 29.7 | - | 28.1 | 28.8 | 26.8 | 25.3 | 25.2 | 25.2 | 35.0 | 28.1 | 26.8 |

*Table 23.* (Continued) GSR Results Across Subsets and Distances (↑ better)

| Method | Dist | BC | BS1 | BS2 | BS3 | BS4 | BS5 | ES1 | HP | HS1 | HS2 | HU1 | HU2 | HU3 | HU4 | HW1 | HW2 | HW3 | HW4 | OC1 | Avg | Avg (Blind) |
|---|---|---|---|---|---|---|---|---|---|---|---|---|---|---|---|---|---|---|---|---|---|---|
| MAE | S | 12.2 | 43.3 | 24.7 | 15.7 | 22.9 | 24.0 | 37.1 | 8.4 | 17.1 | 12.7 | 29.7 | - | 28.1 | 28.8 | 15.1 | 25.3 | 25.2 | 25.2 | 35.0 | 23.1 | 26.8 |
| VC | E | 30.7 | 64.0 | 47.5 | 42.6 | 46.1 | 38.2 | 28.4 | 20.9 | 27.2 | 24.3 | 27.2 | 26.2 | 27.9 | 26.8 | 28.3 | 28.8 | 25.3 | 23.5 | 33.5 | 34.3 | 25.9 |
| W2V D30 | E | 12.4 | 37.8 | 22.0 | 26.2 | 24.6 | 25.0 | 29.3 | 17.0 | 22.3 | 21.9 | 32.9 | 34.0 | 26.3 | 28.2 | 33.0 | 28.1 | 24.0 | 24.7 | 28.4 | 26.3 | 25.8 |
| CLP D30 | S | 27.3 | 45.6 | 37.3 | 34.2 | 37.2 | 30.9 | 50.3 | 21.2 | 28.7 | 21.6 | 19.2 | 27.8 | 27.0 | 27.0 | 25.5 | 25.2 | 24.0 | 23.7 | 34.0 | 31.1 | 25.5 |
| WLM D100 | C | 13.7 | 55.2 | 37.4 | 23.3 | 26.8 | 25.9 | 26.7 | 11.2 | 24.4 | 22.3 | 30.4 | 37.2 | 26.7 | 27.4 | 34.1 | 28.4 | 23.2 | 23.8 | 29.0 | 28.4 | 25.3 |
| WLM | C | 7.9 | 55.3 | 32.1 | 16.4 | 26.8 | 34.3 | 26.7 | 7.1 | 17.5 | 13.7 | 30.4 | 37.2 | 26.7 | 27.4 | 27.0 | 28.4 | 23.2 | 23.8 | 29.0 | 26.0 | 25.3 |
| AC | E | 33.8 | 55.6 | 46.7 | 41.6 | 44.1 | 35.8 | 25.6 | 28.6 | 24.4 | 22.2 | 36.0 | 30.1 | 27.4 | 23.6 | 34.7 | 33.3 | 26.2 | 22.1 | 32.1 | 35.0 | 24.8 |
| ET D30 | S | 27.4 | 45.1 | 33.9 | 32.1 | 35.8 | 34.0 | 29.9 | 20.8 | 27.0 | 20.9 | 20.1 | 37.4 | 26.0 | 26.7 | 23.3 | 24.5 | 23.3 | 23.2 | 32.9 | 29.7 | 24.8 |
| ETF D30 | S | 27.4 | 45.1 | 34.4 | 31.9 | 36.1 | 34.7 | 29.8 | 20.8 | 26.9 | 21.0 | 20.3 | 38.2 | 26.1 | 26.5 | 23.4 | 24.4 | 23.3 | 23.3 | 32.7 | 29.8 | 24.8 |
| EMTF D30 | S | 24.7 | 42.5 | 32.2 | 28.9 | 32.7 | 30.7 | 29.1 | 19.7 | 26.2 | 20.2 | 28.6 | 33.6 | 26.2 | 26.4 | 25.3 | 23.8 | 22.7 | 22.7 | 31.0 | 28.6 | 24.5 |
| EWTF D30 | S | 26.9 | 46.7 | 36.8 | 34.0 | 38.6 | 32.6 | 29.4 | 18.8 | 26.8 | 19.8 | 16.4 | 21.7 | 26.0 | 26.0 | 19.2 | 22.8 | 23.1 | 22.5 | 32.3 | 28.2 | 24.4 |
| EWT D30 | S | 26.9 | 46.3 | 36.6 | 34.1 | 38.5 | 33.0 | 29.5 | 18.8 | 26.8 | 19.7 | 16.3 | 35.3 | 25.8 | 26.0 | 19.3 | 22.8 | 23.1 | 22.4 | 32.0 | 29.1 | 24.3 |
| EWMTF D30 | S | 24.3 | 46.9 | 32.0 | 29.8 | 33.1 | 30.2 | 29.1 | 18.5 | 26.3 | 20.3 | 28.2 | 27.7 | 25.3 | 25.7 | 23.6 | 21.3 | 21.8 | 22.1 | 31.3 | 28.2 | 23.7 |
| EMTF | C | 16.8 | 48.4 | 30.1 | 22.1 | 32.5 | 25.8 | 26.6 | 13.1 | 16.1 | 12.0 | 28.8 | 28.1 | 25.4 | 24.5 | 20.0 | 25.2 | 23.1 | 21.8 | 27.8 | 24.9 | 23.7 |
| EMTF D100 | C | 20.9 | 48.7 | 30.1 | 26.6 | 32.5 | 25.8 | 26.6 | 18.0 | 20.0 | 16.3 | 28.8 | 28.1 | 25.4 | 24.5 | 28.0 | 25.2 | 23.1 | 21.8 | 27.8 | 26.9 | 23.7 |
| E | C | 16.8 | 48.4 | - | 22.1 | 32.5 | 38.5 | 26.6 | 13.1 | 16.1 | 12.0 | 28.8 | 28.1 | 25.4 | 24.5 | 20.0 | 25.2 | 23.1 | 21.8 | 27.8 | 25.4 | 23.7 |
| EW | C | 18.0 | 61.2 | 40.0 | 23.1 | 32.4 | 23.3 | 27.0 | 8.6 | 15.9 | 12.6 | 31.4 | 22.7 | 25.1 | 24.4 | 13.1 | 18.2 | 22.9 | 21.8 | 27.7 | 25.0 | 23.5 |
| EWMTF | C | 18.0 | 58.3 | 27.4 | 23.1 | 32.4 | 23.3 | 27.0 | 8.6 | 15.9 | 12.6 | 31.4 | 22.7 | 25.1 | 24.4 | 13.1 | 18.2 | 22.9 | 21.8 | 27.7 | 24.0 | 23.5 |
| EWMTF D100 | C | 21.4 | 58.0 | 27.6 | 26.7 | 32.4 | 23.3 | 27.0 | 12.4 | 19.4 | 16.9 | 31.4 | 22.7 | 25.1 | 24.4 | 19.9 | 18.2 | 22.9 | 21.8 | 27.7 | 25.7 | 23.5 |
| CLP D30 | C | 23.8 | 61.9 | 38.1 | 34.4 | 35.6 | 26.5 | 51.2 | 18.3 | 24.5 | 20.0 | 15.3 | 27.3 | 24.9 | 24.3 | 24.3 | 23.8 | 22.0 | 21.1 | 32.6 | 30.5 | 23.1 |
| ETF D30 | C | 25.0 | 60.0 | 33.6 | 30.5 | 35.6 | 32.4 | 28.3 | 18.3 | 23.8 | 18.8 | 17.9 | 36.7 | 24.2 | 24.6 | 22.1 | 23.2 | 21.4 | 21.3 | 30.5 | 29.1 | 22.9 |
| ET D30 | C | 25.1 | 59.4 | 33.2 | 30.4 | 35.3 | 32.1 | 28.2 | 18.4 | 24.1 | 18.7 | 17.8 | 36.5 | 24.2 | 24.6 | 22.1 | 23.2 | 21.4 | 21.3 | 30.5 | 29.0 | 22.9 |
| MAE D30 | E | 20.7 | 42.0 | 23.0 | 24.3 | 23.5 | 20.6 | 32.5 | 16.2 | 24.9 | 22.0 | 23.3 | 37.0 | 23.9 | 24.1 | 24.7 | 23.8 | 21.9 | 21.4 | 31.5 | 26.0 | 22.8 |
| EAT | E | 17.9 | 53.9 | 31.6 | 28.5 | 29.2 | 25.6 | 25.7 | 11.4 | 18.3 | 21.9 | 24.7 | 16.8 | 23.3 | 23.8 | 19.7 | 17.0 | 21.5 | 21.8 | 28.4 | 24.7 | 22.6 |
| EF D100 | C | 21.2 | 54.5 | 33.7 | 30.1 | 34.7 | 25.9 | 31.6 | 16.4 | 18.7 | 17.2 | 25.5 | 23.0 | 23.8 | 23.8 | 23.7 | 22.2 | 21.5 | 21.1 | 28.5 | 27.1 | 22.5 |
| EF | C | 13.5 | 54.8 | 33.7 | 23.4 | 34.7 | 25.9 | 31.6 | 10.7 | 12.2 | 10.0 | 25.5 | 23.0 | 23.8 | 23.8 | 16.1 | 22.2 | 21.5 | 21.1 | 28.5 | 24.4 | 22.5 |
| EWTF D30 | C | 24.1 | 64.4 | 37.3 | 32.7 | 37.9 | 29.8 | 27.9 | 14.0 | 23.3 | 17.8 | 13.8 | 20.2 | 23.8 | 23.0 | 17.3 | 20.6 | 21.1 | 19.9 | 29.5 | 27.4 | 21.9 |
| EWT D30 | C | 24.1 | 63.6 | 37.0 | 32.5 | 37.6 | 30.0 | 27.9 | 14.1 | 23.4 | 17.6 | 13.7 | 34.3 | 23.7 | 22.9 | 17.3 | 20.5 | 21.1 | 19.9 | 29.3 | 28.2 | 21.9 |
| EF D30 | S | 18.7 | 44.0 | 28.7 | 28.0 | 31.1 | 23.7 | 26.9 | 16.4 | 21.2 | 15.7 | 18.7 | 29.0 | 22.1 | 23.4 | 19.0 | 18.2 | 19.7 | 19.7 | 27.9 | 24.5 | 21.2 |
| WLM D30 | S | 10.1 | 46.6 | 29.3 | 19.8 | 24.4 | 22.3 | 23.2 | 8.7 | 21.3 | 16.1 | 19.1 | 24.2 | 22.0 | 23.7 | 28.2 | 23.7 | 19.1 | 19.8 | 22.1 | 22.6 | 21.2 |
| EWF | C | 14.1 | 52.9 | 19.7 | 20.0 | 28.0 | 24.0 | 29.9 | 7.4 | 11.4 | 8.1 | 19.8 | 19.7 | 22.8 | 22.3 | 9.1 | 17.1 | 20.2 | 18.7 | 27.8 | 20.6 | 21.0 |
| EWF D100 | C | 19.8 | 54.4 | 25.1 | 25.7 | 28.0 | 24.0 | 29.9 | 12.1 | 17.5 | 14.3 | 19.8 | 19.7 | 22.8 | 22.3 | 16.6 | 17.1 | 20.2 | 18.7 | 27.8 | 23.4 | 21.0 |
| W2V D100 | C | 6.4 | 33.1 | 12.1 | 15.2 | 16.0 | 17.7 | 22.7 | 8.2 | 17.0 | 16.6 | 32.2 | 28.5 | 22.0 | 23.1 | 30.0 | 24.1 | 18.9 | 19.1 | 24.6 | 20.3 | 20.8 |
| W2V | C | 2.7 | 29.3 | 8.5 | 12.1 | 16.0 | 23.8 | 22.7 | 4.9 | 10.3 | 8.6 | 32.2 | 28.5 | 22.0 | 23.1 | 21.6 | 24.1 | 18.9 | 19.1 | 24.6 | 18.0 | 20.8 |
| EWF D30 | S | 18.6 | 44.4 | 25.2 | 24.4 | 24.8 | 23.4 | 26.2 | 12.8 | 20.1 | 13.5 | 16.1 | 13.1 | 20.2 | 20.7 | 12.4 | 13.2 | 17.4 | 16.7 | 27.7 | 21.1 | 18.8 |
| EWMTF D30 | C | 18.0 | 58.3 | 24.5 | 23.1 | 28.8 | 19.8 | 22.3 | 8.6 | 15.9 | 12.6 | 22.9 | 22.7 | 19.5 | 19.4 | 13.1 | 12.2 | 17.3 | 16.7 | 23.9 | 21.8 | 18.2 |
| EMTF D30 | C | 16.8 | 48.4 | 26.2 | 22.1 | 28.5 | 21.7 | 22.2 | 13.1 | 16.1 | 12.0 | 20.6 | 28.1 | 19.7 | 19.4 | 20.0 | 18.3 | 17.3 | 16.4 | 23.9 | 22.5 | 18.2 |
| W2V D30 | S | 5.5 | 31.2 | 12.6 | 16.0 | 13.5 | 15.8 | 22.1 | 8.7 | 16.4 | 12.9 | 22.2 | 30.1 | 19.2 | 20.1 | 24.5 | 19.8 | 15.8 | 15.6 | 20.5 | 18.1 | 17.7 |
| CC | C | 2.6 | 38.0 | 18.3 | 9.7 | 4.4 | 2.9 | 42.7 | 1.5 | 7.4 | 17.0 | 28.9 | 2.0 | 16.4 | 19.2 | 11.5 | 11.2 | 17.6 | 17.3 | 3.5 | 13.4 | 17.6 |
| WLM D30 | C | 7.9 | 55.3 | 32.1 | 16.4 | 22.8 | 20.0 | 18.1 | 7.1 | 17.5 | 13.7 | 16.4 | 37.2 | 18.1 | 19.9 | 27.0 | 21.9 | 14.8 | 16.1 | 19.7 | 22.2 | 17.2 |
| EF D30 | C | 13.5 | 54.8 | 27.0 | 23.4 | 27.7 | 17.5 | 22.4 | 10.7 | 12.2 | 10.0 | 16.1 | 23.0 | 16.0 | 16.5 | 16.1 | 15.0 | 13.9 | 13.8 | 21.9 | 20.7 | 15.1 |
| MAE D30 | S | 12.2 | 38.4 | 14.1 | 15.7 | 13.2 | 12.6 | 25.8 | 8.4 | 17.1 | 12.7 | 13.0 | - | 16.1 | 17.0 | 15.1 | 14.0 | 13.1 | 13.6 | 23.8 | 16.9 | 14.9 |
| MAE | C | 7.3 | 38.9 | 9.6 | 10.1 | 14.9 | 24.2 | 29.4 | 4.5 | 10.7 | 7.9 | 19.6 | 26.3 | 16.1 | 15.9 | 10.4 | 15.5 | 13.7 | 12.7 | 25.9 | 17.0 | 14.6 |
| MAE D100 | C | 12.2 | 38.9 | 14.8 | 15.8 | 14.9 | 13.6 | 29.4 | 7.7 | 16.9 | 14.9 | 19.6 | 26.3 | 16.1 | 15.9 | 16.6 | 15.5 | 13.7 | 12.7 | 25.9 | 18.9 | 14.6 |
| EWF D30 | C | 14.1 | 54.2 | 19.7 | 20.0 | 21.4 | 16.6 | 20.6 | 7.4 | 11.4 | 8.1 | 12.1 | 9.9 | 15.1 | 15.2 | 9.1 | 10.0 | 12.8 | 11.6 | 21.6 | 17.1 | 13.7 |
| W2V D30 | C | 2.7 | 29.3 | 8.5 | 12.1 | 10.5 | 12.0 | 15.7 | 4.9 | 10.3 | 8.6 | 19.0 | 28.5 | 12.7 | 14.7 | 21.6 | 16.3 | 10.4 | 10.9 | 15.8 | 14.4 | 12.2 |
| AC | C | 21.2 | 59.1 | 42.4 | 33.5 | 37.2 | 24.0 | 15.6 | 13.8 | 9.9 | 9.5 | 25.5 | 15.8 | 14.6 | 10.3 | 22.0 | 20.0 | 12.6 | 8.9 | 19.4 | 24.6 | 11.6 |
| EAT | S | 8.4 | 53.6 | 25.2 | 20.5 | 21.5 | 15.8 | 15.5 | 3.1 | 8.1 | 11.0 | 14.7 | 7.6 | 12.4 | 12.5 | 10.4 | 8.1 | 10.4 | 10.3 | 18.7 | 16.2 | 11.4 |
| EAT | C | 7.6 | 53.9 | 23.6 | 17.0 | 18.9 | 15.1 | 14.8 | 2.5 | 7.6 | 10.4 | 13.4 | 7.2 | 11.8 | 11.9 | 10.0 | 7.8 | 9.8 | 9.7 | 18.3 | 15.2 | 10.8 |
| VC | C | 15.8 | **75.8** | 45.3 | 35.4 | 40.9 | 27.0 | 14.1 | 6.9 | 12.3 | 9.2 | 11.5 | 10.4 | 12.6 | 11.4 | 13.3 | 13.3 | 9.8 | 8.2 | 19.3 | 23.4 | 10.5 |
| MAE D30 | C | 7.3 | 35.6 | 9.6 | 10.1 | 9.3 | 7.1 | 21.3 | 4.5 | 10.7 | 7.9 | 9.0 | 26.3 | 10.0 | 10.0 | 10.4 | 9.7 | 8.1 | 7.6 | 18.3 | 13.1 | 8.9 |
| VC | S | 10.4 | 62.0 | 26.1 | 21.3 | 28.1 | 19.5 | 8.8 | 3.7 | 10.2 | 5.9 | 5.7 | 5.1 | 8.7 | 8.1 | 6.6 | 6.1 | 5.7 | 4.9 | 15.5 | 15.7 | 6.9 |

*Table 24.* CSR Results Across Subsets and Distances (↑ better)

| Method | Dist | BC | BS1 | BS2 | BS3 | BS4 | BS5 | ES1 | HP | HS1 | HS2 | HU1 | HU2 | HU3 | HU4 | HW1 | HW2 | HW3 | HW4 | OC1 | Avg | Avg (Blind) |
|---|---|---|---|---|---|---|---|---|---|---|---|---|---|---|---|---|---|---|---|---|---|---|
| ETF | E | - | 42.7 | 33.4 | 28.1 | 32.6 | 28.5 | 34.8 | 24.9 | - | 26.6 | 33.2 | - | **36.6** | **35.8** | 29.6 | **37.4** | **35.1** | **34.3** | 36.8 | 33.1 | **35.4** |
| ETF D100 | E | - | 42.1 | 31.8 | 31.5 | 32.6 | 28.5 | 34.8 | **30.0** | - | **31.9** | 33.2 | - | **36.6** | **35.8** | **36.3** | **37.4** | **35.1** | **34.3** | 36.8 | **34.3** | **35.4** |
| ET D100 | E | - | 41.9 | 31.6 | 31.4 | 32.4 | 28.3 | 34.6 | **30.0** | - | 31.8 | 33.1 | - | 36.5 | **35.8** | **36.3** | **37.4** | **35.1** | **34.3** | 36.8 | 34.2 | **35.4** |
| ET | E | - | 42.7 | 31.6 | 28.0 | 32.4 | 28.3 | 34.6 | 25.0 | - | 26.6 | 33.1 | - | 36.5 | **35.8** | 29.6 | **37.4** | **35.1** | **34.3** | 36.8 | 33.0 | **35.4** |
| EWTF D100 | E | - | 40.5 | 31.4 | 30.5 | 33.5 | 26.8 | 35.1 | 26.1 | - | - | 30.2 | - | 36.0 | 34.8 | 33.4 | 35.6 | 34.6 | 33.0 | 36.2 | 33.2 | 34.6 |
| EWTF | E | - | 40.3 | 29.1 | 27.6 | 33.5 | 26.8 | 35.1 | 21.3 | - | 26.1 | 30.2 | - | 36.0 | 34.8 | 26.7 | 35.6 | 34.6 | 33.0 | 36.2 | 31.7 | 34.6 |
| EWT | E | - | 40.3 | 28.9 | 27.4 | 33.3 | 26.6 | 35.0 | 21.3 | - | 26.0 | 30.1 | - | 36.0 | 34.8 | 26.7 | 35.6 | 34.5 | 33.0 | 36.1 | 31.6 | 34.6 |
| EWT D100 | E | - | 40.2 | 31.2 | 30.3 | 33.3 | 26.6 | 35.0 | 26.1 | - | 31.2 | 30.1 | - | 36.0 | 34.8 | 33.4 | 35.6 | 34.5 | 33.0 | 36.1 | 33.0 | 34.6 |
| CLP D100 | E | - | 41.7 | 32.0 | 31.7 | 31.9 | 26.2 | **42.4** | 27.7 | - | - | 29.0 | - | 35.5 | 34.7 | 34.0 | 34.8 | 34.1 | 32.9 | **37.1** | 33.7 | 34.3 |
| CLP | E | - | 41.8 | 32.4 | 28.8 | 31.9 | 26.8 | **42.4** | 24.2 | - | 26.6 | 29.0 | - | 35.5 | 34.7 | 30.6 | 34.8 | 34.1 | 32.9 | **37.1** | 32.7 | 34.3 |
| CC | E | 24.0 | 33.1 | 27.3 | 20.5 | 23.6 | 29.6 | 41.1 | 16.3 | 24.4 | 31.4 | **41.1** | 26.5 | 33.3 | 34.9 | 32.9 | 31.8 | 32.2 | 33.3 | 34.1 | 30.1 | 33.4 |
| M | E | - | **48.4** | **40.1** | **36.8** | **35.5** | 24.5 | 25.8 | 17.7 | - | - | 32.2 | - | 34.5 | 30.9 | 30.2 | 32.4 | 33.1 | - | 31.9 | 32.4 | 32.9 |
| CLP D100 | S | **30.2** | 38.0 | 32.4 | 31.4 | 31.8 | 28.3 | 39.6 | 26.4 | **32.0** | 26.7 | 26.2 | 20.3 | 33.3 | 33.6 | 30.4 | 31.8 | 31.2 | 30.7 | 36.3 | 31.1 | 32.2 |
| CLP | S | 18.3 | 33.6 | 32.4 | 20.4 | 31.8 | 13.2 | 39.6 | 14.5 | 20.5 | 15.3 | 26.2 | 20.3 | 33.3 | 33.6 | 19.0 | 31.8 | 31.2 | 30.7 | 36.3 | 26.4 | 32.2 |
| EMTF | S | 16.3 | 30.8 | 31.1 | 17.3 | 30.3 | 28.7 | 31.8 | 13.6 | 18.7 | 14.4 | 39.5 | 25.4 | 33.5 | 33.3 | 18.8 | 33.3 | 31.0 | 30.8 | 35.1 | 27.0 | 32.2 |
| EMTF D100 | S | 29.1 | 37.5 | 31.1 | 30.3 | 30.3 | 28.7 | 31.8 | 25.9 | 30.8 | 27.2 | 39.5 | 25.4 | 33.5 | 33.3 | 34.2 | 33.3 | 31.0 | 30.8 | 35.1 | 31.5 | 32.2 |
| EWMTF | S | 16.1 | 38.1 | 13.0 | 17.7 | 30.4 | 29.2 | 31.2 | 12.7 | 18.7 | 14.2 | 39.0 | 20.4 | 33.0 | 33.2 | 17.5 | 32.2 | 30.6 | 30.1 | 34.9 | 25.9 | 31.7 |
| EWMTF D100 | S | 28.4 | 38.1 | 30.8 | 29.8 | 30.4 | 29.2 | 31.2 | 26.3 | - | - | 39.0 | 20.4 | 33.0 | 33.2 | 33.7 | 32.2 | 30.6 | 30.1 | 34.9 | 30.9 | 31.7 |
| ETF D100 | S | 30.0 | 38.0 | 31.7 | 30.8 | 30.5 | 29.1 | 31.3 | 26.3 | 31.3 | 26.7 | 28.1 | 33.4 | 32.8 | 33.0 | 30.9 | 32.4 | 30.9 | 30.2 | 35.5 | 31.2 | 31.7 |
| ETF | S | 18.7 | 38.0 | 31.7 | 19.1 | 30.5 | 29.1 | 31.3 | 14.6 | 19.6 | 15.3 | 28.1 | 33.4 | 32.8 | 33.0 | 17.8 | 32.4 | 30.9 | 30.2 | 35.5 | 27.5 | 31.7 |
| ET | S | 18.7 | 38.0 | 21.0 | 30.8 | 30.4 | 29.0 | 31.3 | 14.6 | 19.6 | 15.3 | 28.0 | 32.6 | 32.7 | 33.0 | 17.8 | 32.4 | 30.8 | 30.2 | 35.3 | 27.5 | 31.7 |
| ET D100 | S | 30.1 | 38.0 | 31.6 | 30.8 | 30.4 | 29.0 | 31.3 | 26.4 | 31.3 | 26.7 | 28.0 | 32.6 | 32.7 | 33.0 | 30.9 | 32.4 | 30.8 | 30.2 | 35.3 | 31.1 | 31.7 |
| WLM D100 | E | - | 41.6 | 28.8 | 24.1 | 23.2 | 22.4 | 31.3 | 17.2 | - | - | 34.4 | - | 32.0 | 32.0 | 33.1 | 32.0 | 30.2 | 30.1 | 30.2 | 29.5 | 31.1 |
| WLM | E | - | 39.5 | 23.2 | 18.5 | 23.2 | 28.1 | 31.3 | 13.5 | - | - | 34.4 | - | 32.0 | 32.0 | 28.4 | 32.0 | 30.2 | 30.1 | 30.2 | 28.4 | 31.1 |
| EWT D100 | S | 28.7 | 37.8 | 31.7 | 30.7 | 31.9 | 28.6 | 30.5 | 24.7 | 30.8 | 25.4 | 24.1 | 30.7 | 32.2 | 32.2 | 26.9 | 30.6 | 30.2 | 29.2 | 34.7 | 30.1 | 31.0 |
| EWT | S | 17.8 | 37.8 | 31.7 | 19.7 | 31.9 | 28.6 | 30.5 | 13.2 | 19.6 | 14.4 | 24.1 | 30.7 | 32.2 | 32.2 | 14.8 | 30.6 | 30.2 | 29.2 | 34.7 | 26.5 | 31.0 |
| EWTF D100 | S | 28.7 | 37.9 | 31.7 | 30.8 | 32.0 | 28.6 | 30.6 | 24.7 | 30.8 | 25.5 | 24.3 | 28.4 | 32.2 | 32.2 | 26.9 | 30.6 | 30.2 | 29.2 | 34.8 | 30.0 | 30.9 |
| EWTF | S | 17.8 | 37.9 | 31.7 | 19.8 | 32.0 | 28.6 | 30.6 | 13.2 | 19.5 | 14.5 | 24.3 | 28.4 | 32.2 | 32.2 | 14.8 | 30.6 | 30.2 | 29.2 | 34.8 | 26.4 | 30.9 |
| ET D30 | E | - | 41.5 | 28.5 | 28.0 | 29.5 | 23.9 | 30.7 | 25.0 | - | 26.6 | 27.0 | - | 31.4 | 31.5 | 29.6 | 31.8 | 29.6 | 29.3 | 33.4 | 29.8 | 30.5 |
| ETF D30 | E | - | 41.7 | 28.6 | 28.1 | 29.6 | 24.1 | 30.8 | 24.9 | - | 26.6 | 27.1 | - | 31.4 | 31.5 | 29.6 | 31.8 | 29.6 | 29.2 | 33.4 | 29.9 | 30.4 |
| CLP D30 | E | - | 41.5 | 29.4 | 28.8 | 29.0 | 23.1 | 40.8 | 24.2 | - | 26.6 | 24.9 | - | 31.8 | 31.0 | 30.6 | 31.4 | 30.0 | 28.9 | 34.4 | 30.4 | 30.4 |
| EWTF D30 | E | - | 40.3 | 29.1 | 27.6 | 30.4 | 23.6 | 31.0 | 21.3 | - | 26.1 | 24.1 | - | 31.2 | 30.5 | 26.7 | 30.1 | 29.5 | 28.2 | 33.1 | 28.9 | 29.9 |
| EWT D30 | E | - | 39.9 | 28.9 | 27.4 | 30.3 | 23.5 | 31.0 | 21.3 | - | 26.0 | 24.1 | - | 31.2 | 30.5 | 26.7 | 30.1 | 29.5 | 28.3 | 33.0 | 28.8 | 29.9 |
| WLM D100 | S | 13.5 | 37.5 | 26.4 | 22.7 | 17.8 | 20.7 | 29.6 | 12.8 | 26.6 | 24.5 | 29.8 | 25.7 | 30.9 | 31.3 | 25.2 | 25.6 | 28.7 | 28.4 | 28.5 | 25.6 | 29.8 |
| WLM | S | 6.3 | 33.0 | 16.8 | 10.2 | 17.8 | 20.0 | 29.6 | 5.7 | 15.8 | 11.8 | 29.8 | 37.9 | 30.9 | 31.3 | 16.5 | 25.6 | 28.7 | 28.4 | 28.5 | 22.3 | 29.8 |
| EF D100 | S | 25.6 | 37.8 | 29.2 | 29.6 | 29.0 | 27.2 | 33.7 | 24.8 | 28.7 | 26.0 | 27.1 | 20.1 | 30.7 | 31.4 | 28.5 | 28.0 | 28.6 | 28.3 | 34.0 | 28.9 | 29.8 |
| EF | S | 12.2 | 32.1 | 16.8 | 16.9 | 29.0 | 27.2 | 33.7 | 11.4 | 15.4 | 11.5 | 27.1 | 20.1 | 30.7 | 31.4 | 14.1 | 28.0 | 28.6 | 28.3 | 34.0 | 23.6 | 29.8 |
| EMTF D100 | E | - | 35.7 | 25.6 | 24.5 | 26.9 | 18.6 | 25.2 | 22.0 | - | - | 30.4 | - | 30.5 | 29.6 | 31.2 | 31.6 | 29.2 | 27.3 | 29.6 | 27.8 | 29.1 |
| E | E | - | 35.7 | 31.6 | 21.6 | 26.9 | **34.7** | 25.2 | 18.2 | - | 17.0 | 30.4 | - | 30.5 | 29.6 | 26.0 | 31.6 | 29.2 | 27.3 | 29.6 | 27.8 | 29.1 |
| EMTF | E | - | 35.7 | 27.0 | 21.6 | 26.9 | 18.6 | 25.2 | 18.2 | - | 17.0 | 30.4 | - | 30.5 | 29.6 | 26.0 | 31.6 | 29.2 | 27.3 | 29.6 | 26.5 | 29.1 |
| W2V D100 | S | 10.2 | 31.9 | 16.9 | 18.2 | 16.2 | 18.6 | 29.5 | 15.2 | 24.5 | 24.3 | 32.7 | 20.6 | 30.6 | 29.8 | 28.0 | 26.4 | 27.2 | 26.3 | 27.8 | 23.9 | 28.5 |
| W2V | S | 3.5 | 22.3 | 7.4 | 9.9 | 16.2 | 17.9 | 29.5 | 6.2 | 12.3 | 9.5 | 32.7 | 20.6 | 30.6 | 29.8 | 17.2 | 26.4 | 27.2 | 26.3 | 27.8 | 19.7 | 28.5 |
| EF | E | - | 39.2 | 29.4 | 22.3 | 29.9 | 24.0 | 33.4 | 15.2 | - | 18.0 | 31.5 | 28.1 | 30.4 | 28.5 | 24.2 | 30.1 | 28.1 | 26.3 | 30.1 | 27.6 | 28.3 |
| EF D100 | E | - | 39.0 | 28.0 | 27.1 | 29.9 | 24.0 | 33.4 | 19.6 | - | 24.0 | 31.5 | 28.1 | 30.4 | 28.5 | 29.7 | 30.1 | 28.1 | 26.3 | 30.1 | 28.7 | 28.3 |
| EWMTF | E | - | 37.3 | 23.2 | 21.2 | 27.5 | 20.2 | 24.8 | 15.0 | - | 17.5 | 32.7 | - | 30.2 | 28.1 | 20.7 | 26.7 | 28.6 | 25.8 | 29.1 | 25.5 | 28.2 |
| EWMTF D100 | E | - | 37.3 | 23.2 | 23.4 | 27.5 | 20.2 | 24.8 | 18.3 | - | - | 32.7 | - | 30.2 | 28.1 | 25.7 | 26.7 | 28.6 | 25.8 | 29.1 | 26.8 | 28.2 |
| EW | E | - | 39.1 | 23.2 | 21.2 | 27.5 | 20.2 | 24.8 | 15.0 | - | 17.5 | 32.7 | - | 30.2 | 28.1 | 20.7 | 26.7 | 28.6 | 25.8 | 29.1 | 25.7 | 28.2 |
| EWF D100 | S | 25.0 | 37.5 | 27.6 | 27.3 | 27.0 | 26.0 | 33.5 | 22.9 | 27.1 | 23.5 | 23.1 | 14.9 | 28.9 | 29.6 | 24.9 | 24.7 | 26.4 | 26.4 | 33.7 | 26.8 | 27.8 |
| EWF | S | 12.4 | 37.5 | 27.6 | 14.3 | 27.0 | 26.0 | 33.5 | 8.9 | 14.6 | 9.8 | 23.1 | 14.9 | 28.9 | 29.6 | 9.2 | 24.7 | 26.4 | 26.4 | 33.7 | 22.6 | 27.8 |
| AC | S | 23.2 | 45.3 | 28.5 | 22.7 | 21.3 | 20.6 | 20.2 | 16.4 | 22.0 | 18.9 | 28.8 | 23.5 | 30.8 | 27.0 | 26.1 | 25.7 | 28.3 | 24.7 | 28.8 | 25.4 | 27.7 |
| M | S | 22.7 | 47.4 | 32.6 | 28.8 | 25.7 | 13.7 | 22.9 | 8.3 | 13.2 | 17.9 | 20.6 | 15.6 | 28.7 | 28.0 | 18.3 | 23.0 | 27.5 | 25.9 | 23.9 | 23.4 | 27.5 |
| EWF | E | - | 36.4 | 18.7 | 19.9 | 26.9 | 24.6 | 29.5 | 13.8 | - | 16.0 | 27.3 | - | 30.0 | 27.2 | 19.0 | 26.8 | 28.0 | 24.5 | 29.7 | 24.9 | 27.4 |
| EWF D100 | E | - | 36.3 | 22.3 | 23.6 | 26.9 | 24.6 | 29.5 | 17.9 | - | 21.5 | 27.3 | - | 30.0 | 27.2 | 25.5 | 26.8 | 28.0 | 24.5 | 29.7 | 26.4 | 27.4 |
| M | C | 26.9 | 46.1 | 30.9 | 27.5 | 23.9 | 14.9 | 17.1 | 11.4 | 15.6 | 17.6 | 20.3 | 19.6 | 29.9 | 26.7 | 20.2 | 24.4 | 27.5 | 24.2 | 21.9 | 23.5 | 27.1 |
| W2V D100 | E | - | 28.9 | 15.4 | 18.2 | 19.3 | 18.8 | 25.6 | 11.7 | - | - | 34.3 | - | 27.7 | 27.8 | 30.3 | 28.1 | 25.1 | 25.0 | 26.3 | 24.2 | 26.4 |
| W2V | E | - | 26.3 | 12.3 | 16.0 | 19.3 | 22.4 | 25.6 | 9.2 | - | 14.5 | 34.3 | - | 27.7 | 27.8 | 24.6 | 28.1 | 25.1 | 25.0 | 26.3 | 22.8 | 26.4 |
| WLM D30 | E | - | 39.5 | 23.2 | 18.5 | 18.9 | 17.1 | 23.6 | 13.5 | - | - | 25.2 | - | 26.1 | 27.3 | 28.4 | 27.6 | 23.7 | 24.3 | 24.1 | 24.1 | 25.4 |
| ET | C | 15.0 | 38.4 | 21.6 | 15.2 | 21.1 | 15.5 | 23.7 | 12.1 | 17.0 | 12.9 | 20.7 | 30.9 | 26.4 | 26.6 | 16.2 | 26.9 | 24.3 | 23.5 | 27.7 | 21.9 | 25.2 |
| ET D100 | C | 20.4 | 38.4 | 21.6 | 20.5 | 21.1 | 15.5 | 23.7 | 18.8 | 24.1 | 19.5 | 20.7 | 30.9 | 26.4 | 26.6 | 25.8 | 26.9 | 24.3 | 23.5 | 27.7 | 24.0 | 25.2 |
| ETF D100 | C | 20.4 | 38.9 | 21.8 | 20.6 | 21.2 | 15.8 | 23.8 | 23.8 | 23.8 | 19.5 | 20.9 | 31.2 | 26.5 | 26.6 | 25.8 | 26.9 | 24.3 | 23.5 | 27.7 | 24.1 | 25.2 |
| ETF | C | 15.0 | 35.9 | 21.8 | 15.3 | 21.2 | 15.8 | 23.8 | 12.0 | 16.8 | 13.0 | 20.9 | 31.2 | 26.5 | 26.6 | 16.2 | 26.9 | 24.3 | 23.5 | 27.7 | 21.8 | 25.2 |
| EMTF D30 | E | - | 34.9 | 23.0 | 21.6 | 24.4 | 16.1 | 22.1 | 18.2 | - | 17.0 | 25.3 | - | 26.4 | 25.8 | 26.0 | 26.8 | 24.9 | 23.3 | 26.9 | 23.9 | 25.1 |
| EWMTF D30 | E | - | 36.9 | 21.2 | 21.2 | 25.1 | 18.1 | 21.5 | 15.0 | - | 17.5 | 27.5 | - | 26.2 | 24.6 | 20.7 | 21.8 | 24.5 | 22.0 | 26.7 | 23.2 | 24.3 |
| MAE D100 | S | 18.9 | 35.8 | 17.9 | 19.5 | 15.3 | 20.1 | 30.9 | 14.4 | 24.6 | 21.9 | 24.9 | - | 25.5 | 25.9 | 23.4 | 22.9 | 22.6 | 22.4 | 31.0 | 23.2 | 24.1 |
| MAE | S | 8.4 | 35.8 | 17.9 | 8.3 | 15.3 | 14.8 | 30.9 | 5.8 | 12.6 | 9.4 | 24.9 | **50.0** | 25.6 | 25.9 | 11.5 | 22.9 | 22.6 | 22.4 | 31.0 | 20.8 | 24.1 |
| CC | C | 14.9 | 20.1 | 12.4 | 6.9 | 10.6 | 19.2 | 33.0 | 6.3 | 16.9 | 21.2 | 35.6 | 18.4 | 23.0 | 26.5 | 22.1 | 21.0 | 21.3 | 24.3 | 26.8 | 20.0 | 23.8 |
| EWTF | C | 13.4 | 32.9 | 19.9 | 14.6 | 22.3 | 14.2 | 24.1 | 9.1 | 16.4 | 12.5 | 16.8 | 22.6 | 25.5 | 24.4 | 12.9 | 24.2 | 23.4 | 21.5 | 26.3 | 19.8 | 23.7 |
| EWTF D100 | C | 17.7 | 35.8 | 20.3 | 18.7 | 22.3 | 14.2 | 24.1 | 14.4 | 22.9 | 18.8 | 16.8 | 22.6 | 25.5 | 24.4 | 21.5 | 24.2 | 23.4 | 21.5 | 26.3 | 21.9 | 23.7 |
| EWT D100 | C | 17.7 | 35.0 | 20.0 | 18.5 | 22.2 | 14.1 | 24.1 | 14.5 | 23.0 | 18.7 | 16.6 | 28.9 | 25.4 | 24.4 | 21.5 | 24.2 | 23.4 | 21.4 | 26.1 | 22.1 | 23.6 |
| EWT | C | 13.4 | 32.2 | 19.9 | 14.5 | 22.2 | 14.1 | 24.1 | 9.1 | 16.6 | 12.4 | 16.6 | 28.9 | 25.4 | 24.4 | 12.9 | 24.2 | 23.4 | 21.4 | 26.1 | 20.1 | 23.6 |
| CLP D100 | C | 17.6 | 40.0 | 21.5 | 19.3 | 20.0 | 14.2 | 36.8 | 15.7 | 21.2 | 18.7 | 14.9 | 19.7 | 25.0 | 24.4 | 22.3 | 23.6 | 22.8 | 21.5 | 28.2 | 22.5 | 23.4 |
| CLP | C | 13.2 | 39.0 | 17.6 | 15.4 | 20.0 | 12.7 | 36.8 | 11.3 | 16.3 | 13.3 | 14.9 | 19.7 | 25.0 | 24.4 | 17.0 | 23.6 | 22.8 | 21.5 | 28.2 | 20.7 | 23.4 |
| EF D30 | E | - | 37.8 | 22.7 | 22.3 | 24.0 | 17.9 | 25.7 | 15.2 | - | 18.0 | 24.5 | 28.1 | 24.4 | 22.9 | 24.2 | 24.5 | 22.0 | 20.6 | 25.2 | 23.5 | 22.5 |
| MAE | E | - | 32.4 | 16.6 | 13.3 | 18.1 | 26.5 | 27.5 | 9.4 | - | 15.1 | 25.2 | - | 23.8 | 22.6 | 18.9 | 25.0 | 21.6 | 20.3 | 28.8 | 21.6 | 22.1 |
| MAE D100 | E | - | 30.0 | 16.6 | 17.2 | 18.1 | 18.0 | 27.5 | 12.7 | - | 21.4 | 25.2 | - | 23.8 | 22.6 | 24.6 | 25.0 | 21.6 | 20.3 | 28.8 | 22.1 | 22.1 |
| EWF D30 | E | - | 35.4 | 18.7 | 19.9 | 22.1 | 18.9 | 22.0 | 13.8 | - | 16.0 | 20.8 | - | 24.0 | 21.8 | 19.0 | 20.6 | 22.0 | 18.8 | 25.2 | 21.2 | 21.6 |

*Table 24.* (Continued) CSR Results Across Subsets and Distances (↑ better)

| Method | Dist | BC | BS1 | BS2 | BS3 | BS4 | BS5 | ES1 | HP | HS1 | HS2 | HU1 | HU2 | HU3 | HU4 | HW1 | HW2 | HW3 | HW4 | OC1 | Avg | Avg (Blind) |
|---|---|---|---|---|---|---|---|---|---|---|---|---|---|---|---|---|---|---|---|---|---|---|
| WLM D100 | C | 8.5 | 39.8 | 15.9 | 12.4 | 10.2 | 11.3 | 19.8 | 8.0 | 20.3 | 17.8 | 23.5 | 32.7 | 22.6 | 23.2 | 21.4 | 21.6 | 19.1 | 19.7 | 23.0 | 19.5 | 21.1 |
| WLM | C | 4.1 | 36.5 | 11.0 | 7.0 | 10.2 | 19.6 | 19.8 | 4.3 | 13.0 | 9.5 | 23.5 | 32.7 | 22.6 | 23.2 | 14.7 | 21.6 | 19.1 | 19.7 | 23.0 | 17.6 | 21.1 |
| CLP D30 | S | 18.3 | 33.6 | 22.5 | 20.4 | 20.1 | 16.1 | 33.8 | 14.5 | 20.5 | 15.3 | 14.5 | 20.3 | 21.8 | 21.9 | 19.0 | 20.2 | 19.1 | 18.6 | 26.1 | 20.9 | 20.3 |
| VC | E | - | 47.7 | 31.3 | 25.8 | 28.0 | 21.3 | 18.4 | 12.1 | - | - | 20.4 | - | 21.9 | 19.9 | 20.1 | 23.1 | 19.0 | - | 24.4 | 23.8 | 20.3 |
| ET D30 | S | 18.7 | 33.3 | 21.0 | 19.3 | 19.1 | 15.1 | 20.5 | 14.6 | 19.6 | 15.3 | 15.2 | 32.6 | 20.9 | 21.6 | 17.8 | 20.0 | 18.5 | 18.3 | 25.1 | 20.3 | 19.8 |
| ETF D30 | S | 18.7 | 33.4 | 21.1 | 19.1 | 19.2 | 15.3 | 20.3 | 14.6 | 19.6 | 15.3 | 15.3 | 33.4 | 21.0 | 21.4 | 17.8 | 20.0 | 18.6 | 18.3 | 24.9 | 20.4 | 19.8 |
| W2V D30 | E | - | 26.3 | 12.3 | 16.0 | 14.6 | 14.6 | 20.2 | 9.2 | - | 14.5 | 25.2 | - | 20.4 | 21.7 | 24.6 | 22.1 | 18.0 | 18.3 | 20.1 | 18.6 | 19.6 |
| EWTF D30 | S | 17.8 | 34.3 | 22.4 | 19.8 | 20.6 | 16.1 | 20.1 | 13.2 | 19.5 | 14.5 | 12.5 | 16.5 | 20.8 | 20.8 | 14.8 | 18.5 | 18.4 | 17.6 | 24.6 | 19.1 | 19.4 |
| EWT D30 | S | 17.8 | 34.1 | 22.3 | 19.7 | 20.7 | 16.3 | 20.1 | 13.2 | 19.6 | 14.4 | 12.3 | 30.7 | 20.7 | 20.8 | 14.8 | 18.5 | 18.3 | 17.6 | 24.3 | 19.8 | 19.4 |
| EMTF D30 | S | 16.3 | 30.8 | 19.3 | 17.3 | 17.6 | 14.6 | 19.8 | 13.6 | 18.7 | 14.4 | 21.1 | 25.4 | 20.9 | 21.0 | 18.8 | 19.1 | 17.8 | 17.6 | 23.4 | 19.3 | 19.3 |
| EWMTF D30 | S | 16.1 | 33.6 | 19.5 | 17.7 | 18.6 | 16.1 | 19.5 | 12.7 | 18.7 | 14.2 | 21.0 | 20.4 | 20.1 | 20.4 | 17.5 | 17.1 | 17.2 | 17.1 | 23.6 | 19.0 | 18.7 |
| E | C | 7.8 | 28.0 | - | 10.0 | 14.7 | 23.5 | 15.5 | 7.5 | 10.0 | 6.4 | 18.9 | 18.1 | 19.1 | 18.5 | 13.3 | 19.5 | 17.4 | 15.9 | 18.6 | 15.7 | 17.7 |
| EMTF D100 | C | 10.5 | 29.5 | 14.2 | 13.2 | 14.7 | 7.9 | 15.5 | 11.2 | 13.2 | 9.4 | 18.9 | 18.1 | 19.1 | 18.5 | 20.3 | 19.5 | 17.4 | 15.9 | 18.6 | 16.1 | 17.7 |
| EMTF | C | 7.8 | 28.0 | 14.2 | 10.0 | 14.7 | 7.9 | 15.5 | 7.5 | 10.0 | 6.4 | 18.9 | 18.1 | 19.1 | 18.5 | 13.3 | 19.5 | 17.4 | 15.9 | 18.6 | 14.8 | 17.7 |
| ET D30 | C | 15.0 | 36.4 | 16.6 | 15.2 | 16.2 | 10.5 | 17.8 | 12.1 | 17.0 | 12.9 | 12.8 | 30.9 | 18.8 | 19.2 | 16.2 | 18.3 | 16.5 | 16.2 | 21.7 | 17.9 | 17.7 |
| ETF D30 | C | 15.0 | 37.0 | 16.7 | 15.3 | 16.3 | 10.6 | 17.9 | 12.0 | 16.8 | 13.0 | 12.9 | 31.2 | 18.8 | 19.2 | 16.2 | 18.3 | 16.5 | 16.2 | 21.7 | 18.0 | 17.7 |
| EWMTF | C | 8.9 | 30.4 | 9.7 | 9.8 | 15.0 | 9.0 | 15.2 | 4.7 | 9.8 | 6.8 | 21.7 | 13.7 | 19.0 | 18.3 | 8.0 | 13.0 | 17.3 | 15.9 | 18.8 | 14.0 | 17.6 |
| EW | C | 8.9 | 35.2 | 19.9 | 9.8 | 15.0 | 9.0 | 15.2 | 4.7 | 9.8 | 6.8 | 21.7 | 13.7 | 19.0 | 18.3 | 8.0 | 13.0 | 17.3 | 15.9 | 18.8 | 14.7 | 17.6 |
| EWMTF D100 | C | 11.3 | 31.3 | 13.2 | 12.3 | 15.0 | 9.0 | 15.2 | 7.2 | 12.7 | 9.8 | 21.7 | 13.7 | 19.0 | 18.3 | 13.1 | 13.0 | 17.3 | 15.9 | 18.8 | 15.2 | 17.6 |
| CLP D30 | C | 13.2 | 39.0 | 17.6 | 15.4 | 15.8 | 10.6 | 33.2 | 11.3 | 16.3 | 13.3 | 10.3 | 19.7 | 19.2 | 18.7 | 17.0 | 18.3 | 16.8 | 15.8 | 23.4 | 18.2 | 17.6 |
| EF D100 | C | 11.8 | 35.2 | 16.6 | 15.7 | 17.1 | 11.7 | 24.0 | 10.1 | 13.0 | 11.5 | 19.0 | 14.7 | 18.5 | 18.1 | 17.6 | 17.7 | 16.3 | 15.6 | 19.0 | 17.0 | 17.1 |
| EF | C | 6.3 | 32.8 | 16.6 | 10.1 | 17.1 | 11.7 | 24.0 | 6.0 | 7.7 | 6.0 | 19.0 | 14.7 | 18.5 | 18.1 | 10.9 | 17.7 | 16.3 | 15.6 | 19.0 | 15.2 | 17.1 |
| WLM D30 | S | 6.3 | 33.0 | 16.8 | 10.2 | 9.4 | 9.4 | 16.7 | 5.7 | 15.8 | 11.8 | 13.7 | 37.9 | 18.0 | 19.2 | 16.5 | 17.1 | 15.3 | 15.7 | 16.1 | 16.0 | 17.0 |
| MAE D30 | E | - | 27.2 | 12.4 | 13.3 | 13.1 | 12.5 | 21.1 | 9.4 | - | 15.1 | 16.6 | - | 18.4 | 17.7 | 18.9 | 19.5 | 16.4 | 15.4 | 23.8 | 16.9 | 17.0 |
| EF D30 | S | 12.2 | 32.1 | 16.8 | 16.9 | 15.8 | 12.5 | 19.9 | 11.4 | 15.4 | 11.5 | 13.8 | 20.1 | 17.9 | 18.8 | 14.1 | 14.6 | 15.7 | 15.4 | 21.0 | 16.6 | 17.0 |
| EWTF D30 | C | 13.4 | 34.6 | 17.0 | 14.6 | 17.7 | 10.4 | 17.9 | 9.1 | 16.4 | 12.5 | 9.9 | 14.8 | 18.3 | 17.6 | 12.9 | 16.2 | 16.2 | 14.9 | 20.9 | 16.1 | 16.8 |
| EWT D30 | C | 13.4 | 33.7 | 16.7 | 14.5 | 17.6 | 10.4 | 18.0 | 9.1 | 16.6 | 12.4 | 9.9 | 28.9 | 18.3 | 17.6 | 12.9 | 16.2 | 16.2 | 14.9 | 20.7 | 16.7 | 16.7 |
| W2V D100 | C | 3.2 | 22.5 | 5.9 | 8.3 | 7.5 | 8.9 | 16.0 | 5.5 | 13.0 | 12.0 | 24.4 | 23.9 | 17.7 | 18.8 | 21.8 | 19.5 | 15.0 | 15.2 | 18.5 | 14.6 | 16.7 |
| W2V | C | 1.2 | 18.7 | 3.6 | 6.0 | 7.5 | 13.9 | 16.0 | 3.1 | 7.0 | 5.6 | 24.4 | 23.9 | 17.7 | 18.8 | 13.6 | 19.5 | 15.0 | 15.2 | 18.5 | 13.1 | 16.7 |
| EWF | C | 7.4 | 26.6 | 8.2 | 8.5 | 13.6 | 12.4 | 21.8 | 4.2 | 7.2 | 4.8 | 13.8 | 12.2 | 17.9 | 17.1 | 6.0 | 13.2 | 15.7 | 13.8 | 18.8 | 12.8 | 16.1 |
| EWF D100 | C | 11.5 | 30.9 | 11.8 | 12.5 | 13.6 | 12.4 | 21.8 | 7.4 | 11.9 | 9.2 | 13.8 | 12.2 | 17.9 | 17.1 | 11.9 | 13.2 | 15.7 | 13.8 | 18.8 | 14.6 | 16.1 |
| AC | E | - | 37.7 | 27.6 | 23.9 | 27.4 | 18.0 | 13.7 | 17.2 | - | 11.3 | 27.2 | - | 17.3 | 13.4 | 25.1 | 26.7 | 15.7 | - | 20.1 | 21.5 | 15.5 |
| EWF D30 | S | 12.4 | 32.6 | 14.7 | 14.3 | 13.7 | 13.4 | 19.0 | 8.9 | 14.6 | 9.8 | 11.6 | 9.7 | 16.5 | 16.7 | 9.2 | 10.6 | 13.9 | 13.1 | 20.5 | 14.5 | 15.0 |
| W2V D30 | S | 3.5 | 22.3 | 7.4 | 9.9 | 7.3 | 8.5 | 16.2 | 6.2 | 12.3 | 9.5 | 15.9 | 20.6 | 15.7 | 16.2 | 17.2 | 15.4 | 12.7 | 12.4 | 14.9 | 12.9 | 14.3 |
| CC | E | 2.0 | 22.3 | 8.7 | 4.0 | 1.3 | 0.9 | 40.0 | 0.8 | 5.5 | 12.9 | 21.2 | 1.3 | 13.4 | 14.7 | 7.0 | 8.9 | 13.9 | 12.9 | 2.1 | 10.2 | 13.8 |
| WLM D30 | C | 4.1 | 36.5 | 11.0 | 7.0 | 6.5 | 6.4 | 11.2 | 4.3 | 13.0 | 9.5 | 11.4 | 32.7 | 14.0 | 15.3 | 14.7 | 15.1 | 11.1 | 12.0 | 13.4 | 13.1 | 13.1 |
| EWMTF D30 | C | 8.9 | 30.4 | 10.9 | 9.8 | 12.1 | 7.0 | 11.2 | 4.7 | 9.8 | 6.8 | 14.7 | 13.7 | 13.9 | 13.7 | 8.0 | 8.2 | 12.3 | 11.4 | 15.3 | 11.7 | 12.8 |
| EMTF D30 | C | 7.8 | 28.0 | 11.2 | 10.0 | 11.6 | 5.8 | 11.8 | 7.5 | 10.0 | 6.4 | 12.5 | 18.1 | 13.9 | 13.7 | 13.3 | 13.3 | 12.2 | 11.3 | 14.9 | 12.3 | 12.8 |
| MAE D30 | S | 8.4 | 27.3 | 7.9 | 8.3 | 6.3 | 8.0 | 17.9 | 5.8 | 12.6 | 9.4 | 9.5 | - | 13.2 | 13.8 | 11.5 | 11.4 | 10.6 | 10.8 | 18.3 | 11.7 | 12.1 |
| MAE | C | 3.9 | 21.3 | 3.9 | 3.4 | 5.9 | 12.9 | 21.2 | 2.7 | 7.3 | 5.0 | 13.8 | 19.8 | 12.5 | 11.9 | 7.0 | 12.1 | 10.3 | 9.2 | 19.8 | 10.7 | 11.0 |
| MAE D100 | C | 7.2 | 21.3 | 7.0 | 6.3 | 5.9 | 8.1 | 21.2 | 5.0 | 12.6 | 10.5 | 13.8 | 19.8 | 12.5 | 11.9 | 12.3 | 12.1 | 10.3 | 9.2 | 19.8 | 11.9 | 11.0 |
| EF D30 | C | 6.3 | 32.8 | 10.7 | 10.1 | 10.3 | 6.0 | 14.1 | 6.0 | 7.7 | 6.0 | 10.5 | 14.7 | 11.3 | 11.5 | 10.9 | 11.0 | 9.7 | 9.4 | 12.8 | 11.2 | 10.5 |
| EWF D30 | C | 7.4 | 29.3 | 8.2 | 8.5 | 8.8 | 6.9 | 12.0 | 4.2 | 7.2 | 4.8 | 7.5 | 6.0 | 10.9 | 10.8 | 6.0 | 7.1 | 9.1 | 7.9 | 12.9 | 9.2 | 9.7 |
| W2V D30 | C | 1.2 | 18.7 | 3.6 | 6.0 | 4.0 | 4.7 | 9.8 | 3.1 | 7.0 | 5.6 | 12.6 | 23.9 | 9.2 | 10.8 | 13.6 | 11.6 | 7.5 | 7.8 | 10.0 | 9.0 | 8.8 |
| VC | C | 7.1 | 46.9 | 19.5 | 12.8 | 14.2 | 7.8 | 7.1 | 2.9 | 6.7 | 5.2 | 6.9 | 5.8 | 8.3 | 6.8 | 7.1 | 8.3 | 6.1 | 4.7 | 10.7 | 10.3 | 6.5 |
| MAE D30 | C | 3.9 | 17.3 | 3.9 | 3.4 | 2.9 | 3.8 | 12.9 | 2.7 | 7.3 | 5.0 | 5.6 | 19.8 | 7.2 | 6.9 | 7.0 | 7.0 | 5.7 | 5.1 | 12.5 | 7.4 | 6.2 |
| AC | C | 7.1 | 29.2 | 13.8 | 10.6 | 14.2 | 5.6 | 5.2 | 4.4 | 2.9 | 2.9 | 15.8 | 7.5 | 6.9 | 4.0 | 11.1 | 12.7 | 5.3 | 3.2 | 8.6 | 9.0 | 4.9 |
| VC | S | 5.5 | 35.3 | 11.0 | 7.6 | 10.5 | 7.0 | 5.0 | 1.7 | 5.8 | 3.5 | 3.6 | 3.1 | 6.1 | 5.3 | 3.7 | 4.0 | 3.8 | 3.1 | 9.8 | 7.1 | 4.6 |

*Table 25.* CS Results Across Subsets and Distances (↓ better)

| Method | Dist | BC | BS1 | BS2 | BS3 | BS4 | BS5 | ES1 | HP | HS1 | HS2 | HU1 | HU2 | HU3 | HU4 | HW1 | HW2 | HW3 | HW4 | OC1 | Avg | Avg (Blind) |
|---|---|---|---|---|---|---|---|---|---|---|---|---|---|---|---|---|---|---|---|---|---|---|
| EWMTF D30 | C | 35.8 | 20.2 | 28.5 | 31.0 | 22.2 | 29.3 | 34.7 | 46.2 | 44.4 | 43.2 | 32.5 | 41.2 | **44.9** | 47.0 | 45.0 | 44.7 | **47.2** | **46.8** | 38.5 | 38.1 | **46.5** |
| EWTF D30 | C | 36.2 | 18.9 | 23.1 | 25.7 | 18.6 | 25.2 | 36.5 | 47.3 | 44.5 | 45.7 | 43.2 | 43.9 | 45.0 | 47.3 | 46.4 | 45.4 | 47.5 | 47.0 | 38.7 | 38.2 | 46.7 |
| EWMTF | C | 35.8 | 20.2 | 29.6 | 31.0 | 23.3 | 30.2 | 36.1 | 46.2 | 44.4 | 43.2 | 32.4 | 41.2 | 45.3 | 47.2 | 45.0 | 44.6 | 47.3 | 47.1 | 39.5 | 38.4 | 46.7 |
| EWMTF D100 | C | 36.6 | 21.2 | 29.2 | 31.7 | 23.3 | 30.2 | 36.1 | 46.3 | 44.7 | 43.7 | 32.4 | 41.2 | 45.3 | 47.2 | 44.5 | 44.6 | 47.3 | 47.1 | 39.5 | 38.5 | 46.7 |
| EW | C | 35.8 | 20.8 | 26.5 | 31.0 | 23.3 | 36.5 | 36.1 | 46.2 | 44.4 | 43.2 | 32.4 | 41.2 | 45.3 | 47.2 | 45.0 | 44.6 | 47.3 | 47.1 | 39.5 | 38.6 | 46.7 |
| EWT D30 | C | 36.3 | 19.1 | 23.2 | 25.8 | 18.8 | 25.2 | 36.6 | 47.3 | 44.6 | 45.9 | 43.3 | 46.2 | 45.0 | 47.4 | 46.5 | 45.5 | 47.5 | 47.1 | 38.8 | 38.4 | 46.7 |
| CLP D30 | C | 36.9 | 25.0 | 22.4 | 25.8 | 20.4 | 28.4 | **22.2** | 43.8 | 42.4 | 43.2 | 42.9 | 42.8 | 45.3 | 47.3 | 40.2 | 41.3 | 47.4 | 47.3 | 36.7 | 36.9 | 46.8 |
| AC | C | **32.5** | 16.5 | 23.9 | 25.1 | 16.8 | 30.0 | 36.8 | 43.8 | **40.7** | **42.3** | 36.2 | 38.9 | 45.4 | 46.9 | 40.7 | 39.3 | 47.7 | 47.3 | 36.3 | 36.2 | 46.8 |
| EMTF D30 | C | 36.5 | 26.4 | 30.0 | 34.8 | 24.2 | 24.5 | 37.0 | 45.4 | 46.0 | 43.7 | 32.3 | 40.5 | 45.2 | 47.3 | 42.9 | 43.0 | 47.5 | 47.3 | 38.8 | 38.6 | 46.8 |
| ETF D30 | C | 38.5 | 24.7 | 28.4 | 31.4 | 22.1 | **23.1** | 35.8 | 45.8 | 45.1 | 45.2 | 41.9 | 45.8 | 45.1 | 47.7 | 44.8 | 44.1 | 47.5 | 47.5 | 37.8 | 39.1 | 47.0 |
| ET D30 | C | 38.8 | 25.0 | 28.7 | 31.5 | 22.4 | 23.3 | 35.9 | 45.9 | 45.1 | 45.3 | 42.1 | 45.8 | 45.2 | 47.7 | 44.9 | 44.2 | 47.6 | 47.5 | 38.0 | 39.2 | 47.0 |
| EMTF | C | 36.5 | 26.4 | 31.1 | 34.8 | 25.6 | 25.6 | 38.1 | 45.4 | 46.0 | 43.7 | **32.1** | 40.5 | 45.6 | 47.6 | 42.9 | 43.2 | 47.6 | 47.6 | 39.8 | 39.0 | 47.1 |
| EMTF D100 | C | 37.2 | 27.8 | 31.1 | 35.5 | 25.6 | 25.6 | 38.1 | 45.7 | 46.2 | 44.1 | **32.1** | 40.5 | 45.6 | 47.6 | 42.6 | 43.2 | 47.6 | 47.6 | 39.8 | 39.1 | 47.1 |
| E | C | 36.5 | 26.4 | 21.5 | 34.8 | 25.6 | 37.8 | 38.1 | 45.4 | 46.0 | 43.7 | **32.1** | 40.5 | 45.6 | 47.6 | 42.9 | 43.2 | 47.6 | 47.6 | 39.8 | 40.1 | 47.1 |
| EWTF D100 | C | 38.1 | 21.3 | 24.9 | 27.8 | 21.4 | 27.8 | 39.1 | 47.6 | 45.5 | 46.6 | 43.2 | 45.0 | 45.8 | 47.8 | 46.5 | 46.0 | 47.8 | 47.6 | 40.7 | 39.5 | 47.3 |
| EWTF | C | 36.2 | 22.3 | 26.5 | 25.7 | 21.4 | 27.8 | 39.1 | 47.3 | 44.5 | 45.7 | 43.2 | 45.0 | 45.8 | 47.8 | 46.4 | 46.0 | 47.8 | 47.6 | 40.7 | 39.3 | 47.3 |
| EF D30 | C | 36.5 | 23.1 | 25.5 | 32.8 | 21.0 | 28.3 | 41.3 | 44.9 | 46.8 | 45.8 | 36.5 | 42.4 | 46.4 | 47.4 | 43.1 | 42.2 | 47.8 | 47.5 | 37.1 | 38.8 | 47.3 |
| EWT D100 | C | 38.2 | 21.4 | 25.0 | 27.8 | 21.5 | 27.9 | 39.2 | 47.6 | 45.6 | 46.7 | 43.4 | 46.2 | 45.9 | 47.9 | 46.6 | 46.0 | 47.8 | 47.6 | 40.8 | 39.6 | 47.3 |
| EWT | C | 36.3 | 22.3 | 26.5 | 25.8 | 21.5 | 27.8 | 39.2 | 47.3 | 44.6 | 45.9 | 43.4 | 46.2 | 45.9 | 47.9 | 46.5 | 46.0 | 47.8 | 47.6 | 40.8 | 39.4 | 47.3 |
| CLP D100 | C | 38.6 | 27.3 | 25.1 | 28.1 | 22.8 | 30.6 | 26.8 | 44.8 | 43.4 | 44.4 | 43.6 | 42.8 | 46.0 | 47.8 | 41.6 | 42.5 | 47.8 | 47.7 | 38.9 | 38.4 | 47.3 |
| CLP | C | 36.9 | 25.0 | 22.4 | 25.8 | 22.8 | 31.2 | 26.8 | 43.8 | 42.4 | 43.2 | 43.6 | 42.8 | 46.0 | 47.8 | 40.2 | 42.5 | 47.8 | 47.7 | 38.9 | 37.8 | 47.3 |
| VC | C | 35.0 | **9.5** | **14.5** | **16.6** | **12.1** | 25.2 | 37.5 | **39.8** | 44.6 | 44.6 | 41.7 | **37.8** | 46.4 | 46.4 | 35.1 | 36.3 | 48.0 | 47.8 | **35.8** | **34.5** | 47.4 |
| EWMTF D30 | S | 42.7 | 38.8 | 33.9 | 35.6 | 29.5 | 34.0 | 38.7 | 47.1 | 45.7 | 45.2 | 38.0 | 44.2 | 46.1 | 47.9 | 43.8 | 44.9 | 47.9 | 47.7 | 43.7 | 41.9 | 47.4 |
| EWTF D30 | S | 41.7 | 39.1 | 31.4 | 32.0 | 24.9 | 31.3 | 39.3 | 47.7 | 45.3 | 46.5 | 44.1 | 45.2 | 46.2 | 48.0 | 46.7 | 46.1 | 47.8 | 47.8 | 43.2 | 41.8 | 47.5 |
| EWT D30 | S | 41.9 | 39.2 | 31.7 | 31.9 | 25.1 | 31.4 | 39.3 | 47.7 | 45.2 | 46.5 | 44.2 | 46.5 | 46.3 | 48.1 | 46.7 | 46.1 | 47.9 | 47.8 | 43.1 | 41.9 | 47.5 |
| ETF D100 | C | 40.5 | 27.9 | 30.7 | 33.3 | 25.0 | 27.1 | 38.3 | 46.6 | 45.9 | 46.1 | 42.1 | 45.8 | 46.1 | 48.2 | 45.2 | 45.0 | 47.9 | 48.0 | 40.3 | 40.5 | 47.5 |
| ETF | C | 38.5 | 29.2 | 30.7 | 31.4 | 25.0 | 27.1 | 38.3 | 45.8 | 45.1 | 45.2 | 42.1 | 45.8 | 46.1 | 48.2 | 44.8 | 45.0 | 47.9 | 48.0 | 40.3 | 40.2 | 47.5 |
| ET | C | 38.8 | 28.1 | 30.9 | 31.5 | 25.3 | 27.1 | 38.4 | 45.9 | 45.1 | 45.3 | 42.3 | 45.8 | 46.1 | 48.2 | 44.9 | 45.1 | 48.0 | 48.0 | 40.4 | 40.3 | 47.6 |
| ET D100 | C | 40.7 | 28.1 | 30.9 | 33.4 | 25.3 | 27.2 | 38.4 | 46.7 | 45.9 | 46.2 | 42.3 | 45.8 | 46.1 | 48.2 | 45.3 | 45.1 | 48.0 | 48.0 | 40.4 | 40.6 | 47.6 |
| CLP D30 | S | 41.6 | 40.3 | 29.9 | 33.7 | 28.4 | 33.2 | 25.8 | 46.1 | 43.7 | 45.2 | 44.1 | 43.0 | 46.5 | 48.0 | 42.6 | 43.7 | 47.9 | 48.0 | 42.5 | 40.7 | 47.6 |
| ETF D30 | S | 42.2 | 40.4 | 34.8 | 35.7 | 28.2 | 28.5 | 39.5 | 46.9 | 45.6 | 46.4 | 43.0 | 46.2 | 46.4 | 48.2 | 45.5 | 45.4 | 48.0 | 48.1 | 42.8 | 42.2 | 47.6 |
| EMTF D30 | S | 43.6 | 41.8 | 35.7 | 38.7 | 31.0 | 31.2 | 40.5 | 46.9 | 46.1 | 46.0 | 38.1 | 44.0 | 46.4 | 48.1 | 43.6 | 44.6 | 48.0 | 48.0 | 43.3 | 42.4 | 47.6 |
| ET D30 | S | 42.3 | 40.5 | 35.2 | 35.9 | 28.6 | 29.0 | 39.5 | 46.9 | 45.7 | 46.5 | 43.2 | 46.2 | 46.4 | 48.2 | 45.5 | 45.4 | 48.0 | 48.1 | 42.7 | 42.3 | 47.7 |
| EWF D30 | C | 37.4 | 21.5 | 28.5 | 32.3 | 25.3 | 33.2 | 40.8 | 45.9 | 45.2 | 45.6 | 35.9 | 40.7 | 46.8 | 47.8 | 45.1 | 43.4 | 48.1 | 47.9 | 37.4 | 39.4 | 47.7 |
| EF | C | 36.5 | 23.1 | 28.9 | 32.8 | 25.1 | 31.2 | 43.9 | 44.9 | 46.8 | 45.8 | 38.9 | 42.4 | 47.1 | 47.9 | 43.1 | 43.4 | 48.2 | 48.0 | 39.1 | 39.8 | 47.8 |
| EF D100 | C | 38.5 | 25.9 | 28.9 | 35.0 | 25.1 | 31.2 | 43.9 | 45.4 | 47.1 | 46.4 | 38.9 | 42.4 | 47.1 | 47.9 | 44.1 | 43.4 | 48.2 | 48.0 | 39.1 | 40.3 | 47.8 |
| VC | S | 39.3 | 17.7 | 20.3 | 23.3 | 17.9 | 29.5 | 40.4 | 41.6 | 45.6 | 45.9 | 43.2 | 40.3 | 47.4 | 47.4 | 38.4 | 39.4 | 48.4 | 48.1 | 38.6 | 37.5 | 47.8 |
| EWMTF D100 | S | 46.1 | 46.5 | 39.1 | 40.0 | 36.3 | 40.7 | 43.3 | 47.7 | 47.1 | 47.0 | 36.3 | 44.2 | 46.9 | 48.4 | 42.7 | 44.9 | 48.2 | 48.3 | 46.8 | 44.2 | 48.0 |
| EWMTF | S | 42.7 | 46.5 | 32.5 | 35.6 | 36.3 | 40.7 | 43.3 | 47.1 | 45.7 | 45.2 | 36.3 | 44.2 | 46.9 | 48.4 | 43.8 | 44.9 | 48.2 | 48.3 | 46.8 | 43.3 | 48.0 |
| EWMTF D30 | E | - | 31.6 | 37.5 | 39.1 | 33.1 | 37.8 | 41.2 | 47.7 | - | 46.1 | 39.3 | - | 47.2 | 48.4 | 47.1 | 47.0 | 48.4 | 48.3 | 43.7 | 42.7 | 48.1 |
| EWF D100 | C | 39.1 | 22.9 | 30.0 | 33.8 | 27.4 | 35.7 | 43.5 | 46.2 | 45.6 | 46.1 | 37.9 | 42.7 | 47.4 | 48.3 | 45.5 | 44.3 | 48.5 | 48.2 | 39.3 | 40.7 | 48.1 |
| EWF | C | 37.4 | 23.2 | 28.5 | 32.3 | 27.4 | 35.7 | 43.5 | 45.9 | 45.2 | 45.6 | 37.9 | 42.7 | 47.4 | 48.3 | 45.1 | 44.3 | 48.5 | 48.2 | 39.3 | 40.3 | 48.1 |
| AC | E | - | 28.4 | 34.5 | 35.0 | 28.7 | 37.6 | 40.6 | 46.7 | - | 45.2 | 40.8 | - | 47.4 | 48.2 | 45.1 | 44.4 | 48.8 | - | 42.0 | 40.9 | 48.1 |
| WLM D30 | C | 35.1 | 20.4 | 19.1 | 28.0 | 14.8 | 25.3 | 40.8 | 42.3 | 46.7 | 45.5 | 41.2 | 44.6 | 47.7 | 48.1 | **28.1** | **36.2** | 48.6 | 48.4 | 38.9 | 36.8 | 48.2 |
| EWTF D30 | E | - | 31.2 | 34.2 | 35.9 | 30.7 | 35.1 | 42.5 | 48.5 | - | 47.6 | 45.8 | - | 47.3 | 48.5 | 48.0 | 47.5 | 48.6 | 48.4 | 43.7 | 42.7 | 48.2 |
| EWT D30 | E | - | 31.3 | 34.3 | 35.9 | 30.8 | 35.1 | 42.6 | 48.5 | - | 47.7 | 45.9 | - | 47.3 | 48.5 | 48.0 | 47.5 | 48.6 | 48.4 | 43.8 | 42.8 | 48.2 |
| EWMTF | E | - | 32.5 | 38.1 | 39.1 | 34.0 | 38.4 | 42.1 | 47.7 | - | 46.1 | 39.4 | - | 47.4 | 48.5 | 47.1 | 47.0 | 48.5 | 48.4 | 44.2 | 43.0 | 48.2 |
| EWMTF D100 | E | - | 32.5 | 38.1 | 39.6 | 34.0 | 38.4 | 42.1 | 47.8 | - | - | 39.4 | - | 47.4 | 48.5 | 46.9 | 47.0 | 48.5 | 48.4 | 44.2 | 42.9 | 48.2 |
| EW | E | - | 32.0 | 38.1 | 39.1 | 34.0 | 42.5 | 42.1 | 47.7 | - | 46.1 | 39.4 | - | 47.4 | 48.5 | 47.1 | 47.0 | 48.5 | 48.4 | 44.2 | 43.3 | 48.2 |
| CLP D30 | E | - | 35.7 | 33.8 | 36.1 | 32.0 | 36.9 | 33.9 | 46.7 | - | 46.3 | 45.9 | - | 47.4 | 48.6 | 44.7 | 45.4 | 48.6 | 48.5 | 42.7 | 42.1 | 48.3 |
| EMTF D30 | E | - | 36.4 | 38.7 | 41.5 | 34.8 | 34.1 | 43.4 | 47.9 | - | 46.3 | 39.2 | - | 47.4 | 48.6 | 46.2 | 46.2 | 48.7 | 48.5 | 43.9 | 43.1 | 48.3 |
| EWTF D100 | S | 45.8 | 46.7 | 37.8 | 38.6 | 33.9 | 40.1 | 44.0 | 48.4 | 46.8 | 47.9 | 44.0 | 46.5 | 47.4 | 48.7 | 46.9 | 46.9 | 48.4 | 48.7 | 46.8 | 45.0 | 48.3 |
| EWTF | S | 41.7 | 46.7 | 37.8 | 32.0 | 33.9 | 40.1 | 44.0 | 47.7 | 45.3 | 46.5 | 44.0 | 46.5 | 47.4 | 48.7 | 46.7 | 46.9 | 48.4 | 48.7 | 46.8 | 44.2 | 48.3 |
| EMTF D100 | S | 46.6 | 47.3 | 41.0 | 42.0 | 37.7 | 40.0 | 44.3 | 47.8 | 47.3 | 47.5 | 36.4 | 44.0 | 47.4 | 48.8 | 42.9 | 45.0 | 48.4 | 48.7 | 46.7 | 44.7 | 48.3 |
| EMTF | S | 43.6 | 41.8 | 41.0 | 38.7 | 37.7 | 40.0 | 44.3 | 46.9 | 46.1 | 46.0 | 36.4 | 44.0 | 47.4 | 48.8 | 43.6 | 45.0 | 48.4 | 48.7 | 46.7 | 43.9 | 48.3 |
| EWT | S | 41.9 | 46.7 | 37.8 | 31.9 | 34.1 | 40.1 | 44.0 | 47.7 | 45.2 | 46.5 | 44.1 | 46.5 | 47.4 | 48.8 | 46.7 | 46.9 | 48.4 | 48.6 | 46.8 | 44.2 | 48.3 |
| EWT D100 | S | 45.9 | 46.7 | 38.0 | 38.7 | 34.1 | 40.2 | 44.0 | 48.4 | 46.9 | 47.9 | 44.1 | 46.5 | 47.4 | 48.8 | 46.9 | 46.9 | 48.4 | 48.6 | 46.8 | 45.0 | 48.3 |
| ETF D30 | E | - | 35.3 | 37.9 | 39.6 | 33.0 | 33.4 | 42.2 | 47.7 | - | 47.4 | 45.2 | - | 47.3 | 48.8 | 47.2 | 46.8 | 48.6 | 48.7 | 43.2 | 43.3 | 48.3 |
| ET D30 | E | - | 35.5 | 38.0 | 39.7 | 33.2 | 33.6 | 42.2 | 47.8 | - | 47.4 | 45.3 | - | 47.3 | 48.8 | 47.2 | 46.9 | 48.7 | 48.7 | 43.3 | 43.3 | 48.4 |
| VC | E | - | 24.0 | 28.0 | 29.1 | 25.0 | 34.7 | 41.8 | 44.5 | - | 44.0 | - | - | 48.1 | 48.2 | 42.1 | 42.4 | 48.8 | - | 42.2 | 38.8 | 48.4 |
| W2V D30 | C | 35.6 | 32.5 | 30.6 | 36.6 | 26.0 | 31.5 | 44.9 | 46.9 | 46.9 | 46.7 | 39.0 | 45.1 | 48.0 | 47.9 | 36.7 | 41.2 | 48.8 | 48.9 | 40.0 | 40.7 | 48.4 |
| E | E | - | 37.3 | 39.5 | 41.5 | 35.8 | 43.2 | 43.2 | 47.4 | - | 46.3 | 39.3 | - | 47.6 | 48.7 | 46.2 | 46.4 | 48.7 | 48.7 | 44.5 | 44.0 | 48.4 |
| EMTF D100 | E | - | 37.3 | 39.5 | 42.0 | 35.8 | 35.1 | 43.2 | 47.6 | - | - | 39.3 | - | 47.6 | 48.7 | 46.1 | 46.4 | 48.7 | 48.7 | 44.5 | 43.4 | 48.4 |
| EMTF | E | - | 37.3 | 40.1 | 41.5 | 35.8 | 35.1 | 43.2 | 47.4 | - | 46.3 | 39.3 | - | 47.6 | 48.7 | 46.2 | 46.4 | 48.7 | 48.7 | 44.5 | 43.6 | 48.4 |
| ETF | S | 42.2 | 47.0 | 40.1 | 35.7 | 36.1 | 39.7 | 44.0 | 46.9 | 45.6 | 46.4 | 43.1 | 46.2 | 47.6 | 48.9 | 45.5 | 46.5 | 48.6 | 48.8 | 46.7 | 44.5 | 48.4 |
| ETF D100 | S | 46.2 | 47.0 | 40.1 | 40.8 | 36.1 | 39.7 | 44.0 | 48.2 | 47.0 | 47.9 | 43.1 | 46.2 | 47.6 | 48.9 | 46.0 | 46.5 | 48.6 | 48.8 | 46.7 | 45.2 | 48.4 |
| ET D100 | S | 46.1 | 47.0 | 40.3 | 40.9 | 36.4 | 39.8 | 44.0 | 48.2 | 47.0 | 47.9 | 43.2 | 46.2 | 47.6 | 48.9 | 46.0 | 46.5 | 48.6 | 48.8 | 46.7 | 45.3 | 48.5 |
| ET | S | 42.3 | 47.0 | 35.2 | 40.9 | 36.4 | 39.8 | 44.0 | 46.9 | 45.7 | 46.5 | 43.2 | 46.2 | 47.6 | 48.9 | 45.5 | 46.5 | 48.6 | 48.8 | 46.7 | 44.6 | 48.5 |
| EF D30 | E | - | 33.9 | 35.5 | 40.0 | 32.0 | 36.6 | 45.0 | 47.0 | - | 47.5 | 42.1 | 45.7 | 47.9 | 48.6 | 46.3 | 45.6 | 48.8 | 48.6 | 42.7 | 43.2 | 48.5 |
| CC | S | 34.3 | 24.8 | 24.2 | 33.9 | 19.9 | 30.7 | 49.0 | 45.8 | 46.5 | 47.3 | 41.0 | 39.0 | 48.4 | 47.8 | 43.1 | 43.9 | 49.0 | 48.7 | 37.2 | 39.7 | 48.5 |
| EWTF D100 | E | - | 33.2 | 35.6 | 37.4 | 33.0 | 37.1 | 44.1 | 48.7 | - | - | 45.9 | - | 47.7 | 48.8 | 48.1 | 47.8 | 48.8 | 48.7 | 45.0 | 43.3 | 48.5 |
| EWTF | E | - | 31.2 | 34.2 | 35.9 | 33.0 | 37.1 | 44.1 | 48.7 | - | 47.6 | 45.9 | - | 47.7 | 48.8 | 48.0 | 47.8 | 48.7 | 48.7 | 45.0 | 43.3 | 48.5 |
| EWT D100 | E | - | 33.2 | 35.7 | 37.4 | 33.1 | 37.1 | 44.2 | 48.7 | - | 48.2 | 46.0 | - | 47.8 | 48.8 | 48.1 | 47.9 | 48.8 | 48.7 | 45.0 | 43.7 | 48.5 |
| EWT | E | - | 31.2 | 34.3 | 35.9 | 33.1 | 37.1 | 44.2 | 48.5 | - | 47.7 | 46.0 | - | 47.8 | 48.8 | 48.0 | 47.9 | 48.8 | 48.7 | 45.0 | 43.3 | 48.5 |
| CLP | E | - | 37.5 | 36.2 | 36.1 | 33.9 | 39.0 | 37.0 | 46.7 | - | 46.3 | 46.4 | - | 47.8 | 48.8 | 44.7 | 46.1 | 48.8 | 48.8 | 44.0 | 43.0 | 48.6 |
| CLP D100 | E | - | 37.3 | 35.8 | 37.7 | 33.9 | 38.6 | 37.0 | 47.2 | - | - | 46.4 | - | 47.8 | 48.8 | 45.5 | 46.1 | 48.8 | 48.8 | 44.0 | 42.9 | 48.6 |

*Table 25.* (Continued) CS Results Across Subsets and Distances (↓ better)

| Method | Dist | BC | BS1 | BS2 | BS3 | BS4 | BS5 | ES1 | HP | HS1 | HS2 | HU1 | HU2 | HU3 | HU4 | HW1 | HW2 | HW3 | HW4 | OC1 | Avg | Avg (Blind) |
|---|---|---|---|---|---|---|---|---|---|---|---|---|---|---|---|---|---|---|---|---|---|---|
| WLM D100 | C | 38.3 | 24.5 | 22.1 | 30.5 | 17.5 | 28.7 | 42.8 | 43.9 | 47.3 | 46.7 | 41.4 | 44.6 | 48.2 | 48.5 | 31.0 | 38.4 | 48.9 | 48.7 | 42.1 | 38.6 | 48.6 |
| WLM | C | 35.1 | 20.4 | 19.1 | 28.0 | 17.5 | 35.8 | 42.8 | 42.3 | 46.7 | 45.5 | 41.4 | 44.6 | 48.2 | 48.5 | **28.1** | 38.4 | 48.9 | 48.7 | 42.1 | 38.0 | 48.6 |
| CLP D100 | S | 45.9 | 46.9 | 38.7 | 40.7 | 37.1 | 41.0 | 36.4 | 48.1 | 46.3 | 47.4 | 46.1 | 43.0 | 48.0 | 48.9 | 45.6 | 46.4 | 48.8 | 48.9 | 46.8 | 44.8 | 48.6 |
| CLP | S | 41.6 | 40.3 | 38.7 | 33.7 | 37.1 | 31.5 | 36.4 | 46.1 | 43.7 | 45.2 | 46.1 | 43.0 | 48.0 | 48.9 | 42.6 | 46.4 | 48.8 | 48.9 | 46.8 | 42.8 | 48.6 |
| EF D30 | S | 43.8 | 40.7 | 34.1 | 37.8 | 28.8 | 35.1 | 45.7 | 47.3 | 47.2 | 47.8 | 43.4 | 43.4 | 48.3 | 48.5 | 45.1 | 45.7 | 49.0 | 48.7 | 43.4 | 43.3 | 48.6 |
| ETF D100 | E | - | 37.6 | 39.4 | 40.9 | 35.3 | 36.6 | 43.7 | 48.2 | - | 47.9 | 45.4 | - | 47.9 | 49.0 | 47.5 | 47.4 | 48.9 | 48.9 | 44.8 | 44.3 | 48.7 |
| ETF | E | - | 38.5 | 40.8 | 39.6 | 35.3 | 36.6 | 43.7 | 47.7 | - | 47.4 | 45.4 | - | 47.9 | 49.0 | 47.2 | 47.4 | 48.9 | 48.9 | 44.8 | 44.3 | 48.7 |
| ET | E | - | 38.5 | 39.5 | 39.7 | 35.5 | 36.6 | 43.7 | 47.8 | - | 47.4 | 45.4 | - | 47.9 | 49.0 | 47.2 | 47.4 | 48.9 | 49.0 | 44.9 | 44.3 | 48.7 |
| ET D100 | E | - | 37.7 | 39.5 | 41.0 | 35.5 | 36.7 | 43.7 | 48.2 | - | 47.9 | 45.4 | - | 47.9 | 49.0 | 47.5 | 47.4 | 48.9 | 49.0 | 44.9 | 44.4 | 48.7 |
| EWF D30 | E | - | 32.4 | 37.3 | 39.8 | 34.9 | 40.1 | 44.6 | 47.6 | - | 47.4 | 41.3 | - | 48.2 | 48.8 | 47.3 | 46.3 | 49.0 | 48.8 | 42.9 | 43.5 | 48.7 |
| WLM D30 | S | 42.3 | 39.4 | 28.9 | 32.8 | 20.1 | 31.3 | 44.9 | 44.1 | 47.4 | 46.9 | 42.3 | 48.0 | 48.3 | 48.7 | 30.0 | 37.3 | 49.0 | 49.0 | 42.6 | 40.7 | 48.7 |
| M | C | 43.3 | 24.1 | 26.3 | 29.6 | 22.9 | 34.4 | 40.1 | 44.2 | 46.5 | 47.5 | 40.9 | 43.5 | 48.5 | 48.6 | 42.2 | 44.0 | 49.0 | 48.8 | 41.7 | 40.3 | 48.7 |
| EWF D30 | S | 44.5 | 40.3 | 33.9 | 37.3 | 33.1 | 38.2 | 45.4 | 47.7 | 46.9 | 47.8 | 42.3 | 45.3 | 48.2 | 48.8 | 46.6 | 46.6 | 49.0 | 48.9 | 43.5 | 43.9 | 48.7 |
| EF D100 | E | - | 36.0 | 38.0 | 41.5 | 35.3 | 39.0 | 46.6 | 47.4 | - | 47.9 | 43.7 | 45.7 | 48.4 | 48.9 | 46.8 | 46.4 | 49.0 | 48.8 | 44.0 | 44.3 | 48.8 |
| EF | E | - | 36.4 | 38.7 | 40.0 | 35.3 | 39.0 | 46.6 | 47.0 | - | 47.5 | 43.7 | 45.7 | 48.4 | 48.9 | 46.3 | 46.4 | 49.0 | 48.8 | 44.0 | 44.2 | 48.8 |
| W2V | C | 35.6 | 32.5 | 30.6 | 36.6 | 29.1 | 38.9 | 45.7 | 46.9 | 46.9 | 46.7 | 40.4 | 45.1 | 48.5 | 48.5 | 36.7 | 43.0 | 49.0 | 49.1 | 42.3 | 41.7 | 48.8 |
| W2V D100 | C | 37.4 | 34.3 | 32.3 | 38.2 | 29.1 | 33.7 | 45.7 | 47.3 | 47.4 | 47.2 | 40.4 | 45.1 | 48.5 | 48.5 | 38.8 | 43.0 | 49.0 | 49.1 | 42.3 | 42.0 | 48.8 |
| M | S | 41.6 | 26.3 | 29.8 | 33.5 | 24.9 | 34.3 | 45.0 | 45.1 | 45.4 | 47.0 | 40.4 | 42.1 | 48.6 | 48.6 | 42.6 | 43.9 | 49.1 | 48.9 | 41.9 | 41.0 | 48.8 |
| WLM D30 | E | - | 30.4 | 28.8 | 34.3 | 23.4 | 33.0 | 44.5 | 45.1 | - | - | 44.2 | - | 48.6 | 48.9 | 36.9 | 42.3 | 49.2 | 49.1 | 43.3 | 40.1 | 48.9 |
| EWF D100 | E | - | 33.7 | 38.6 | 40.8 | 36.6 | 41.8 | 46.3 | 47.8 | - | 47.7 | 42.7 | - | 48.6 | 49.1 | 47.6 | 46.8 | 49.2 | 49.0 | 44.0 | 44.4 | 49.0 |
| EWF | E | - | 33.9 | 37.3 | 39.8 | 36.6 | 41.8 | 46.3 | 47.6 | - | 47.4 | 42.7 | - | 48.6 | 49.1 | 47.3 | 46.8 | 49.2 | 49.0 | 44.0 | 44.2 | 49.0 |
| W2V D30 | E | - | 39.9 | 38.3 | 42.0 | 34.9 | 38.0 | 46.9 | 48.0 | - | 48.0 | 42.4 | - | 48.7 | 48.8 | 42.3 | 44.9 | 49.2 | 49.3 | 43.8 | 44.1 | 49.0 |
| AC | S | 44.0 | 22.2 | 30.9 | 30.4 | 23.1 | 36.7 | 43.4 | 47.6 | 44.9 | 47.2 | 44.6 | 46.3 | 48.7 | 49.0 | 46.3 | 46.1 | 49.4 | 49.1 | 46.0 | 41.9 | 49.1 |
| CC | C | 42.8 | 37.2 | 36.1 | 41.0 | 34.0 | 40.6 | 48.8 | 47.7 | 47.6 | 48.1 | 45.7 | 44.7 | 49.0 | 48.7 | 46.3 | 46.7 | 49.4 | 49.3 | 44.6 | 44.7 | 49.1 |
| W2V D30 | S | 43.6 | 42.7 | 39.0 | 43.2 | 37.4 | 37.8 | 47.1 | 48.7 | 48.4 | 47.7 | 41.6 | 43.6 | 48.7 | 49.0 | 39.7 | 43.2 | 49.3 | 49.4 | 44.8 | 44.5 | 49.1 |
| M | E | - | 34.9 | 35.4 | 38.5 | 31.5 | 37.8 | 43.2 | 46.8 | - | - | 44.1 | - | 49.0 | 49.1 | 45.8 | 46.6 | 49.3 | - | 45.1 | 42.7 | 49.1 |
| WLM D100 | E | - | 33.9 | 32.0 | 36.7 | 26.5 | 36.2 | 45.9 | 46.2 | - | - | 44.3 | - | 48.9 | 49.1 | 39.0 | 43.6 | 49.3 | 49.2 | 44.9 | 41.7 | 49.1 |
| WLM | E | - | 30.4 | 28.8 | 34.3 | 26.5 | 40.2 | 45.9 | 45.1 | - | - | 44.3 | - | 48.9 | 49.1 | 36.9 | 43.6 | 49.3 | 49.2 | 44.9 | 41.2 | 49.1 |
| WLM D100 | S | 46.4 | 45.9 | 37.2 | 39.8 | 28.0 | 39.6 | 47.4 | 47.0 | 47.9 | 48.3 | 42.2 | 43.8 | 48.9 | 49.2 | 33.7 | 40.4 | 49.4 | 49.3 | 46.3 | 43.7 | 49.2 |
| WLM | S | 42.3 | 39.4 | 28.9 | 32.8 | 28.0 | 35.3 | 47.4 | 44.1 | 47.4 | 46.9 | 42.2 | 48.0 | 48.9 | 49.2 | 30.0 | 40.4 | 49.4 | 49.3 | 46.3 | 41.9 | 49.2 |
| MAE D30 | C | 42.3 | 29.0 | 30.2 | 30.1 | 22.5 | 42.5 | 38.4 | 46.0 | 47.3 | 47.1 | 41.4 | 47.7 | 48.6 | 49.5 | 47.0 | 46.7 | 49.3 | 49.4 | 43.8 | 42.0 | 49.2 |
| W2V D100 | E | - | 41.1 | 39.6 | 43.1 | 37.4 | 39.9 | 47.5 | 48.3 | - | - | 43.6 | - | 49.0 | 49.1 | 43.5 | 45.9 | 49.4 | 49.4 | 45.1 | 44.8 | 49.2 |
| W2V | E | - | 39.9 | 38.3 | 42.0 | 37.4 | 42.0 | 47.5 | 48.0 | - | 48.0 | 43.6 | - | 49.0 | 49.1 | 42.3 | 45.9 | 49.4 | 49.4 | 45.1 | 44.8 | 49.2 |
| EF D100 | S | 47.2 | 47.1 | 41.0 | 43.1 | 38.4 | 42.7 | 48.2 | 48.5 | 48.3 | 48.9 | 45.9 | 43.4 | 49.1 | 49.2 | 46.5 | 47.5 | 49.5 | 49.3 | 47.3 | 46.4 | 49.3 |
| EF | S | 43.8 | 40.7 | 34.1 | 37.8 | 38.4 | 42.7 | 48.2 | 47.3 | 47.2 | 47.8 | 45.9 | 43.4 | 49.1 | 49.2 | 45.1 | 47.5 | 49.5 | 49.3 | 47.3 | 45.0 | 49.3 |
| EWF D100 | S | 47.4 | 46.9 | 40.4 | 42.3 | 39.2 | 44.0 | 48.0 | 48.7 | 48.0 | 48.9 | 44.8 | 43.4 | 49.0 | 49.4 | 47.2 | 47.9 | 49.4 | 49.4 | 47.3 | 46.4 | 49.3 |
| EWF | S | 44.5 | 46.9 | 40.4 | 37.3 | 39.2 | 44.0 | 48.0 | 47.7 | 46.9 | 47.8 | 44.8 | 43.4 | 49.0 | 49.4 | 46.6 | 47.9 | 49.4 | 49.4 | 47.3 | 45.8 | 49.3 |
| MAE D100 | C | 43.1 | 31.1 | 31.4 | 31.3 | 24.6 | 43.0 | 40.9 | 46.3 | 47.5 | 47.5 | 42.4 | 47.7 | 48.8 | 49.6 | 47.4 | 47.2 | 49.3 | 49.5 | 45.1 | 42.8 | 49.3 |
| MAE | C | 42.3 | 31.1 | 30.2 | 30.1 | 24.6 | 45.0 | 40.9 | 46.0 | 47.3 | 47.1 | 42.4 | 47.7 | 48.8 | 49.6 | 47.0 | 47.2 | 49.3 | 49.5 | 45.1 | 42.7 | 49.3 |
| MAE D30 | S | 45.9 | 40.9 | 36.7 | 36.4 | 30.4 | 42.9 | 41.9 | 47.7 | 47.5 | 47.8 | 43.9 | - | 48.9 | 49.3 | 47.7 | 47.6 | 49.5 | 49.5 | 45.7 | 44.5 | 49.3 |
| CC | E | 43.8 | 42.3 | 41.7 | 44.3 | 38.6 | 43.1 | 49.2 | 48.6 | 47.7 | 48.7 | 47.5 | 46.0 | 49.3 | 49.2 | 47.8 | 47.9 | 49.6 | 49.6 | 46.3 | 46.4 | 49.4 |
| MAE D30 | E | - | 37.1 | 37.0 | 37.5 | 30.9 | 45.5 | 43.6 | 47.5 | - | 48.3 | 44.4 | - | 49.2 | 49.6 | 48.3 | 48.1 | 49.5 | 49.6 | 46.6 | 44.5 | 49.5 |
| MAE | E | 45.9 | 46.7 | 40.3 | 36.4 | 37.1 | 45.4 | 45.3 | 47.7 | 47.5 | 47.8 | 45.4 | 50.0 | 49.2 | 49.5 | 47.7 | 48.5 | 49.6 | 49.7 | 47.8 | 46.2 | 49.5 |
| MAE D100 | S | 47.7 | 46.7 | 40.3 | 40.3 | 37.1 | 44.4 | 45.3 | 48.6 | 47.9 | 48.5 | 45.4 | - | 49.2 | 49.5 | 48.4 | 48.5 | 49.6 | 49.7 | 47.8 | 46.4 | 49.5 |
| W2V D100 | S | 47.4 | 47.5 | 44.9 | 46.8 | 44.2 | 43.3 | 48.6 | 49.3 | 48.4 | 48.8 | 43.0 | 43.6 | 49.3 | 49.6 | 41.9 | 45.6 | 49.6 | 49.7 | 47.6 | 46.8 | 49.5 |
| W2V | S | 43.6 | 42.7 | 39.0 | 43.2 | 44.2 | 37.6 | 48.6 | 48.7 | 48.4 | 47.7 | 43.0 | 43.6 | 49.3 | 49.6 | 39.7 | 45.6 | 49.6 | 49.7 | 47.6 | 45.3 | 49.5 |
| MAE | E | - | 40.0 | 38.2 | 37.5 | 33.1 | 47.1 | 45.0 | 47.5 | - | 48.3 | 45.4 | - | 49.3 | 49.6 | 48.3 | 48.4 | 49.6 | 49.7 | 47.4 | 45.3 | 49.5 |
| MAE D100 | E | - | 38.7 | 38.2 | 38.6 | 33.1 | 45.9 | 45.0 | 47.8 | - | 48.6 | 45.4 | - | 49.3 | 49.6 | 48.5 | 48.4 | 49.6 | 49.7 | 47.4 | 45.2 | 49.5 |

*Table 26.* CSCF Results Across Subsets and Distances (↓ better)

| Method | Dist | BC | BS1 | BS2 | BS3 | BS4 | BS5 | ES1 | HP | HS1 | HS2 | HU1 | HU2 | HU3 | HU4 | HW1 | HW2 | HW3 | HW4 | OC1 | Avg | Avg (Blind) |
|---|---|---|---|---|---|---|---|---|---|---|---|---|---|---|---|---|---|---|---|---|---|---|
| EWMTF D100 | S | **0.1** | **0.0** | 0.2 | 0.8 | **0.0** | 0.1 | **0.0** | 0.2 | **0.0** | **0.4** | **0.0** | 9.4 | **5.5** | 11.6 | **0.0** | 0.4 | **8.1** | **8.9** | 0.2 | 2.4 | **8.6** |
| EWMTF | S | 1.5 | **0.0** | 1.2 | 2.3 | **0.0** | 0.1 | **0.0** | 1.3 | 2.7 | 1.3 | 0.0 | 9.4 | **5.5** | 11.6 | 1.7 | 0.4 | **8.1** | **8.9** | 0.2 | 3.0 | **8.6** |
| EMTF | S | 1.1 | **0.0** | 0.2 | 2.1 | **0.0** | 0.1 | **0.0** | 0.7 | 3.8 | 1.7 | 0.0 | 6.7 | 9.7 | **8.3** | 1.3 | 0.4 | 10.3 | 10.0 | 0.2 | 3.0 | 9.6 |
| EMTF D100 | S | 0.4 | **0.0** | 0.2 | 1.5 | **0.0** | 0.1 | **0.0** | **0.2** | **0.0** | 0.7 | 0.0 | 6.7 | 9.7 | **8.3** | 0.0 | 0.4 | 10.3 | 10.0 | 0.2 | 2.6 | 9.6 |
| EWTF D100 | S | 0.3 | **0.0** | 0.1 | 1.2 | **0.0** | 0.1 | **0.0** | 0.2 | **0.0** | 0.5 | 0.0 | 2.7 | 8.9 | 13.8 | 1.8 | 2.3 | 9.1 | 10.4 | 0.2 | 2.7 | 10.6 |
| EWTF | S | 0.7 | **0.0** | 0.1 | 1.7 | **0.0** | 0.1 | **0.0** | 1.3 | **0.0** | 1.6 | 0.0 | 2.7 | 8.9 | 13.8 | 8.7 | 2.3 | 9.1 | 10.4 | 0.2 | 3.2 | 10.6 |
| EWT D100 | S | 0.3 | **0.0** | 0.2 | 1.0 | **0.0** | 0.1 | **0.0** | 0.3 | **0.0** | 0.5 | 0.0 | 4.9 | 8.8 | 13.8 | 1.8 | 2.4 | 9.3 | 10.5 | 0.2 | 2.9 | 10.6 |
| EWT | S | 0.8 | **0.0** | 0.1 | 1.5 | **0.0** | 0.1 | **0.0** | 1.4 | **0.0** | 1.6 | 0.0 | 4.9 | 8.8 | 13.8 | 8.7 | 2.4 | 9.3 | 10.5 | 0.2 | 3.4 | 10.6 |
| ETF | S | 0.7 | **0.0** | 0.1 | 2.4 | **0.0** | 0.1 | **0.0** | 0.7 | 6.6 | 1.6 | 0.0 | 3.5 | 8.6 | 12.0 | 3.1 | 1.3 | 10.4 | 13.8 | **0.0** | 3.4 | 11.2 |
| ETF D100 | S | **0.1** | **0.0** | 0.1 | 2.5 | **0.0** | 0.1 | **0.0** | 0.2 | 1.1 | 0.7 | 0.0 | 3.5 | 8.6 | 12.0 | 0.4 | 1.3 | 10.4 | 13.8 | **0.0** | 2.9 | 11.2 |
| ET D100 | S | 0.3 | **0.0** | 0.1 | 2.5 | **0.0** | 0.1 | **0.0** | 0.2 | 1.1 | 0.9 | 0.0 | 3.7 | 9.0 | 11.8 | 0.4 | 1.3 | 10.8 | 13.4 | **0.0** | 2.9 | 11.3 |
| ET | S | 0.7 | **0.0** | 0.9 | 2.5 | **0.0** | 0.1 | **0.0** | 0.7 | 8.2 | 1.9 | 0.0 | 3.7 | 9.0 | 11.8 | 3.1 | 1.3 | 10.8 | 13.4 | **0.0** | 3.6 | 11.3 |
| ETF D100 | C | 0.4 | **0.0** | 0.8 | 2.8 | 0.0 | 0.1 | 0.2 | 1.3 | 2.2 | 1.8 | 0.2 | 4.9 | 11.8 | 15.1 | 0.9 | 2.7 | 13.4 | 15.7 | 1.4 | 4.0 | 14.0 |
| ETF | C | 0.9 | **0.0** | 0.8 | 3.2 | 0.0 | 0.1 | 0.2 | 2.2 | 6.0 | 2.3 | 0.2 | 4.9 | 11.8 | 15.1 | 3.6 | 2.7 | 13.4 | 15.7 | 1.4 | 4.5 | 14.0 |
| EWTF | C | 1.6 | **0.0** | 0.2 | 2.5 | 0.0 | 0.1 | 0.0 | 3.7 | 5.5 | 2.1 | 0.4 | 3.9 | 10.6 | 17.2 | 9.8 | 4.6 | 13.8 | 14.9 | 1.4 | 4.9 | 14.1 |
| EWTF D100 | C | 1.2 | **0.0** | 0.2 | 2.3 | 0.0 | 0.1 | 0.0 | 2.2 | 2.2 | 1.4 | 0.4 | 3.9 | 10.6 | 17.2 | 3.3 | 4.6 | 13.8 | 14.9 | 1.4 | 4.2 | 14.1 |
| ET D100 | C | 0.4 | **0.0** | 0.7 | 2.9 | 0.0 | 0.1 | 0.2 | 1.2 | 2.2 | 1.9 | 0.2 | 6.0 | 12.0 | 15.5 | 0.9 | 2.8 | 13.4 | 15.9 | 1.7 | 4.1 | 14.2 |
| ET | C | 0.7 | **0.0** | 0.7 | 3.2 | 0.0 | 0.1 | 0.2 | 2.0 | 6.0 | 2.4 | 0.2 | 6.0 | 12.0 | 15.5 | 3.7 | 2.8 | 13.4 | 15.9 | 1.7 | 4.6 | 14.2 |
| EWT | C | 1.5 | **0.0** | 0.2 | 2.5 | 0.0 | 0.1 | 0.0 | 3.7 | 4.9 | 2.2 | 0.4 | 8.9 | 10.8 | 17.5 | 10.0 | 4.6 | 13.9 | 14.9 | 1.7 | 5.1 | 14.3 |
| EWT D100 | C | 1.2 | **0.0** | 0.2 | 2.3 | 0.0 | 0.1 | 0.0 | 2.2 | 2.2 | 1.5 | 0.4 | 8.9 | 10.8 | 17.5 | 3.4 | 4.6 | 13.9 | 14.9 | 1.7 | 4.5 | 14.3 |
| EWMTF D30 | S | 1.5 | **0.0** | 0.2 | 2.3 | 0.1 | 0.1 | **0.0** | 1.3 | 2.7 | 1.3 | 0.1 | 9.4 | 9.0 | 19.3 | 1.7 | 4.8 | 15.2 | 13.7 | 0.5 | 4.4 | 14.3 |
| CLP D100 | S | **0.1** | **0.0** | **0.0** | 0.9 | **0.0** | 0.1 | **0.0** | 0.4 | 1.1 | 0.7 | 0.4 | 5.8 | 12.5 | 17.7 | 0.7 | 1.8 | 14.4 | 14.3 | 0.2 | 3.7 | 14.7 |
| CLP | S | 0.4 | **0.0** | **0.0** | 2.9 | **0.0** | 0.2 | **0.0** | 0.9 | 2.7 | 1.7 | 0.4 | 5.8 | 12.5 | 17.7 | 1.3 | 1.8 | 14.4 | 14.3 | 0.2 | 4.1 | 14.7 |
| EWTF D30 | S | 0.7 | **0.0** | 0.2 | 1.7 | 0.0 | 0.1 | 0.0 | 1.3 | **0.0** | 1.6 | 1.4 | 6.0 | 12.6 | 17.1 | 8.7 | 7.3 | 14.6 | 14.7 | 0.5 | 4.7 | 14.7 |
| ETF D30 | S | 0.7 | **0.0** | 0.8 | 2.4 | 0.0 | 0.1 | **0.0** | 0.7 | 6.6 | 1.6 | 0.8 | 3.5 | 12.4 | 15.7 | 3.1 | 5.2 | 15.1 | 16.5 | 0.2 | 4.5 | 14.9 |
| EWT D30 | S | 0.8 | **0.0** | 0.3 | 1.5 | 0.0 | 0.1 | 0.0 | 1.4 | **0.0** | 1.6 | 1.3 | 4.9 | 12.5 | 17.8 | 8.7 | 7.3 | 15.3 | 15.6 | 0.7 | 4.7 | 15.3 |
| EMTF D30 | S | 1.1 | **0.0** | 1.0 | 2.1 | 0.1 | 0.1 | 0.1 | 0.7 | 3.8 | 1.7 | 0.1 | 6.7 | 12.9 | 15.0 | 1.3 | 3.9 | 16.5 | 16.8 | 0.2 | 4.4 | 15.3 |
| ET D30 | S | 0.7 | **0.0** | 0.9 | 1.9 | 0.0 | 0.1 | **0.0** | 0.7 | 8.2 | 1.9 | 0.8 | 3.7 | 12.9 | 17.0 | 3.1 | 5.2 | 15.5 | 17.4 | 0.2 | 4.8 | 15.7 |
| CLP | C | 1.1 | **0.0** | 0.2 | 3.2 | 0.1 | 0.2 | **0.0** | 2.0 | 4.9 | 2.5 | 3.2 | 6.0 | 12.7 | 19.1 | 2.1 | 3.0 | 14.9 | 16.4 | 0.7 | 4.9 | 15.8 |
| CLP D100 | C | 0.9 | **0.0** | 0.2 | 2.9 | 0.1 | 0.1 | **0.0** | 1.7 | 4.4 | 2.3 | 3.2 | 6.0 | 12.7 | 19.1 | 1.5 | 3.0 | 14.9 | 16.4 | 0.7 | 4.7 | 15.8 |
| CLP D30 | S | 0.4 | **0.0** | 0.1 | 2.9 | **0.0** | 0.1 | **0.0** | 0.9 | 2.7 | 1.7 | 2.2 | 5.8 | 13.6 | 20.0 | 1.3 | 2.8 | 15.2 | 16.1 | 0.4 | 4.6 | 16.2 |
| ETF D30 | C | 0.9 | **0.0** | 1.0 | 3.2 | 0.1 | 0.2 | 0.2 | 2.2 | 6.0 | 2.3 | 1.7 | 4.9 | 13.9 | 18.4 | 3.6 | 5.9 | 16.2 | 17.8 | 1.2 | 5.3 | 16.6 |
| EWMTF | C | 6.7 | **0.0** | 0.9 | 2.9 | 0.3 | 0.5 | 0.5 | 6.3 | 5.5 | 4.9 | 0.8 | 10.2 | 12.6 | 19.7 | 12.0 | 12.9 | 17.3 | 16.8 | 2.9 | 7.0 | 16.6 |
| EWMTF D100 | C | 6.3 | **0.0** | 0.5 | 2.9 | 0.3 | 0.5 | 0.5 | 5.2 | 4.4 | 4.5 | 0.8 | 10.2 | 12.6 | 19.7 | 5.8 | 12.9 | 17.3 | 16.8 | 2.9 | 6.5 | 16.6 |
| EW | C | 6.7 | **0.0** | 0.2 | 2.9 | 0.3 | 1.6 | 0.5 | 6.3 | 5.5 | 4.9 | 0.8 | 10.2 | 12.6 | 19.7 | 12.0 | 12.9 | 17.3 | 16.8 | 2.9 | 7.1 | 16.6 |
| CLP D30 | C | 1.1 | **0.0** | 0.2 | 3.2 | 0.1 | 0.1 | **0.0** | 2.0 | 4.9 | 2.5 | 4.6 | 6.0 | 13.2 | 20.3 | 2.1 | 3.6 | 15.9 | 17.3 | 0.7 | 5.2 | 16.7 |
| EWTF D30 | C | 1.6 | **0.0** | 0.3 | 2.5 | 0.1 | 0.1 | 0.1 | 3.7 | 5.5 | 2.1 | 2.7 | 6.3 | 13.3 | 20.2 | 9.8 | 8.5 | 17.1 | 17.0 | 1.9 | 5.9 | 16.9 |
| ET D30 | C | 0.7 | **0.0** | 1.0 | 3.2 | 0.1 | 0.2 | 0.2 | 2.0 | 6.0 | 2.4 | 1.7 | 6.0 | 14.2 | 19.1 | 3.7 | 5.9 | 16.3 | 18.1 | 1.4 | 5.4 | 16.9 |
| EWT D30 | C | 1.5 | **0.0** | 0.3 | 2.5 | 0.1 | 0.1 | 0.1 | 3.7 | 4.9 | 2.2 | 2.5 | 8.9 | 13.5 | 20.5 | 10.0 | 8.6 | 17.1 | 17.3 | 1.7 | 6.1 | 17.1 |
| EWMTF D30 | C | 6.7 | **0.0** | 0.6 | 2.9 | 0.3 | 0.6 | 0.7 | 6.3 | 5.5 | 4.9 | 2.8 | 10.2 | 14.3 | 21.9 | 12.0 | 17.7 | 20.2 | 18.5 | 3.1 | 7.9 | 18.8 |
| EWTF | E | 2.6 | **0.0** | 0.2 | 3.2 | 0.0 | 0.1 | 0.0 | 16.6 | 2.7 | 3.4 | 0.4 | 6.0 | 13.1 | 24.3 | 15.7 | 9.0 | 19.0 | 21.9 | 2.9 | 7.4 | 19.6 |
| EWTF D100 | E | 4.9 | **0.0** | 0.2 | 3.2 | 0.0 | 0.1 | 0.0 | 22.8 | 1.1 | 3.0 | 0.4 | 6.0 | 13.1 | 24.3 | 9.8 | 9.0 | 19.0 | 21.9 | 2.9 | 7.5 | 19.6 |
| EWT D100 | E | 4.8 | **0.0** | 0.2 | 3.3 | 0.0 | 0.1 | 0.0 | 25.2 | 1.6 | 3.3 | 0.5 | 10.3 | 13.3 | 24.6 | 10.1 | 9.4 | 19.2 | 22.0 | 2.9 | 7.9 | 19.8 |
| EWT | E | 2.6 | **0.0** | 0.2 | 3.4 | 0.0 | 0.1 | 0.0 | 18.1 | 3.3 | 3.6 | 0.5 | 10.3 | 13.3 | 24.6 | 16.0 | 9.4 | 19.2 | 22.0 | 2.9 | 7.9 | 19.8 |
| EMTF D100 | C | 7.7 | **0.0** | 1.4 | 3.1 | 0.2 | 1.4 | 0.7 | 4.0 | 8.2 | 6.2 | 2.3 | 7.6 | 15.9 | 22.2 | 1.3 | 5.7 | 20.2 | 21.5 | 2.1 | 6.9 | 19.9 |
| E | C | 7.8 | **0.0** | **0.0** | 3.5 | 0.2 | 2.3 | 0.7 | 5.0 | 9.3 | 6.7 | 2.3 | 7.6 | 15.9 | 22.2 | 4.1 | 5.7 | 20.2 | 21.5 | 2.1 | 7.2 | 19.9 |
| EMTF | C | 7.8 | **0.0** | 1.4 | 3.5 | 0.2 | 1.4 | 0.7 | 5.0 | 9.3 | 6.7 | 2.3 | 7.6 | 15.9 | 22.2 | 4.1 | 5.7 | 20.2 | 21.5 | 2.1 | 7.2 | 19.9 |
| EWTF D30 | E | 2.6 | **0.0** | 0.2 | 3.2 | 0.0 | 0.1 | 0.0 | 16.6 | 2.7 | 3.4 | 2.3 | 7.0 | 14.0 | 24.3 | 15.7 | 10.6 | 20.6 | 21.2 | 1.2 | 7.7 | 20.0 |
| EWT D30 | E | 2.6 | **0.0** | 0.2 | 3.4 | 0.0 | 0.1 | 0.0 | 18.1 | 3.3 | 3.6 | 2.5 | 10.3 | 14.3 | 24.1 | 16.0 | 10.9 | 20.8 | 21.4 | 1.4 | 8.1 | 20.2 |
| ETF | E | 0.4 | **0.0** | 1.1 | 3.6 | 0.0 | 0.1 | 0.2 | 3.4 | 1.6 | 3.4 | 0.1 | 6.0 | 13.8 | 25.2 | 5.4 | 4.5 | 20.3 | 22.6 | 1.0 | 5.9 | 20.5 |
| ETF D100 | E | 0.5 | **0.0** | 1.0 | 3.7 | 0.0 | 0.1 | 0.2 | 5.7 | 0.5 | 3.3 | 0.1 | 6.0 | 13.8 | 25.2 | 2.0 | 4.5 | 20.3 | 22.6 | 1.0 | 5.8 | 20.5 |
| ET D100 | E | 0.7 | **0.0** | 1.0 | 3.7 | 0.0 | 0.1 | 0.2 | 6.0 | 1.1 | 3.4 | 0.2 | 6.4 | 13.9 | 25.5 | 2.1 | 4.8 | 20.7 | 23.2 | 1.2 | 6.0 | 20.8 |
| ET | E | 0.4 | **0.0** | 1.0 | 3.5 | 0.0 | 0.1 | 0.2 | 3.5 | 2.2 | 3.5 | 0.2 | 6.4 | 13.9 | 25.5 | 5.5 | 4.8 | 20.7 | 23.2 | 1.2 | 6.1 | 20.8 |
| ETF D30 | E | 0.4 | **0.0** | 1.0 | 3.6 | 0.0 | 0.1 | 0.1 | 3.4 | 1.6 | 3.4 | 1.2 | 6.0 | 15.6 | 24.9 | 5.4 | 6.8 | 21.4 | 22.3 | 1.0 | 6.2 | 21.1 |
| CLP D30 | E | 0.9 | **0.0** | 0.2 | 3.5 | 0.0 | 0.1 | **0.0** | 4.0 | 1.6 | 3.9 | 3.9 | 5.3 | 16.2 | 24.0 | 1.8 | 4.3 | 20.3 | 23.9 | 0.7 | 6.0 | 21.1 |
| ET D30 | E | 0.4 | **0.0** | 1.1 | 3.5 | 0.0 | 0.1 | 0.1 | 3.5 | 2.2 | 3.5 | 1.3 | 6.4 | 15.8 | 25.1 | 5.5 | 7.0 | 21.7 | 22.6 | 1.0 | 6.4 | 21.3 |
| EMTF D30 | C | 7.8 | **0.0** | 1.4 | 3.5 | 0.4 | 1.7 | 0.9 | 5.0 | 9.3 | 6.7 | 5.4 | 7.6 | 17.8 | 23.5 | 4.1 | 9.6 | 22.4 | 23.0 | 2.9 | 8.1 | 21.7 |
| CLP | E | 0.9 | **0.0** | 0.3 | 3.5 | 0.0 | 0.3 | **0.0** | 4.0 | 1.6 | 3.9 | 3.1 | 5.3 | 17.1 | 24.8 | 1.8 | 4.7 | 21.1 | 24.8 | 0.7 | 6.2 | 22.0 |
| CLP D100 | E | 0.9 | **0.0** | 0.3 | 3.5 | 0.0 | 0.2 | **0.0** | 4.1 | 1.6 | 5.2 | 3.1 | 5.3 | 17.1 | 24.8 | 2.1 | 4.7 | 21.1 | 24.8 | 0.7 | 6.3 | 22.0 |
| WLM D100 | S | 2.1 | **0.0** | 1.0 | 3.0 | 0.7 | 0.1 | 0.1 | 0.4 | 1.1 | 1.7 | 0.0 | 32.6 | 21.3 | 23.0 | 0.0 | **0.2** | 23.2 | 21.0 | 0.2 | 6.9 | 22.1 |
| WLM | S | 5.0 | **0.0** | 1.2 | 3.6 | 0.7 | 4.6 | 0.1 | 0.9 | 3.3 | 3.5 | 0.0 | 3.7 | 21.3 | 23.0 | 0.1 | **0.2** | 23.2 | 21.0 | 0.2 | 6.1 | 22.1 |
| EF D100 | S | 0.8 | **0.0** | 0.2 | 3.1 | **0.0** | 0.1 | 0.8 | 0.2 | **0.0** | 2.6 | 0.3 | 12.7 | 24.2 | 17.6 | 0.8 | 5.3 | 26.9 | 22.4 | **0.0** | 6.2 | 22.8 |
| EF | S | 1.9 | **0.0** | 0.8 | 3.6 | **0.0** | 0.1 | 0.8 | 1.1 | 2.2 | 4.5 | 0.3 | 12.7 | 24.2 | 17.6 | 4.3 | 5.3 | 26.9 | 22.4 | **0.0** | 6.8 | 22.8 |
| EWF D100 | S | 0.8 | **0.0** | 0.2 | 2.5 | **0.0** | 0.1 | 0.5 | 0.5 | 0.5 | 1.9 | 0.1 | 13.0 | 19.9 | 29.3 | 1.6 | 6.2 | 24.3 | 24.0 | 0.2 | 6.6 | 24.4 |
| EWF | S | 2.8 | **0.0** | 0.2 | 3.2 | **0.0** | 0.1 | 0.5 | 2.0 | 2.7 | 4.1 | 0.1 | 13.0 | 19.9 | 29.3 | 9.2 | 6.2 | 24.3 | 24.0 | 0.2 | 7.5 | 24.4 |
| EF D30 | S | 1.9 | **0.0** | 0.8 | 3.6 | 0.0 | 0.2 | 1.6 | 1.1 | 2.2 | 4.5 | 2.8 | 12.7 | 24.7 | 22.9 | 4.3 | 9.6 | 28.6 | 26.2 | 1.4 | 7.8 | 25.6 |
| EWMTF D100 | E | 8.3 | **0.0** | 0.4 | 4.7 | 0.0 | 0.6 | 3.1 | 10.5 | 4.9 | 10.5 | 0.7 | 8.7 | 19.9 | 30.0 | 9.1 | 15.3 | 27.2 | 28.5 | 8.6 | 10.0 | 26.4 |
| EWMTF | E | 7.1 | **0.0** | 0.4 | 4.8 | 0.0 | 0.6 | 3.1 | 11.0 | 4.9 | 10.3 | 0.7 | 8.7 | 19.9 | 30.0 | 14.2 | 15.3 | 27.2 | 28.5 | 8.6 | 10.3 | 26.4 |
| EW | E | 7.1 | **0.0** | 0.4 | 4.8 | 0.0 | 0.8 | 3.1 | 11.0 | 4.9 | 10.3 | 0.7 | 8.7 | 19.9 | 30.0 | 14.2 | 15.3 | 27.2 | 28.5 | 8.6 | 10.3 | 26.4 |
| E | E | 15.1 | **0.0** | 1.0 | 6.8 | 0.2 | 1.4 | 2.9 | 9.0 | 6.6 | 11.3 | 1.5 | 6.9 | 21.4 | 29.0 | 7.6 | 10.4 | 27.3 | 27.9 | 8.1 | 10.2 | 26.4 |
| EMTF D100 | E | 10.6 | **0.0** | 1.5 | 6.5 | 0.2 | 0.8 | 2.9 | 9.1 | 6.0 | 11.4 | 1.5 | 6.9 | 21.4 | 29.0 | 4.3 | 10.4 | 27.3 | 27.9 | 8.1 | 9.8 | 26.4 |
| EMTF | E | 15.1 | **0.0** | 1.6 | 6.8 | 0.2 | 0.8 | 2.9 | 9.0 | 6.6 | 11.3 | 1.5 | 6.9 | 21.4 | 29.0 | 7.6 | 10.4 | 27.3 | 27.9 | 8.1 | 10.2 | 26.4 |
| WLM D100 | C | 8.2 | **0.0** | 2.3 | 4.2 | 2.1 | 2.0 | 4.0 | 1.2 | 3.3 | 4.9 | 0.1 | 0.6 | 24.2 | 26.5 | 0.0 | 0.5 | 28.0 | 28.1 | 1.2 | 7.4 | 26.7 |
| WLM | C | 8.5 | **0.0** | 2.3 | 4.5 | 2.1 | 4.0 | 4.0 | 1.6 | 4.4 | 5.9 | 0.1 | 0.6 | 24.2 | 26.5 | 0.1 | 0.5 | 28.0 | 28.1 | 1.2 | 7.7 | 26.7 |
| WLM D30 | S | 5.0 | **0.0** | 1.2 | 3.6 | 2.0 | 0.8 | 1.2 | 0.9 | 3.3 | 3.5 | 1.3 | 3.7 | 25.3 | 29.0 | 0.1 | 0.7 | 27.2 | 27.1 | 0.5 | 7.2 | 27.2 |
| EWMTF D30 | E | 7.1 | **0.0** | 0.4 | 4.8 | 0.1 | 0.6 | 3.1 | 11.0 | 4.9 | 10.3 | 1.6 | 8.7 | 21.0 | 30.6 | 14.2 | 18.3 | 28.3 | 29.0 | 7.9 | 10.6 | 27.2 |

*Table 26.* (Continued) CSCF Results Across Subsets and Distances (↓ better)

| Method | Dist | BC | BS1 | BS2 | BS3 | BS4 | BS5 | ES1 | HP | HS1 | HS2 | HU1 | HU2 | HU3 | HU4 | HW1 | HW2 | HW3 | HW4 | OC1 | Avg | Avg (Blind) |
|---|---|---|---|---|---|---|---|---|---|---|---|---|---|---|---|---|---|---|---|---|---|---|
| EMTF D30 | E | 15.1 | **0.0** | 1.5 | 6.8 | 0.2 | 0.9 | 2.7 | 9.0 | 6.6 | 11.3 | 2.9 | 6.9 | 23.1 | 29.4 | 7.6 | 13.3 | 28.7 | 28.7 | 8.1 | 10.7 | 27.5 |
| EWF D30 | S | 2.8 | **0.0** | 0.2 | 3.2 | 0.0 | **0.1** | 1.3 | 2.0 | 2.7 | 4.1 | 1.9 | 9.5 | 24.4 | 32.7 | 9.2 | 12.6 | 27.8 | 28.8 | 1.0 | 8.6 | 28.4 |
| EF D100 | C | 6.9 | **0.0** | 1.7 | 4.0 | 0.2 | 0.8 | 5.1 | 5.5 | 6.0 | 7.9 | 4.8 | 15.2 | 26.3 | 29.5 | 4.5 | 11.6 | 31.0 | 29.6 | 5.5 | 10.3 | 29.1 |
| EF | C | 7.3 | **0.0** | 1.7 | 4.2 | 0.2 | 0.8 | 5.1 | 6.1 | 7.7 | 8.4 | 4.8 | 15.2 | 26.3 | 29.5 | 7.0 | 11.6 | 31.0 | 29.6 | 5.5 | 10.6 | 29.1 |
| EWF D100 | C | 7.3 | **0.0** | 0.9 | 4.0 | 0.3 | 0.5 | 4.5 | 5.7 | 9.9 | 7.8 | 4.6 | 16.2 | 25.1 | 32.3 | 10.3 | 16.1 | 29.1 | 31.2 | 3.3 | 11.0 | 29.4 |
| EWF | C | 8.1 | **0.0** | 0.9 | 4.2 | 0.3 | 0.5 | 4.5 | 6.1 | 11.5 | 8.4 | 4.6 | 16.2 | 25.1 | 32.3 | 14.4 | 16.1 | 29.1 | 31.2 | 3.3 | 11.4 | 29.4 |
| WLM D30 | C | 8.5 | **0.0** | 2.3 | 4.5 | 2.7 | 2.5 | 4.7 | 1.6 | 4.4 | 5.9 | 2.0 | 0.6 | 26.5 | 30.8 | 0.1 | 1.0 | 30.4 | 30.7 | 2.1 | 8.5 | 29.6 |
| EF D30 | C | 7.3 | **0.0** | 2.0 | 4.2 | 0.2 | 1.0 | 5.6 | 6.1 | 7.7 | 8.4 | 6.9 | 15.2 | 27.0 | 29.7 | 7.0 | 13.5 | 31.8 | 30.1 | 6.0 | 11.0 | 29.6 |
| EWF D30 | C | 8.1 | **0.0** | 0.9 | 4.2 | 0.5 | 0.7 | 5.1 | 6.1 | 11.5 | 8.4 | 7.1 | 13.2 | 26.2 | 32.6 | 14.4 | 18.4 | 30.4 | 32.0 | 3.1 | 11.7 | 30.3 |
| W2V | S | 9.5 | **0.0** | 3.0 | 7.4 | 0.7 | 7.6 | 1.3 | 5.5 | 11.0 | 5.3 | 0.0 | 28.0 | 27.7 | 29.3 | 0.8 | 3.5 | 31.4 | 35.9 | 3.8 | 11.3 | 31.1 |
| W2V D100 | S | 6.3 | **0.0** | 2.1 | 5.7 | 0.7 | 0.9 | 1.3 | 2.8 | 2.2 | 2.8 | 0.0 | 28.0 | 27.7 | 29.3 | 0.2 | 3.5 | 31.4 | 35.9 | 3.8 | 9.7 | 31.1 |
| MAE | S | 5.0 | **0.0** | 0.4 | 2.9 | 0.3 | 2.3 | **0.0** | 4.5 | 7.1 | 4.6 | 0.2 | **0.0** | 30.2 | 30.4 | 14.4 | 14.4 | 34.5 | - | - | 8.9 | 31.7 |
| MAE D100 | S | 2.4 | **0.0** | 0.4 | **0.7** | 0.3 | **0.1** | **0.0** | 1.6 | 3.8 | 1.5 | 0.2 | - | 30.2 | 30.4 | 8.4 | 14.4 | 34.5 | 36.0 | 0.2 | 9.2 | 32.8 |
| EF D30 | E | 10.6 | **0.0** | 1.5 | 6.8 | 0.2 | 0.2 | 7.5 | 11.4 | 4.9 | 13.3 | 5.1 | 14.0 | 30.0 | 34.2 | 9.2 | 14.6 | 33.8 | 34.6 | 7.9 | 12.6 | 33.1 |
| EF D100 | E | 9.9 | **0.0** | 1.6 | 7.2 | 0.2 | 0.2 | 8.6 | 11.7 | 5.5 | 14.5 | 4.0 | 14.0 | 30.0 | 34.2 | 7.2 | 14.0 | 33.6 | 35.0 | 9.5 | 12.7 | 33.2 |
| EF | E | 10.6 | **0.0** | 1.5 | 6.8 | 0.2 | 0.2 | 8.6 | 11.4 | 4.9 | 13.3 | 4.0 | 14.0 | 30.0 | 34.2 | 9.2 | 14.0 | 33.6 | 35.0 | 9.5 | 12.7 | 33.2 |
| VC | S | 7.3 | **0.0** | 0.7 | 4.3 | 0.0 | 0.7 | 5.7 | 5.1 | 7.1 | 9.4 | 10.2 | 7.6 | 33.4 | 33.5 | 3.1 | 6.9 | 33.3 | 33.6 | 4.0 | 10.8 | 33.5 |
| VC | C | 7.9 | **0.0** | 0.6 | 3.3 | 0.0 | 0.5 | 6.5 | 3.2 | 6.0 | 9.3 | 8.4 | 7.1 | 32.0 | 33.7 | 2.0 | 5.1 | 33.9 | 34.5 | 3.8 | 10.4 | 33.5 |
| EWF | E | 7.8 | **0.0** | 1.0 | 5.8 | 0.1 | 0.8 | 14.1 | 8.4 | 5.5 | 13.1 | 3.1 | 14.5 | 28.3 | 37.1 | 14.0 | 16.2 | 32.7 | 36.2 | 7.1 | 12.9 | 33.6 |
| EWF D100 | E | 11.0 | **0.0** | 1.2 | 6.1 | 0.1 | 0.8 | 14.1 | 8.4 | 4.9 | 13.7 | 3.1 | 14.5 | 28.3 | 37.1 | 11.2 | 16.2 | 32.7 | 36.2 | 7.1 | 13.0 | 33.6 |
| EWF D30 | E | 7.8 | **0.0** | 1.0 | 5.8 | 0.2 | 0.7 | 14.2 | 8.4 | 5.5 | 13.1 | 5.0 | 13.3 | 28.6 | 36.8 | 14.0 | 16.8 | 32.8 | 36.2 | 6.7 | 13.0 | 33.6 |
| W2V D30 | S | 9.5 | 3.3 | 3.0 | 7.4 | 3.0 | 1.3 | 3.2 | 5.5 | 11.0 | 5.3 | 1.2 | 28.0 | 31.4 | 30.6 | 0.8 | 5.8 | 34.8 | 38.0 | 2.6 | 11.9 | 33.7 |
| AC | S | 5.3 | **0.0** | 0.2 | 3.8 | 0.1 | **0.1** | 12.7 | 10.7 | 7.1 | 9.8 | 1.3 | 10.6 | 33.2 | 36.8 | 6.3 | 10.2 | 38.2 | 38.4 | 6.9 | 12.2 | 36.7 |
| M | C | 1.9 | **0.0** | 0.2 | 2.0 | 0.0 | 2.4 | 20.7 | 13.0 | 18.1 | 24.8 | 6.0 | 9.0 | 36.5 | 34.4 | 3.7 | 8.8 | 40.9 | 37.1 | 4.5 | 13.9 | 37.2 |
| MAE D30 | S | 5.0 | **0.0** | 1.3 | 2.9 | 1.4 | 0.6 | 0.2 | 4.5 | 7.1 | 4.6 | 3.9 | - | 33.7 | 36.9 | 14.4 | 19.1 | 40.5 | 39.3 | 0.2 | 12.0 | 37.6 |
| VC | E | 9.4 | **0.0** | 0.3 | 2.8 | 0.0 | 0.3 | 11.5 | 6.1 | 8.8 | 19.9 | 11.7 | 6.6 | 35.6 | 37.2 | 3.3 | 5.4 | 39.5 | 39.6 | 7.9 | 12.9 | 38.0 |
| W2V | C | 13.0 | **0.0** | 4.0 | 7.6 | 3.5 | 2.9 | 8.4 | 13.8 | 11.0 | 10.1 | 0.9 | 6.8 | 38.4 | 33.6 | 1.8 | 9.8 | 40.7 | 40.3 | 7.9 | 13.4 | 38.3 |
| W2V D100 | C | 12.6 | **0.0** | 4.0 | 7.3 | 3.5 | 3.6 | 8.4 | 13.2 | 10.4 | 9.5 | 0.9 | 6.8 | 38.4 | 33.6 | 0.8 | 9.8 | 40.7 | 40.3 | 7.9 | 13.2 | 38.3 |
| WLM D30 | E | 20.6 | **0.0** | 2.5 | 5.4 | 1.7 | 1.5 | 6.5 | 8.7 | 19.2 | 24.6 | 1.9 | 20.3 | 37.8 | 37.9 | 0.1 | 3.9 | 39.6 | 38.2 | 23.1 | 15.5 | 38.4 |
| M | S | 37.8 | **0.0** | 0.5 | 3.4 | 0.1 | 4.5 | 16.1 | 29.5 | 21.4 | 22.8 | 9.7 | 21.2 | 38.8 | 35.5 | 6.5 | 14.3 | 42.1 | 39.7 | 31.9 | 19.8 | 39.0 |
| W2V D30 | C | 13.0 | **0.0** | 4.0 | 7.6 | 4.3 | 3.9 | 8.4 | 13.8 | 11.0 | 10.1 | 0.4 | 6.8 | 39.0 | 34.2 | 1.8 | 11.8 | 41.9 | 41.6 | 5.5 | 13.8 | 39.1 |
| WLM | E | 20.6 | **0.0** | 2.5 | 5.4 | 1.6 | 3.6 | 5.8 | 8.7 | 19.2 | 24.6 | 0.4 | 20.3 | 39.3 | 42.0 | 0.1 | 4.4 | 40.7 | 40.8 | 32.1 | 16.4 | 40.7 |
| WLM D100 | E | 32.7 | **0.0** | 2.5 | 6.2 | 1.6 | 1.1 | 5.8 | 12.7 | 15.9 | 29.2 | 0.4 | 20.3 | 39.3 | 42.0 | 0.0 | 4.4 | 40.7 | 40.8 | 32.1 | 17.3 | 40.7 |
| AC | C | 11.5 | **0.0** | 0.6 | 6.4 | 0.1 | 6.3 | 23.1 | 14.2 | 22.0 | 25.8 | 10.2 | 20.0 | 40.5 | 43.0 | 13.6 | 16.8 | 43.4 | 42.3 | 11.4 | 18.5 | 42.3 |
| CC | C | 10.8 | **0.0** | 1.7 | 11.4 | 3.2 | 9.0 | - | 22.2 | 22.5 | 26.6 | 11.3 | 20.6 | 42.4 | 38.2 | 16.1 | 27.7 | 46.5 | 42.0 | 13.3 | 20.3 | 42.4 |
| AC | E | 10.2 | **0.0** | 0.3 | 6.0 | 0.1 | 3.0 | 11.3 | 13.5 | 6.6 | 19.0 | 7.0 | 15.1 | 40.6 | 44.4 | 12.8 | 16.3 | 45.3 | 43.5 | 14.3 | 16.3 | 43.4 |
| M | E | 33.7 | **0.0** | 0.2 | 5.2 | 0.0 | 3.8 | 11.9 | 18.2 | 12.1 | 28.8 | 4.6 | 27.1 | 42.3 | 44.4 | 6.2 | 23.2 | 44.9 | 43.5 | 20.7 | 19.5 | 43.8 |
| W2V D30 | E | 25.0 | **0.0** | 4.4 | 10.7 | 4.5 | 6.7 | 19.7 | 32.8 | 16.5 | 27.1 | 3.8 | 29.1 | 44.5 | 40.3 | 3.7 | 25.0 | 47.6 | 46.4 | 23.3 | 21.6 | 44.7 |
| W2V D100 | E | 24.7 | **0.0** | 4.0 | 11.1 | 4.1 | 8.6 | 21.1 | 38.6 | 19.2 | 28.9 | 1.3 | 29.1 | 45.8 | 43.6 | 4.3 | 29.5 | 48.5 | 47.5 | 38.8 | 23.6 | 46.3 |
| W2V | E | 25.0 | **0.0** | 4.4 | 10.7 | 4.1 | 14.6 | 21.1 | 32.8 | 16.5 | 27.1 | 1.3 | 29.1 | 45.8 | 43.6 | 3.7 | 29.5 | 48.5 | 47.5 | 38.8 | 23.4 | 46.3 |
| CC | S | 22.2 | 3.3 | 3.7 | 15.5 | 5.1 | 11.8 | - | 28.3 | 33.5 | 30.2 | 17.8 | 25.5 | 46.8 | 42.8 | 22.0 | 30.5 | 50.4 | 48.0 | 20.5 | 25.4 | 47.0 |
| CC | E | 34.1 | **0.0** | 2.3 | 13.3 | 3.7 | 13.3 | - | 38.6 | 33.5 | 35.9 | 16.6 | 31.1 | 47.3 | 43.3 | 22.1 | 35.2 | 50.5 | 50.6 | 36.4 | 28.2 | 47.9 |
| MAE | C | 15.6 | **0.0** | 2.6 | 5.2 | 1.5 | 3.2 | 0.9 | 10.6 | 14.3 | 9.7 | 5.6 | 31.3 | 42.9 | 53.0 | 21.9 | 23.7 | 48.1 | 50.4 | 2.4 | 18.0 | 48.6 |
| MAE D100 | C | 14.0 | **0.0** | 1.8 | 4.7 | 1.5 | 0.7 | 0.9 | 10.3 | 11.0 | 7.4 | 5.6 | 31.3 | 42.9 | 53.0 | 18.9 | 23.7 | 48.1 | 50.4 | 2.4 | 17.3 | 48.6 |
| MAE D30 | C | 15.6 | **0.0** | 2.6 | 5.2 | 2.3 | 1.5 | 1.4 | 10.6 | 14.3 | 9.7 | 10.3 | 31.3 | 44.4 | 53.6 | 21.9 | 25.4 | 49.2 | 50.6 | 2.9 | 18.6 | 49.4 |
| MAE | E | 13.6 | **0.0** | 1.5 | 6.1 | 1.2 | 3.2 | 5.0 | 22.1 | 11.5 | 19.1 | 11.3 | 26.9 | 48.6 | 51.8 | 27.6 | 30.9 | 51.1 | 51.6 | 20.2 | 21.2 | 50.8 |
| MAE D100 | E | 15.1 | **0.0** | 1.5 | 6.6 | 1.2 | 1.5 | 5.0 | 27.0 | 12.1 | 19.3 | 11.3 | 26.9 | 48.6 | 51.8 | 27.0 | 30.9 | 51.1 | 51.6 | 20.2 | 21.5 | 50.8 |
| MAE D30 | E | 13.6 | **0.0** | 2.1 | 6.1 | 1.5 | 1.7 | 3.3 | 22.1 | 11.5 | 19.1 | 14.6 | 26.9 | 48.5 | 52.2 | 27.6 | 30.4 | 51.0 | 51.7 | 10.0 | 20.7 | 50.9 |

*Table 27.* Weighted Purity Results Across Subsets and Distances (↑better)

| Method | Dist | BC | BS1 | BS2 | BS3 | BS4 | BS5 | ES1 | HP | HS1 | HS2 | HU1 | HU2 | HU3 | HU4 | HW1 | HW2 | HW3 | HW4 | OC1 | Avg | Avg (Blind) |
|---|---|---|---|---|---|---|---|---|---|---|---|---|---|---|---|---|---|---|---|---|---|---|
| EWMTF D100 | S | 59.5 | 97.7 | 87.5 | 82.1 | 81.4 | 81.4 | 54.9 | 16.2 | 66.8 | 2.2 | 33.6 | 8.7 | 11.9 | **15.5** | 16.9 | 6.3 | 8.2 | **10.1** | 35.1 | 40.8 | **11.4** |
| EWMTF | S | 51.2 | 97.7 | 72.9 | 73.2 | 81.4 | 81.4 | 54.9 | 11.7 | 66.3 | 11.7 | 33.6 | 8.7 | 11.9 | **15.5** | 1.0 | 6.3 | 8.2 | **10.1** | 35.1 | 38.6 | **11.4** |
| VC | S | 59.2 | 96.6 | 83.8 | 82.0 | 82.6 | 81.4 | 24.8 | 17.8 | 59.4 | 12.7 | 9.9 | 11.3 | 12.3 | 14.5 | 14.0 | 8.1 | **8.9** | 9.1 | 40.4 | 38.4 | 11.2 |
| EWT | S | 56.2 | 98.9 | 90.3 | 83.1 | **89.2** | 86.7 | 52.5 | 11.5 | 64.7 | 13.1 | 2.7 | 9.4 | **13.0** | 14.7 | 6.9 | 6.0 | 7.9 | 9.0 | 32.4 | 39.4 | 11.1 |
| EWT D100 | S | 62.0 | 98.9 | 93.8 | 89.6 | **89.2** | 86.5 | 52.5 | 12.3 | 62.9 | 14.4 | 2.7 | 9.4 | **13.0** | 14.7 | 7.4 | 6.0 | 7.9 | 9.0 | 32.4 | 40.2 | 11.1 |
| EWTF | S | 57.8 | 98.9 | 90.3 | 80.9 | 87.6 | 86.7 | 50.8 | 1.9 | 45.1 | 12.9 | 2.9 | 9.3 | 12.5 | 14.6 | 6.9 | 6.7 | 7.7 | 9.2 | 36.7 | 37.9 | 11.0 |
| EWTF D100 | S | 63.7 | 98.9 | 90.3 | 86.1 | 87.6 | 86.7 | 50.8 | 2.0 | 64.9 | 14.6 | 2.9 | 9.3 | 12.5 | 14.6 | 7.6 | 6.7 | 7.7 | 9.2 | 36.7 | 39.6 | 11.0 |
| ETF | S | 56.2 | 98.7 | 86.1 | 74.2 | 81.2 | 89.0 | 52.9 | 12.6 | 60.8 | 11.6 | 4.0 | 0.8 | 12.5 | 15.2 | 7.9 | 0.1 | 7.5 | 8.1 | 9.1 | 36.2 | 10.8 |
| ETF D100 | S | 64.5 | 98.7 | 86.1 | 88.3 | 81.2 | 89.0 | 52.9 | 1.8 | 63.6 | 2.2 | 4.0 | 0.8 | 12.5 | 15.2 | 9.0 | 0.1 | 7.5 | 8.1 | 9.1 | 36.6 | 10.8 |
| ET | E | 54.4 | 99.2 | 83.6 | 84.3 | 80.3 | 86.9 | 44.5 | 12.4 | 64.6 | 2.5 | 4.1 | 0.8 | 12.2 | 14.3 | 1.0 | 0.1 | 8.1 | 8.6 | 39.2 | 36.9 | 10.8 |
| ET D100 | E | 57.9 | 99.2 | 83.6 | 80.2 | 80.3 | 86.1 | 44.5 | 1.8 | 63.6 | 13.7 | 4.1 | 0.8 | 12.2 | 14.3 | 8.8 | 0.1 | 8.1 | 8.6 | 39.2 | 37.2 | 10.8 |
| ETF | E | 53.4 | 99.2 | 84.0 | 82.9 | 79.7 | 86.9 | 47.6 | 13.2 | 66.2 | 2.6 | 4.2 | 9.6 | 12.3 | 14.5 | 1.0 | 6.1 | 7.5 | 8.6 | 40.6 | 37.9 | 10.7 |
| ETF D100 | E | 58.5 | 99.2 | 70.9 | 88.0 | 79.7 | 86.9 | 47.6 | 1.9 | 65.9 | 13.4 | 4.2 | 9.6 | 12.3 | 14.5 | 8.9 | 6.1 | 7.5 | 8.6 | 40.6 | 38.1 | 10.7 |
| CLP | S | 58.3 | 82.9 | 88.1 | 82.5 | 88.8 | **90.1** | 90.8 | 13.4 | 68.9 | 14.6 | 5.1 | 9.3 | 12.3 | 13.8 | 0.9 | 6.8 | 7.8 | 8.8 | 42.9 | 41.4 | 10.7 |
| CLP D100 | S | 65.5 | 82.9 | 88.1 | 90.5 | 88.8 | 86.9 | 90.8 | 13.9 | 68.3 | 16.9 | 5.1 | 9.3 | 12.3 | 13.8 | 1.2 | 6.8 | 7.8 | 8.8 | 42.9 | 42.1 | 10.7 |
| ETF | C | 55.6 | 99.2 | 70.9 | 75.7 | 78.8 | 87.7 | 47.5 | 12.7 | 66.2 | 2.4 | 4.4 | 0.8 | 11.6 | 14.2 | 8.5 | 0.1 | 7.8 | 8.7 | 35.4 | 36.2 | 10.6 |
| ETF D100 | C | 60.9 | 99.2 | 70.9 | 88.1 | 78.8 | 87.7 | 47.5 | 1.9 | 64.5 | 2.4 | 4.4 | 0.8 | 11.6 | 14.2 | 8.9 | 0.1 | 7.8 | 8.7 | 35.4 | 36.5 | 10.6 |
| EWTF D30 | E | 58.4 | 99.2 | 89.6 | 74.5 | 83.0 | 86.7 | 45.4 | 11.1 | 45.7 | 13.3 | 4.0 | 8.5 | 11.6 | 14.2 | 7.4 | 0.2 | 7.4 | 8.9 | 41.3 | 37.4 | 10.5 |
| EWT | C | 58.5 | 98.9 | 90.6 | 82.8 | 87.2 | 89.9 | 47.0 | 10.1 | 65.3 | 13.2 | 3.2 | 0.8 | 12.4 | 13.5 | 7.2 | 5.0 | 7.4 | 8.7 | 9.1 | 37.4 | 10.5 |
| EWT D100 | C | 47.6 | 98.9 | 91.1 | 87.8 | 87.2 | 87.7 | 47.0 | 11.0 | 66.1 | 14.4 | 3.2 | 0.8 | 12.4 | 13.5 | 7.7 | 5.0 | 7.4 | 8.7 | 9.1 | 37.2 | 10.5 |
| CLP D30 | C | 60.5 | 99.6 | 86.3 | 87.0 | 81.3 | 88.5 | 92.3 | 2.0 | **70.8** | 16.0 | 3.3 | 0.8 | 11.2 | 13.2 | 10.5 | 6.6 | 8.2 | 9.1 | 14.5 | 40.1 | 10.4 |
| EWT D30 | S | 56.2 | 99.2 | 90.3 | 83.1 | 79.8 | 85.4 | 42.9 | 11.5 | 64.7 | 13.1 | 3.6 | 9.4 | 10.9 | 14.4 | 6.9 | 5.9 | 7.7 | 8.7 | 34.5 | 38.3 | 10.4 |
| CLP | C | 60.5 | 99.6 | 86.3 | 87.0 | 85.4 | 87.7 | **92.7** | 2.0 | **70.8** | 16.0 | 6.5 | 0.8 | 11.2 | 14.0 | 10.5 | 6.3 | 7.4 | 8.8 | 42.0 | 41.9 | 10.3 |
| CLP D100 | C | 64.7 | 99.6 | 87.4 | 87.5 | 85.4 | 88.3 | **92.7** | 14.9 | 70.1 | 16.7 | 6.5 | 0.8 | 11.2 | 14.0 | 1.2 | 6.3 | 7.4 | 8.8 | 42.0 | **42.4** | 10.3 |
| MAE | S | 41.4 | 60.9 | 66.0 | 60.8 | 56.5 | 48.3 | 41.1 | 11.4 | 55.6 | 8.5 | 6.3 | **33.8** | 8.6 | 12.0 | 1.0 | 1.0 | 1.0 | 1.0 | 1.0 | 36.5 | 10.3 |
| ETF D30 | S | 56.2 | 98.5 | 80.3 | 74.2 | 74.2 | 84.0 | 43.6 | 12.6 | 60.8 | 11.6 | 5.9 | 0.8 | 10.7 | 14.5 | 7.9 | 5.7 | 7.0 | 8.7 | 40.8 | 36.7 | 10.2 |
| EW | E | 50.1 | 96.4 | 78.6 | 72.5 | 76.6 | 78.4 | 44.6 | 2.5 | 62.0 | 10.5 | 13.7 | 9.3 | 10.9 | 13.6 | 0.9 | 0.4 | 7.4 | 9.0 | 32.9 | 35.3 | 10.2 |
| EWMTF | E | 50.1 | 97.0 | 78.6 | 72.5 | 76.6 | 76.5 | 44.6 | 2.5 | 62.0 | 10.5 | 13.7 | 9.3 | 10.9 | 13.6 | 0.9 | 0.4 | 7.4 | 9.0 | 32.9 | 35.2 | 10.2 |
| EWMTF D100 | E | 49.1 | 97.0 | 78.6 | 75.7 | 76.6 | 76.5 | 44.6 | 2.3 | 62.0 | 11.4 | 13.7 | 9.3 | 10.9 | 13.6 | 0.9 | 0.4 | 7.4 | 9.0 | 32.9 | 35.4 | 10.2 |
| ETF D30 | C | 55.6 | 99.2 | 69.3 | 75.7 | 75.5 | 85.4 | 44.0 | 12.7 | 66.2 | 2.4 | 5.5 | 0.8 | 11.3 | 13.5 | 8.5 | 0.1 | 7.3 | 8.7 | 41.7 | 36.0 | 10.2 |
| CLP D30 | S | 58.3 | 82.9 | 85.3 | 82.5 | 79.3 | 88.0 | 88.5 | 13.4 | 68.9 | 14.6 | 3.0 | 9.3 | 11.3 | 14.0 | 0.9 | 6.3 | 7.2 | 8.3 | 42.4 | 40.2 | 10.2 |
| EWTF | C | 56.6 | 97.5 | 90.6 | 84.3 | 86.3 | 89.9 | 48.7 | 10.9 | 64.9 | 13.8 | 2.9 | 9.1 | 11.6 | 12.3 | 7.3 | 5.3 | 7.7 | 8.9 | 9.1 | 37.8 | 10.1 |
| EWTF D100 | C | 60.1 | 98.9 | 93.4 | 87.0 | 86.3 | 89.9 | 48.7 | 1.9 | 64.4 | 14.1 | 2.9 | 9.1 | 11.6 | 12.3 | 7.6 | 5.3 | 7.7 | 8.9 | 9.1 | 37.8 | 10.1 |
| ETF D30 | E | 53.4 | 99.2 | 79.5 | 82.9 | 76.8 | 88.3 | 41.5 | 13.2 | 66.2 | 2.6 | 5.4 | 9.6 | 10.8 | 13.5 | 1.0 | 5.8 | 7.3 | 8.8 | 39.9 | 37.1 | 10.1 |
| ET D30 | S | 54.1 | 98.7 | 80.2 | 80.9 | 74.4 | 85.9 | 44.9 | 12.0 | 64.6 | 2.5 |  | 9.5 | 11.1 | 13.4 | 7.9 | 5.4 | 7.5 | 8.2 | 42.0 | 36.6 | 10.0 |
| ET D30 | E | 54.4 | 99.2 | 69.1 | 84.3 | 73.8 | 87.2 | 40.8 | 12.4 | 64.6 | 2.5 | 5.8 | 0.8 | 11.6 | 13.7 | 1.0 | 0.1 | 6.7 | 8.1 | 41.3 | 35.6 | 10.0 |
| EWTF D30 | C | 56.6 | 98.9 | 89.0 | 84.3 | 82.6 | 85.7 | 43.9 | 10.9 | 64.9 | 13.8 | 4.4 | 8.4 | 10.7 | 13.3 | 7.3 | 4.7 | 7.2 | 8.5 | 32.2 | 38.3 | 9.9 |
| EMTF D30 | S | 26.3 | 97.7 | 60.5 | 68.6 | 66.9 | 73.8 | 36.8 | 12.8 | 61.6 | 6.7 | 0.7 | 0.8 | 10.2 | 14.5 | 8.2 | 5.5 | 6.7 | 8.1 | 36.5 | 31.7 | 9.9 |
| EWMTF D30 | S | 51.2 | 93.9 | 73.9 | 73.2 | 74.0 | 76.0 | 44.5 | 12.4 | 66.3 | 11.7 | 5.8 | 8.7 | 10.9 | 13.5 | 1.0 | 3.2 | 6.5 | 8.5 | 36.5 | 35.4 | 9.8 |
| E | C | 44.1 | 82.2 | 45.0 | 66.3 | 74.5 | 83.4 | 36.1 | 12.0 | 59.6 | 9.9 | 11.9 | 0.8 | 10.9 | 13.3 | 8.4 | 6.0 | 6.9 | 8.0 | 32.0 | 32.2 | 9.8 |
| EMTF | C | 44.1 | 82.2 | 67.6 | 66.3 | 74.5 | 84.1 | 36.1 | 12.0 | 59.6 | 9.9 | 11.9 | 0.8 | 10.9 | 13.3 | 8.4 | 6.0 | 6.9 | 8.0 | 32.0 | 33.4 | 9.8 |
| EMTF D100 | C | 44.7 | 97.5 | 67.6 | 72.1 | 74.5 | 84.1 | 36.1 | 13.3 | 60.5 | 10.0 | 11.9 | 0.8 | 10.9 | 13.3 | 9.4 | 6.0 | 6.9 | 8.0 | 32.0 | 34.7 | 9.8 |
| E | E | 43.1 | 98.1 | 83.6 | 67.0 | 71.0 | 82.9 | 4.7 | 1.9 | 60.2 | 9.6 | 1.0 | 1.0 | 10.7 | 11.9 | 1.2 | 0.2 | 7.3 | 8.6 | 31.3 | 31.3 | 9.6 |
| EMTF | E | 43.1 | 98.1 | 80.9 | 67.0 | 71.0 | 83.6 | 4.7 | 1.9 | 60.2 | 9.6 | 1.0 | 1.0 | 10.7 | 11.9 | 1.2 | 0.2 | 7.3 | 8.6 | 31.3 | 31.2 | 9.6 |
| EMTF D100 | E | 44.8 | 98.1 | 79.8 | 72.3 | 71.0 | 83.6 | 4.7 | 1.9 | 58.3 | 1.8 | 1.0 | 1.0 | 10.7 | 11.9 | 1.2 | 0.2 | 7.3 | 8.6 | 31.3 | 31.0 | 9.6 |
| EMTF D30 | C | 44.1 | 82.2 | 74.4 | 66.3 | 71.4 | 75.6 | 33.4 | 12.0 | 59.6 | 9.9 | 8.7 | 0.8 | 10.7 | 12.5 | 8.4 | 5.3 | 6.7 | 8.2 | 31.5 | 32.7 | 9.5 |
| EWF D30 | C | 46.6 | 94.9 | 70.6 | 70.4 | 64.3 | 66.3 | 3.5 | 10.2 | 56.9 | 8.4 | 7.1 |  | 8.9 | 13.7 | 6.0 | 4.2 | 6.5 | 8.6 | 37.0 | 31.2 | 9.4 |
| EW | C | 49.2 | 92.6 | 90.6 | 69.8 | 75.1 | 74.9 | 45.8 | 2.1 | 61.8 | 11.0 | 12.4 | 0.9 | 9.3 | 12.1 | 1.0 | 3.2 | 7.3 | 8.9 | 31.1 | 34.7 | 9.4 |
| EWMTF | C | 49.2 | 96.0 | 78.3 | 69.8 | 75.1 | 76.8 | 45.8 | 2.1 | 61.8 | 11.0 | 12.4 | 0.9 | 9.3 | 12.1 | 1.0 | 3.2 | 7.3 | 8.9 | 31.1 | 34.3 | 9.4 |
| EWMTF D100 | C | 51.4 | 93.9 | 78.0 | 74.7 | 75.1 | 76.8 | 45.8 | 2.1 | 61.0 | 12.0 | 12.4 | 0.9 | 9.3 | 12.1 | 1.0 | 3.2 | 7.3 | 8.9 | 31.1 | 34.6 | 9.4 |
| EMTF D30 | E | 43.1 | 97.9 | 75.2 | 67.0 | 71.1 | 75.9 | 35.0 | 1.9 | 60.2 | 9.6 | 9.3 | 1.0 | 10.7 | 11.3 | 1.2 | 0.2 | 7.3 | 8.1 | 32.7 | 32.6 | 9.4 |
| W2V D30 | E | 39.1 | 75.1 | 48.4 | 27.0 | 22.9 | 50.9 | 17.8 | 5.2 | 55.0 | 1.3 | 10.6 | 7.5 | 8.8 | 13.7 | 1.1 | 6.3 | 6.6 | 8.3 | 34.5 | 23.2 | 9.3 |
| EWMTF D30 | C | 49.2 | 96.0 | 74.8 | 69.8 | 72.9 | 75.6 | 39.5 | 2.1 | 61.8 | 11.0 | 0.6 | 0.9 | 10.4 | 11.7 | 1.0 | 3.0 | 6.8 | 8.3 | 32.0 | 33.0 | 9.3 |
| WLM D30 | E | 44.2 | 96.6 | 70.2 | 65.6 | 72.4 | 77.4 | 22.5 | 18.7 | 59.0 | 10.3 | 10.9 | 14.5 | 8.3 | 13.9 | 44.6 | 10.1 | 6.7 | 8.2 | 35.4 | 36.3 | 9.3 |
| WLM D30 | S | 46.5 | 96.8 | 75.6 | 65.8 | 71.0 | 79.0 | 21.9 | 18.6 | 60.9 | 9.8 | 10.4 | 27.0 | 8.7 | 13.3 | 44.5 | 9.5 | 6.3 | 8.6 | 32.4 | 37.2 | 9.2 |
| EWF | C | 46.6 | 96.0 | 70.6 | 70.4 | 69.3 | 68.4 | 3.6 | 10.0 | 56.9 | 1.1 | 6.6 | 6.0 | 8.5 | 13.5 | 6.0 | 4.1 | 6.3 | 8.4 | 36.1 | 31.4 | 9.2 |
| EWF D100 | C | 45.9 | 94.7 | 74.1 | 70.2 | 69.3 | 68.4 | 3.6 | 10.0 | 55.2 | 1.1 | 6.6 | 6.0 | 8.5 | 13.5 | 6.2 | 4.1 | 6.3 | 8.4 | 36.1 | 31.0 | 9.2 |
| EWF D30 | E | 44.9 | 96.2 | 72.8 | 67.3 | 63.6 | 65.3 | 3.6 | 9.1 | 55.7 | 8.6 | 5.1 | 7.1 | 9.2 | 13.2 | 6.0 | 3.5 | 6.3 | 7.9 | 32.9 | 30.4 | 9.1 |
| MAE D30 | S | 41.4 | 89.2 | 50.2 | 60.8 | 45.1 | 44.3 | 36.5 | 11.4 | 55.6 | 8.5 | 1.7 | 1.0 | 9.1 | 13.2 | 6.2 | 5.1 | 6.1 | 8.0 | 28.3 | 28.9 | 9.1 |
| WLM | C | 45.6 | 99.2 | 69.8 | 65.4 | 74.8 | 82.2 | 3.8 | **18.9** | 59.0 | 10.2 | 12.8 | 15.2 | 8.6 | 13.1 | 45.8 | 0.2 | 6.5 | 8.2 | 38.5 | 35.7 | 9.1 |
| WLM D100 | C | 46.7 | 99.2 | 76.9 | 71.1 | 74.8 | 80.6 | 3.8 | 16.9 | 57.8 | 9.7 | 12.8 | 15.2 | 8.6 | 13.1 | 51.6 | 0.2 | 6.5 | 8.2 | 38.5 | 36.4 | 9.1 |
| W2V D30 | S | 39.0 | 75.3 | 49.5 | 27.3 | 23.9 | 53.1 | 17.4 | 5.4 | 49.5 | 1.4 | 0.8 | 6.8 | 8.9 | 12.8 | 15.4 | 6.3 | 6.5 | 7.9 | 32.0 | 23.1 | 9.0 |
| WLM D30 | C | 45.6 | 99.2 | 69.8 | 65.4 | 71.7 | 77.9 | 3.8 | **18.9** | 59.0 | 10.2 | 11.3 | 15.2 | 9.3 | 11.5 | 45.8 | 10.2 | 6.9 | 8.3 | 39.9 | 35.8 | 9.0 |
| EWF | E | 44.9 | 96.8 | 72.8 | 67.3 | 67.0 | 66.2 | 3.7 | 9.1 | 55.7 | 8.6 | 8.0 | 6.4 | 8.9 | 12.5 | 6.0 | 3.7 | 6.3 | 8.3 | 33.3 | 30.8 | 9.0 |
| EWF D100 | E | 45.9 | 96.0 | 74.1 | 71.0 | 67.0 | 66.2 | 3.7 | 9.5 | 55.6 | 8.3 | 8.0 | 6.4 | 8.9 | 12.5 | 5.5 | 3.7 | 6.3 | 8.3 | 33.3 | 31.1 | 9.0 |
| W2V D30 | C | 40.1 | 75.7 | 49.3 | 27.6 | 23.9 | 52.3 | 17.2 | 4.5 | 54.9 | 8.6 | 5.0 | 7.8 | 9.3 | 12.2 | 16.1 | 6.5 | 6.5 | 7.9 | 31.1 | 24.0 | 9.0 |
| M | C | **67.9** | 99.8 | **96.3** | 93.2 | 86.6 | 84.7 | 19.8 | 6.7 | 59.6 | 8.4 | 8.5 | 5.5 | 8.6 | 12.8 | 2.4 | 0.2 | 6.5 | 7.9 | 38.5 | 37.6 | 8.9 |
| WLM | E | 44.2 | 96.6 | 70.2 | 65.6 | 76.2 | 81.3 | 3.8 | 18.7 | 59.0 | 10.3 | 0.5 | 14.5 | 8.5 | 12.7 | 44.6 | 0.2 | 6.5 | 7.9 | 36.3 | 34.6 | 8.9 |
| WLM D100 | E | 45.8 | 99.2 | 77.4 | 72.4 | 76.2 | 80.8 | 3.8 | 15.7 | 43.8 | 10.1 | 0.5 | 14.5 | 8.5 | 12.7 | 50.6 | 0.2 | 6.5 | 7.9 | 36.3 | 34.9 | 8.9 |
| ET | S | 54.1 | 98.7 | 80.2 | 88.7 | 81.6 | 89.0 | 50.0 | 12.6 | 44.9 | 12.0 | 3.7 | 9.5 | 12.4 | 13.7 | 7.9 | 0.1 | 0.6 | 8.8 | 9.1 | 35.7 | 8.9 |
| ET D100 | S | 62.1 | 96.8 | 86.1 | 88.7 | 81.6 | 88.2 | 50.0 | 1.9 | 46.5 | 2.0 | 3.7 | 9.5 | 12.4 | 13.7 | 9.2 | 0.1 | 0.6 | 8.8 | 9.1 | 35.3 | 8.9 |
| MAE D30 | C | 34.1 | 89.9 | 52.9 | 58.0 | 45.7 | 41.0 | 36.1 | 1.7 | 56.3 | 8.7 | 2.3 | 8.0 | 8.5 | 13.2 | 6.9 | 5.3 | 6.2 | 7.5 | 32.0 | 27.1 | 8.8 |
| AC | C | 57.2 | 98.1 | 91.4 | 87.8 | 80.8 | 83.3 | 18.8 | 1.7 | 57.4 | 10.1 | 8.5 | 8.5 | 8.2 | 12.7 | 8.5 | 6.0 | 6.6 | 6.8 | 29.5 | 36.0 | 8.8 |
| CLP D30 | E | 61.9 | 99.6 | 87.9 | 86.9 | 82.9 | 89.7 | 92.5 | 1.8 | 46.0 | 15.9 | 3.4 | 0.8 | 12.1 | 14.0 | 0.9 | 6.5 | 7.5 | 1.6 | 14.5 | 38.2 | 8.8 |
| W2V | C | 40.1 | 75.7 | 49.3 | 27.6 | 23.2 | 53.4 | 17.2 | 4.5 | 54.9 | 8.6 | 10.0 | 7.8 | 8.7 | 12.1 | 16.1 | 6.4 | 6.5 | 7.8 | 31.3 | 24.3 | 8.8 |

*Continued on next page*

*Table 27.* (Continued) Weighted Purity Results Across Subsets and Distances (↑ better)

| Method | Dist | BC | BS1 | BS2 | BS3 | BS4 | BS5 | ES1 | HP | HS1 | HS2 | HU1 | HU2 | HU3 | HU4 | HW1 | HW2 | HW3 | HW4 | OC1 | Avg | Avg (Blind) |
|---|---|---|---|---|---|---|---|---|---|---|---|---|---|---|---|---|---|---|---|---|---|---|
| W2V D100 | C | 39.2 | 77.0 | 49.4 | 27.6 | 23.2 | 55.8 | 17.2 | 5.0 | 54.9 | 8.4 | 10.0 | 7.8 | 8.7 | 12.1 | 21.4 | 6.4 | 6.5 | 7.8 | 31.3 | 24.7 | 8.8 |
| EF D30 | S | 40.4 | 94.5 | 77.4 | 69.0 | 68.0 | 74.9 | 5.5 | 5.3 | 55.4 | 1.4 | 3.0 | 8.7 | 9.3 | 11.8 | 4.2 | 4.1 | 6.2 | 7.6 | 37.6 | 30.8 | 8.7 |
| EWF D30 | S | 45.2 | 94.3 | 70.9 | 68.0 | 64.7 | 68.8 | 6.2 | 10.5 | 57.1 | 1.6 | 4.6 | 6.9 | 9.2 | 11.3 | 6.4 | 4.5 | 6.3 | 8.1 | 39.0 | 30.7 | 8.7 |
| MAE D30 | E | 40.9 | 88.4 | 56.0 | 57.5 | 43.0 | 40.8 | 31.9 | 11.1 | 55.9 | 8.6 | 1.9 | 7.7 | 8.7 | 12.1 | 7.0 | 5.3 | 6.4 | 7.7 | 27.9 | 27.3 | 8.7 |
| EF D30 | E | 44.1 | 91.3 | 75.6 | 69.0 | 67.6 | 74.8 | 3.7 | 3.7 | 55.0 | 8.7 | 2.3 | 7.4 | 9.7 | 11.2 | 3.4 | 2.2 | 5.2 | 8.7 | 33.6 | 30.4 | 8.7 |
| MAE | C | 34.1 | 88.6 | 52.9 | 58.0 | 49.7 | 52.8 | 36.0 | 1.7 | 56.3 | 8.7 | 1.7 | 8.0 | 8.9 | 11.5 | 6.9 | 5.2 | 6.2 | 8.1 | 26.8 | 27.5 | 8.7 |
| MAE D100 | C | 40.1 | 88.6 | 62.6 | 63.3 | 49.7 | 51.4 | 36.0 | 11.6 | 56.4 | 8.8 | 1.7 | 8.0 | 8.9 | 11.5 | 7.1 | 5.2 | 6.2 | 8.1 | 26.8 | 29.1 | 8.7 |
| W2V | E | 39.1 | 75.1 | 48.4 | 27.0 | 22.8 | 53.1 | 16.3 | 5.2 | 55.0 | 1.3 | 11.5 | 7.5 | 8.0 | 11.4 | 1.1 | 6.4 | 6.3 | 8.5 | 26.1 | 22.6 | 8.6 |
| W2V D100 | E | 39.4 | 77.2 | 50.2 | 27.4 | 22.8 | 54.1 | 16.3 | 5.9 | 53.4 | 8.3 | 11.5 | 7.5 | 8.0 | 11.4 | 19.8 | 6.4 | 6.3 | 8.5 | 26.1 | 24.2 | 8.6 |
| EWT | E | 57.1 | 99.2 | 89.8 | 81.6 | 86.1 | 88.9 | 47.2 | 2.0 | 65.1 | 13.8 | 2.8 | 8.8 | 12.2 | 12.6 | 7.2 | 0.2 | 7.6 | 1.5 | 37.9 | 38.0 | 8.4 |
| EWT D100 | E | 60.0 | 99.4 | 90.3 | 88.2 | 86.1 | 88.9 | 47.2 | 11.7 | 45.7 | 14.4 | 2.8 | 8.8 | 12.2 | 12.6 | 8.3 | 0.2 | 7.6 | 1.5 | 37.9 | 38.1 | 8.4 |
| EWTF | E | 58.4 | 99.2 | 89.6 | 74.5 | 88.1 | 88.9 | 47.6 | 11.1 | 45.7 | 13.3 | 3.5 | 8.7 | 1.9 | 14.5 | 7.4 | 0.1 | 7.9 | 9.5 | 37.6 | 37.2 | 8.4 |
| EWTF D100 | E | 60.8 | 99.4 | 92.2 | 87.2 | 88.1 | 88.9 | 47.6 | 1.8 | 63.2 | 15.0 | 3.5 | 8.7 | 1.9 | 14.5 | 7.6 | 0.1 | 7.9 | 9.5 | 37.6 | 38.7 | 8.4 |
| EF | E | 44.1 | 96.0 | 78.0 | 69.0 | 69.8 | 75.3 | 3.7 | 3.7 | 55.0 | 8.7 | 0.7 | 7.4 | 8.6 | 10.4 | 3.4 | 3.9 | 6.3 | 8.4 | 12.2 | 29.7 | 8.4 |
| EF D100 | E | 39.4 | 95.1 | 73.1 | 73.7 | 69.8 | 75.3 | 3.7 | 6.1 | 45.4 | 8.5 | 0.7 | 7.4 | 8.6 | 10.4 | 5.9 | 3.9 | 6.3 | 8.4 | 12.2 | 29.2 | 8.4 |
| EF D30 | C | 42.2 | 93.2 | 77.9 | 70.4 | 69.3 | 74.4 | 3.8 | 6.9 | 56.0 | 8.6 | 1.4 | 6.0 | 6.6 | 12.3 | 4.1 | 2.2 | 6.2 | 8.4 | 34.9 | 30.8 | 8.4 |
| EF | S | 40.4 | 94.5 | 77.4 | 69.0 | 72.7 | 79.1 | 18.4 | 5.3 | 55.4 | 1.4 | 6.4 | 8.7 | 8.7 | 12.7 | 4.2 | 2.9 | 4.8 | 6.8 | 32.0 | 31.6 | 8.3 |
| EF D100 | S | 44.1 | 96.6 | 77.6 | 79.5 | 72.7 | 79.1 | 18.4 | 13.3 | 53.5 | 1.1 | 6.4 | 8.7 | 8.7 | 12.7 | 5.5 | 2.9 | 4.8 | 6.8 | 32.0 | 32.9 | 8.3 |
| MAE | E | 40.9 | 90.9 | 61.7 | 57.5 | 46.7 | 51.7 | 33.9 | 11.1 | 55.9 | 8.6 | 0.8 | 7.7 | 7.4 | 11.6 | 7.0 | 5.2 | 6.2 | 7.8 | 28.6 | 28.5 | 8.2 |
| MAE D100 | E | 23.0 | 88.2 | 61.7 | 62.8 | 46.7 | 50.0 | 33.9 | 11.4 | 45.9 | 1.4 | 0.8 | 7.7 | 7.4 | 11.6 | 7.1 | 5.2 | 6.2 | 7.8 | 28.6 | 26.7 | 8.2 |
| ET D30 | C | 55.5 | 99.2 | 80.0 | 82.4 | 77.0 | 88.8 | 42.6 | 1.8 | 64.3 | 2.1 | 5.4 | 0.9 | 10.5 | 13.6 | 8.0 | 5.9 | 0.6 | 8.3 | 39.5 | 36.1 | 8.2 |
| VC | C | 62.2 | 96.4 | 93.3 | 91.1 | 82.9 | 82.5 | 24.8 | 15.7 | 60.2 | 11.4 | 10.4 | 9.9 | 9.9 | 13.4 | 12.1 | 0.1 | 0.6 | 8.8 | 41.5 | 38.3 | 8.2 |
| EWMTF D30 | E | 50.1 | 96.6 | 74.0 | 72.5 | 73.0 | 74.2 | 43.8 | 2.5 | 62.5 | 10.5 | 0.5 | 9.3 | 10.0 | 13.4 | 0.9 | 2.9 | 0.8 | 8.4 | 32.4 | 33.6 | 8.1 |
| M | E | 66.9 | 99.8 | 96.1 | **93.3** | 87.5 | 84.6 | 20.2 | 11.7 | 57.3 | 1.1 | 2.5 | 0.8 | 6.0 | 12.3 | 9.3 | 6.3 | 6.2 | 7.9 | 36.3 | 37.2 | 8.1 |
| EWF | S | 45.2 | 91.8 | 76.1 | 68.0 | 71.2 | 69.8 | 2.1 | 10.5 | 57.1 | 1.6 | 5.2 | 1.1 | 8.3 | 12.8 | 6.4 | 2.1 | 4.3 | 6.7 | 30.6 | 30.0 | 8.0 |
| EWF D100 | S | 45.9 | 91.8 | 76.1 | 76.2 | 71.2 | 69.8 | 2.1 | 10.6 | 56.0 | 6.2 | 5.2 | 1.1 | 8.3 | 12.8 | 4.7 | 2.1 | 4.3 | 6.7 | 30.6 | 30.6 | 8.0 |
| EWT D30 | E | 57.1 | 97.7 | 89.8 | 81.6 | 82.9 | 85.4 | 42.2 | 2.0 | 65.1 | 13.8 | 4.2 | 8.8 | 10.9 | 5.0 | 7.2 | 0.3 | 7.5 | 9.3 | 32.4 | 37.0 | 8.0 |
| EMTF | S | 26.3 | 97.7 | 84.6 | 68.6 | 75.1 | 85.9 | 46.9 | 12.8 | 61.6 | 6.7 | **37.5** | 0.8 | 2.0 | 13.4 | 8.2 | 7.0 | 7.4 | 9.2 | 37.6 | 36.3 | 8.0 |
| EMTF D100 | S | 57.6 | 98.9 | 84.6 | 87.4 | 75.1 | 85.9 | 46.9 | 17.8 | 65.9 | 2.1 | **37.5** | 0.8 | 2.0 | 13.4 | 20.0 | 7.0 | 7.4 | 9.2 | 37.6 | 39.8 | 8.0 |
| EWT D30 | C | 58.5 | 98.9 | 89.8 | 82.8 | 82.1 | 82.5 | 43.8 | 10.1 | 65.3 | 13.2 | 3.3 | 0.8 | 11.3 | 4.5 | 7.2 | 0.1 | 6.6 | 9.3 | 32.4 | 37.0 | 7.9 |
| EWTF D30 | C | 57.8 | 98.9 | 88.9 | 80.9 | 82.3 | 82.4 | 45.2 | 1.9 | 45.1 | 12.9 | 3.8 | 0.8 | 9.9 | 4.9 | 6.9 | 0.1 | 6.9 | 9.3 | 39.7 | 36.1 | 7.7 |
| AC | S | 62.0 | **100.0** | 92.7 | 87.8 | 82.0 | 83.3 | 2.7 | 11.7 | 57.8 | 1.2 | 9.1 | 8.6 | 9.4 | 13.2 | 8.5 | 6.3 | 6.9 | 1.5 | 9.1 | 34.4 | 7.7 |
| EF | C | 42.2 | 93.2 | 73.7 | 70.4 | 71.5 | 76.5 | 14.8 | 6.9 | 56.0 | 8.6 | 4.3 | 6.0 | 7.3 | 9.0 | 4.1 | 3.9 | 5.8 | 8.3 | 34.7 | 31.4 | 7.6 |
| EF D100 | C | 39.2 | 95.1 | 73.7 | 74.3 | 71.5 | 76.5 | 14.8 | 3.8 | 55.0 | 1.1 | 4.3 | 6.0 | 7.3 | 9.0 | 6.1 | 3.9 | 5.8 | 8.3 | 34.7 | 31.1 | 7.6 |
| VC | E | 62.5 | **100.0** | 94.5 | 92.4 | 85.3 | 80.9 | 25.5 | 2.3 | 59.7 | 11.5 | 9.5 | 1.6 | 13.1 | 11.3 | 7.2 | 0.3 | 8.1 | 7.6 | 37.8 | 37.8 | 7.3 |
| WLM | S | 46.5 | 96.8 | 75.6 | 65.8 | 77.5 | 80.4 | 22.1 | 18.6 | 60.9 | 9.8 | 13.6 | 27.0 | 1.5 | 13.0 | 44.5 | **11.5** | 6.4 | 8.2 | 30.6 | 37.4 | 7.3 |
| WLM D100 | S | 47.5 | 98.7 | 79.4 | 71.2 | 77.5 | 80.4 | 22.1 | 16.2 | 61.3 | 10.4 | 13.6 | 0.9 | 1.5 | 13.0 | **53.5** | **11.5** | 6.4 | 8.2 | 30.6 | 37.0 | 7.3 |
| CLP | E | 61.9 | 99.6 | 88.3 | 86.9 | 84.9 | 88.9 | 92.5 | 1.8 | 46.0 | 15.9 | 6.5 | 0.8 | 12.6 | 13.5 | 0.9 | 6.8 | 0.9 | 1.7 | **43.1** | 39.7 | 7.2 |
| CLP D100 | E | 62.1 | 99.8 | 87.7 | 84.6 | 84.9 | 90.0 | 92.5 | 14.0 | 46.0 | **17.1** | 6.5 | 0.8 | 12.6 | 13.5 | 1.2 | 6.8 | 0.9 | 1.7 | **43.1** | 40.3 | 7.2 |
| M | S | 63.3 | **100.0** | 91.1 | 90.3 | 87.2 | 82.1 | 16.2 | 5.1 | 57.2 | 7.7 | 3.6 | 6.4 | 1.8 | 12.5 | 4.0 | 5.4 | 6.4 | 7.7 | 35.1 | 36.0 | 7.1 |
| MAE D100 | S | 42.2 | 60.9 | 66.0 | 76.4 | 56.5 | 57.4 | 41.1 | 11.5 | 56.2 | 9.0 | 6.3 | 1.0 | 8.6 | 12.0 | 1.0 | 0.1 | 6.2 | 1.5 | 26.1 | 29.9 | 7.1 |
| AC | E | 58.3 | 98.1 | 87.1 | 85.4 | 79.5 | 80.9 | 18.9 | 11.9 | 56.5 | 9.5 | 7.9 | 8.4 | 6.1 | 5.5 | 0.9 | 5.9 | 6.6 | 7.8 | 10.0 | 34.0 | 6.5 |
| ET | C | 55.5 | 99.2 | 84.4 | 82.4 | 79.9 | 87.7 | 48.9 | 1.8 | 64.3 | 2.1 | 4.1 | 0.9 | 2.4 | 13.9 | 8.0 | 0.1 | 7.7 | 1.4 | 37.2 | 35.9 | 6.4 |
| ET D100 | C | 61.2 | 99.2 | 84.4 | 85.2 | 79.9 | 87.8 | 48.9 | 1.9 | 63.5 | 13.4 | 4.1 | 0.9 | 2.4 | 13.9 | 8.7 | 0.1 | 7.7 | 1.4 | 37.2 | 36.9 | 6.4 |
| CC | E | 37.8 | 63.8 | 46.7 | 37.4 | 43.9 | 31.6 | 1.0 | 3.9 | 46.8 | 1.1 | 0.7 | 1.0 | 1.0 | 1.0 | 1.5 | 4.2 | 5.2 | 7.0 | 26.5 | 23.9 | 6.1 |
| W2V | S | 39.0 | 75.3 | 49.5 | 27.3 | 24.9 | 53.5 | 16.2 | 5.4 | 49.5 | 1.4 | 0.6 | 6.8 | 1.5 | 12.3 | 15.4 | 6.8 | 0.6 | 8.1 | 27.2 | 22.2 | 5.6 |
| W2V D100 | S | 39.1 | 61.7 | 47.4 | 27.8 | 24.9 | 58.2 | 16.2 | 5.9 | 46.5 | 7.6 | 0.6 | 6.8 | 1.5 | 12.3 | 22.3 | 6.8 | 0.6 | 8.1 | 27.2 | 22.2 | 5.6 |
| CC | S | 39.0 | 87.3 | 54.5 | 34.6 | 36.2 | 39.1 | 1.0 | 6.0 | 56.0 | 7.0 | 3.7 | 1.0 | 1.0 | 1.0 | 3.4 | 1.8 | 4.2 | 6.8 | 33.6 | 27.5 | 5.5 |
| CC | C | 38.1 | 70.4 | 56.3 | 36.2 | 40.8 | 8.1 | 1.0 | 7.4 | 47.4 | 5.7 | 6.7 | 1.0 | 1.0 | 1.0 | 5.8 | 4.0 | 0.6 | 6.3 | 18.6 | 23.5 | 3.5 |

*Table 28.* Avian Perception Alignment: Triplet Accuracy (High Consistency)

| Method | Triplet Acc. (%) |
| --- | --- |
| EW (C) | **80.9** |
| EW (E) | **80.9** |
| EW (S) | 80.6 |
| EWTF (S) | 74.8 |
| EWF (S) | 73.0 |
| EWTF (C) | 73.0 |
| EWTF (E) | 72.9 |
| EWF (C) | 72.8 |
| EWF (E) | 72.7 |
| EMB-LUA * | 72.7 |
| ETF (D=100, PCA) (E) | 72.1 |
| ETF (E) | 71.7 |
| ETF (S) | 71.7 |
| ETF (C) | 71.6 |
| EF (C) | 71.6 |
| EF (E) | 71.4 |
| E (S) | 71.2 |
| EF (S) | 71.0 |
| E (E) | 70.1 |
| E (C) | 69.9 |
| Luscinia-U * | 69.8 |
| CLAP (C) | 67.6 |
| CLAP (E) | 67.6 |
| CLAP (S) | 67.4 |
| CLP D100 (E) | 67.3 |
| ETF (D=100, PCA) (C) | 66.5 |
| CLP D30 (E) | 66.2 |
| ET (S) | 66.0 |
| Luscinia * | 66.0 |
| ET (C) | 64.8 |
| ET (E) | 64.4 |
| SAP * | 64.0 |
| CC (E) | 62.8 |
| CC (C) | 62.6 |
| CLP D30 (C) | 62.1 |
| CLP D100 (C) | 61.9 |
| CLP D30 (S) | 61.9 |
| M (E) | 61.4 |
| ETF (D=100, PCA) (S) | 60.4 |
| CLP D100 (S) | 60.2 |
| AC (S) | 60.0 |
| CC (S) | 59.2 |
| MAE D100 (E) | 59.0 |
| AC (C) | 58.8 |
| MAE D30 (E) | 58.4 |
| M (C) | 57.9 |
| MAE (S) | 57.7 |
| MAE (C) | 57.3 |

*Continued on next page*

*Table 28.* (Continued) Avian Perception Alignment: Triplet Accuracy (High Consistency)

| Method | Triplet Acc. (%) |
| --- | --- |
| VC (S) | 57.2 |
| WLM (S) | 57.1 |
| MAE (E) | 57.1 |
| Raven ∗ | 57.0 |
| VC (E) | 56.1 |
| MAE D100 (C) | 56.1 |
| MAE D30 (C) | 56.0 |
| WLM (C) | 55.8 |
| MAE D30 (S) | 55.3 |
| VC (C) | 55.3 |
| WLM (E) | 54.9 |
| W2V D100 (S) | 54.3 |
| W2V D30 (C) | 54.3 |
| W2V D100 (C) | 54.2 |
| MAE D100 (S) | 53.9 |
| WLM D30 (C) | 53.7 |
| WLM D100 (C) | 53.7 |
| WLM D30 (S) | 53.6 |
| WLM D100 (E) | 53.3 |
| W2V D30 (S) | 52.9 |
| W2V D100 (E) | 52.0 |
| WLM D100 (S) | 52.0 |
| W2V D30 (E) | 51.9 |
| WLM D30 (E) | 51.8 |
| AC (E) | 51.8 |

[*]Reference values from Zandberg et al. (2024) (Zandberg et al., 2024).

*Table 29.* Predicting mouse strain. Classification accuracy (Top-1, %) and std over 5 splits. Only MLP at $\alpha = 0.001$ is shown.

| Method | k-NN | | | RF | | | MLP ($\alpha = 0.001$) |
|---|---|---|---|---|---|---|---|
| | k=3 | k=10 | k=30 | depth=10 | depth=15 | depth=20 | Top-1 (%) ± std |
| EWTF | 98.00 | 96.96 | 95.14 | 93.25 | 96.12 | 96.57 | 99.49 (0.12) |
| EWTF D30 | 97.19 | 96.30 | 94.82 | 89.87 | 94.86 | 94.99 | 97.90 (0.27) |
| EWTF D100 | 99.04 | 98.61 | 97.77 | 92.77 | 96.18 | 96.40 | 98.77 (0.22) |
| EWT | 98.00 | 96.96 | 95.14 | 93.25 | 96.12 | 96.57 | **99.49 (0.12)** |
| EWT D30 | 97.95 | 97.19 | 95.64 | 89.61 | 94.96 | 95.09 | 98.14 (0.11) |
| EWT D100 | 99.34 | 98.99 | 97.99 | 93.34 | 96.69 | 96.76 | 99.22 (0.13) |
| EWF | 97.18 | 95.82 | 93.73 | 91.49 | 95.25 | 95.80 | 99.18 (0.08) |
| EWF D30 | 96.06 | 94.14 | 91.61 | 89.53 | 94.41 | 94.76 | 95.95 (0.10) |
| EWF D100 | 97.66 | 96.55 | 94.28 | 90.60 | 95.02 | 95.27 | 97.46 (0.18) |
| EWMTF | 97.01 | 95.77 | 93.45 | 92.44 | 95.46 | 95.97 | 99.25 (0.19) |
| EWMTF D30 | 96.89 | 95.75 | 93.67 | 89.38 | 94.43 | 94.57 | 96.79 (0.31) |
| EWMTF D100 | 98.80 | 98.29 | 96.90 | 92.44 | 95.79 | 96.10 | 98.39 (0.09) |
| Spectrogram D=10* | 68.1 | 71.0 | 72.8 | 72.8 | 73.1 | 73.2 | 72.4 (0.4) |
| Spectrogram D=30* | 76.4 | 78.2 | 78.5 | 76.6 | 78.0 | 78.3 | 78.5 (0.8) |
| Spectrogram D=100* | 82.3 | 82.7 | 81.3 | 79.1 | 80.5 | 80.7 | 82.8 (0.1) |
| MUPET D=9* | 86.1 | 87.0 | 86.8 | 87.4 | 87.9 | 87.9 | 87.9 (0.2) |
| DeepSqueak D=10* | 79.0 | 80.7 | 81.0 | 81.2 | 82.1 | 81.9 | 82.4 (0.3) |
| Latent D=7* | 89.8 | 90.7 | 90.3 | 88.1 | 89.6 | 89.6 | 90.4 (0.3) |

Results with (*) are from (Goffinet et al., 2021). Values are Top-1 accuracy (%) ± std.

*Table 30.* Predicting mouse identity. Classification accuracy (Top-1, %) and std over 5 splits.

| Feature Set | $\alpha = 0.01$ | | $\alpha = 0.001$ | | $\alpha = 0.0001$ | |
|---|---|---|---|---|---|---|
| | Top-1 | Top-5 | Top-1 | Top-5 | Top-1 | Top-5 |
| EF (D=100) | 35.0 (0.5) | 65.9 (0.3) | 35.3 (0.5) | 66.0 (0.2) | 35.0 (0.4) | 65.9 (0.4) |
| EF (D=30) | 30.1 (1.1) | 66.0 (0.5) | 30.0 (0.4) | 66.0 (0.4) | 30.4 (0.5) | 66.2 (0.4) |
| ETF (D=100) | 51.9 (0.9) | 79.3 (0.5) | 52.1 (0.4) | 79.5 (0.2) | 52.0 (0.4) | 79.6 (0.2) |
| ETF (D=30) | 41.6 (0.6) | 74.0 (0.4) | 42.5 (0.4) | 74.4 (0.5) | 42.5 (0.2) | 74.3 (0.2) |
| ET (D=100) | **53.1 (0.4)** | **80.0 (0.5)** | 52.8 (0.4) | 79.9 (0.3) | 52.6 (0.5) | 79.8 (0.3) |
| ET (D=30) | 42.8 (1.1) | 74.5 (0.5) | 42.9 (0.4) | 74.5 (0.2) | 42.9 (0.5) | 74.5 (0.4) |
| EMTF (D=100) | 48.2 (0.4) | 76.6 (0.3) | 48.2 (0.5) | 76.6 (0.2) | 47.9 (0.5) | 76.4 (0.2) |
| EMTF (D=30) | 38.3 (0.6) | 71.7 (0.3) | 37.9 (0.7) | 71.5 (0.4) | 38.1 (0.1) | 71.7 (0.2) |
| Spectrogram D=10* | 9.9 (0.2) | 36.6 (0.4) | 10.8 (0.1) | 38.6 (0.2) | 10.7 (0.2) | 38.7 (0.5) |
| Spectrogram D=30* | 14.9 (0.2) | 45.1 (0.5) | 17.3 (0.4) | 50.7 (0.6) | 17.3 (0.3) | 50.8 (0.3) |
| Spectrogram D=100* | 20.4 (0.4) | 55.0 (0.3) | 25.3 (0.3) | 62.9 (0.4) | 25.1 (0.3) | 63.2 (0.4) |
| MUPET D=9* | 14.7 (0.2) | 46.5 (0.3) | 19.0 (0.3) | 54.0 (0.2) | 20.6 (0.4) | 57.3 (0.4) |
| Latent D=8* | 17.0 (0.3) | 49.9 (0.4) | 22.7 (0.5) | 59.2 (0.6) | 24.0 (0.2) | 61.6 (0.4) |

Results with (*) are from (Goffinet et al., 2021). Values are accuracy (%) $\pm$ std over 5 folds.

