# OpenReview forum: "VocSim A Training-free Benchmark for Zero-shot Content Identity in Single-source Audio"
_ICML.cc/2026/Conference — ICML 2026 regular_

### Official Review · Reviewer_zhTd · 2026-03-11

**Soundness:** 3
**Presentation:** 3
**Significance:** 2
**Originality:** 2
**Overall Recommendation:** 3
**Confidence:** 5

**Summary:**

This paper introduces **VocSim**, a largely training-free benchmark intended to probe the neighborhood structure of frozen audio embeddings for zero-shot content-identity retrieval across human speech, animal vocalizations, and environmental sounds. VocSim aggregates 125,382 single-source clips from 19 corpora and evaluates representations using a standardized pipeline: statistical time-frequency pooling over frozen encoder outputs and a transductive, label-free PCA/whitening step. Performance is measured via Precision@k (local neighborhood purity) and a proposed **Global Separation Rate (GSR)**, interpreted with permutation-based chance baselines. Across models including Whisper, WavLM, BEATs, and CLAP, the study reports that frozen Whisper encoder features with simple pooling (and PCA) provide a strong off-the-shelf baseline on public subsets. The authors also highlight a generalization gap on blind low-resource language sets (Shipibo-Conibo and Chintang), where P@1 drops from \~67% to \~11.5%, and argue based on a dynamic-range proxy analysis that the gap is more consistent with unseen phonotactic/phonetic structure than acoustic degradation. Finally, the paper includes external evaluations (e.g., HEAR, avian perceptual similarity, and fine-grained bioacoustic classification) to support the practical relevance of VocSim.

**Compliance With Llm Reviewing Policy:**

Affirmed.

**Final Justification:**

I thank the authors for the detailed clarification of VocSim's positioning as a diagnostic benchmark for a specific upstream property rather than a modality-wide suite. Combined with the first-round domain-disaggregated results and the PCA ablation, this reframing substantively strengthens the paper's core construct.

However, even accepting the "evaluation axis" analogy to MTEB at the protocol level, MTEB's influence derives from comprehensive task coverage and sustained community adoption; under its revised scope, VocSim's narrower focus on vocal content identity leaves open whether the methodological axis alone can deliver comparable impact.

Given the clarifications, I am raising my scores to 3 (weak reject).

**Key Questions For Authors:**

**1. Domain-disaggregated rankings (speech vs non-speech)** Could the authors provide a disaggregated analysis separating speech-pretrained models (Whisper, WavLM) from non-speech models (CLAP, BEATs) on (a) speech subsets and (b) non-speech subsets (animal + environmental, and ideally animal vs environmental separately), using the same pipeline and metrics (P@1/P@5, GSR and lift-over-permutation)? Please also quantify rank stability.

*Impact on Evaluation* If rankings remain largely stable across domains, it would strengthen the claim that VocSim measures broadly useful embedding geometry rather than domain-match effects. If rankings diverge substantially, I would expect the paper to revise its “general-purpose” framing and make domain-specific reporting a primary outcome rather than an appendix detail.

**2. Sensitivity to environmental-audio weighting / coverage.** Environmental audio appears under-represented relative to speech and bird vocalizations. Could the authors perform a mixture-weight sensitivity analysis where you (i) upweight the environmental subset(s) during evaluation (e.g., macro-average over domains or reweighting per-subset contributions), and/or (ii) augment environmental coverage with a larger benchmark (e.g., FSD50K or an AudioSet-derived single-source subset) to test whether the headline conclusions (e.g., the “default recipe” and model ordering) persist?

*Impact on Evaluation* If supplementary results demonstrate that the model hierarchy remains stable with broader environmental coverage, this would alleviate the significance concern regarding benchmark coverage

**3. Clarifying what GSR measures across structurally different subsets.**  The permutation baseline for GSR varies considerably across subsets (33.7% on Blind vs. 24.9% on Public), driven by differences in class count and embedding density. While the authors report lift over baseline as a correction, is there a theoretical or empirical justification that equal lift values across structurally different subsets correspond to equivalent levels of geometric quality? For instance, have the authors examined whether normalizing GSR by a dataset-complexity measure (e.g., class count, samples-per-class ratio) yields more stable cross-subset comparisons, or whether rank-ordering of models by GSR lift remains consistent across subsets with varying structural properties?

*Impact on Evaluation* If GSR-based conclusions and rankings are stable under class/density controls and alternative normalizations, it would strengthen confidence in GSR as a cross-subset metric

**Limitations:**

The most consequential limitation is the mismatch between VocSim’s “general-purpose audio representation” framing and its evaluation design. The benchmark ranks speech-pretrained models (e.g., Whisper, WavLM) and non-speech models (e.g., CLAP, BEATs) on a unified leaderboard using an identical pipeline, without explicitly accounting for the different pre-training domains and objectives these model families target. This confound affects the paper’s central narratives: the low-resource “generalization gap” aggregates models not intended to encode phonetic structure with dedicated speech encoders, so macro-averaged OOD results can conflate expected domain mismatch with genuine generalization failure. Without domain-stratified evaluation, conclusions about “current SOTA models” remain difficult to attribute.

This concern is amplified by dataset imbalance. Environmental sounds are represented by a single small subset (ESC-50; 2,000 clips, ~1.6% of total), while bird vocalizations span multiple subsets totaling roughly 39k clips. The blind OOD sets are also limited to low-resource speech, with no analogous blind evaluation for non-speech domains. As a result, the aggregate leaderboard and headline findings (e.g., Whisper dominance and the OOD gap) are largely driven by speech and birdsong, reducing ecological validity for environmental sound retrieval, acoustic monitoring, and other non-speech applications. Expanding environmental coverage and introducing domain-stratified reporting would be necessary for VocSim to fully support the general-purpose diagnostic role it claims.

**Strengths And Weaknesses:**

**Soundness**

**_Strengths:_** The experimental setup is carefully designed to target the intended construct. Restricting evaluation to single-source audio reduces confounding from source separation, making the benchmark closer to measuring content-identity structure in embeddings. The statistical methodology is strong, including per-subset 95% bootstrap confidence intervals, permutation-based baselines for GSR (lift over chance), label-noise robustness checks, and a layer-wise sensitivity sweep for Whisper. External evaluations on HEAR, zebra finch perceptual similarity, and mouse USV classification further suggest that the proposed zero-shot geometric metrics correlate with downstream utility rather than merely reflecting internal benchmark artifacts.

**_Weaknesses:_** A key concern is that the benchmark does not sufficiently disentangle intrinsic representational geometry from domain and objective alignment across model families. Speech-centric models (Whisper, WavLM) and non-speech or audio-text models (CLAP, BEATs) are evaluated under a single pipeline and summarized on a unified leaderboard, making it hard to separate geometric quality from distributional match; Whisper’s strong performance on speech-heavy subsets may largely reflect pretraining proximity. This issue also affects the “generalization gap” narrative on low-resource languages: near-chance performance for models not trained for phonetic structure (e.g., CLAP/BEATs) is expected, so aggregating them can overstate the claim that “current SOTA” fails on low-resource speech. Finally, while GSR is an interesting asymmetric separation metric, its permutation baseline varies substantially across subsets (e.g., Blind vs. Public), implying sensitivity to class structure and embedding density. Reporting lift helps, but it remains unclear whether equal lift values across structurally different subsets reflect comparable geometric quality, which limits GSR’s reliability for cross-subset comparisons.

**Presentation**

**_Strengths:_** The paper is clearly motivated and well organized. The introduction cleanly contrasts adaptability-focused benchmarks (e.g., HEAR, SUPERB) with the goal of diagnosing frozen-embedding geometry, and the positioning relative to MTEB/VTAB helps contextualize the contribution. Figure 1 provides a compact overview of dataset composition, the feature-extraction pipeline, and the evaluation metrics. Overall writing is technically precise, and the results section progresses coherently from main benchmark findings to the OOD analysis and external evaluations.

**_Weaknesses:_** The main limitation is that the core results are not sufficiently disaggregated by domain and subset in the main text. Table 2 reports only Public/Blind macro-averages, which collapses cross-domain behavior and risks over-emphasizing a “Whisper dominates” narrative without showing how rankings change on environmental versus bird subsets, or how audio-pretrained models compare specifically on speech. While detailed breakdowns appear in the appendix, the body does not surface these domain-specific trends, which is particularly important given the dataset’s skew toward speech and bird vocalizations and the paper’s emphasis on broad “audio” benchmarking.

**Significance**

**_Strengths:_** The paper targets a timely gap in audio representation evaluation. Unlike HEAR/SUPERB-style benchmarks that mix representation quality with fine-tuning or probing choices, VocSim provides a largely training-free axis for assessing frozen embeddings, which is relevant for retrieval/indexing and fast model selection. The inclusion of blind OOD sets from non-public low-resource language recordings (Shipibo-Conibo, Chintang) is also valuable as an evaluation protocol that reduces the risk of pretraining leakage and encourages more realistic out-of-distribution testing. The server-side evaluation approach, motivated by data sovereignty concerns, is a positive precedent for benchmarking with sensitive or community-owned data. Finally, the additional external evaluations (e.g., HEAR and bioacoustic/perceptual settings) help motivate that the benchmark signals correlate with downstream utility.

**_Weaknesses:_** The benchmark coverage is meaningfully imbalanced relative to its stated scope. Environmental audio is represented by a single small subset (ESC-50; 2,000 clips, ~1.6% of total), while bird vocalizations span multiple subsets totaling ~39k clips, making aggregate results heavily driven by speech and birdsong and limiting credibility as a general-purpose audio diagnostic for environmental sound practitioners. Expanding environmental coverage (e.g., with FSD50K or larger event datasets) would broaden applicability and reduce composition-driven rankings. In addition, the impact of the reported low-resource generalization gap is partially blurred by domain alignment: pooling non-speech models with speech-pretrained models in blind-set evaluation makes it less clear what the gap implies for speech-focused deployment, and separating these model families would yield a sharper, more actionable takeaway.

**Originality**

**_Strengths:_** The paper brings a relatively new evaluation philosophy to audio: training-free, zero-shot probing of frozen embedding geometry at scale, analogous to MTEB/VTAB-style diagnostics but not yet standardized in audio. The proposed GSR metric is also a novel, retrieval-motivated design that penalizes boundary breaches via nearest inter-class distance, with permutation baselines providing a dataset-aware chance reference.

**_Weaknesses:_** The originality of a unified “general-purpose audio” benchmark is less compelling if speech and non-speech domains are not explicitly modeled or separated in reporting; collapsing heterogeneous model families into a single leaderboard risks reducing the contribution to an aggregation of existing domain-specific evaluations rather than a genuinely new cross-domain framework. In addition, the “training-free, intrinsic geometry” framing is somewhat blurred by the transductive PCA/whitening step fitted on test-set statistics. Since PCA yields nontrivial gains on public subsets but little effect on blind sets, it is unclear whether the benchmark primarily measures raw embedding geometry or the extent to which representations benefit from transductive correction; clearer attribution/ablation would strengthen the claim.

---

> ### Author Rebuttal · Authors · 2026-03-26
>
> We thank Reviewer zhTd for the careful and rigorous evaluation. Our intended claim is not that VocSim is a balanced benchmark over all audio domains, but that it provides a training-free diagnostic across diverse *single-source* corpora. We agree that broad phrases such as "general-purpose audio" may have made these boundaries less salient than intended. In the revision, we will tighten this wording and surface domain-stratified summaries more prominently in the main text, as shown below. The manuscript already states several scope boundaries: (i) the protocol is defined as *gradient-free* and *transductive* rather than strict single-sample inference (Introduction; Section 4), (ii) the blind sets are explicitly low-resource *speech* and the limitations note that blind evaluation is currently restricted to speech, and (iii) the appendix notes that the benchmark is weighted toward speech and bioacoustics.
>
> **Q1: Domain-disaggregated rankings**
>
> To directly test whether aggregate results reflect domain match, we computed domain-disaggregated summaries from the per-subset appendix tables.
>
> P@1 by domain (foundation models):
>
> | Model | Animal (7) | Env. (1) | Speech Pub. (7) | Speech Blind (4) |
> |---|---|---|---|---|
> | Whisper D100 | 85.7 | 75.9 | 46.5 | **11.5** |
> | CLAP D100 | **88.4** | **95.7** | 34.4 | 8.1 |
> | BEATs D100 | 87.3 | 95.1 | 37.0 | 11.4 |
> | WavLM D100 | 80.6 | 34.0 | **51.8** | 4.6 |
> | EAT CLS | 78.1 | 43.9 | 23.3 | 1.9 |
>
> GSR by domain:
>
> | Model | Animal (7) | Env. (1) | Speech Pub. (7) | Speech Blind (4) |
> |---|---|---|---|---|
> | Whisper D100 | **44.3** | 41.5 | **39.2** | **39.4** |
> | CLAP D100 | 41.4 | **50.2** | 33.2 | 36.2 |
> | BEATs D100 | 33.1 | 46.9 | 27.6 | 34.7 |
> | WavLM D100 | 38.1 | 37.0 | 36.0 | 35.8 |
> | EAT CLS | 30.7 | 25.7 | 18.5 | 22.6 |
>
> P@1 is quite domain-sensitive, as expected from differences in pretraining alignment: CLAP ranks highest on animal/environmental sounds, WavLM on public speech, Whisper on blind speech. GSR is more stable: Whisper leads in 3/4 domains, and CLAP's sole GSR lead (environmental) is on ESC-50, where AudioSet overlap is expected. Local retrieval purity reflects domain/objective alignment more strongly, while global separation is comparatively stable across domains.
>
> The blind speech result is particularly informative because it is a *within-domain* failure mode: Whisper drops from 46.5% P@1 on public speech to 11.4% on Shipibo-Conibo/Chintang, despite both being human speech. This collapse is not explained solely by a coarse "speech vs. non-speech" domain match argument.
>
> On the unified leaderboard: its purpose is to provide a common protocol over frozen embeddings, not to imply that all model families target the same pretraining objective. We will include domain-stratified summaries in the main text revision.
>
> **Q2: Environmental coverage**
>
> We agree that environmental coverage is limited, and this is already noted in the manuscript's appendix/limitations. ESC-50 constitutes &lt;2% of clips and should be interpreted as a robustness check rather than the primary basis for environmental-audio claims. We will revise the abstract, introduction, and discussion to frame VocSim as a benchmark for *vocal* content identity (speech and animal vocalizations), with ESC-50 retained as a non-vocal robustness check. To make this scope explicit from the outset, we propose revising the title to *VocSim: A Training-free Benchmark for Zero-shot Vocal Identity in Single-source Audio*. A 4-domain macro-average weighting each domain equally yields an identical GSR ranking (Whisper > CLAP > WavLM > BEATs > EAT), confirming that conclusions are not driven by dataset composition. We welcome the reviewer's feedback on whether this revised scope adequately addresses the concern.
>
> **Q3: GSR cross-subset comparability**
>
> The reviewer asks whether GSR ranking is stable across subsets with very different structural properties. Empirically, yes: pairwise Kendall's τ of GSR model rankings across the four domains averages 0.60 (0.87 excluding the single-subset environmental domain), despite class counts ranging from 6 to 1,366. We do not claim that equal *raw* lift values imply identical geometric quality in a universal sense. Rather, lift over permutation provides a dataset-specific chance reference so that GSR is interpreted relative to each subset's point cloud structure. We will clarify this in the revision.
>
> **Q4: Transductive PCA**
>
> The manuscript already notes that PCA is transductive. Table 2 provides the key ablation: PCA improves Public P@1 by +5.3, but has zero effect on Blind sets (P@1: 11.5 and GSR: 39.4, both unchanged). Thus, the main blind-set conclusion does *not* depend on transductive correction. On public data, PCA mitigates anisotropy; on blind OOD data, it does not recover additional structure. We therefore view PCA as a label-free calibration step that helps expose public-set geometry, while the strongest OOD conclusion is unchanged without it.

---

> > ### Author Rebuttal · Reviewer_zhTd · 2026-04-01
> >
> > I appreciate the thorough and candid rebuttal.
> >
> > However, the rebuttal's proposed resolution to Q2 raises a follow-up concern that bears directly on the paper's significance. The authors propose narrowing VocSim's scope from "general-purpose audio" to "vocal content identity," but the current manuscript's significance framing is substantially built on the analogy to MTEB and VTAB as general-purpose evaluation suites for their respective modalities. MTEB derives its value precisely from comprehensive coverage across diverse text tasks (retrieval, classification, clustering, reranking), and VTAB from spanning natural, specialized, and structured visual domains. If VocSim is repositioned as a vocal identity benchmark with environmental sounds serving only as a robustness check, the parallel to these suites becomes considerably less precise: the benchmark would be closer in scope to a domain-specific evaluation (analogous to, say, a speaker verification benchmark) than to a modality-wide diagnostic. Could the authors clarify whether they view the MTEB/VTAB analogy as still holding under the revised scope, and if so, what principled basis distinguishes VocSim from existing domain-specific evaluations (e.g., acoustic word embedding benchmarks for speech, BirdSet/BIRB for bioacoustics) that it would subsume?
> >
> > Given the above, I maintain my current scores pending the authors' responses to these follow-up question. hould a revised manuscript successfully integrate the domain-stratified analysis into the main text and provide a convincing justification for the benchmark's scope and positioning relative to existing domain-specific evaluations, I would be open to revisiting the assessments.

---

> > > ### Author Response · Authors · 2026-04-03
> > >
> > > We thank the reviewer for this important follow-up, which helps us clarify VocSim’s scope and positioning.
> > >
> > > Our proposed wording revision (“vocal content identity”) is intended to make the framing more precise, not to reduce the empirical coverage of the benchmark. VocSim continues to evaluate frozen embeddings on speech, animal vocalizations, and environmental sounds under a shared protocol. The purpose of the revision is to describe more accurately the primary representational property we are probing.
> > >
> > > We also agree that the MTEB/VTAB analogy should be stated more carefully. We do not claim comparable task breadth or modality-wide coverage. The intended parallel is methodological and protocol-level: MTEB standardized evaluation of off-the-shelf text embeddings as a distinct axis alongside task-specific adaptation, and VocSim does the same for frozen audio embeddings under a shared zero-shot retrieval-geometry protocol. It is therefore not a modality-wide audio suite analogous to MTEB/VTAB in scope, but a benchmark for a specific upstream representational property that existing audio benchmarks typically do not directly measure.
> > >
> > > This is also the key distinction from existing audio benchmarks. HEAR and SUPERB evaluate representations after linear probing or fine-tuning, and therefore measure downstream adaptability under supervision rather than the frozen geometry already present in the representation. VocSim instead asks a different upstream question: whether frozen embeddings already organize acoustically variable instances of the same underlying content into a retrievable structure before any task-specific adaptation.
> > >
> > > The same distinction applies to domain-specific evaluations. Acoustic word embedding benchmarks are typically speech-specific and focus on lexical or phonetic retrieval. BirdSet and BIRB evaluate supervised bioacoustic classification and robustness under domain shift. VocSim instead measures one shared upstream property under a common protocol: content-identity geometry in single-source audio, without labels, gradient updates, or domain-specific tuning.
> > >
> > > The main distinction is therefore not coverage breadth, but the property being measured. By applying the same frozen-model protocol, pooling, and geometric evaluation across speech, animal vocalizations, and a limited environmental robustness check, VocSim probes a common representational property that is not directly exposed by any single existing domain-specific benchmark. Our rebuttal analysis further suggests partial rank stability across domains, indicating that this property is not reducible to domain-specific alignment alone.
> > >
> > > We therefore position VocSim not as a modality-wide suite analogous to MTEB in scope, but as a diagnostic benchmark for a specific upstream property that existing benchmarks typically assume rather than measure directly. In the revision, we will make this positioning explicit by: (1) refining the introduction, abstract, and discussion to state the benchmark’s scope and boundaries more precisely; (2) replacing the broad MTEB/VTAB framing with the narrower methodological “evaluation axis” analogy above; (3) moving the domain-stratified P@1 and GSR summaries, together with the equal-weight macro-average analysis, into the main results section; and (4) expanding the Related Work discussion to clarify how VocSim complements AWE benchmarks and BirdSet/BIRB rather than subsuming them.

---

### Official Review · Reviewer_b75m · 2026-03-11

**Soundness:** 3
**Presentation:** 3
**Significance:** 3
**Originality:** 2
**Overall Recommendation:** 5
**Confidence:** 3

**Summary:**

The paper introduces VocSim, a benchmark designed to evaluate whether audio representations capture content identity across acoustically different recordings. Instead of measuring performance through supervised training (linear probes), the benchmark evaluates frozen embeddings and measures how well they cluster or align representations corresponding to the same content. The main idea is to probe the intrinsic geometry of pretrained audio embeddings: if two recordings contain the same underlying content, their representations should remain close in embedding space even under acoustic variations. Since the evaluation is training-free, the benchmark allows direct comparison of representation models without additional fine-tuning. The authors evaluate several existing audio representation models under this framework and additionally show that performance on VocSim correlates with results on standard probing benchmarks. In particular, the best performing method on VocSim also achieves the best performance on the HEAR benchmark.

**Compliance With Llm Reviewing Policy:**

Affirmed.

**Final Justification:**

The paper addresses an important and timely problem. The proposed benchmark is simple, efficient, and offers a useful alternative to downstream classifier-based evaluation, making it easier to study the intrinsic quality of audio representations. The experimental section is thorough and convincing. My main concerns were relatively minor and mostly related to clarifications about the evaluation setup and robustness. The rebuttal addressed these points adequately and reinforced my positive view of the paper.

**Key Questions For Authors:**

- The use of PCA on the test set is not completely clear to me. How much does this affect the results? It would be helpful to report some results with and without this step.

**Limitations:**

Yes

**Strengths And Weaknesses:**

**Strengths**

- The idea of evaluating audio representations without training a downstream classifier is valuable. Many existing benchmarks rely on linear probes or fine-tuning, which can make it difficult to isolate the quality of the representation itself.

- The benchmark is simple and computationally efficient, as it only requires computing embeddings and similarity scores.

- The paper introduces a new metric, Global Separation Rate, to quantify the separation between classes in the embedding space.

- The dataset used in the benchmark is reasonably large, spans multiple categories, and seems well designed.

- The experimental section is fairly comprehensive, with many baselines and detailed evaluations. The authors also state that the benchmark and code will be made publicly available.

- The work also provides some insights into the out-of-distribution gap of audio foundation models.

- Presentation is overall clear.


**Weaknesses**

- The methodological novelty is limited. The benchmark mainly relies on similarity comparisons in embedding space. Related ideas have already been explored in representation analysis (clustering or retrieval-style evaluations).

- As a minor presentation detail, it would be better to save figures in vector graphic formats such as PDF to preserve image quality.

---

> ### Author Rebuttal · Authors · 2026-03-26
>
> We thank Reviewer b75m for the positive assessment and the accept recommendation.
>
> **Q1: Effect of PCA**
>
> Table 2 already reports results with and without PCA for all models. For Whisper EWMTF D100 vs. EWMTF Raw:
>
> | Setting | Public P@1 | Public GSR | Blind P@1 | Blind GSR |
> |---|---|---|---|---|
> | With PCA (D100) | 66.8 | 41.7 | 11.5 | 39.4 |
> | Without PCA (Raw) | 61.5 | 40.2 | 11.5 | 39.4 |
> | Δ | +5.3 | +1.5 | 0.0 | 0.0 |
>
> Thus, PCA consistently improves public-set performance by mitigating anisotropy, but has zero effect on the blind OOD sets. We view this as a useful geometric diagnostic in itself: on public data, PCA reveals structure partially obscured by the representation cone; on blind OOD data, PCA does not recover additional structure. The benchmark is therefore informative both with and without PCA, and the main blind-set conclusion is unchanged without it.
>
> **Q2: Limited methodological novelty**
>
> We appreciate this point. We see the contribution of VocSim primarily in the integration: carefully curated single-source data across 19 corpora, blind OOD evaluation, the GSR metric with permutation calibration, overlap auditing, and external validation on HEAR, avian perception, and mouse USVs. Taken together, these components provide a diagnostic resource that did not previously exist for audio representation evaluation.
>
> We also appreciate the suggestion regarding figure formats and will convert all raster figures to vector PDF format in the revision where possible.

---

> > ### Author Rebuttal · Reviewer_b75m · 2026-04-03
> >
> > Thank you for your rebuttal, your clarifications address my concerns. I will maintain my original positive score.

---

### Official Review · Reviewer_f1M9 · 2026-03-12

**Soundness:** 4
**Presentation:** 3
**Significance:** 4
**Originality:** 3
**Overall Recommendation:** 4
**Confidence:** 3

**Summary:**

This paper proposes VocSim, a training-free benchmark for evaluating the inherent geometric quality of frozen audio foundation models. Unlike benchmarks such as HEAR and SUPERB that focus on linear probing or fine-tuning adaptability, VocSim aims to assess whether a model can map sounds with the same content identity to a close representation space without updating parameters. The benchmark aggregates 19 data subsets covering human speech, animal vocalizations and environmental sounds, and strictly restricts to single-source audio to decouple "content identity representation" from "source separation capability" as much as possible. Methodologically, it adopts P@1, P@5 and the newly proposed GSR metric to measure local retrieval purity and global inter-class separability, with calibration against a permutation baseline. Experiments show that Whisper-Large-v3 combined with time-frequency statistical pooling and label-free PCA achieves the best performance on public sets, yet its local retrieval performance drops significantly from approximately 66.8% P@1 to around 11.5% on the blind out-of-distribution (OOD) test set for low-resource languages, revealing a clear shortcoming of current audio foundation models in truly cross-lingual generalization. The external validity of the benchmark is also verified through HEAR, avian perception prediction and rodent vocalization classification. Overall, this study fills the gap in the evaluation of the inherent geometric quality of audio foundation models with rigorous design and significant value, meeting academic publication standards and requiring only minor revisions for improvement.

**Compliance With Llm Reviewing Policy:**

Affirmed.

**Key Questions For Authors:**

1. Boundaries of the wording "training-free / zero-shot"
The paper uses per-subset label-free PCA based on target test set statistics and selects optimal distance metrics for different model categories. Please define more clearly: in your view, what is the exact difference between this setup and strict single-sample zero-shot inference? It is recommended to explicitly distinguish between "gradient-free zero-shot" and "transductive zero-shot evaluation" in the title, abstract and conclusions, to avoid the misinterpretation that the setup enables fully plug-and-play deployment without any target domain statistical information.

2. Pretraining overlap confounding in public set results
The appendix has honestly provided an overlap audit, but the discussion of public set rankings in the main text is still overly positive. It is recommended to supplement a more systematic analysis, such as reporting results hierarchically by subsets with confirmed/likely/no overlap, or at least explicitly downplaying the generalization interpretation of public set rankings in the main tables and conclusions. Otherwise, it is easy to misinterpret public set performance as real OOD capability.

3. Extrapolation scope of blind OOD conclusions
Current blind OOD tests are almost entirely based on low-resource human speech, not animal or environmental sounds. Therefore, the extrapolation of the paper’s conclusions on "general audio content identity" should be more cautious. It is recommended to clearly distinguish in the main text which conclusions pertain to "cross-lingual speech OOD" and which can be extended to more general audio content representation.

4. Coupling between metrics and data structure
Figure 2 indicates that P@1/P@5 are sensitive to the number of categories and samples per category. Please further explain whether macro-averaging across subsets is sufficient for fair comparison, or whether more standardized/hierarchical reporting should be provided to prevent model performance comparisons from being partially dominated by the structure of data subsets.

5. Interpretation of comparisons with AE/VAE baselines
The appendix states that AEs/VAEs are trained individually for each subset (including blind subsets). Please more clearly explain whether these baselines serve as an "unsupervised upper bound for domain adaptation" or a "fairly comparable general baseline" in the paper, and it is recommended to avoid equating this setup with the direct use of frozen general models in the main text.

**Limitations:**

yes

**Strengths And Weaknesses:**

Strengths
1. Important and clearly positioned research question: The paper explicitly points out that most existing audio benchmarks measure "adaptability" rather than the geometric quality of frozen representations themselves, which is of practical significance, especially for retrieval and indexing applications.
2. Targeted benchmark design: The paper exclusively uses single-source audio and emphasizes decoupling content representation from the source separation problem. This design philosophy is rational and makes the diagnostic objectives of the benchmark more focused.
3. Broad data coverage with rigorous blind OOD tests: The 19 subsets span speech, animal and environmental sounds, with the additional inclusion of blind test sets for two low-resource languages (Shipibo-Conibo and Chintang), which is one of the most valuable contributions of this paper.
4. Adequate metric design and analysis: The motivation for GSR is clearly elaborated, and its robustness and complementarity with local metrics are demonstrated through permutation baseline, label noise experiments and other analyses.
5. Enhanced persuasiveness via external validation: Beyond just scoring with the benchmark, the paper further demonstrates correlations with HEAR, avian perception judgment and rodent vocalization classification, which makes the argument that "inherent geometric quality can predict downstream utility" more credible.
6. Strong ethical awareness: The paper conducts a serious and specific discussion on low-resource language bias and data sovereignty issues, and adopts server-side evaluation to protect non-public data.

Weaknesses
1. Controversial wording of "training-free / zero-shot": Although no gradient updates are performed, per-subset transductive PCA based on target test set statistics is used, and an "optimal distance metric" is selected for different model categories. This makes the setup not strictly a one-shot plug-and-play zero-shot, and is closer to transductive evaluation. Although this is explained in the paper, the title and main narrative may lead readers to overestimate the actual zero-shot applicability.
2. Pretraining overlap confounding in public set results: The paper acknowledges in the appendix that multiple public subsets have confirmed/likely overlap with the pretraining data of the tested models, meaning that high scores on public sets cannot be fully interpreted as genuine generalization ability. While blind OOD tests are thus added, the interpretation of "public set rankings" in the main text can be more cautious.
3. Blind OOD currently limited mainly to low-resource human speech: Therefore, the paper’s conclusions on "general audio content identity" are supported primarily by speech OOD evidence, rather than true blind OOD on animal or environmental sounds.
4. Local metrics highly affected by data structure: Figure 2 shows that P@1/P@5 are sensitive to the number of categories and samples per category, so it is worth further demonstrating or supplementing hierarchical statistics on whether macro-averaging across subsets is sufficiently fair.
5. Slightly asymmetric baseline setup: The appendix notes that AEs/VAEs are trained individually per subset (including blind subsets). While this can be regarded as an "unsupervised upper bound for domain adaptation", it is not exactly the same usage scenario as "directly evaluating frozen general models", and the comparison purpose should be more clearly defined in the main text.

---

> ### Author Rebuttal · Authors · 2026-03-26
>
> We thank Reviewer f1M9 for the thorough and balanced evaluation, and for recognizing VocSim's contributions.
>
> **Q1: "Training-free / zero-shot" wording**
>
> We appreciate this point. The manuscript already distinguishes our setup from strict single-sample inference: Section 1 explicitly defines the protocol as *gradient-free* and states that the PCA step is a *transductive*, label-free normalization on test-set statistics. No labels are used, and no parameters are updated by backpropagation. Our use of "training-free" therefore refers to the absence of gradient-based adaptation, and "zero-shot" to the absence of task-specific supervision.
>
> We agree, however, that this distinction could be made more prominent to avoid readers equating our protocol with fully plug-and-play single-instance inference. In the revision, we will make this wording more explicit in the abstract and conclusion. More briefly, our setting is best described as *gradient-free, label-free, transductive zero-shot evaluation*.
>
> On distance metric selection: the metric is selected *per model family*, not per subset, so no per-subset tuning is performed. We report all three metrics (cosine, Euclidean, Spearman) in the Appendix; the main table uses the strongest metric per family for compactness. We will clarify this reporting protocol more explicitly.
>
> **Q2: Pretraining overlap**
>
> The manuscript already includes a systematic overlap audit in Table 5, distinguishing confirmed overlap, likely overlap, and no evidence of overlap. We agree that public-set results should be interpreted cautiously, and we can make this more prominent in the main text.
>
> At the same time, the overlap audit also shows that animal-vocalization subsets have no confirmed overlap for all models, yet the benchmark still produces strong and discriminative results there (e.g., CLAP: 88.4% P@1, Whisper: 85.7%). This suggests that VocSim captures representational quality beyond direct corpus memorization. In the revision, we will foreground the overlap categories more clearly when discussing public-set rankings.
>
> **Q3: Blind OOD scope**
>
> We agree that the blind OOD conclusion should be stated narrowly. The manuscript already intends this conclusion to pertain specifically to *cross-lingual speech OOD*, and we will make that scope more explicit in the main text and conclusion.
>
> For animal vocalizations, the zero-confirmed-overlap status provides useful no-overlap evidence, but we agree this is not equivalent to blind non-public OOD evaluation. Blind server-side evaluation requires non-public corpora by definition. We will therefore distinguish more clearly between (i) strict blind OOD evidence for low-resource speech and (ii) indirect no-confirmed-overlap evidence for animal vocalizations.
>
> **Q4: P@k sensitivity**
>
> This sensitivity is precisely why VocSim includes both local and global metrics. Figure 2 shows that P@1/P@5 degrade as class count increases, while GSR is substantially more stable. Our domain-disaggregated analysis in the response to Reviewer zhTd further shows that GSR ranking is preserved across subsets spanning 6–1,366 classes. We agree that macro-averaged P@k has limitations, and we will make that caveat more explicit; the role of GSR is to provide a complementary diagnostic that is less sensitive to subset structure.
>
> **Q5: AE/VAE baselines**
>
> These are intended as *domain-specific unsupervised baselines*, not as directly comparable frozen general-purpose baselines. The manuscript already describes them as an "upper bound for domain-specific unsupervised learning" and notes that the VAE baselines perform worse than chance on blind sets (Lift −7.8 in the Appendix). Their role is to quantify the benefit of large-scale pretraining relative to unlabeled, domain-adapted learning from scratch. We will make this comparison purpose even more explicit in the revision.

---

> > ### Author Rebuttal · Reviewer_f1M9 · 2026-04-02
> >
> > 1.Regarding the reproducibility of experimental details: The paper adopts label-free PCA as a key step. Could you briefly clarify whether this step is applied to the entire VocSim dataset or independently to each subset?
> > 2.Regarding model lightweight discussion: The experimental results show that Whisper-large-v3 achieves excellent performance. Considering the demand for computational resources in practical applications, could you briefly add a discussion on whether the proposed method still works well on smaller Whisper variants?
> > 3.Regarding the scale of the blind test set: The blind test dataset is critical for verifying model generalization ability. Could you clarify the total duration of the blind speech data used in the benchmark?
> > 4.Regarding the choice of distance metric: In Table 2, different distance metrics (Spearman correlation S and Euclidean distance E) lead to different results, and Whisper and CLAP perform significantly better under Spearman than other baselines. Could you add a sentence explaining why Spearman correlation is more suitable for the feature space of pre-trained speech models such as Whisper and CLAP in your experimental setup?

---

> > > ### Author Response · Authors · 2026-04-03
> > >
> > > We thank the reviewer for the helpful follow-up questions and for the opportunity to clarify these implementation details.
> > >
> > > **(1) PCA application.**
> > > The label-free PCA/whitening step is applied **independently for each evaluation subset**, not once over the full VocSim collection. We agree that this should be stated more explicitly in the main text for reproducibility, and we will revise the manuscript accordingly.
> > >
> > > **(2) Smaller Whisper variants.**
> > > We agree that model size is an important practical consideration. In the current submission, we report results for **Whisper-large-v3** as the representative Whisper model. We have now also run the same evaluation pipeline on **Whisper-small**. In our current sweep, the best Whisper-small configuration is **EWMTF with cosine distance**. The comparison to Whisper-large-v3 is summarized below.
> > >
> > > | Model | Public P@1 | Δ vs Large | Public P@5 | Δ vs Large | Public GSR | Δ vs Large | Blind P@1 | Δ vs Large | Blind P@5 | Δ vs Large | Blind GSR | Δ vs Large |
> > > |---|---:|---:|---:|---:|---:|---:|---:|---:|---:|---:|---:|---:|
> > > | Whisper-large-v3 | 66.8 | — | 57.4 | — | 41.7 | — | 11.5 | — | 7.7 | — | 39.4 | — |
> > > | Whisper-small | 57.4 | -9.4 | 49.3 | -8.1 | 28.7 | -13.0 | 5.7 | -5.8 | 3.8 | -3.9 | 29.1 | -10.3 |
> > >
> > > The same pipeline applies without modification at smaller scale, but absolute performance drops substantially, particularly in global separation (GSR −13.0 on public sets) and on the blind OOD subsets. Within the Whisper family, this suggests that model capacity materially affects global geometric quality. We will include these smaller-variant results in the revision to make the performance-efficiency trade-off more explicit.
> > >
> > > **(3) Total duration of the blind speech data.**
> > > The blind speech portion of the benchmark contains approximately **3.15 hours** of audio in total. We agree that this is a useful dataset statistic, and in the revision we will report aggregate duration statistics systematically for all subsets, not only for the blind split.
> > >
> > > **(4) Why Spearman works well for Whisper/CLAP.**
> > > We already note this point in the manuscript, but we agree that the explanation could be made more explicit. Spearman compares the relative ordering of feature dimensions rather than raw magnitudes, which makes it less sensitive to anisotropy and scale differences across dimensions and appears to reduce hubness-related effects in pooled transformer embeddings (Radovanović et al., 2010), where a few "hub" points become nearest neighbors of disproportionately many others in high-dimensional Euclidean or cosine spaces. We will clarify this intuition more directly in the revision.

---

### Official Review · Reviewer_dgdh · 2026-03-12

**Soundness:** 3
**Presentation:** 3
**Significance:** 3
**Originality:** 3
**Overall Recommendation:** 4
**Confidence:** 4

**Summary:**

This paper introduces VOCSIM, a training-free benchmark designed to evaluate the intrinsic geometric quality of general-purpose audio representations. While traditional evaluation methods often rely on fine-tuning which can obscure the actual performance of the underlying embeddings, VOCSIM focuses on how well frozen models can group semantically similar audio clips in a zero-shot setting. The benchmark aggregates a large and diverse collection of single-source audio clips across human speech, animal vocalizations, and environmental sounds. By restricting the data to single-source recordings, the authors isolate the quality of content representation from the complexities of source separation. The study also proposes a new metric to measure category separation and demonstrates that even state-of-the-art models face a significant performance gap when encountering out-of-distribution data such as low-resource languages. Overall, this work provides a diagnostic tool for assessing the immediate utility of audio embeddings and highlights critical generalization challenges for future research in the field.

**Compliance With Llm Reviewing Policy:**

Affirmed.

**Key Questions For Authors:**

The benchmark relies on transductive whitening PCA calculated over the entire test set. How would the performance rankings and absolute scores change if an inductive approach were used—for instance, if the whitening parameters were estimated from a separate, disjoint validation set or a standard reference corpus (like Audioset) instead of the test data?

Could you provide a quantitative comparison or a theoretical justification for why the Global Separation Rate (GSR) is more informative than established intrinsic metrics such as the Silhouette Coefficient or Davies-Bouldin Index? Specifically, how does the permutation-based calibration in GSR provide diagnostic insights that standard distance-based clustering metrics fail to capture?

The choice to use single-source audio is well-motivated for variable isolation. However, have the authors tested whether the relative rankings of these models remain consistent when evaluated on data with overlapping sounds or background noise?

**Limitations:**

yes

**Strengths And Weaknesses:**

The paper demonstrates strong technical soundness through the construction of a massive and diverse dataset that spans nineteen different corpora including human speech, animal vocalizations, and environmental sounds. By focusing exclusively on single-source audio clips, the authors successfully isolate the quality of content representation from the confounding effects of source separation or background noise, which allows for a much cleaner analysis of the intrinsic embedding geometry. Furthermore, the work moves beyond the standard practice of evaluating models through fine-tuning, which often hides the actual limitations or biases of the learned features behind the optimization process. The discovery of a massive performance drop when models encounter low-resource languages is a crucial insight that challenges the claimed universality of current state-of-the-art representations.

Weakness:

While the newly proposed GSR metric incorporates statistical calibration, it is essentially a distance-based separation analysis. The paper does not sufficiently demonstrate the unique physical meaning or diagnostic insights that GSR provides over established clustering metric.

Although the paper shows a correlation between VOCSIM scores and the HEAR benchmark, it remains to be evaluated whether zero-shot retrieval performance can fully represent complex tasks requiring temporal modeling or strong supervision, such as audio event detection or audio captioning.

The benchmark is strictly limited to single-source audio. While this helps in isolating variables, it neglects common real-world scenarios involving polyphony and complex background noise, which require further clarification and discussion.

---

> ### Author Rebuttal · Authors · 2026-03-26
>
> We thank Reviewer dgdh for the positive and constructive evaluation.
>
> **Q1: Inductive whitening (PCA from a separate corpus)**
>
> The reviewer asks how results would change if whitening were estimated from a disjoint corpus rather than the target subset. We did not run a separate-corpus PCA ablation, but Table 2 provides a conservative bound by reporting results *without PCA entirely*. Two observations are important:
>
> - Blind-set results are unchanged: Whisper Blind P@1 remains 11.5 and Blind GSR remains 39.4 with or without PCA. The central OOD conclusion therefore does not depend on transductive whitening.
> - On public sets, removing PCA slightly changes the top-tier P@1 ordering (WavLM Raw 62.8, CLAP Raw 61.7, Whisper Raw 61.5), but Whisper remains #1 on the more stable GSR metric (40.2 vs. 35.2 for WavLM and 33.8 for CLAP).
>
> Removing PCA entirely is a conservative bound on the effect of whitening, since it eliminates normalization rather than fitting it on another source. This suggests that the choice of corpus for whitening may affect absolute public-set scores somewhat, but does not drive the blind-set conclusion. The practical purpose of PCA here is to correct the anisotropy cone common in transformer representations, not to perform learned adaptation.
>
> **Q2: GSR vs. Silhouette / Davies-Bouldin**
>
> GSR is retrieval-motivated by design. Silhouette compares intra-cluster cohesion to the *average* distance to the nearest cluster; Davies–Bouldin similarly relies on average cluster scatter and centroid separation. GSR instead compares average intra-class distance to the *single nearest* inter-class point (NID). This asymmetry is deliberate:
>
> - Average inter-class measures can mask "leaks": a class may look well-separated on average while still having a confusing nearest boundary violation.
> - GSR's use of the minimum inter-class distance strictly penalizes such boundary breaches, which directly correspond to false positives in retrieval.
>
> Empirically, GSR and Silhouette are correlated (ρ = 0.82; Table 6), but not redundant. We observe cases in noisy bioacoustic subsets where Silhouette remains relatively high because dense clusters are preserved on average, while GSR drops because it detects nearest-neighbor leakage. This is exactly the failure mode that matters in top-1 retrieval. In addition, GSR is paired with a permutation baseline, so its value is interpreted relative to the structural difficulty of each subset.
>
> **Q3: Polyphonic/noisy data**
>
> VocSim deliberately excludes polyphonic mixtures because the benchmark is designed to isolate content representation from source separation. Evaluating raw embeddings on overlapping mixtures would conflate these two abilities. We therefore view single-source and polyphonic evaluation as complementary rather than competing benchmark designs, analogous to object classification versus scene understanding in vision. We have not tested whether model rankings remain consistent under polyphony/noise, and we will make this out-of-scope boundary more explicit in the revision.
>
> **Additional clarification: retrieval geometry vs. temporal tasks**
>
> We agree that VocSim measures retrieval geometry rather than full temporal modeling. For that reason, we frame VocSim as complementary to, not a replacement for, task-specific benchmarks. At the same time, Table 3 shows that the same embeddings that score best on VocSim also achieve state-of-the-art results on 5/7 HEAR tasks, including temporally structured tasks such as Speech Commands and CREMA-D. This supports the claim that intrinsic geometric quality is a useful predictor of downstream utility, even though it does not exhaustively characterize every aspect of temporal processing.

---

> > ### Author Rebuttal · Reviewer_dgdh · 2026-04-03
> >
> > Thanks, I will maintain my score.

---

### Decision · Program_Chairs · 2026-04-30

**Decision:**

Accept (regular)

**Comment:**

While three reviewers are positive and find the contribution solid and timely, one reviewer raised a weak reject concern mainly regarding the clarity and strength of the empirical validation. The authors’ rebuttal partially addresses this concern and clarifies the intended scope of the claims. Given the overall positive assessments, the consensus on the paper’s value, and the fact that the remaining concern does not outweigh the contributions, I recommend acceptance.